# Risk Phase Transitions in Spiked Regression: Alignment Driven Benign and Catastrophic Overfitting

**Jiping Li**
Department of Mathematics
University of California, Los Angeles
jipingli0324@g.ucla.edu

**Rishi Sonthalia**
Department of Mathematics
Boston College
rishi.sonthalia@bc.edu

## Abstract

This paper analyzes the generalization error of minimum-norm interpolating solutions in linear regression using spiked covariance data models. The paper characterizes how varying spike strengths and target-spike alignments can affect risk, especially in overparameterized settings. The study presents an exact expression for the generalization error, leading to a comprehensive classification of benign, tempered, and catastrophic overfitting regimes based on spike strength, the aspect ratio $c = d/n$ (particularly as $c \to \infty$), and target alignment. Notably, in well-specified aligned problems, increasing spike strength can surprisingly induce catastrophic overfitting before achieving benign overfitting. The paper also reveals that target-spike alignment is not always advantageous, identifying specific, sometimes counterintuitive, conditions for its benefit or detriment. Alignment with the spike being detrimental is empirically demonstrated to persist in nonlinear models.

## 1 Introduction

Understanding the generalization error of overparameterized models is a central challenge in modern machine learning. Phenomena such as double descent (Belkin et al., 2019; Hastie et al., 2022) and benign overfitting (Bartlett et al., 2020; Mallinar et al., 2022; Tsigler and Bartlett, 2023) have spurred research underscoring the critical role of the data's spectral structure (Bartlett et al., 2020; Dobriban and Wager, 2018; Hastie et al., 2022; Kausik et al., 2024; Mei et al., 2022; Sonthalia and Nadakuditi, 2023; Tsigler and Bartlett, 2023; Wang et al., 2024a). The spiked covariance model is one commonly considered spectral structure Couillet and Liao (2022). In this model, the data matrix $\boldsymbol{X} = \boldsymbol{Z} + \boldsymbol{A} \in \mathbb{R}^{d \times n}$, comprising $n$ data points in $\mathbb{R}^d$, is decomposed into a rank-one signal component ("spike") $\boldsymbol{Z}$ and an isotropic noise component ("bulk") $\boldsymbol{A}$. Spiked covariance models emerge naturally in practice, for instance, in the features learned by neural networks during training Sonthalia et al. (2025); Ba et al. (2022; 2023); Damian et al. (2022); Dandi et al. (2024); Martin and Mahoney (2021); Moniri et al. (2023); Wang et al. (2024b). While recent studies have examined benign overfitting in spiked models (Ba et al., 2023; Kausik et al., 2024), they lack a systematic taxonomy spanning spike strength, target–spike alignment, model misspecification, and train–test covariate shift. This paper closes the gap for linear regression.

This work explores how general spike sizes and target alignments affect generalization error in least squares linear regression. We consider targets $y$ generated by:

$$y = \alpha_Z \boldsymbol{\beta}_*^\top \boldsymbol{z} + \alpha_A \boldsymbol{\beta}_*^\top \boldsymbol{a} + \varepsilon$$

Here, $\boldsymbol{z} \in \mathbb{R}^d$ represents the signal (spike) component, $\boldsymbol{a} \in \mathbb{R}^d$ corresponds to the noise (bulk) component, $\varepsilon$ is observation noise, and $\boldsymbol{\beta}_* \in \mathbb{R}^d$. The coefficients $\alpha_Z$ and $\alpha_A$ model the target's dependence on the spike and bulk components. Notably, if $\alpha_A \neq \alpha_Z$, the targets are non-linear functions of $\boldsymbol{x} = \boldsymbol{z} + \boldsymbol{a}$, introducing model mis-specification. We address two fundamental questions:

- **Q1:** For a fixed aspect ratio $c = d/n$, in the asympototically proportional regime, under what conditions does alignment of the target signal with the data spike improve or impair generalization?

- **Q2:** In the high-dimensional limit where $c \to \infty$, when do we observe benign, tempered, or catastrophic overfitting regimes?

**Contributions.** We present precise characterization of the generalization performance of minimum-norm interpolating solutions in linear regression. Our exact risk decomposition pinpoints conditions for transitions between benign and catastrophic overfitting. This reveals alignment-dependent phenomena obscured by isotropic theories, clarifying how signal structure, data scaling, and overparameterization shape generalization. Our primary contributions are as follows:

- **Precise Risk Characterization:** We derive an exact generalization error decomposition (Theorem 5) into interpretable bias, variance, data noise, and alignment terms.

- **Comprehensive Categorization of Overfitting Regimes:** We precisely classify benign, tempered, or catastrophic overfitting regimes based on spike strength, overparameterization ($c = d/n$), and target alignment (Table 1). Surprisingly, for well-specified aligned problems, increasing spike strength can induce catastrophic overfitting before achieving benign overfitting. Misspecified problems show distinct transitions, often precluding benign overfitting.

- **Conditions for Beneficial Alignment:** Challenging conventional wisdom, we show spike alignment is not always beneficial and depends on spike strength meeting critical thresholds (Table 2). For misspecified problems, beneficial alignment requires $\alpha_Z/\alpha_A$ in a specific, non-trivial range. Counterintuitively, very strong spike dependence ($\alpha_Z/\alpha_A$) can render alignment detrimental.

- **Empirical Validation:** [1] Empirical validation confirms our theoretical phenomena, including surprising negative alignment impacts, persist in nonlinear models, underscoring broader relevance.

**Benign Overfitting in Linear Regression.** Significant research has explored benign overfitting in linear regression (Bartlett et al., 2020; Cao et al., 2021; Chatterji and Long, 2021; Karhadkar et al., 2024; Koehler et al., 2021; Liang and Rakhlin, 2020; Mallinar et al., 2022; Muthukumar et al., 2020; Shamir, 2022; Tsigler and Bartlett, 2023; Wu and Xu, 2020). Many studies assume a uniformly bounded largest covariance eigenvalue or lack precise characterizations of its interplay with target alignment and generalization. *Our work allows this eigenvalue to grow, offering precise performance characterizations based on this growth and alignment.* While Kausik et al. (2024) considers spiked models, their focus is on noiseless, well-specified scenarios with specific spike scaling. *Our analysis is broader, encompassing observation noise, misspecification, and general spike scaling.*

Many prior works (Karhadkar et al., 2024; Shamir, 2022; Tsigler and Bartlett, 2023) on benign overfitting with low-rank signals plus isotropic noise require near-orthogonality between signal and noise, sometimes imposing strong conditions like $d = \Omega(n^2 \log n)$. *We instead consider the proportional regime $d/n \to c = \Theta(1)$, subsequently examining $c \to \infty$.* This setting is morally similar to allowing $d = \omega(n)$ and aligns with approaches like (Karhadkar et al., 2024) which, for classification, shows misclassification probability can be upper bounded by $Ce^{-d/n}$, vanishing as $d/n \to \infty$.

**Generalization Error with Spiked Covariance.** While recovering spike properties (Sonthalia and Nadakuditi, 2023; Kausik et al., 2024; Nadakuditi, 2014; Benaych-Georges and Nadakuditi, 2011; 2012) and analyzing generalization error in spiked models (Kobak et al., 2020; Ba et al., 2022; 2023; Mousavi-Hosseini et al., 2023; Moniri et al., 2023; Freeman, 2025) are active research areas, existing analyses often characterize generalization implicitly (e.g., via fixed-point equations) or focus on specific spike strengths/alignments. *In contrast, we provide explicit, generic formulae for generalization error, enabling precise categorization of overfitting regimes and conditions for beneficial spike alignment.*

We further discuss connections to selected prior works, direct theoretical extensions to other settings, and proof techniques in Appendix C.

**Notation** The subscript on $o, O, \omega, \Omega, \Theta$ will denote which quantity is being sent to infinity.

---

[1] Our code is available at the anonymous GitHub repository: link

**Table 1: Asymptotic Generalization Regimes.** This table summarizes conditions for when overfitting is benign, tempered, or catastrophic in the limit where $d/n \to c$ and subsequently $c \to \infty$. The behavior depends on the spike scaling relative to the bulk, target alignment ($\boldsymbol{\beta}_*$ relative to spike direction $\boldsymbol{u}$), and target specifications $\alpha_A, \alpha_Z$ (train) and $\tilde{\alpha}_A, \tilde{\alpha}_Z$ (test). Here, $\theta^2$ quantifies the scaled spike strength and $\tau^2$ the scaled bulk variance; the two primary scaling regimes are operator norm based ($\theta^2 = \gamma\tau^2$) and Frobenius norm based ($\theta^2 = d\tau^2$). The $\omega, o, O, \Theta$ are all as we send $c \to \infty$.

| Scaling | Benign | Tempered | Catastrophic |
|---|---|---|---|
| **Well-Specified, No Covariate Shift:** $\alpha_A = \tilde{\alpha}_A = \alpha_Z = \tilde{\alpha}_Z = \alpha > 0$ | | | |
| $\theta^2 = \gamma\tau^2$ | $\gamma = \omega_c(c^2), \boldsymbol{\beta}_* \parallel \boldsymbol{u}$ | All other cases | $o_c(c^2) \geq \gamma \geq \omega_c(1), \boldsymbol{\beta}_* \not\perp \boldsymbol{u}$ |
| $\theta^2 = d\tau^2$ | $\boldsymbol{\beta}_* \parallel \boldsymbol{u}$ | $\boldsymbol{\beta}_* \nparallel \boldsymbol{u}$ | Never |
| **Misspecified, No Covariate Shift:** $\alpha_A = \tilde{\alpha}_A, \alpha_Z = \tilde{\alpha}_Z, \alpha_A \neq \alpha_Z$ | | | |
| $\theta^2 = \gamma\tau^2$ | Never | All other cases | $o_c(c^2) \geq \gamma \geq \omega_c(1), \boldsymbol{\beta}_* \not\perp \boldsymbol{u}$ |
| $\theta^2 = d\tau^2$ | Never | Always | Never |
| **Misspecified with Covariate Shift:** $\alpha_A \neq \tilde{\alpha}_A$ or $\alpha_Z \neq \tilde{\alpha}_Z$ | | | |
| $\theta^2 = \gamma\tau^2$ | Never | All other cases | $\alpha_Z \neq \tilde{\alpha}_Z, \boldsymbol{\beta}_* \not\perp \boldsymbol{u}, \gamma = \omega_c(1)$ 
 or 
 $\alpha_Z = \tilde{\alpha}_Z, \boldsymbol{\beta}_* \not\perp \boldsymbol{u},$ 
 $\omega_c(1) \leq \gamma \leq o_c(c^2)$ |
| $\theta^2 = d\tau^2$ | $\alpha_Z = \tilde{\alpha}_Z = \tilde{\alpha}_A,$ 
 $\boldsymbol{\beta}_* \parallel \boldsymbol{u}$ | All other cases | $\alpha_Z \neq \tilde{\alpha}_Z$ and $\boldsymbol{\beta}_* \not\perp \boldsymbol{u}$ |
| **Spike Recovery:** $\alpha_A = \tilde{\alpha}_A = 0, \alpha_Z = \tilde{\alpha}_Z$ | | | (Appendix D) |
| $\theta^2 = \gamma\tau^2$ | $\gamma\tau^2 = o_c(1)$ | $\gamma\tau^2 = \Theta_c(1)$ | $\gamma\tau^2 = \omega_c(1)$ |
| $\theta^2 = d\tau^2$ | $\tau^2 = o_c(1)$ | $\tau^2 = \Theta_c(1)$ | Never |

## 2 PROBLEM SETTING

We study the generalization of minimum-norm interpolators in high-dimensional linear regression. Using a spiked covariance data model, we quantify how spike strength and alignment influence generalization and the emergence of benign, tempered, or catastrophic overfitting.

**Data Model.** We consider a data matrix $\boldsymbol{X} = \boldsymbol{Z} + \boldsymbol{A} \in \mathbb{R}^{d \times n}$ with *signal component $\boldsymbol{Z}$* and *isotropic noise component $\boldsymbol{A}$* that satisfy the following assumptions. Specifically, we shall that the population feature covariance is $\boldsymbol{\Sigma} = \theta^2 \boldsymbol{u}\boldsymbol{u}^\top + \tau^2 \boldsymbol{I}_d$, modeling a rank-one perturbation of isotropic noise.

**Assumption 1** (Signal). *Let $\boldsymbol{u} \in \mathbb{R}^d$ be a fixed unit vector representing the spike direction. Then*

$$\boldsymbol{Z} = \theta \boldsymbol{u}\boldsymbol{v}^\top, \tag{1}$$

*where $\theta > 0$ controls the spike strength, and the vector $\boldsymbol{v} \in \mathbb{R}^n$ has i.i.d. standard normal entries.*

**Assumption 2** (Noise). *The entries of $\boldsymbol{A}$ have zero mean and variance $\tau^2$. The matrix $\boldsymbol{A}$ satisfies:*

- *Its entries are uncorrelated and possess finite fourth moments.*
- *Its distribution is invariant under left and right orthogonal transformations.*
- *The empirical spectral distribution of $\frac{1}{\tau^2 d}\boldsymbol{A}\boldsymbol{A}^\top$ converges to the Marchenko–Pastur law as $n, d \to \infty$ with $d/n \to c \in (0, \infty)$.*

**Spike Strength Normalizations.** We consider two key scaling regimes for the spike strength relative to the bulk noise. These lead to distinct generalization behaviors.

**Table 2: Conditions for Beneficial Spike Alignment at Finite Aspect Ratios ($c = d/n$).** This table outlines the specific regions where alignment of the target signal with the data's principal spike direction improves generalization. Conditions depend on the problem setting (well-specified vs. mis-specified), the spike scaling regime (operator or frobenius norm based), the overparameterization level $c = d/n$, and the relative dependence of the targets $y$ on the spike versus the bulk $\alpha_Z/\alpha_A$.

| Setting | Alignment Beneficial Region |
|---|---|
| Well-Specified, Operator Norm | $\gamma > c(c-2)$ |
| Well-Specified, Frobenius Norm | $c > 1$ |
| Misspecified, No Covariate Shift, Operator Norm | $\frac{1}{c} \le \frac{\alpha_Z}{\alpha_A} \le \frac{1}{c}\left(\frac{3c^2 - \gamma + 2c\gamma - 2c}{(c^2 + \gamma)}\right)$ |
| Misspecified, No Covariate Shift, Frobenius Norm | $\frac{1}{c} < \frac{\alpha_Z}{\alpha_A} < 2 - \frac{1}{c}$ |

1. **Operator Norm Scaling ($\theta^2 = \gamma\tau^2$):** Here $\gamma$ tunes the spike strength $\theta^2$ relative to the noise variance $\tau^2$. When $\gamma = (1 + \sqrt{c})^2$, the spectral norm of the signal component $Z$ is comparable to that of the noise component $A$. If $\gamma > (1 + \sqrt{c})^2$, the spike emerges as an isolated eigenvalue beyond the bulk spectrum established by $A$, a phenomenon known as the Baik–Ben Arous–Péché (BBP) transition (Baik et al., 2005). This scaling reflects spikes in learned neural network features (Ba et al., 2022; Moniri et al., 2023).

2. **Frobenius Norm Scaling ($\theta^2 = d\tau^2$):** Here $\theta^2 = d\tau^2$ matches expected signal and noise Frobenius norms ($\mathbb{E}[\|Z\|_F^2] = \mathbb{E}[\|A\|_F^2]$) and the spike has macroscopic proportion of the energy. Such strong signals can lead to improved sample complexity, potentially overcoming limitations observed in purely isotropic models (Ba et al., 2023; Mei et al., 2022).

**Target Model.** Given $x_i = z_i + a_i$, the targets $y$ are obtained as follows:

$$y_i = \alpha_Z z_i^\top \beta_* + \alpha_A a_i^\top \beta_* + \varepsilon_i, \tag{2}$$

where $\beta_* \in \mathbb{R}^d$ in uniformly distributed in the subspace $\{\beta \in \mathbb{S}^{d-1} : \beta^\top u = \text{ fixed constant}\}$ is the true underlying parameter vector. The terms $z_i$ and $a_i$ are the $i$-th columns of $Z$ and $A$ respectively. The observation noise $\varepsilon_i$ are i.i.d. with $\mathbb{E}[\varepsilon_i] = 0$, $\mathbb{E}[\varepsilon_i^2] = \tau_\varepsilon^2$. The coefficients $\alpha_Z, \alpha_A \in \mathbb{R}$ control the target's dependence on the signal and noise components. If $\alpha_Z \ne \alpha_A$, the true data generating process for $y$ differentially weights components of $x_i$, causing model misspecification.

**Generalization Risk.** We study the minimum-norm interpolating ordinary least squares estimator:

$$\beta_{int} = X^\dagger y, \qquad \text{with} \qquad \hat{y} = (\tilde{z} + \tilde{a})\beta_{int} \tag{3}$$

where $X^\dagger$ denotes the pseudoinverse. Given a new test data point $(\tilde{x}, \tilde{y})$, where $\tilde{x} = \tilde{z} + \tilde{a}$ and targets $\tilde{y} = \tilde{\alpha}_Z \tilde{z}^\top \beta_* + \tilde{\alpha}_A \tilde{a}^\top \beta_* + \tilde{\varepsilon}$ with potentially with different coefficients $\tilde{\alpha}_Z, \tilde{\alpha}_A$ and model parameters $\tilde{\tau}, \tilde{\tau}_\varepsilon$, the generalization risk is defined as the expected squared prediction error:

$$\mathcal{R}(\beta_{int}) = \mathbb{E}_{X,\varepsilon,\{\tilde{x},\tilde{\varepsilon}\}}\left[(\tilde{y} - \hat{y})^2\right] = \mathbb{E}_{X,\varepsilon,\tilde{x},\tilde{\varepsilon}}\left[(\tilde{y} - \tilde{x}^T \beta_{int})^2\right]. \tag{4}$$

The expectation is over the training data $(X, \varepsilon)$ and the test data realization $(\tilde{x}, \tilde{\varepsilon})$. We shall denote the asymptotic excess risk in the proportional regime as follows:

$$\mathcal{R}_c = \lim_{n,d\to\infty,\, d/n\to c} \mathcal{R}(\beta_{int}) - \tilde{\tau}_\varepsilon^2.$$

**Remark 1** (Generalizing Prior Work). *This problem formulation encompasses several existing models as special cases. For instance, isotropic regression settings studied in Hastie et al. (2022) are recovered by setting $\theta = 0$ (no spike) and $\alpha_Z = 0$. Spike recovery models, such as in Sonthalia and Nadakuditi (2023), correspond to specific choices like $\tau^2 = 1/d$, $\tau_\varepsilon^2 = 0$, and $\alpha_A = 0$. Our generalized setup allows for a nuanced investigation of the interplay between signal structure, target alignment, and overparameterization.*

**Remark 2.** *Although stylized for theoretical simplicity, our setting naturally arises as a low-order Hermite approximation of nonlinear multi-index models, captured by our $(\alpha_A, \alpha_Z)$-parameterization.*

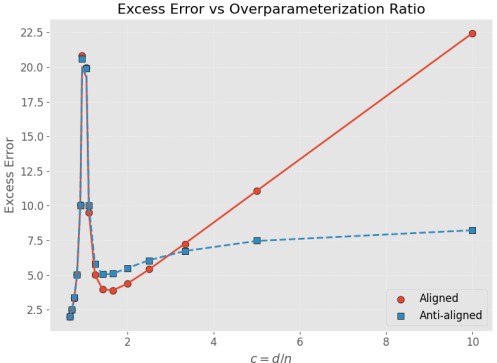 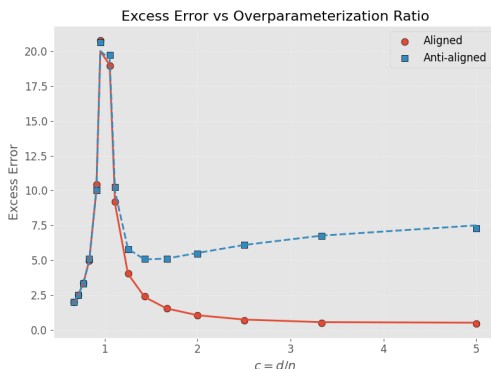

**(a)** Operator norm scaling ($\theta^2 = c\tau^2$). Alignment initially improves generalization, but have catastrophic risk as $c \to \infty$. Anti-alignment yields tempered risk.

**(b)** Equal Frobenius norm scaling ($\theta^2 = d\tau^2$). Alignment leads to benign overfitting, while anti-alignment results in tempered risk.

**Figure 1:** Excess error vs. overparameterization ratio $c = d/n$ in the well-specified case. Each plot shows the risk for aligned and anti-aligned targets under different spike scaling regimes. **The scatter plots are empirically obtained and the lines are theory.**

*This essentially makes our model a tractable surrogate for a broad class of nonlinear targets, while still being simple enough to allow a complete, closed-form risk analysis. Additionally, recent one-step feature-learning analyses like Moniri et al. (2024) show that learned features are **polynomials** of the inputs and the learned spikes. Motivated by this, we study arguably the **simplest non-trivial polynomial target** $y = \alpha_A a + \alpha_Z z$*

**Quantifying the Benefit of Alignment.** A key aspect of our investigation is to determine when the alignment of the true parameter vector $\boldsymbol{\beta}_*$ with the data's principal spike direction $\boldsymbol{u}$ is beneficial for generalization. We define alignment as *beneficial* if the generalization risk $\mathcal{R}(\boldsymbol{\beta}_{int})$ (or $\mathcal{R}_c$), is monotonically decreasing as a function of $(\boldsymbol{\beta}_*^\top \boldsymbol{u})^2 \in [0, 1]$. Conversely, alignment is *detrimental* if the risk is a monotonically increasing function of $(\boldsymbol{\beta}_*^\top \boldsymbol{u})^2$.

**Characterizing Overfitting Regimes.** Following Bartlett et al. (2020); Mallinar et al. (2022), we classify the asymptotic behavior of the excess risk, $\mathcal{R}_c$ as $c \to \infty$ as benign, tempered or catastrophic. We say the overfitting is **benign** if $\lim_{c \to \infty} \mathcal{R}_c$ is zero, **tempered** if this limit is positive and finite, **catastrophic** if this limit is infinite.

## 3 THEORETICAL RESULTS

Our core theoretical contribution is a precise analytical formula for excess risk in the spiked covariance model. This result relies on Assumption 3, which encompasses both the operator norm scaling ($\theta^2 = \gamma\tau^2$) and Frobenius norm scaling ($\theta^2 = d\tau^2$) regimes. We develop our general risk theorem by analyzing progressively complex scenarios. Specifically, our forthcoming theorems provide specific conditions for benign, tempered, or catastrophic overfitting (as $c \to \infty$), and determine when, for finite $c$, alignment of $\boldsymbol{\beta}_*$ with spike $\boldsymbol{u}$ is beneficial or detrimental.

**Assumption 3** (Scaling). *As $n, d \to \infty$ with $d/n \to c \in (0, \infty)$, we assume that $\theta^2$ and $\tau^2$ satisfy $\Omega(\tau^2) \leq \theta^2 \leq O(d\tau^2)$ and $\tau^2 = \Theta(1)$.*

### 3.1 WELL SPECIFIED PROBLEM

We begin by analyzing the well-specified case, where the target $\boldsymbol{y}$ is a direct linear function of the observed covariates $\boldsymbol{X} = \boldsymbol{Z} + \boldsymbol{A}$. This scenario is realized by setting:

$$\alpha_Z = \alpha_A = \tilde{\alpha}_Z = \tilde{\alpha}_A = \alpha > 0.$$

Consequently, $y_i = \alpha \boldsymbol{x}_i^\top \boldsymbol{\beta}_* + \varepsilon_i$, and the model is properly specified.

**Theorem 1** (Well-Specified Risk). *Given data $(\boldsymbol{X}, \boldsymbol{y})$ and $(\tilde{\boldsymbol{X}}, \tilde{\boldsymbol{y}})$ generated according to Assumptions 1 (Signal), 2 (Noise), Equation 2 (Target Model), and Assumption 3 (Scaling). If the well-specification condition $\alpha_Z = \alpha_A = \tilde{\alpha}_Z = \tilde{\alpha}_A = \alpha > 0$ holds, the asymptotic excess risk $\mathcal{R}_c$ is:*

$$
\mathcal{R}_c = \begin{cases} \tau_\varepsilon^2 \frac{c}{1-c} & \text{if } c < 1 \\ \tau_\varepsilon^2 \frac{1}{c-1} + \alpha^2 \tau^2 \left(1 - \frac{1}{c}\right) \left[ \|\boldsymbol{\beta}_*\|^2 + (\boldsymbol{\beta}_*^\top \boldsymbol{u})^2 \frac{\theta^2 \tau^2 c^2 - 2\theta^2 \tau^2 c - \theta^4}{(\theta^2 + \tau^2 c)^2} \right] & \text{if } c > 1 \end{cases}
$$

*where $\boldsymbol{u}$ is the unit vector defining the spike direction.*

**Remark 3.** *If $\theta^2 = \gamma\tau^2$ with $\gamma = o(1)$ (a regime not allowed by Assumption 3 but useful for sanity checks), the coefficient of $(\boldsymbol{\beta}_*^\top \boldsymbol{u})^2$ vanishes, the risk expression aligns with that of isotropic models, such as in (Hastie et al., 2022, Theorem 1).*

**Operator Norm Scaling** ($\theta^2 = \gamma\tau^2$). In this regime, the excess risk for $c > 1$ becomes:

$$
\mathcal{R}_c = \alpha^2 \tau^2 \left(1 - \frac{1}{c}\right) \left( \|\boldsymbol{\beta}_*\|^2 + \frac{\gamma c^2 - 2\gamma c - \gamma^2}{(\gamma + c)^2} (\boldsymbol{\beta}_*^\top \boldsymbol{u})^2 \right) + \tau_\varepsilon^2 \frac{1}{c-1}.
$$

The formula shows that alignment with the spike direction $\boldsymbol{u}$ is beneficial if and only if the coefficient of $(\boldsymbol{\beta}_*^\top \boldsymbol{u})^2$ is negative, which occurs when $\gamma > c(c-2)$. We consider different scalings for $\gamma$.

*Case 1: $\gamma = \Theta_c(1)$ (constant with respect to c).* The condition for beneficial alignment, $\gamma > c(c-2)$, interacts intricately with the BBP phase transition condition, $\gamma > (1 + \sqrt{c})^2$. Let $c_* \approx 4.212$ be the unique solution to $c(c-2) = (1 + \sqrt{c})^2$ for $c > 1$.

- For $1 < c < c_*$: Here, $c(c-2) < (1+\sqrt{c})^2$. If $c(c-2) < \gamma < (1+\sqrt{c})^2$, alignment is beneficial even though the BBP transition has *not* occurred (the spike is not resolved from the bulk).

- For $c > c_*$: Here, $c(c-2) > (1 + \sqrt{c})^2$. For alignment to be beneficial ($\gamma > c(c-2)$), the BBP transition must have occurred (as $\gamma > c(c-2) \implies \gamma > (1+\sqrt{c})^2$). However, the BBP transition occurring is not sufficient for beneficial alignment. If $(1+\sqrt{c})^2 < \gamma < c(c-2)$, the BBP transition occurs, yet alignment is detrimental.

Regarding the type of overfitting as $c \to \infty$ (while $\gamma$ remains constant):

$$
\lim_{c \to \infty} \mathcal{R}_c = \alpha^2 \tau^2 \left( \|\boldsymbol{\beta}_*\|^2 + \gamma(\boldsymbol{\beta}_*^\top \boldsymbol{u})^2 \right).
$$

Since this limit is a positive constant, we consistently observe *tempered overfitting* when $\gamma = \Theta_c(1)$.

*Case 2: $\gamma = \omega_c(1)$ ($\gamma$ grows with c).* The behavior depends on the growth rate of $\gamma$ relative to $c$. The limit of the excess risk for $\boldsymbol{\beta}_*^\top \boldsymbol{u} \neq 0$ as $c \to \infty$ is:

$$
\lim_{c \to \infty} \mathcal{R}_c = \alpha^2 \tau^2 \cdot \begin{cases} \infty & \text{if } \omega_c(1) \leq \gamma \leq o_c(c^2) \\ \|\boldsymbol{\beta}_*\|^2 + (\frac{1}{\phi} - 1)(\boldsymbol{\beta}_*^\top \boldsymbol{u})^2 & \text{if } \gamma = \phi c^2 \text{ for const. } \phi > 0 \\ \|\boldsymbol{\beta}_*\|^2 - (\boldsymbol{\beta}_*^\top \boldsymbol{u})^2 & \text{if } \gamma = \omega_c(c^2) \end{cases}
$$

Surprisingly, while $\gamma = \Theta_c(1)$ gives tempered overfitting, increasing spike strength to $\omega_c(1) \leq \gamma \leq o_c(c^2)$ results in *catastrophic overfitting*, even though morally, this version of the problem has less noise. Additionally, we see that this catastrophic overfitting is not present in the anti-aligned $(\boldsymbol{\beta}_*^\top \boldsymbol{u})$ case. More, aligned with intuition, we see that further increasing the size of the spike improves the generalization performance. Specifically, we get *tempered overfitting* if $\gamma = \phi c^2$ and *benign overfitting* if $\gamma = \omega_c(c^2)$, $\boldsymbol{\beta}_* \parallel \boldsymbol{u}$ and $\|\boldsymbol{\beta}_*\| = 1$.

For $\gamma = c$, the $(\boldsymbol{\beta}_*^\top \boldsymbol{u})^2$ coefficient is $(c-3)/4$. Thus, for $1 < c < 3$, alignment is beneficial and for $c > 3$, alignment becomes detrimental. As $c \to \infty$, if $\boldsymbol{\beta}_* \parallel \boldsymbol{u}$, the excess risk grows approximately as $\alpha^2 \tau^2 \frac{c}{4} (\boldsymbol{\beta}_*^\top \boldsymbol{u})^2$, indicating *catastrophic overfitting*. In contrast, if $\boldsymbol{\beta}_* \perp \boldsymbol{u}$, the excess risk grows like $\alpha^2 \tau^2 (1 - 1/c) \|\boldsymbol{\beta}_*\|^2$, leading to *tempered overfitting*. This transition is illustrated in Figure 1a.

**Intuition:** This result is a special case of a more general result Theorem 5. At a high level, the complicated phase transitions are a direct consequence of competitions among the four terms seen in Theorem 5. In particular among (1) **Bias**, (2) **Variance**, (3) **Target Alignment**, where (1) and (2) are non-negative (harmful) and (3) is negative (beneficial). For the well specified case, alignment completely cancels the variance, but not the bias, which then drives catastrophic overfitting.

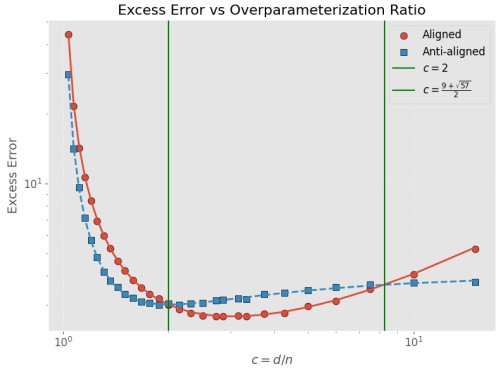 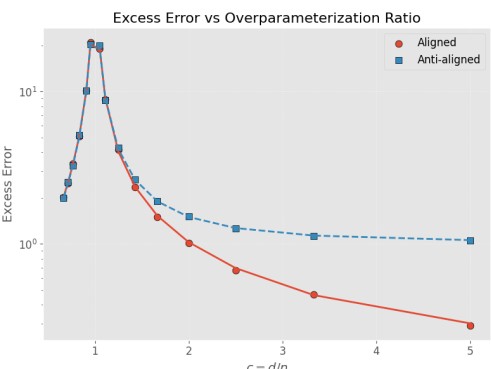

**(a)** Under operator norm scaling ($\theta^2 = c\tau^2$) with $\alpha_Z = 1$, $\alpha_A = 2$, alignment initially improves generalization for small $c$, but becomes harmful beyond a critical point, leading to catastrophic overfitting.

**(b)** Under Frobenius norm scaling ($\theta = \sqrt{d}\tau$) with $\alpha_A = 1$ and $\alpha_Z = 1.1$, alignment remains better than anti-alignment across all $c$, but benign overfitting is not achieved unless $\alpha_Z = \alpha_A$.

**Figure 2:** Transition from beneficial to harmful alignment under mild misspecification. **The scatter plots are empirically obtained and the lines are theory.**

**Frobenius Norm Scaling ($\theta^2 = d\tau^2$).** The excess risk for $c > 1$ simplifies to:

$$\mathcal{R}_{c>1} = \alpha^2\tau^2\left(1 - \frac{1}{c}\right)\left(\|\boldsymbol{\beta}_*\|^2 - (\boldsymbol{\beta}_*^\top\boldsymbol{u})^2\right) + \tau_\varepsilon^2\frac{1}{c-1}.$$

We have a few observations. First, if $\boldsymbol{\beta}_* \parallel \boldsymbol{u}$ and $\|\boldsymbol{\beta}_*\| = 1$, the excess risk $\mathcal{R}_c$ tends to 0 as $c \to \infty$ (*benign overfitting*). Second, if $\boldsymbol{\beta}_*$ is not perfectly aligned with $\boldsymbol{u}$, $\mathcal{R}_c \to \alpha^2\tau^2(\|\boldsymbol{\beta}_*\|^2 - (\boldsymbol{\beta}_*^\top\boldsymbol{u})^2) > 0$ as $c \to \infty$ (*tempered overfitting*). Finally, the coefficient of $(\boldsymbol{\beta}_*^\top\boldsymbol{u})^2$ in the risk formula is negative. Hence, in contrast with the operator norm regime, *alignment is always beneficial* in this regime for $c > 1$, and we visualize these behaviors in Figure 1b.

**Takeaways for the Well-Specified Case.** As a result, spike scaling profoundly impacts overfitting, especially with target alignment, which is not always beneficial. For aligned targets, increasing spike strength can drive transitions from tempered $\to$ catastrophic $\to$ tempered $\to$ benign overfitting, while anti-alignment ($\boldsymbol{\beta}_* \perp \boldsymbol{u}$) can mitigate catastrophic overfitting.

## 3.2 MISSPECIFIED CASE AND NO COVARIATE SHIFT

We next consider misspecified targets $\boldsymbol{y}$ with differing dependence on spike $\boldsymbol{Z}$ and noise $\boldsymbol{A}$ feature components. Specifically, we assume $\alpha_Z \neq \alpha_A$ but introduce no covariate shift between training and test distributions, i.e., $\tilde{\alpha}_Z = \alpha_Z$ and $\tilde{\alpha}_A = \alpha_A$. This scenario models situations where intrinsic feature properties lead to differential correlations with the target, a common occurrence in practice. For notational convenience, we define $\Delta_c := \alpha_Z - \frac{\alpha_A}{c}$ with $\Delta_1 := \alpha_Z - \alpha_A$.

**Theorem 2** (Misspecified). *Let $\boldsymbol{Z}, \tilde{\boldsymbol{Z}}$ satisfy Assumption 1, $\boldsymbol{A}, \tilde{\boldsymbol{A}}$ satisfy Assumption 2 and $\boldsymbol{y}, \tilde{\boldsymbol{y}}$ according to Equation (2). If Assumption 3 holds with $\alpha_Z = \tilde{\alpha}_Z$, $\alpha_A = \tilde{\alpha}_A$, then*

$$\mathcal{R}_c = \begin{cases} \tau_\varepsilon^2\frac{c}{1-c} + \tau^2(\boldsymbol{\beta}_*^\top\boldsymbol{u})^2\frac{\Delta_1^2}{1-c}\frac{\theta^2}{\theta^2+\tau^2} & c < 1 \\[2ex] \tau_\varepsilon^2\frac{1}{c-1} + \alpha_A^2\tau^2\|\boldsymbol{\beta}_*\|^2\left(1 - \frac{1}{c}\right) + \tau^2(\boldsymbol{\beta}_*^\top\boldsymbol{u})^2\Delta_c^2\frac{\theta^2}{\theta^2+\tau^2c}\left[\frac{c}{c-1}\frac{\theta^2+\tau^2c^2}{\theta^2+\tau^2c} - 2\frac{\alpha_A}{\Delta_c}\right] & c > 1 \end{cases}$$

A key observation is that misspecification ($\alpha_Z \neq \alpha_A$) can itself induce double descent, even if $\tau_\varepsilon^2 = 0$. This contrasts with the well-specified case where, if $\tau_\varepsilon^2 = 0$, double descent is absent. However, in the misspecified case, we do not observe double descent if there is no alignment $\boldsymbol{\beta}_*^\top\boldsymbol{u} = 0$.

**Equal Operator Norm Case.** For $\theta^2 = \gamma\tau^2$, the excess risk is

$$\mathcal{R} = \begin{cases} \tau^2(\boldsymbol{\beta}_*^\top\boldsymbol{u})^2 \frac{\Delta_1^2}{1-c}\frac{\gamma}{\gamma+1} + \tau_\varepsilon^2\frac{c}{1-c} & c < 1 \\ \tau^2\frac{\gamma}{\gamma+c}(\boldsymbol{\beta}_*^\top\boldsymbol{u})^2\Delta_c^2\left[\left(\frac{c^2+\gamma}{\gamma+c}\frac{c}{c-1}\right) - 2\frac{\alpha_A}{\Delta_c}\right] + \alpha_A^2\tau^2\|\boldsymbol{\beta}_*\|^2\left(1 - \frac{1}{c}\right) + \tau_\varepsilon^2\frac{1}{c-1} & c > 1 \end{cases}$$

For $c < 1$, the spike is *detrimental*. For $c > 1$, the behavior depends on $\alpha_Z/\alpha_A$. In particular, if

$$\frac{1}{c} \leq \frac{\alpha_Z}{\alpha_A} \leq \frac{1}{c}\left(\frac{3c^2 - \gamma + 2c\gamma - 2c}{(c^2 + \gamma)}\right),$$

then we have that the coefficient in front of $(\boldsymbol{\beta}_*^\top\boldsymbol{u})^2$ is negative. Thus, when $\alpha_Z/\alpha_A$ lies between these thresholds, the spike *helps*, but the spike *is harmful* outside this range. As $c \to \infty$, if $\gamma = o_c(c^2)$, the beneficial region shrinks and *alignment increasingly harms generalization*. On the other hand, if the spike is big enough ($\gamma = \omega_c(c^2)$), we have that the beneficial region limits to $0 \leq \frac{\alpha_Z}{\alpha_A} \leq 2$. Figures 3a and 3b plot the coefficient of $(\boldsymbol{\beta}_*^\top\boldsymbol{u})^2$ for $c = 2$ and $c = 20$ for $\gamma = c$.

The upper bound on beneficial $\alpha_Z/\alpha_A$ is surprising, as stronger target dependence on the spike might be expected to always favor alignment. Additionally, the dependence on the level of overparameterization $c$ also offers new insights. Consider the example of $\gamma = c$, and $\alpha_Z/\alpha_A = 2$. Then when $c < 2$ or $c > (9 + \sqrt{57})/2$, we have that the ratio is outside the beneficial region. Figure 2a shows that in the beneficial region, the aligned risk is lower than the anti-aligned risk. However, outside the beneficial region, the aligned risk becomes strictly larger than the anti-aligned counterpart.

Next, in terms of benign vs. tempered vs. catastrophic overfitting, we have that

$$\lim_{c\to\infty} \mathcal{R}_c = \begin{cases} \tau^2\left[\gamma\alpha_Z^2(\boldsymbol{\beta}_*^\top\boldsymbol{u})^2 + \alpha_A^2\|\boldsymbol{\beta}_*\|^2\right] & \boldsymbol{\beta}_* \not\perp \boldsymbol{u}, \gamma = \Theta_c(1) \\ \infty & \boldsymbol{\beta}_* \not\perp \boldsymbol{u}, \omega_c(1) \leq \gamma \leq o_c(c^2) \\ \tau^2\left[\alpha_A^2\|\boldsymbol{\beta}_*\|^2 + \left(\alpha_Z^2\left(1 + \frac{1}{\phi}\right) - 2\alpha_Z\alpha_A\right)(\boldsymbol{\beta}_*^\top\boldsymbol{u})^2\right] & \boldsymbol{\beta}_* \not\perp \boldsymbol{u}, \gamma = \phi c^2 \\ \tau^2(\alpha_A^2\|\boldsymbol{\beta}_*\|^2 + (\alpha_Z^2 - 2\alpha_Z\alpha_A)(\boldsymbol{\beta}_*^\top\boldsymbol{u})^2) & \boldsymbol{\beta}_* \not\perp \boldsymbol{u}, \gamma = \omega_c(c^2) \\ \alpha_A^2\tau^2\|\boldsymbol{\beta}_*\|^2 & \boldsymbol{\beta}_* \perp \boldsymbol{u} \end{cases}.$$

For $\boldsymbol{\beta}_* \not\perp \boldsymbol{u}$, if $\omega_c(1) \leq \gamma \leq o_c(c^2)$ we have *catastrophic overfitting*. If $\gamma = \Theta_c(c^2)$, overfitting is tempered, with benign overfitting precluded (Appendix Proposition 3). If $\gamma = \omega_c(c^2)$, overfitting is again tempered with benign requiring returning to the well-specified case ($\alpha_A = \alpha_Z$).

**Ridge Extension.** As done in Li and Sonthalia (2024), it is also possible to analyze the ridge regularized version by looking at the resolvent away from zero. Since the general proof structure remains the same, we leave this as a future direction and empirically test this case. In particular, we follow the setting of Figure 1a ($\theta^2 = c\tau^2$) and vary the regularization strengths ($\lambda \in \{0, 1, c, c^2, dc\}$). In all cases the catastrophic band in our phase diagram persists, indicating that generally regularization does not remove alignment-driven catastrophic overfitting. The plot can be seen in Appendix B.2.

**Equal Frobenius Norm Case.** For $\theta^2 = d\tau^2$, the excess risk becomes:

$$\mathcal{R}_{c>1} = \alpha_A^2\|\boldsymbol{\beta}_*\|^2\left(1 - \frac{1}{c}\right) + (\boldsymbol{\beta}_*^\top\boldsymbol{u})^2\left[\frac{c}{c-1}\left(\alpha_Z - \frac{\alpha_A}{c}\right)^2 - 2\alpha_A\left(\alpha_Z - \frac{\alpha_A}{c}\right)\right] + \frac{\tau_\varepsilon^2}{c-1}.$$

For $c > 1$, the beneficial region for the ratio $\alpha_Z/\alpha_A$ is defined by: $\frac{1}{c} \leq \frac{\alpha_Z}{\alpha_A} \leq 2 - \frac{1}{c}$. The beneficial region expands with $c$, making alignment increasingly beneficial in extreme overparameterization (Figure 3c). Beneficial alignment can also be seen in Figure 2b. Here $\alpha_Z/\alpha_A = 1.1$, which is in the beneficial region for $c > 10/9$. Finally, the overfitting is tempered unless $\alpha_A = \alpha_Z$.

### 3.3 MISSPECIFIED TARGET AND COVARIATE SHIFT

Lastly, in addition to misspecification, we also have covariate shift between train and test. Specifically, $\alpha_Z \neq \tilde{\alpha}_Z$ or $\alpha_A \neq \tilde{\alpha}_A$, hence we have the spike/noise importance differ between train and test. For the **equal operator norm** case, we show the following.

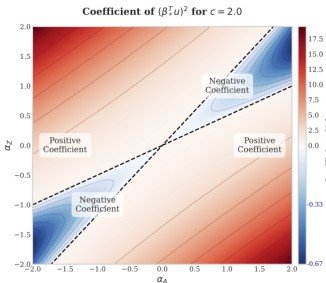 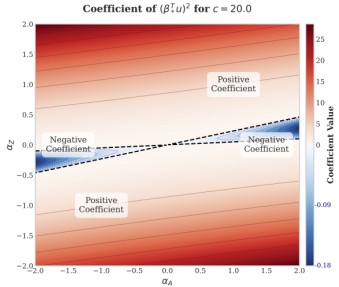 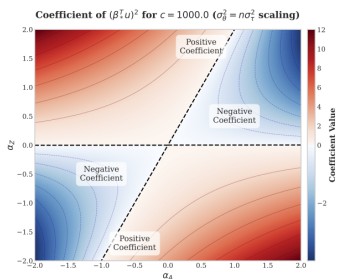

(a) **Operator norm scaling**, $c = 2$. Large beneficial region.

(b) **Operator norm scaling**, $c = 20$. Smaller beneficial region

(c) **Frobenius norm scaling**, $c = 1000$. The beneficial region persists at extreme overparameterization.

**Figure 3: Phase boundaries for spike alignment impact.** Coefficient of $(\boldsymbol{\beta}_*^\top \boldsymbol{u})^2$ as a function of $\alpha_Z/\alpha_A$, indicating whether alignment improves or harms generalization.

**Theorem 3.** *Given data $\boldsymbol{Z}, \tilde{\boldsymbol{Z}}$ that satisfy Assumption 1, $\boldsymbol{A}, \tilde{\boldsymbol{A}}$ that satisfy Assumption 2 and $\boldsymbol{y}, \tilde{\boldsymbol{y}}$ according to Equation (2). If Assumption 3 holds, catastrophic overfitting occurs if $\tilde{\alpha}_Z = \alpha_Z$, $\boldsymbol{\beta}_* \not\perp \boldsymbol{u}$, and $\omega_c(1) \leq \gamma \leq o_c(c^2)$. Additionally, if $\tilde{\alpha}_Z \neq \alpha_Z$ with $\gamma = \omega_c(1)$ and $\boldsymbol{\beta}_* \not\perp \boldsymbol{u}$ we get catastrophic overfitting. Other scenarios yield tempered overfitting.*

Different covariate shifts pose varying challenges. In particular, if $\alpha_Z \neq \tilde{\alpha}_Z$, (target's spike dependence shifts), then catastrophic overfitting becomes unavoidable for sufficiently large spikes. This contradicts the earlier theoretical intuition, as increasing the spike size in this setting actually induces catastrophic overfitting instead of mitigating it.

**Equal Frobenius Norm.** In this case, we have the following theorem.

**Theorem 4.** *Let $\boldsymbol{Z}, \tilde{\boldsymbol{Z}}$ satisfy Assumption 1, $\boldsymbol{A}, \tilde{\boldsymbol{A}}$ satisfy Assumption 2 and $\boldsymbol{y}, \tilde{\boldsymbol{y}}$ according to Equation (2). If Assumption 3 holds and $\alpha_Z \neq \tilde{\alpha}_Z$ then $\mathcal{R}_c = \infty$ for all $c \neq 1$. For $\alpha_Z = \tilde{\alpha}_Z$:*

$$\lim_{c \to \infty} \mathcal{R}_c = \tau^2 \left[ (\boldsymbol{\beta}_*^\top \boldsymbol{u})^2 (\alpha_Z^2 - 2\tilde{\alpha}_A \alpha_Z) + \|\boldsymbol{\beta}_*\|^2 \tilde{\alpha}_A^2 \right].$$

If $\alpha_Z \neq \tilde{\alpha}_Z$, catastrophic overfitting occurs. When $\boldsymbol{\beta}_*$ and $\boldsymbol{u}$ are parallel, we have that $\tau^2 \|\boldsymbol{\beta}_*\|^2 (\alpha_Z - \tilde{\alpha}_A)^2$. This is benign if and only if $\alpha_Z = \tilde{\alpha}_A$. Notably, if training data is misspecified ($\alpha_A \neq \alpha_Z$) but test data is well-specified and matches the training spike dependence ($\alpha_Z = \tilde{\alpha}_Z = \tilde{\alpha}_A$), benign overfitting becomes achievable.

### 3.4 GENERAL THEOREM

Prior results are special cases of our main theorem (Theorem 5). Its full form is complex (Appendix E). We present a high-level decomposition here.

**Theorem 5** (Generalization Risk). *Suppose Assumption 1, Assumption 2, and Assumption 3 hold.*

$$\mathcal{R} = \mathbb{E}\left[ \underbrace{\left\| \tilde{\alpha}_z \boldsymbol{\beta}_*^\top \tilde{\boldsymbol{Z}} - \boldsymbol{\beta}_{int}^\top \tilde{\boldsymbol{Z}} \right\|_F^2}_{Bias} + \underbrace{\tau^2 \left\| \boldsymbol{\beta}_{int}^\top \tilde{\boldsymbol{A}} \right\|_F^2}_{Variance} + \underbrace{\tilde{\alpha}_A^2 \left\| \boldsymbol{\beta}_*^\top \tilde{\boldsymbol{A}} \right\|_F^2}_{Data\ Noise} + \underbrace{\left( -2\tilde{\alpha}_A \boldsymbol{\beta}_*^\top \tilde{\boldsymbol{A}} \tilde{\boldsymbol{A}}^\top \boldsymbol{\beta}_{int} \right)}_{Target\ Alignment} \right].$$

- **Bias.** This is the squared error between the learned predictor $\boldsymbol{\beta}_{int}$ and the true parameter $\boldsymbol{\beta}_*$ *projected onto the spike direction $\boldsymbol{u}$.* In particular, the risk penalizes discrepancies only along the top eigen-direction of the population covariance $\Sigma$, reflecting the anistropic influence of the spike.

- **Variance.** The variance is equivalent to $\tau^2 \|\boldsymbol{\beta}_{int}\|_2$. This mirrors classical isotropic regression results (Hastie et al., 2022; Bartlett et al., 2020), but the norm $\|\boldsymbol{\beta}_{int}\|^2$ itself is dependent upon the interaction between signal and noise, the alignment between $\boldsymbol{\beta}_*$ and $\boldsymbol{u}$, and the scaling parameters.

- **Data Noise.** The data noise term quantifies the contribution of the noise matrix $\boldsymbol{A}$ to the target outputs $y_i$ through $\alpha_A$. Even in the absence of observation noise ($\tau_\varepsilon^2 = 0$), target corruption via data noise can create an irreducible error floor.

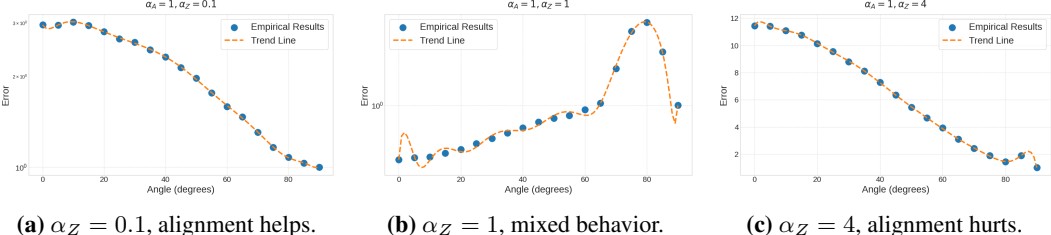

**(a)** $\alpha_Z = 0.1$, alignment helps.    **(b)** $\alpha_Z = 1$, mixed behavior.    **(c)** $\alpha_Z = 4$, alignment hurts.

**Figure 4: Alignment-phase transitions persist in deep networks.** Generalization error vs. angle between spike direction $u$ and ground-truth parameter $\beta_*$ when fitting data with a 3-layer ReLU networks. The effect of alignment switches as $\alpha_Z$ increases, consistent with the phase transitions predicted by our theory. Experimental details are in Appendix B.

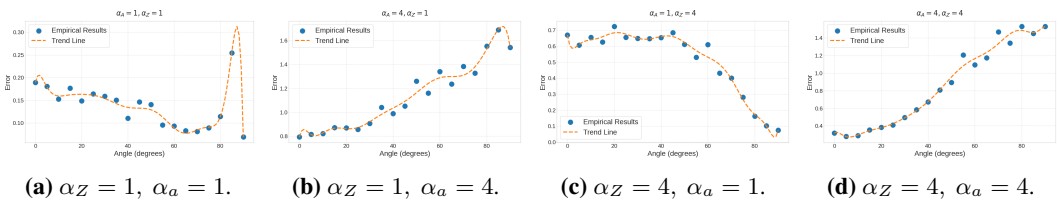

**(a)** $\alpha_Z = 1$, $\alpha_a = 1$.    **(b)** $\alpha_Z = 1$, $\alpha_a = 4$.    **(c)** $\alpha_Z = 4$, $\alpha_a = 1$.    **(d)** $\alpha_Z = 4$, $\alpha_a = 4$.

**Figure 5: Generalization error vs. alignment for deep networks.** Generalization error vs. the angle between spike direction $u$ and ground-truth parameter $\beta_*$ when fitting MNIST-derived data with a ReLU network, for a sweep over $(\alpha_Z, \alpha_a) \in \{1, 4\}^2$.

- **Target Alignment.** The alignment term measures the inner product between $\beta_{int}$ and $\beta_*$ with respect to the sample noise covariance. This cross-term captures how mismatch between $\beta_{int}$ and $\beta_*$, especially when mediated by $A$, can amplify or dampen generalization error.

### 3.5 NONLINEAR MODELS

While our theoretical focus is on linear regression, key phenomena like $\alpha_Z$ dependent non-monotonic alignment effects appear in nonlinear models as well. We test this by training 3-layer ReLU networks to predict $y$ (Equation (2)) given $X$, where we vary the alignment angle between spike $u$ and $\beta_*$ and record the generalization error. Figure 4, shows our results for three $\alpha_Z$ values. For $\alpha_Z = 0.1$, increasing alignment with the spike is detrimental. For $\alpha_Z = 1$, alignment is beneficial, while for $\alpha_Z = 10$, alignment is detrimental again. This mirrors our theoretical findings that there is a region for beneficial alignment and a nuanced phase transition for different $\alpha_Z$ values.

To move beyond purely synthetic inputs, we perform an experiment on MNIST where we artificially inject a spiked direction into the input space. Please see Appendix B.1 for more details. We then train model as we vary $(\alpha_A, \alpha_Z) \in \{1, 4\}^2$ and the target–spike angle. We again observe a similar pattern: the phase transitions we recover potentially generalize to nonlinear deep networks trained on real data, broadening the theoretical robustness of our setting. Results can seen in Figure 5.

## 4 CONCLUSION

This work provided a precise analytical characterization of the generalization error for minimum-norm interpolators in spiked covariance models. We decomposed the risk into interpretable components and comprehensively classified overfitting regimes based on spike strength, target alignment, and overparameterization. We reveal surprising phenomena, such as the potential for increasing spike strength to induce catastrophic overfitting before benign overfitting in well-specified aligned problems, and that strong target-spike alignment is not universally beneficial, especially under model misspecification. These alignment-dependent phase transitions, theoretically derived for linear models, were also empirically observed in nonlinear neural networks, suggesting broader relevance. Our results offer a more nuanced understanding of generalization in the presence of data anisotropy, challenging conventional intuitions and providing a detailed map of risk behaviors in overparameterized settings.

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

CONTENTS

# A    NOTATION

| Symbol | Description / Role | Typical scaling / range | First used |
|---|---|---|---|
| $d, n$ | Data dimension and sample size | $d, n \to \infty$ with $c = d/n$ fixed | Sec. 2 |
| $c$ | Aspect ratio $d/n$ | $(0, \infty)$ | Sec. 2 |
| $\tau^2/d$ | Noise variance in ambient bulk $A$ | $\tau^2 = \Theta(1)$ | Sec. 2 |
| $\theta^2$ | Spike (signal) variance | $\theta^2 = \gamma\tau^2$ (operator-norm) or $\theta^2 = d\tau^2$ (Frobenius) | Sec. 2 |
| $\gamma$ | Spike-to-noise ratio $\gamma = \theta^2/\tau^2$ (effective outlier eigenvalue) | $[0, \infty)$; critical line $\gamma = (1 + \sqrt{c})^2$ | Sec. 2 |
| $\alpha_Z, \alpha_A$ | Coeffs. weighting spike vs. bulk in *targets y* | $\Theta(1)$ | Eq. (2) |
| $\tilde{\alpha}_Z, \tilde{\alpha}_A$ | Same coefficients for *test* data (covariate shift) | $\Theta(1)$ | Sec. 3 |
| $\boldsymbol{\beta}_*$ | True parameter vector | $\|\boldsymbol{\beta}_*\|_2 = 1$ | Sec. 2 |
| $\boldsymbol{u}$ | Spike direction in data covariance | $\|\boldsymbol{u}\|_2 = 1$ | Sec. 2 |
| $\boldsymbol{A}, \boldsymbol{Z}$ | Bulk noise matrix, rank-one signal matrix | $A_{ij} \sim \mathcal{N}(0, \tau^2/d)$, $\boldsymbol{Z} = \theta\,\boldsymbol{u}\boldsymbol{v}^\top$ | Sec. 2 |
| $\boldsymbol{\varepsilon}, \tau_\varepsilon^2$ | Label noise and its variance | IID, $\mathcal{N}(0, \tau_\varepsilon^2)$ | Sec. 2 |

**Table 3:** Glossary of recurrent parameters and symbols. All $\Theta(1)$ constants are independent of $n, d$.

**Other Notations.**    We use lowercase $a$, lowercase bold $\boldsymbol{a}$, and uppercase bold $\boldsymbol{A}$ letters to denote scalars, vectors, and matrices respectively. We use $\|\cdot\|_2$ to denote the Euclidean norm if the argument is a vector and the operator norm if the argument is a matrix. We use $\|\cdot\|_F$ to denote the Frobenius norm. When slicing one entry from a vector or matrix, we use both $a_i$, $A_{ij}$ and $\boldsymbol{a}_i$, $\boldsymbol{A}_{ij}$, where the latter intends to emphasize the source of the scalar.

# B    MORE NON-LINEAR EXPERIMENTS AND SETTINGS

We used 500 data points in 750 dimensional space, with a hidden width of 1000. We used full batch gradient descent for 100 epochs with a learning rate of 1e-4. Each data point is averaged over 50 trials. Equal Frobenius norm scaling was used for the size of the spike.

## B.1    MNIST DATA

For each trial, we construct a spiked design matrix by combining a rank-one signal with a "bulk" component drawn from MNIST. Fix sample size $n$ and ambient dimension $d$ (here $d = 784$). First, we sample a unit-norm ground-truth vector $\boldsymbol{\beta}_\star \in \mathbb{R}^d$ and, for each prescribed angle, construct a unit vector $\boldsymbol{u} \in \mathbb{R}^d$ at that angle to $\boldsymbol{\beta}_\star$. Independently sample unit vectors $\boldsymbol{v}, \boldsymbol{v}_t \in \mathbb{R}^n$ and set spike strengths $\theta = \theta_t = \sqrt{n}\,\tau$. The rank-one signal matrices are

$$\boldsymbol{Z} = \theta\boldsymbol{u}\boldsymbol{v}^\top \in \mathbb{R}^{d\times n}, \qquad \boldsymbol{Z}_t = \theta_t\boldsymbol{u}\boldsymbol{v}_t^\top \in \mathbb{R}^{d\times n}.$$

The bulk matrices are generated by sampling $n$ MNIST images (train split for training, test split for testing), vectorizing each image into $\mathbb{R}^{784}$, and stacking these as columns to obtain

$$\boldsymbol{A} \in \mathbb{R}^{d\times n} \quad \text{and} \quad \boldsymbol{A}_t \in \mathbb{R}^{d\times n},$$

with raw pixel intensities in $[0, 1]$. The training features are $\boldsymbol{X} = \boldsymbol{Z} + \boldsymbol{A}$, and responses are generated as a noisy linear measurement of a weighted combination of spike and bulk:

$$\boldsymbol{y} = \boldsymbol{\beta}_\star^\top(\alpha_Z\boldsymbol{Z} + \alpha_A\boldsymbol{A}) + \boldsymbol{\varepsilon}, \qquad \boldsymbol{\varepsilon} \sim \mathcal{N}(0, \tau_\varepsilon^2\boldsymbol{I}_n).$$

**Network architecture.**    We use the same network structure as previously, but trained for 1000 epochs.

## B.2    RIDGE FIGURE

Figure 6 shows the risk that still manifests alignment-driven catastrophic overfitting with ridge regularization.

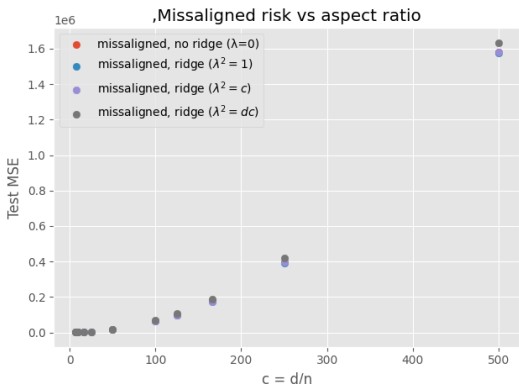

**Figure 6:** Catastrophic Overfitting even in the presence of Ridge Regularizer.

## C    PRIOR WORKS & NATURAL EXTENSIONS

**Li and Sonthalia (2024).**  This paper mainly shows that least-squares regression can exhibit a *double-descent peak in the under-parameterized regime*, by carefully analyzing the ridge-regularized resolvent bounded away from zero. They provide two examples where these spectral/alignment properties move the peak to $c < 1$ and argue that existing double-descent explanations, which focus on $c \geq 1$, are incomplete. Our setting recovers theirs when the regularization $\mu = 0$, $\alpha_Z = \alpha_A = 1$ (equal dependence), and learning a denoiser (target is the noiseless data $\boldsymbol{X}$). Our model significantly generalizes beyond the data model with interpolating parameters $\alpha_A, \alpha_Z$, uncovering different insights about overfitting behaviors.

**Demir and Dogan (2024) and Demir and Dogan (2025).**   Demir and Dogan (2025) analyzes the Gaussian-mixture/spiked data after one gradient step.  In contrast, we provide closed-form generalization risk for the minimum-norm linear interpolator. In particular, this paper serves as the motivation for studying their setting, which is the analytically tractable "base case" that isolates spike–alignment mechanisms.  On the other hand, Demir and Dogan (2024) considers whether substituting one activation function for another results in the same generalization error; it does not characterize the error itself, which is the goal of this paper.

In terms of proof techniques, our paper requires more sophisticated bounding approaches, as we argue in the subsequent sections.

### C.1    NEW TECHNIQUES & POSSIBLE THEORETICAL EXTENSIONS

Our analysis combines ingredients that, to the best of our knowledge, have not been put together in prior work on benign overfitting or spiked regression.

In particular, we develop a *mixed spherical hypercontractivity* argument that controls products of random terms involving both bulk and spiked components. Instead of assuming strict Gaussianity, we only require zero mean, unit variance, and a mild rotational invariance; hypercontractivity up to order $2k$ then yields concentration for multilinear forms of the type that appear in our risk formula.

Two natural generalizations from our data model include (1) adding ridge regularization and (2) incorporating general low-rank data.

**Ridge Regression.** As discussed above for Li and Sonthalia (2024), how regularization affects the data resolvent can be tracked and a similar Stieljties transform argument applies. In particular, check their Lemmas 13-15 for modified statements that can be generalized.

**General Low-Rank Data.** Our proof requires a Sherman-Morrison-style expansion for pseudo-inverse in Meyer (1973). This result from perturbation theory allows us to decouple spike and bulk and makes the subsequent analysis more tractable. A Woodbury-style formula for pseudo-inverse (extending Meyer (1973)) exists in Wei (2001), and the same proof structure follows. We can then bound contributions along each singular direction, which requires significantly more bookkeeping.

Due to the significant effort involved rigorously doing these calculations. We leave this as future work. However, we believe that an extension to ridge with multiple spikes is feasible.

## D  SPIKE RECOVERY CASE

We consider the special case where the goal is to recover the spike direction $\boldsymbol{u}$. In this setting, the target $\boldsymbol{y}$ depends only on the spike component $\boldsymbol{Z}$, with no contribution from the noise $\boldsymbol{A}$:

$$\alpha_A = \tilde{\alpha}_A = 0, \qquad \alpha_Z = \tilde{\alpha}_Z = \alpha > 0.$$

Thus, the target $\boldsymbol{y}$ is proportional to the signal $\boldsymbol{Z}$ plus possible observation noise $\varepsilon$.

**Equal Operator Norm**    In this regime, we have that the risk is

$$\mathcal{R}_{c<1} = \frac{\gamma \alpha_Z^2 \tau^2}{(1-c)(\gamma+1)}(\boldsymbol{\beta}^\top \boldsymbol{u})^2 + \frac{c}{1-c}\tau_\varepsilon^2, \qquad \mathcal{R}_{c>1} = \frac{\gamma c(c^2+\gamma)\alpha_Z^2 \tau^2}{(c-1)(\gamma+c)^2}(\boldsymbol{\beta}^\top \boldsymbol{u})^2 + \frac{1}{c-1}\tau_\varepsilon^2$$

Here again, we see that when $\gamma = \Theta_c(1)$, we have tempered overfitting and $\omega_c(1) \leq \gamma \leq o_c(c^2)$, we have catastrophic overfitting and for $\gamma = \Omega_c(c^2)$ we get tempered overfitting again.

**Equal Frobenius Norm**    . In this regime, we have that

$$R_{c<1} = \frac{\alpha_Z^2 \tau^2}{1-c}(\boldsymbol{\beta}^\top \boldsymbol{u})^2 + \frac{c}{1-c}\tau_\varepsilon^2 \quad R_{c>1} = \frac{c\alpha_Z^2 \tau^2}{c-1}(\boldsymbol{\beta}^\top \boldsymbol{u})^2 + \frac{1}{c-1}\tau_\varepsilon^2.$$

This generalizes the spike recovery setting studied in Sonthalia and Nadakuditi (2023), which assumed noiseless targets ($\tau_\varepsilon = 0$) and the equal Frobenius norm scaling. Our formula allows for observation noise and thus captures the more realistic case where the target $y$ itself contains randomness not aligned with the spike. Here we see that we have tempered overfitting unless $\tau^2 = o(1)$, which is the case considered in Sonthalia and Nadakuditi (2023).

# E    PROOF OF THEOREM 5

**Theorem 5** (Generalization Risk). *Suppose Assumption 1, Assumption 2, and Assumption 3 hold.*

$$\mathcal{R} = \mathbb{E}\left[\underbrace{\left\|\tilde{\alpha}_z \boldsymbol{\beta}_*^\top \tilde{\boldsymbol{Z}} - \boldsymbol{\beta}_{int}^\top \tilde{\boldsymbol{Z}}\right\|_F^2}_{Bias} + \underbrace{\tau^2 \left\|\boldsymbol{\beta}_{int}^\top \tilde{\boldsymbol{A}}\right\|_F^2}_{Variance} + \underbrace{\tilde{\alpha}_A^2 \left\|\boldsymbol{\beta}_*^\top \tilde{\boldsymbol{A}}\right\|_F^2}_{Data\ Noise} + \underbrace{\left(-2\tilde{\alpha}_A \boldsymbol{\beta}_*^\top \tilde{\boldsymbol{A}}\tilde{\boldsymbol{A}}^\top \boldsymbol{\beta}_{int}\right)}_{Target\ Alignment}\right].$$

In particular, as $n, d \to \infty$ with $d/n \to c \in (0, \infty)$, we have the following expressions for each term.

**Bias:** For $c < 1$, we have that the bias term is

$$\tilde{\theta}^2 \left[(\boldsymbol{\beta}_*^\top \boldsymbol{u})^2 \left(\tilde{\alpha}_Z - \alpha_Z + (\alpha_Z - \alpha_A) + \frac{\tau^2}{\theta^2 + \tau^2}\right)^2 + \tau_\varepsilon^2 \frac{c}{1-c}\frac{1}{d(\theta^2 + \tau^2)}\right].$$

If $c > 1$, we that the bias term is

$$\tilde{\theta}^2(\boldsymbol{\beta}_*^\top \boldsymbol{u})^2 \left(\tilde{\alpha}_Z - \alpha_Z + \left(\alpha_Z - \frac{\alpha_A}{c}\right)\frac{\tau^2 c}{\theta^2 + \tau^2 c}\right)^2 + \tilde{\theta}^2 \left[\alpha_A^2 \frac{\|\boldsymbol{\beta}_*\|^2}{d}\frac{c-1}{c}\frac{\theta^2 \tau^2 c}{(\theta^2 + \tau^2 c)^2} + \tau_\varepsilon^2 \frac{c}{c-1}\frac{\theta^2 + \tau^2}{n(\theta^2 + \tau^2 c)^2}\right].$$

**Variance:** For $c < 1$, we have that the variance term is

$$\alpha_A^2 \tilde{\tau}^2 \|\boldsymbol{\beta}_*\|^2 + \tilde{\tau}^2(\boldsymbol{\beta}_*^\top \boldsymbol{u})^2 \left[\frac{1}{1-c}\frac{\theta^4 + \theta^2 \tau^2 c}{(\theta^2 + \tau^2)^2}(\alpha_Z - \alpha_A)^2 + 2\alpha_A(\alpha_Z - \alpha_A)\frac{\theta^2}{\theta^2 + \tau^2}\right]$$

$$+ \tau_\varepsilon^2 \frac{\tilde{\tau}^2}{\tau^2}\left[\frac{c}{1-c} - \frac{\theta^2}{d(\theta^2 + \tau^2)}\frac{c}{1-c}\right].$$

For $c > 1$, we have that the variance term is

$$\tilde{\tau}^2 \|\boldsymbol{\beta}_*\|^2 \left(\frac{\alpha_A^2}{c} - \frac{\alpha_A^2}{d}\frac{\theta^2}{\theta^2 + \tau^2 c}\right) + \tilde{\tau}^2(\boldsymbol{\beta}_*^\top \boldsymbol{u})^2 \frac{c}{(c-1)}\frac{\theta^2}{\theta^2 + \tau^2 c}\left(\alpha_Z - \frac{\alpha_A}{c}\right)^2$$

$$+ \tau_\varepsilon^2 \frac{\tilde{\tau}^2}{\tau^2}\left(\frac{1}{c-1} - \frac{\theta^2}{d(\theta^2 + \tau^2 c)}\frac{c}{c-1}\right).$$

**Data Noise:** For all $c$, we have that

$$\tilde{\alpha}_A^2 \tilde{\tau}^2 \|\boldsymbol{\beta}_*\|^2.$$

**Target Alignment:** For $c < 1$, we have that the alignment term is

$$-2\tilde{\alpha}_A \tilde{\tau}^2 \left((\alpha_Z - \alpha_A)\frac{\theta^2}{\theta^2 + \tau^2}(\boldsymbol{\beta}_*^\top \boldsymbol{u})^2 + \alpha_A \|\boldsymbol{\beta}_*\|^2\right).$$

For $c > 1$, we have that the alignment term is

$$-2\tilde{\alpha}_A \tilde{\tau}^2 \left(\left(\alpha_Z - \frac{\alpha_A}{c}\right)\frac{\theta^2}{\theta^2 + \tau^2 c}(\boldsymbol{\beta}_*^\top \boldsymbol{u})^2 + \alpha_A \|\boldsymbol{\beta}_*\|^2 \left(\frac{1}{c} - \frac{1}{d}\frac{\theta^2}{\theta^2 + \tau^2 c}\right)\right).$$

**Error terms:** The largest error terms for all $c$ are:

$$o(1) + O\left(\frac{1}{n}\right) = o(1).$$

**Remark:** We note that the above theorem is very general and captures all of the theorems in the main text as special cases. It is worth noting that the theorem also incorporates different signal and bulk strengths for test data, namely for $\tilde{\theta}$ and $\tilde{\tau}$.

The proof will be broken up into roughly 6 steps

    0. **Rescale the problem** To apply standard results we rescale the problem. Section E.1

1. **Decompose the error into four terms.** We shall refer to these terms as the 1) bias, 2) variance, 3) data noise, and 4) target alignment. Section E.2

2. **Simplify the expressions.** We shall then use the result from Meyer (1973) to simplify the expression for each of the four terms. In particular, we shall express each term as the product of dependent functions of the eigenvalues of $\boldsymbol{X}$. Section E.3

3. **Random matrix theory estimate.** We then use standard results from random matrix theory such as Marchenko and Pastur (1967); Bai and Zhou (2008); Baik and Silverstein (2006) to obtain a closed-form formula of the building blocks for the risk. Section E.4

4. **Bound Products.** We then show that products of our building blocks concentrate. Step 4 (Section E.5) then collects the final terms.

5. **Undo Scaling** Step 5 (Section E.6) gives us back the correct scaling.

Section F has some generic probability lemmas that we need.

### E.1 STEP 0: RESCALING

In order to better align with existing results and use them accordingly, we change our scalings for now and switch back after our derivation. That is, we divide everything by $\sqrt{d}$. Hence, we shall use

$$\frac{\theta}{\sqrt{d}}\boldsymbol{u}\boldsymbol{w}^{\top} = \theta\frac{\|\boldsymbol{w}\|}{\sqrt{d}}\boldsymbol{u}\frac{\boldsymbol{w}^{\top}}{\|\boldsymbol{w}\|}$$

as the spike. We shall let

$$\eta^2 := \theta^2\frac{\|\boldsymbol{w}\|^2}{d} \quad \text{and} \quad \boldsymbol{v} := \frac{\boldsymbol{w}^{\top}}{\|\boldsymbol{w}\|}$$

Here, we treat $\boldsymbol{v}$ as fixed unit norm vector and our spike is

$$\boldsymbol{Z}_r := \eta\boldsymbol{u}\boldsymbol{v}^T$$

The $\boldsymbol{A}$ noise after dividing by $\sqrt{d}$ is

$$\boldsymbol{A}_r := \frac{\tau}{\sqrt{d}}N$$

where $N$ are mean zero variance 1 entries. Here the appendix, we shall use the letter $\rho$ for $\tau$. Finally let

$$\boldsymbol{X}_r = \boldsymbol{Z}_r + \boldsymbol{A}_r$$

We can note that $\boldsymbol{\beta}_{int}$, is still the solution to

$$\left\|\frac{\boldsymbol{y}}{\sqrt{d}} - \boldsymbol{\beta}^{\top}\boldsymbol{X}_r\right\|^2, \quad \text{where} \quad \frac{\boldsymbol{y}}{\sqrt{d}} = \boldsymbol{\beta}_*^{\top}(\boldsymbol{Z}_r + \boldsymbol{A}_r) + \frac{\boldsymbol{\varepsilon}}{\sqrt{d}}.$$

We define

$$\frac{\boldsymbol{\varepsilon}}{\sqrt{d}} =: \boldsymbol{\varepsilon}_r \sim \mathcal{N}\left(0, \frac{\tau_{\varepsilon}^2}{d}\right), \quad \tau_{\varepsilon,r}^2 := \frac{\tau_{\varepsilon}^2}{d}.$$

Then when we want to test, we shall look at the rescaled error

$$\frac{1}{\tilde{n}}\left\|\boldsymbol{\beta}_*^{\top}(\tilde{\alpha}_Z\tilde{\boldsymbol{Z}}_r + \tilde{\alpha}_A\tilde{\boldsymbol{A}}_r) - \boldsymbol{\beta}_{int}^{\top}(\tilde{\boldsymbol{Z}}_r + \tilde{\boldsymbol{A}}_r)\right\|_F^2$$

**Through Steps 1 - 4, we shall drop the subscript** $\mathbf{r}$.

## E.2 STEP 1: DECOMPOSE ERROR

Using the fact that $\tilde{A}$ has been zero entries and is independent of $\tilde{Z}$, we see that we can decompose the error as follows. Again here we consider $\tilde{n}$ samples of test data and take the average (in expectation, this is the same as one test point).

$$\mathbb{E}\left[\frac{1}{\tilde{n}}\left\|\boldsymbol{\beta}_*^\top(\tilde{\alpha}_z\tilde{\boldsymbol{Z}} + \tilde{\alpha}_A\tilde{\boldsymbol{A}}) - \boldsymbol{\beta}_{int}^\top(\tilde{\boldsymbol{Z}} + \tilde{\boldsymbol{A}})\right\|_F^2\right]$$

$$= \mathbb{E}\left[\frac{1}{\tilde{n}}\left\|\tilde{\alpha}_z\boldsymbol{\beta}_*^\top\tilde{\boldsymbol{Z}} - \boldsymbol{\beta}_{int}^\top\tilde{\boldsymbol{Z}}\right\|_F^2\right] + \mathbb{E}\left[\frac{1}{\tilde{n}}\left\|\tilde{\alpha}_A\boldsymbol{\beta}_*^\top\tilde{\boldsymbol{A}} - \boldsymbol{\beta}_{int}^\top\tilde{\boldsymbol{A}}\right\|_F^2\right]$$

$$= \mathbb{E}\left[\underbrace{\frac{1}{\tilde{n}}\left\|\tilde{\alpha}_z\boldsymbol{\beta}_*^\top\tilde{\boldsymbol{Z}} - \boldsymbol{\beta}_{int}^\top\tilde{\boldsymbol{Z}}\right\|_F^2}_{\text{Bias}} + \underbrace{\frac{1}{\tilde{n}}\left\|\boldsymbol{\beta}_{int}^\top\tilde{\boldsymbol{A}}\right\|_F^2}_{\text{Variance}} + \underbrace{\frac{1}{\tilde{n}}\tilde{\alpha}_A^2\left\|\boldsymbol{\beta}_*^\top\tilde{\boldsymbol{A}}\right\|_F^2}_{\text{Data Noise}} + \underbrace{\left(-\frac{2}{\tilde{n}}\tilde{\alpha}_A\boldsymbol{\beta}_*^\top\tilde{\boldsymbol{A}}\tilde{\boldsymbol{A}}^\top\boldsymbol{\beta}_{int}\right)}_{\text{Target Alignment}}\right].$$

We compute these four terms one by one in the following sections.

## E.3 STEP 2: SIMPLIFYING TERMS

This section simplifies the four terms. We begin by recalling results from prior work. We state them here for completeness.

**Theorem 6** (Theorems 3, 5 of Meyer (1973)). *Define the following helper functions* $\boldsymbol{h} = \boldsymbol{v}^\top\boldsymbol{A}^\dagger$, $\boldsymbol{k} = \boldsymbol{A}^\dagger\boldsymbol{u}$, $\boldsymbol{t} = \boldsymbol{v}^\top(\boldsymbol{I} - \boldsymbol{A}^\dagger\boldsymbol{A})$, $\xi = 1 + \eta\boldsymbol{v}^\top\boldsymbol{A}^\dagger\boldsymbol{u}$, $\boldsymbol{s} = (\boldsymbol{I} - \boldsymbol{A}\boldsymbol{A}^\dagger)\boldsymbol{u}$, $\gamma_1 = \eta^2\|\boldsymbol{t}\|^2\|\boldsymbol{k}\|^2 + \xi^2$, $\gamma_2 = \eta^2\|\boldsymbol{s}\|^2\|\boldsymbol{h}\|^2 + \xi^2$ *and*

$$\boldsymbol{p}_1 = -\frac{\eta^2\|\boldsymbol{k}\|^2}{\xi}\boldsymbol{t}^\top - \eta\boldsymbol{k}, \qquad\qquad \boldsymbol{q}_1^\top = -\frac{\eta\|\boldsymbol{t}\|^2}{\xi}\boldsymbol{k}^\top\boldsymbol{A}^\dagger - \boldsymbol{h}.$$

$$\boldsymbol{p}_2 = -\frac{\eta^2\|\boldsymbol{s}\|^2}{\xi}\boldsymbol{A}^\dagger\boldsymbol{h}^\top - \eta\boldsymbol{k}, \qquad\qquad \boldsymbol{q}_2^\top = -\frac{\eta\|\boldsymbol{h}\|^2}{\xi}\boldsymbol{s}^\top - \boldsymbol{h},$$

*Then we have that*

$$(\boldsymbol{Z} + \boldsymbol{A})^\dagger = \begin{cases} \boldsymbol{A}^\dagger + \frac{\eta}{\xi}\boldsymbol{t}^\top\boldsymbol{k}^\top\boldsymbol{A}^\dagger - \frac{\xi}{\gamma_1}\boldsymbol{p}_1\boldsymbol{q}_1^\top, & c < 1 \\ \boldsymbol{A}^\dagger + \frac{\eta}{\xi}\boldsymbol{A}^\dagger\boldsymbol{h}^\top\boldsymbol{s}^\top - \frac{\xi}{\gamma_2}\boldsymbol{p}_2\boldsymbol{q}_2^\top, & c > 1 \end{cases}.$$

The following subsections - Bias E.3.1, Variance E.3.2, Data Noise E.3.3, and Target Alignment E.3.4 - present the linear algebraic simplifications of the results. To derive this results. We shall need some helper results that are presented in Section E.3.5.

### E.3.1 BIAS

Using Lemma 5, we have that if $c < 1$

$$\tilde{\alpha}_z\boldsymbol{\beta}_*^\top\tilde{\boldsymbol{Z}} - \boldsymbol{\beta}_{int}^\top\tilde{\boldsymbol{Z}} = \left[\tilde{\alpha}_Z - \alpha_Z + \frac{\xi}{\gamma_1}(\alpha_Z - \alpha_A)\right]\boldsymbol{\beta}_*^\top\tilde{\boldsymbol{Z}} + \frac{\tilde{\eta}}{\eta}\frac{\xi}{\gamma_1}\boldsymbol{\varepsilon}^\top\boldsymbol{p}_1\tilde{\boldsymbol{v}}^\top,$$

and if $c > 1$

$$\tilde{\alpha}_z\boldsymbol{\beta}_*^\top\tilde{\boldsymbol{Z}} - \boldsymbol{\beta}_{int}^\top\tilde{\boldsymbol{Z}} = \boldsymbol{\beta}_*^\top\left[(\tilde{\alpha}_Z - \alpha_Z)\boldsymbol{I} + \frac{\xi}{\gamma_2}(\alpha_Z\boldsymbol{I} - \alpha_A\boldsymbol{A}\boldsymbol{A}^\dagger)\right]\tilde{\boldsymbol{Z}} - \alpha_A\frac{\eta\|\boldsymbol{s}\|^2}{\gamma_2}\boldsymbol{\beta}_*^\top\boldsymbol{h}^\top\boldsymbol{u}^\top\tilde{\boldsymbol{Z}} + \frac{\tilde{\eta}}{\eta}\frac{\xi}{\gamma_2}\boldsymbol{\varepsilon}^\top\boldsymbol{p}_2\tilde{\boldsymbol{v}}^\top.$$

The bias equals the expected squared norm of this term (divided by $\tilde{n}$).

### E.3.2 VARIANCE

Lemma 8 gives us that

$$\mathbb{E}\left[\frac{1}{\tilde{n}}\left\|\boldsymbol{\beta}_{int}^\top\tilde{\boldsymbol{A}}\right\|_F^2\right] = \mathbb{E}\left[\frac{\tilde{\tau}^2\alpha_z^2}{d}\boldsymbol{\beta}_*^\top\boldsymbol{Z}(\boldsymbol{Z} + \boldsymbol{A})^\dagger(\boldsymbol{Z} + \boldsymbol{A})^{\dagger\top}\boldsymbol{Z}\boldsymbol{\beta}_* + \frac{\tilde{\tau}^2\alpha_A^2}{d}\boldsymbol{\beta}_*^\top\boldsymbol{A}(\boldsymbol{Z} + \boldsymbol{A})^\dagger(\boldsymbol{Z} + \boldsymbol{A})^{\dagger\top}\boldsymbol{A}^\top\boldsymbol{\beta}_*\right.$$

$$\left. + \frac{2\tilde{\tau}^2\alpha_A\alpha_z}{d}\boldsymbol{\beta}_*^\top\boldsymbol{Z}(\boldsymbol{Z} + \boldsymbol{A})^\dagger(\boldsymbol{Z} + \boldsymbol{A})^{\dagger\top}\boldsymbol{A}^\top\boldsymbol{\beta}_* + \frac{\tilde{\tau}^2}{d}\boldsymbol{\varepsilon}^\top(\boldsymbol{Z} + \boldsymbol{A})^\dagger(\boldsymbol{Z} + \boldsymbol{A})^{\dagger\top}\boldsymbol{\varepsilon}\right].$$

### E.3.3 DATA NOISE

The data noise term is the simplest to understand. Preliminary calculation gives us:

$$\frac{1}{\tilde{n}}\tilde{\alpha}_A^2 \mathbb{E}_{\tilde{A}}\left[\left\|\boldsymbol{\beta}_*^\top \tilde{\boldsymbol{A}}\right\|_F^2\right] = \frac{\tilde{\alpha}_A^2}{\tilde{n}}\frac{\tilde{\rho}^2 \tilde{n}}{d}\|\boldsymbol{\beta}_*\|^2 = \frac{\tilde{\alpha}_A^2 \tilde{\rho}^2}{d}\|\boldsymbol{\beta}_*\|^2.$$

### E.3.4 TARGET ALIGNMENT

To understand this term, we first note that $\tilde{\boldsymbol{A}}$ is independent of everything else. Hence we replace $\tilde{\boldsymbol{A}}\tilde{\boldsymbol{A}}^\top$ with its expectation $\frac{\tilde{\rho}^2 \tilde{n}}{d}\boldsymbol{I}$.

$$\mathbb{E}_{\tilde{A}}\left[-\frac{2}{\tilde{n}}\tilde{\alpha}_A \boldsymbol{\beta}_*^\top \tilde{\boldsymbol{A}}\tilde{\boldsymbol{A}}^\top \boldsymbol{\beta}_{int}\right] = -\frac{2}{\tilde{n}}\frac{\tilde{\rho}^2 \tilde{n}}{d}\tilde{\alpha}_A \boldsymbol{\beta}_*^\top \boldsymbol{\beta}_{int} = -\frac{2\tilde{\alpha}_A \tilde{\rho}^2}{d}\boldsymbol{\beta}_*^\top \boldsymbol{\beta}_{int}.$$

Since $\boldsymbol{\varepsilon}$ has mean-zero entries that are independent of everything else. We see that

$$\mathbb{E}_{\varepsilon}\left[\boldsymbol{\beta}_*^\top \boldsymbol{\beta}_{int}\right] = \mathbb{E}_{\varepsilon}\left[\boldsymbol{\beta}_*^\top \left((\alpha_z \boldsymbol{\beta}_*^\top \boldsymbol{Z} + \boldsymbol{\varepsilon}^\top)(\boldsymbol{Z} + \boldsymbol{A})^\dagger + \alpha_A \boldsymbol{\beta}_*^\top \boldsymbol{A}(\boldsymbol{Z} + \boldsymbol{A})^\dagger\right)^\top\right] \quad (5)$$

$$= \boldsymbol{\beta}_*^\top \left(\alpha_z \boldsymbol{\beta}_*^\top \boldsymbol{Z}(\boldsymbol{Z} + \boldsymbol{A})^\dagger - \alpha_A \boldsymbol{\beta}_*^\top \boldsymbol{A}(\boldsymbol{Z} + \boldsymbol{A})^\dagger\right)^\top \quad (6)$$

$$= \alpha_z \boldsymbol{\beta}_*^\top (\boldsymbol{Z} + \boldsymbol{A})^{\dagger\top}\boldsymbol{Z}^\top \boldsymbol{\beta}_* + \alpha_A \boldsymbol{\beta}_*^\top (\boldsymbol{Z} + \boldsymbol{A})^{\dagger\top}\boldsymbol{A}^\top \boldsymbol{\beta}_*. \quad (7)$$

### E.3.5 HELPER LEMMAS

**Proposition 1** (Proposition 2 from Sonthalia and Nadakuditi (2023)). *In the setting from Section 2*

$$\boldsymbol{Z}(\boldsymbol{Z}+\boldsymbol{A})^\dagger = \begin{cases} \frac{\eta\xi}{\gamma_1}\boldsymbol{u}\boldsymbol{h} + \frac{\eta^2\|\boldsymbol{t}\|^2}{\gamma_1}\boldsymbol{u}\boldsymbol{k}^\top \boldsymbol{A}^\dagger, & c < 1 \\ \frac{\eta\xi}{\gamma_2}\boldsymbol{u}\boldsymbol{h} + \frac{\eta^2\|\boldsymbol{h}\|^2}{\gamma_2}\boldsymbol{u}\boldsymbol{s}^\top, & c > 1 \end{cases}.$$

**Lemma 1.** *If $\xi \neq 0$ and $\boldsymbol{A}$ has full rank, we have:*

$$\boldsymbol{\varepsilon}^\top (\boldsymbol{Z}+\boldsymbol{A})^\dagger \tilde{\boldsymbol{Z}} = \begin{cases} -\frac{\tilde{\eta}\xi}{\eta\gamma_1}\boldsymbol{\varepsilon}^\top \boldsymbol{p}_1 \tilde{\boldsymbol{v}}^\top & c < 1 \\ -\frac{\tilde{\eta}\xi}{\eta\gamma_2}\boldsymbol{\varepsilon}^\top \boldsymbol{p}_2 \tilde{\boldsymbol{v}}^\top & c > 1 \end{cases}.$$

*Proof.* After substitutions, Proposition 1 implies that for $c < 1$, $\boldsymbol{\varepsilon}^\top (\boldsymbol{Z}+\boldsymbol{A})^\dagger \tilde{\boldsymbol{Z}}$ becomes:

$$\boldsymbol{\varepsilon}^\top \left(\boldsymbol{A}^\dagger + \frac{\eta}{\xi}\boldsymbol{t}^\top \boldsymbol{k}^\top \boldsymbol{A}^\dagger - \frac{\xi}{\gamma_1}\boldsymbol{p}_1\left(-\frac{\eta\|\boldsymbol{t}\|^2}{\xi}\boldsymbol{k}^\top \boldsymbol{A}^\dagger - \boldsymbol{h}\right)\right)\tilde{\boldsymbol{Z}}$$

$$= \tilde{\eta}\boldsymbol{\varepsilon}^\top \left(\boldsymbol{A}^\dagger \boldsymbol{u}\tilde{\boldsymbol{v}}^\top + \frac{\eta}{\xi}\boldsymbol{t}^\top \boldsymbol{k}^\top \boldsymbol{A}^\dagger \boldsymbol{u}\tilde{\boldsymbol{v}}^\top - \frac{\xi}{\gamma_1}\boldsymbol{p}_1\left(-\frac{\eta\|\boldsymbol{t}\|^2}{\xi}\boldsymbol{k}^\top \boldsymbol{A}^\dagger \boldsymbol{u} - \boldsymbol{h}\boldsymbol{u}\right)\tilde{\boldsymbol{v}}^\top\right) \quad \text{by } \tilde{\boldsymbol{Z}} = \tilde{\eta}\boldsymbol{u}\tilde{\boldsymbol{v}}^\top.$$

Since $\boldsymbol{k} = \boldsymbol{A}^\dagger \boldsymbol{u}$ and $\boldsymbol{h}\boldsymbol{u} = \boldsymbol{v}^\top \boldsymbol{A}^\dagger \boldsymbol{u} = \frac{\xi-1}{\eta}$, we then have that

$$\tilde{\eta}\boldsymbol{\varepsilon}^\top \left(\boldsymbol{A}^\dagger \boldsymbol{u}\tilde{\boldsymbol{v}}^\top + \frac{\eta}{\xi}\boldsymbol{t}^\top \boldsymbol{k}^\top \boldsymbol{A}^\dagger \boldsymbol{u}\tilde{\boldsymbol{v}}^\top - \frac{\xi}{\gamma_1}\boldsymbol{p}_1\left(-\frac{\eta\|\boldsymbol{t}\|^2}{\xi}\boldsymbol{k}^\top \boldsymbol{A}^\dagger \boldsymbol{u} - \boldsymbol{h}\boldsymbol{u}\right)\tilde{\boldsymbol{v}}^\top\right)$$

$$= \tilde{\eta}\boldsymbol{\varepsilon}^\top \left(\boldsymbol{k}\tilde{\boldsymbol{v}}^\top + \frac{\eta\|\boldsymbol{k}\|^2}{\xi}\boldsymbol{t}^\top \tilde{\boldsymbol{v}}^\top + \frac{\xi}{\gamma_1}\boldsymbol{p}_1\left(\frac{\eta^2\|\boldsymbol{t}\|^2\|\boldsymbol{k}\|^2 + \xi^2 - \xi}{\xi\eta}\right)\tilde{\boldsymbol{v}}^\top\right)$$

$$= \tilde{\eta}\boldsymbol{\varepsilon}^\top \left(\boldsymbol{k}\tilde{\boldsymbol{v}}^\top + \frac{\eta\|\boldsymbol{k}\|^2}{\xi}\boldsymbol{t}^\top \tilde{\boldsymbol{v}}^\top + \frac{1}{\gamma_1}\boldsymbol{p}_1\left(\frac{\gamma_1 - \xi}{\eta}\right)\tilde{\boldsymbol{v}}^\top\right)$$

$$= \tilde{\eta}\boldsymbol{\varepsilon}^\top \left(\frac{1}{\eta}\left(\frac{\eta^2\|\boldsymbol{k}\|^2}{\xi}\boldsymbol{t}^\top + \eta\boldsymbol{k}\right)\tilde{\boldsymbol{v}}^\top + \frac{1}{\eta}\boldsymbol{p}_1\tilde{\boldsymbol{v}}^\top - \frac{\xi}{\eta\gamma_1}\boldsymbol{p}_1\tilde{\boldsymbol{v}}^\top\right)$$

$$= \boldsymbol{\varepsilon}^\top \left(-\frac{\tilde{\eta}}{\eta}\boldsymbol{p}_1\tilde{\boldsymbol{v}}^\top + \frac{\tilde{\eta}}{\eta}\boldsymbol{p}_1\tilde{\boldsymbol{v}}^\top - \frac{\tilde{\eta}\xi}{\eta\gamma_1}\boldsymbol{p}_1\tilde{\boldsymbol{v}}^\top\right)$$

$$= -\frac{\tilde{\eta}\xi}{\eta\gamma_1}\boldsymbol{\varepsilon}^\top \boldsymbol{p}_1\tilde{\boldsymbol{v}}^\top.$$

For $c > 1$, we note that the calculation is exactly the same. An example of such a calculation can be seen in the proof of Lemma 4. $\qquad\square$

**Lemma 2.** *In the setting of Section 2, we have:*

$$
\boldsymbol{A}(\boldsymbol{Z} + \boldsymbol{A})^\dagger = \begin{cases} \boldsymbol{I} - \frac{\eta\xi}{\gamma_1}\boldsymbol{u}\boldsymbol{h} + \frac{\eta^2\|\boldsymbol{t}\|^2}{\gamma_1}\boldsymbol{u}\boldsymbol{k}^\top\boldsymbol{A}^\dagger, & c < 1 \\ \boldsymbol{A}\boldsymbol{A}^\dagger + \frac{\eta\xi}{\gamma_2}\boldsymbol{h}^\top\boldsymbol{s}^\top - \frac{\eta^2\|\boldsymbol{s}\|^2}{\gamma_2}\boldsymbol{h}^\top\boldsymbol{h} - \frac{\eta^2\|\boldsymbol{h}\|^2}{\gamma_2}\boldsymbol{A}\boldsymbol{A}^\dagger\boldsymbol{u}\boldsymbol{s}^\top - \frac{\eta\xi}{\gamma_2}\boldsymbol{A}\boldsymbol{A}^\dagger\boldsymbol{u}\boldsymbol{h}, & c > 1 \end{cases}.
$$

*Proof.* For $c < 1$, $\boldsymbol{Z}, \boldsymbol{A}$ are $d \times n$ with $d < n$. Since $\boldsymbol{A}$ is assumed to have full rank, $\boldsymbol{Z} + \boldsymbol{A}$ has full rank with probability 1, and hence

$$
(\boldsymbol{Z} + \boldsymbol{A})(\boldsymbol{Z} + \boldsymbol{A})^\dagger = \boldsymbol{I}.
$$

Thus, from Proposition 1,

$$
\boldsymbol{A}(\boldsymbol{Z} + \boldsymbol{A})^\dagger = (\boldsymbol{Z} + \boldsymbol{A})(\boldsymbol{Z} + \boldsymbol{A})^\dagger - \boldsymbol{Z}(\boldsymbol{Z} + \boldsymbol{A})^\dagger = \boldsymbol{I} - \frac{\eta\xi}{\gamma_1}\boldsymbol{u}\boldsymbol{h} - \frac{\eta^2\|\boldsymbol{t}\|^2}{\gamma_1}\boldsymbol{u}\boldsymbol{k}^\top\boldsymbol{A}^\dagger.
$$

For $c > 1$, since $(\boldsymbol{Z} + \boldsymbol{A})(\boldsymbol{Z} + \boldsymbol{A})^\dagger$ is no longer the identity matrix, we directly expand using Theorem 6:

$$
\boldsymbol{A}(\boldsymbol{Z} + \boldsymbol{A})^\dagger = \boldsymbol{A}\left(\boldsymbol{A}^\dagger + \frac{\eta}{\xi}\boldsymbol{A}^\dagger\boldsymbol{h}^\top\boldsymbol{s}^\top - \frac{\xi}{\gamma_2}\left(\frac{\eta^2\|\boldsymbol{s}\|^2}{\xi}\boldsymbol{A}^\dagger\boldsymbol{h}^\top + \eta\boldsymbol{k}\right)\left(\frac{\eta\|\boldsymbol{h}\|^2}{\xi}\boldsymbol{s}^\top + \boldsymbol{h}\right)\right)
$$

$$
= \boldsymbol{A}\boldsymbol{A}^\dagger + \frac{\eta}{\xi}\boldsymbol{A}\boldsymbol{A}^\dagger\boldsymbol{h}^\top\boldsymbol{s}^\top - \frac{\xi}{\gamma_2}\left(\frac{\eta^2\|\boldsymbol{s}\|^2}{\xi}\boldsymbol{A}\boldsymbol{A}^\dagger\boldsymbol{h}^\top + \eta\boldsymbol{A}\boldsymbol{A}^\dagger\boldsymbol{u}\right)\left(\frac{\eta\|\boldsymbol{h}\|^2}{\xi}\boldsymbol{s}^\top + \boldsymbol{h}\right).
$$

Noting that $\boldsymbol{A}\boldsymbol{A}^\dagger\boldsymbol{h}^\top = \boldsymbol{A}\boldsymbol{A}^\dagger\boldsymbol{A}^{\dagger\top}\boldsymbol{v} = \boldsymbol{A}^{\dagger\top}\boldsymbol{v} = \boldsymbol{h}^\top$, we have

$$
\boldsymbol{A}(\boldsymbol{Z} + \boldsymbol{A})^\dagger = \boldsymbol{A}\boldsymbol{A}^\dagger + \frac{\eta}{\xi}\boldsymbol{h}^\top\boldsymbol{s}^\top - \frac{\xi}{\gamma_2}\left(\frac{\eta^2\|\boldsymbol{s}\|^2}{\xi}\boldsymbol{h}^\top + \eta\boldsymbol{A}\boldsymbol{A}^\dagger\boldsymbol{u}\right)\left(\frac{\eta\|\boldsymbol{h}\|^2}{\xi}\boldsymbol{s}^\top + \boldsymbol{h}\right)
$$

$$
= \boldsymbol{A}\boldsymbol{A}^\dagger + \frac{\eta}{\xi}\boldsymbol{h}^\top\boldsymbol{s}^\top - \frac{\eta^3\|\boldsymbol{s}\|^2\|\boldsymbol{h}\|^2}{\xi\gamma_2}\boldsymbol{h}^\top\boldsymbol{s}^\top - \frac{\eta^2\|\boldsymbol{s}\|^2}{\gamma_2}\boldsymbol{h}^\top\boldsymbol{h} - \frac{\eta^2\|\boldsymbol{h}\|^2}{\gamma_2}\boldsymbol{A}\boldsymbol{A}^\dagger\boldsymbol{u}\boldsymbol{s}^\top - \frac{\eta\xi}{\gamma_2}\boldsymbol{A}\boldsymbol{A}^\dagger\boldsymbol{u}\boldsymbol{h}.
$$

We can combine the coefficients in front of $\boldsymbol{h}^\top\boldsymbol{s}^\top$ to get

$$
\frac{\eta}{\xi} - \frac{\eta^3\|\boldsymbol{s}\|^2\|\boldsymbol{h}\|^2}{\xi\gamma_2} = \frac{\eta(\eta^2\|\boldsymbol{s}\|^2\|\boldsymbol{h}\|^2 + \xi^2) - \eta^3\|\boldsymbol{s}\|^2\|\boldsymbol{h}\|^2}{\xi\gamma_2} = \frac{\eta\xi}{\gamma_2}.
$$

The statement follows from here. $\qquad\square$

**Lemma 3.** *If $\xi \neq 0$ and $\boldsymbol{A}$ has full rank, we have:*

$$
\boldsymbol{\beta}_*^\top\boldsymbol{Z}(\boldsymbol{Z} + \boldsymbol{A})^\dagger\tilde{\boldsymbol{Z}} = \begin{cases} \left(1 - \frac{\xi}{\gamma_1}\right)\boldsymbol{\beta}_*^\top\tilde{\boldsymbol{Z}} & c < 1 \\ \left(1 - \frac{\xi}{\gamma_2}\right)\boldsymbol{\beta}_*^\top\tilde{\boldsymbol{Z}} & c > 1 \end{cases}.
$$

*Proof.* Using Proposition 1 for $c < 1$ and $\tilde{\boldsymbol{Z}} = \tilde{\eta}\boldsymbol{u}\tilde{\boldsymbol{v}}^\top$, we have that

$$
\boldsymbol{\beta}_*^\top\boldsymbol{Z}(\boldsymbol{Z} + \boldsymbol{A})^\dagger\tilde{\boldsymbol{Z}} = \boldsymbol{\beta}_*^\top\left(\frac{\eta\xi}{\gamma_1}\boldsymbol{u}\boldsymbol{h} + \frac{\eta^2\|\boldsymbol{t}\|^2}{\gamma_1}\boldsymbol{u}\boldsymbol{k}^\top\boldsymbol{A}^\dagger\right)\tilde{\boldsymbol{Z}}
$$

$$
= \tilde{\eta}\boldsymbol{\beta}_*^\top\left(\frac{\eta\xi}{\gamma_1}\boldsymbol{u}\boldsymbol{h}\boldsymbol{u}\tilde{\boldsymbol{v}}^\top + \frac{\eta^2\|\boldsymbol{t}\|^2}{\gamma_1}\boldsymbol{u}\boldsymbol{k}^\top\boldsymbol{A}^\dagger\boldsymbol{u}\tilde{\boldsymbol{v}}^\top\right)
$$

$$
= \tilde{\eta}\boldsymbol{\beta}_*^\top\left(\frac{\eta\xi}{\gamma_1}\boldsymbol{u}\boldsymbol{v}^\top\boldsymbol{A}^\dagger\boldsymbol{u}\tilde{\boldsymbol{v}}^\top + \frac{\eta^2\|\boldsymbol{t}\|^2}{\gamma_1}\boldsymbol{u}\boldsymbol{k}^\top\boldsymbol{A}^\dagger\boldsymbol{u}\tilde{\boldsymbol{v}}^\top\right).
$$

Note $\xi - 1 = \eta\boldsymbol{v}^\top\boldsymbol{A}^\dagger\boldsymbol{u}$, $\boldsymbol{k}\boldsymbol{A}^\dagger\boldsymbol{u} = \boldsymbol{k}^\top\boldsymbol{k} = \|\boldsymbol{k}\|^2$. The above equation becomes

$$
\tilde{\eta}\boldsymbol{\beta}_*^\top\left(\frac{\xi(\xi - 1)}{\gamma_1} + \frac{\eta^2\|\boldsymbol{t}\|^2\|\boldsymbol{k}\|^2}{\gamma_1}\right)\boldsymbol{u}\tilde{\boldsymbol{v}}^\top = \boldsymbol{\beta}_*^\top\left(\frac{\xi(\xi - 1)}{\gamma_1} + \frac{\eta^2\|\boldsymbol{t}\|^2\|\boldsymbol{k}\|^2}{\gamma_1}\right)\tilde{\boldsymbol{Z}}^\top.
$$

Using $\gamma_1 = \eta^2\|\boldsymbol{t}\|^2\|\boldsymbol{k}\|^2 + \xi^2$ to combine the coefficients, we have that

$$
\frac{\xi(\xi - 1)}{\gamma_1} + \frac{\eta^2\|\boldsymbol{t}\|^2\|\boldsymbol{k}\|^2}{\gamma_1} = \frac{-\xi + \xi^2 + \eta^2\|\boldsymbol{t}\|^2\|\boldsymbol{k}\|^2}{\gamma_1} = \frac{-\xi + \gamma_1}{\gamma_1} = 1 - \frac{\xi}{\gamma_1}.
$$

This completes the proof for $c < 1$. Similarly, for $c > 1$, we obtain

$$
\begin{aligned}
\boldsymbol{\beta}_*^\top \boldsymbol{Z} (\boldsymbol{Z} + \boldsymbol{A})^\dagger \tilde{\boldsymbol{Z}} &= \boldsymbol{\beta}_*^\top \left( \frac{\eta \xi}{\gamma_2} \boldsymbol{u} \boldsymbol{h} + \frac{\eta^2 \|\boldsymbol{h}\|^2}{\gamma_2} \boldsymbol{u} \boldsymbol{s}^\top \right) \tilde{\boldsymbol{Z}} \\
&= \tilde{\eta} \boldsymbol{\beta}_*^\top \left( \frac{\eta \xi}{\gamma_2} \boldsymbol{u} \boldsymbol{h} \boldsymbol{u} \tilde{\boldsymbol{v}}^\top + \frac{\eta^2 \|\boldsymbol{h}\|^2}{\gamma_2} \boldsymbol{u} \boldsymbol{s}^\top \boldsymbol{u} \tilde{\boldsymbol{v}}^\top \right) \\
&= \tilde{\eta} \boldsymbol{\beta}_*^\top \left( \frac{\eta \xi}{\gamma_2} \boldsymbol{u} \boldsymbol{v}^\top \boldsymbol{A}^\dagger \boldsymbol{u} \tilde{\boldsymbol{v}}^\top + \frac{\eta^2 \|\boldsymbol{h}\|^2}{\gamma_2} \boldsymbol{u} \boldsymbol{s}^\top \boldsymbol{u} \tilde{\boldsymbol{v}}^\top \right).
\end{aligned}
$$

Note $\xi - 1 = \eta \boldsymbol{v}^\top \boldsymbol{A}^\dagger \boldsymbol{u}$, $\boldsymbol{s}^\top \boldsymbol{u} = \|\boldsymbol{s}\|^2$. The above equation becomes

$$
\tilde{\eta} \boldsymbol{\beta}_*^\top \left( \frac{\xi(\xi - 1)}{\gamma_2} + \frac{\eta^2 \|\boldsymbol{s}\|^2 \|\boldsymbol{h}\|^2}{\gamma_2} \right) \boldsymbol{u} \tilde{\boldsymbol{v}}^\top = \boldsymbol{\beta}_*^\top \left( \frac{\xi(\xi - 1)}{\gamma_2} + \frac{\eta^2 \|\boldsymbol{s}\|^2 \|\boldsymbol{h}\|^2}{\gamma_2} \right) \tilde{\boldsymbol{Z}}^\top.
$$

Using $\gamma_2 = \eta^2 \|\boldsymbol{s}\|^2 \|\boldsymbol{h}\|^2 + \xi^2$ to combine the coefficients, we have that

$$
\frac{\xi(\xi - 1)}{\gamma_2} + \frac{\eta^2 \|\boldsymbol{s}\|^2 \|\boldsymbol{h}\|^2}{\gamma_2} = \frac{-\xi + \xi^2 + \eta^2 \|\boldsymbol{t}\|^2 \|\boldsymbol{k}\|^2}{\gamma_2} = \frac{-\xi + \gamma_2}{\gamma_2} = 1 - \frac{\xi}{\gamma_2}.
$$

The target expression follows. $\qquad\square$

**Lemma 4.** *If $\xi \neq 0$ and $\boldsymbol{A}$ has full rank, we have:*

$$
\boldsymbol{\beta}_*^\top \boldsymbol{A} (\boldsymbol{Z} + \boldsymbol{A})^\dagger \tilde{\boldsymbol{Z}} = \begin{cases} \frac{\xi}{\gamma_1} \boldsymbol{\beta}_*^\top \tilde{\boldsymbol{Z}} & c < 1 \\ \frac{\eta \|\boldsymbol{s}\|^2}{\gamma_2} \boldsymbol{\beta}_*^\top \boldsymbol{h}^\top \boldsymbol{u}^\top \tilde{\boldsymbol{Z}} + \frac{\xi}{\gamma_2} \boldsymbol{\beta}_*^\top \boldsymbol{A} \boldsymbol{A}^\dagger \tilde{\boldsymbol{Z}} & c > 1 \end{cases}.
$$

*Proof.* We begin with $c < 1$. Since $\boldsymbol{A}$ is assumed to have full rank, $\boldsymbol{Z} + \boldsymbol{A}$ has full column rank with probability 1, and hence

$$
(\boldsymbol{Z} + \boldsymbol{A})(\boldsymbol{Z} + \boldsymbol{A})^\dagger = \boldsymbol{I}.
$$

It follows from Lemma 3 that

$$
\begin{aligned}
\boldsymbol{\beta}_*^\top \boldsymbol{A} (\boldsymbol{Z} + \boldsymbol{A})^\dagger \tilde{\boldsymbol{Z}} &= \boldsymbol{\beta}_*^\top (\boldsymbol{Z} + \boldsymbol{A})(\boldsymbol{Z} + \boldsymbol{A})^\dagger \tilde{\boldsymbol{Z}} - \boldsymbol{\beta}_*^\top \boldsymbol{Z} (\boldsymbol{Z} + \boldsymbol{A})^\dagger \tilde{\boldsymbol{Z}} \\
&= \boldsymbol{\beta}_*^\top \tilde{\boldsymbol{Z}} - \left( 1 - \frac{\xi}{\gamma_1} \right) \boldsymbol{\beta}_*^\top \tilde{\boldsymbol{Z}} = \frac{\xi}{\gamma_1} \boldsymbol{\beta}_*^\top \tilde{\boldsymbol{Z}}.
\end{aligned}
$$

For $c > 1$, $\boldsymbol{Z} + \boldsymbol{A}$ now has full row rank instead of full column rank. Hence, we do not have $(\boldsymbol{Z} + \boldsymbol{A})(\boldsymbol{Z} + \boldsymbol{A})^\dagger = \boldsymbol{I}$ and need to directly expand it using Theorem 6 and its helper variables:

$$\begin{aligned}
\boldsymbol{\beta}_*^\top \boldsymbol{A}(\boldsymbol{Z}+\boldsymbol{A})^\dagger \tilde{\boldsymbol{Z}} &= \boldsymbol{\beta}_*^\top \boldsymbol{A}\left(\boldsymbol{A}^\dagger + \frac{\eta}{\xi}\boldsymbol{A}^\dagger \boldsymbol{h}^\top \boldsymbol{s}^\top - \frac{\xi}{\gamma_2}\boldsymbol{p}_2\boldsymbol{q}_2^\top\right)\tilde{\boldsymbol{Z}} \\
&= \tilde{\eta}\boldsymbol{\beta}_*^\top \boldsymbol{A}\left(\boldsymbol{k}\tilde{\boldsymbol{v}}^\top + \frac{\eta\|\boldsymbol{s}\|^2}{\xi}\boldsymbol{A}^\dagger \boldsymbol{h}^\top \tilde{\boldsymbol{v}}^\top - \frac{\xi}{\gamma_2}\boldsymbol{p}_2\boldsymbol{q}_2^\top \boldsymbol{u}\tilde{\boldsymbol{v}}^\top\right) \\
&= \tilde{\eta}\boldsymbol{\beta}_*^\top \boldsymbol{A}\left(-\frac{1}{\eta}\boldsymbol{p}_2\tilde{\boldsymbol{v}}^\top - \frac{\xi}{\gamma_2}\boldsymbol{p}_2\left(-\frac{\eta\|\boldsymbol{h}\|^2}{\xi}\boldsymbol{s}^\top - \boldsymbol{h}\right)\boldsymbol{u}\tilde{\boldsymbol{v}}^\top\right) \\
&= \tilde{\eta}\boldsymbol{\beta}_*^\top \boldsymbol{A}\left(-\frac{1}{\eta}\boldsymbol{p}_2\tilde{\boldsymbol{v}}^\top + \frac{\xi}{\gamma_2}\boldsymbol{p}_2\left(\frac{\eta\|\boldsymbol{s}\|^2\|\boldsymbol{h}\|^2}{\xi} + \frac{\xi-1}{\eta}\right)\tilde{\boldsymbol{v}}^\top\right) \\
&= \tilde{\eta}\boldsymbol{\beta}_*^\top \boldsymbol{A}\left(-\frac{1}{\eta}\boldsymbol{p}_2\tilde{\boldsymbol{v}}^\top + \frac{\xi}{\gamma_2}\boldsymbol{p}_2\left(\frac{\eta^2\|\boldsymbol{s}\|^2\|\boldsymbol{h}\|^2 + \xi^2 - \xi}{\xi\eta}\right)\tilde{\boldsymbol{v}}^\top\right) \\
&= \tilde{\eta}\boldsymbol{\beta}_*^\top \boldsymbol{A}\left(-\frac{1}{\eta}\boldsymbol{p}_2\tilde{\boldsymbol{v}}^\top + \frac{\xi}{\gamma_2}\boldsymbol{p}_2\left(\frac{\gamma_2-\xi}{\xi\eta}\right)\tilde{\boldsymbol{v}}^\top\right) \\
&= \tilde{\eta}\boldsymbol{\beta}_*^\top \boldsymbol{A}\left(-\frac{1}{\eta}\boldsymbol{p}_2\tilde{\boldsymbol{v}}^\top + \frac{1}{\eta}\boldsymbol{p}_2\tilde{\boldsymbol{v}}^\top - \frac{\xi}{\eta\gamma_2}\boldsymbol{p}_2\tilde{\boldsymbol{v}}^\top\right) \\
&= -\frac{\tilde{\eta}\xi}{\eta\gamma_2}\boldsymbol{\beta}_*^\top \boldsymbol{A}\boldsymbol{p}_2\tilde{\boldsymbol{v}}^\top \\
&= \frac{\tilde{\eta}\xi}{\eta\gamma_2}\boldsymbol{\beta}_*^\top\left(\frac{\eta^2\|\boldsymbol{s}\|^2}{\xi}\boldsymbol{h}^\top + \eta\boldsymbol{A}\boldsymbol{k}\right)\tilde{\boldsymbol{v}}^\top \quad \text{by plugging in the expression of } \boldsymbol{p}_2 \\
&= \frac{\tilde{\eta}\eta\|\boldsymbol{s}\|^2}{\gamma_2}\boldsymbol{\beta}_*^\top \boldsymbol{h}^\top \tilde{\boldsymbol{v}}^\top + \frac{\xi}{\gamma_2}\boldsymbol{\beta}_*^\top \boldsymbol{A}\boldsymbol{A}^\dagger \tilde{\boldsymbol{Z}} \quad \text{by } \tilde{\eta}\boldsymbol{k}\tilde{\boldsymbol{v}}^\top = \boldsymbol{A}^\dagger \tilde{\eta}\boldsymbol{u}\tilde{\boldsymbol{v}}^\top = \boldsymbol{A}^\dagger \tilde{\boldsymbol{Z}}.
\end{aligned}$$

Noting that $\boldsymbol{\beta}_*^\top \boldsymbol{h}^\top$ is a scalar, we then introduce $1 = \boldsymbol{u}^\top \boldsymbol{u}$ and get that

$$\frac{\tilde{\eta}\eta\|\boldsymbol{s}\|^2}{\gamma_2}\boldsymbol{\beta}_*^\top \boldsymbol{h}^\top \boldsymbol{u}^\top \boldsymbol{u}\tilde{\boldsymbol{v}}^\top = \frac{\eta\|\boldsymbol{s}\|^2}{\gamma_2}\boldsymbol{\beta}_*^\top \boldsymbol{h}^\top \boldsymbol{u}^\top \tilde{\boldsymbol{Z}} \quad \text{since } \tilde{\eta}\boldsymbol{u}\tilde{\boldsymbol{v}}^\top = \tilde{\boldsymbol{Z}}.$$

Thus, the final expression is

$$\frac{\eta\|\boldsymbol{s}\|^2}{\gamma_2}\boldsymbol{\beta}_*^\top \boldsymbol{h}^\top \boldsymbol{u}^\top \tilde{\boldsymbol{Z}} + \frac{\xi}{\gamma_2}\boldsymbol{\beta}_*^\top \boldsymbol{A}\boldsymbol{A}^\dagger \tilde{\boldsymbol{Z}}.$$

$\square$

**Lemma 5** (Bias Term). *In the setting of Section 2, we have that if $c < 1$,*

$$\tilde{\alpha}_z\boldsymbol{\beta}_*^\top \tilde{\boldsymbol{Z}} - \boldsymbol{\beta}_{int}^\top \tilde{\boldsymbol{Z}} = \left[\tilde{\alpha}_Z - \alpha_Z + \frac{\xi}{\gamma_1}(\alpha_Z - \alpha_A)\right]\boldsymbol{\beta}_*^\top \tilde{\boldsymbol{Z}} + \frac{\tilde{\eta}}{\eta}\frac{\xi}{\gamma_1}\boldsymbol{\varepsilon}^\top \boldsymbol{p}_1\tilde{\boldsymbol{v}}^\top,$$

*and if $c > 1$,*

$$\tilde{\alpha}_z\boldsymbol{\beta}_*^\top \tilde{\boldsymbol{Z}} - \boldsymbol{\beta}_{int}^\top \tilde{\boldsymbol{Z}} = \boldsymbol{\beta}_*^\top\left[(\tilde{\alpha}_Z - \alpha_Z)\boldsymbol{I} + \frac{\xi}{\gamma_2}(\alpha_Z\boldsymbol{I} - \alpha_A\boldsymbol{A}\boldsymbol{A}^\dagger)\right]\tilde{\boldsymbol{Z}} - \alpha_A\frac{\eta\|\boldsymbol{s}\|^2}{\gamma_2}\boldsymbol{\beta}_*^\top \boldsymbol{h}^\top \boldsymbol{u}^\top \tilde{\boldsymbol{Z}} + \frac{\tilde{\eta}}{\eta}\frac{\xi}{\gamma_2}\boldsymbol{\varepsilon}^\top \boldsymbol{p}_2\tilde{\boldsymbol{v}}^\top.$$

*Proof.* To simplify the bias term, we first need the following expansion:

$$\begin{aligned}
\tilde{\alpha}_z\boldsymbol{\beta}_*^\top \tilde{\boldsymbol{Z}} - \boldsymbol{\beta}_{int}^\top \tilde{\boldsymbol{Z}} &= \tilde{\alpha}_z\boldsymbol{\beta}_*^\top \tilde{\boldsymbol{Z}} - (\boldsymbol{\beta}_*^\top(\alpha_z\boldsymbol{Z} + \alpha_A\boldsymbol{A}) + \boldsymbol{\varepsilon}^\top)(\boldsymbol{Z}+\boldsymbol{A})^\dagger \tilde{\boldsymbol{Z}} \\
&= \tilde{\alpha}_z\boldsymbol{\beta}_*^\top \tilde{\boldsymbol{Z}} - \alpha_z\boldsymbol{\beta}_*^\top \boldsymbol{Z}(\boldsymbol{Z}+\boldsymbol{A})^\dagger - \alpha_A\boldsymbol{\beta}_*^\top \boldsymbol{A}(\boldsymbol{Z}+\boldsymbol{A})^\dagger \tilde{\boldsymbol{Z}} - \boldsymbol{\varepsilon}^\top(\boldsymbol{Z}+\boldsymbol{A})^\dagger \tilde{\boldsymbol{Z}}.
\end{aligned}$$

From Lemmas 1, 3, 4, we get simplified expressions for $\boldsymbol{\varepsilon}^\top(\boldsymbol{Z}+\boldsymbol{A})^\dagger \tilde{\boldsymbol{Z}}, \boldsymbol{\beta}_*^\top \boldsymbol{A}(\boldsymbol{Z}+\boldsymbol{A})^\dagger \tilde{\boldsymbol{Z}}, \boldsymbol{\beta}_*^\top \boldsymbol{Z}(\boldsymbol{Z}+\boldsymbol{A})^\dagger$ and plug them in. For $c < 1$, we get

$$\begin{aligned}
&\tilde{\alpha}_Z\boldsymbol{\beta}_*^\top \tilde{\boldsymbol{Z}} - \alpha_Z\left(1 - \frac{\xi}{\gamma_1}\right)\boldsymbol{\beta}_*^\top \tilde{\boldsymbol{Z}} - \alpha_A\frac{\xi}{\gamma_1}\boldsymbol{\beta}_*^\top \tilde{\boldsymbol{Z}} + \frac{\tilde{\eta}}{\eta}\frac{\xi}{\gamma_1}\boldsymbol{\varepsilon}^\top \boldsymbol{p}_1\tilde{\boldsymbol{v}}^\top \\
&= \left[\tilde{\alpha}_Z - \alpha_Z + \frac{\xi}{\gamma_1}(\alpha_Z - \alpha_A)\right]\boldsymbol{\beta}_*^\top \tilde{\boldsymbol{Z}} + \frac{\tilde{\eta}}{\eta}\frac{\xi}{\gamma_1}\boldsymbol{\varepsilon}^\top \boldsymbol{p}_1\tilde{\boldsymbol{v}}^\top.
\end{aligned}$$

On the other hand, for $c > 1$, we have

$$\tilde{\alpha}_Z \boldsymbol{\beta}_*^\top \tilde{\boldsymbol{Z}} - \alpha_Z \left(1 - \frac{\xi}{\gamma_2}\right) \boldsymbol{\beta}_*^\top \tilde{\boldsymbol{Z}} - \alpha_A \left[\frac{\eta \|\boldsymbol{s}\|^2}{\gamma_2} \boldsymbol{\beta}_*^\top \boldsymbol{h}^\top \boldsymbol{u}^\top \tilde{\boldsymbol{Z}} + \frac{\xi}{\gamma_2} \boldsymbol{\beta}_*^\top \boldsymbol{A} \boldsymbol{A}^\dagger \tilde{\boldsymbol{Z}}\right] + \frac{\tilde{\eta}}{\eta} \frac{\xi}{\gamma_2} \boldsymbol{\varepsilon}^\top \boldsymbol{p}_2 \tilde{\boldsymbol{v}}^\top$$

$$= \boldsymbol{\beta}_*^\top \left[(\tilde{\alpha}_Z - \alpha_Z)\boldsymbol{I} + \frac{\xi}{\gamma_2}(\alpha_Z \boldsymbol{I} - \alpha_A \boldsymbol{A} \boldsymbol{A}^\dagger)\right] \tilde{\boldsymbol{Z}} - \alpha_A \frac{\eta \|\boldsymbol{s}\|^2}{\gamma_2} \boldsymbol{\beta}_*^\top \boldsymbol{h}^\top \boldsymbol{u}^\top \tilde{\boldsymbol{Z}} + \frac{\tilde{\eta}}{\eta} \frac{\xi}{\gamma_2} \boldsymbol{\varepsilon}^\top \boldsymbol{p}_2 \tilde{\boldsymbol{v}}^\top.$$

$\square$

**Lemma 6** (Squared Norms of $\boldsymbol{p}_1$ and $\boldsymbol{p}_2$). *Recall* $\boldsymbol{p}_1 = -\frac{\eta^2 \|\boldsymbol{k}\|^2}{\xi} \boldsymbol{t}^\top - \eta \boldsymbol{k}$ *and* $\boldsymbol{p}_2 = -\frac{\eta^2 \|\boldsymbol{s}\|^2}{\xi} \boldsymbol{A}^\dagger \boldsymbol{h} - \eta \boldsymbol{k}$.

1. $\|\boldsymbol{p}_1\|^2 = \frac{\eta^2 \|\boldsymbol{k}\|^2}{\xi^2} \gamma_1.$

2. $\|\boldsymbol{p}_2\|^2 = \frac{\eta^4 \|\boldsymbol{s}\|^4}{\xi^2} \boldsymbol{h} \boldsymbol{A}^{\dagger\top} \boldsymbol{A}^\dagger \boldsymbol{h}^\top + \frac{2\eta^3 \|\boldsymbol{s}\|^2}{\xi} \boldsymbol{k}^\top \boldsymbol{A}^\dagger \boldsymbol{h}^\top + \eta^2 \|\boldsymbol{k}\|^2.$

*Proof.* For $\boldsymbol{p}_1$, we have

$$\|\boldsymbol{p}_1\|^2 = \left(-\frac{\eta^2 \|\boldsymbol{k}\|^2}{\xi} \boldsymbol{t} - \eta \boldsymbol{k}\right)\left(-\frac{\eta^2 \|\boldsymbol{k}\|^2}{\xi} \boldsymbol{t}^\top - \eta \boldsymbol{k}^\top\right) = \left(\frac{\eta^2 \|\boldsymbol{k}\|^2}{\xi}\right)^2 \|\boldsymbol{t}\|^2 + 2\frac{\eta^3 \|\boldsymbol{k}\|^2}{\xi} \boldsymbol{t} \boldsymbol{k} + \eta^2 \|\boldsymbol{k}\|^2.$$

Using $\boldsymbol{t}\boldsymbol{k} = \boldsymbol{0}$ yields the first result, which we can further simplify as

$$\frac{\eta^2 \|\boldsymbol{k}\|^2}{\xi^2} \left(\eta^2 \|\boldsymbol{k}\|^2 \|\boldsymbol{t}\|^2 + \xi^2\right) = \frac{\eta^2 \|\boldsymbol{k}\|^2}{\xi^2} \gamma_1.$$

For $\boldsymbol{p}_2$, similarly, we have

$$\|\boldsymbol{p}_2\|^2 = \left(-\frac{\eta^2 \|\boldsymbol{s}\|^2}{\xi} \boldsymbol{h} \boldsymbol{A}^{\dagger\top} - \eta \boldsymbol{k}^\top\right)\left(-\frac{\eta^2 \|\boldsymbol{s}\|^2}{\xi} \boldsymbol{A}^\dagger \boldsymbol{h}^\top - \eta \boldsymbol{k}\right)$$

$$= \frac{\eta^4 \|\boldsymbol{s}\|^4}{\xi^2} \boldsymbol{h} \boldsymbol{A}^{\dagger\top} \boldsymbol{A}^\dagger \boldsymbol{h}^\top + \frac{2\eta^3 \|\boldsymbol{s}\|^2}{\xi} \boldsymbol{k}^\top \boldsymbol{A}^\dagger \boldsymbol{h}^\top + \eta^2 \|\boldsymbol{k}\|^2.$$

$\square$

**Lemma 7** (Squared Norms of $\boldsymbol{q}_1$ and $\boldsymbol{q}_2$). *Let* $\boldsymbol{q}_1^\top = -\frac{\eta \|\boldsymbol{t}\|^2}{\xi} \boldsymbol{k}^\top \boldsymbol{A}^\dagger - \boldsymbol{h}$ *and* $\boldsymbol{q}_2^\top = -\frac{\eta \|\boldsymbol{h}\|^2}{\xi} \boldsymbol{s}^\top - \boldsymbol{h}$.

1. $\|\boldsymbol{q}_1\|^2 = \frac{\eta^2 \|\boldsymbol{t}\|^4}{\xi^2} \boldsymbol{k}^\top \boldsymbol{A}^\dagger \boldsymbol{A}^{\dagger\top} \boldsymbol{k} + \frac{2\eta \|\boldsymbol{t}\|^2}{\xi} \boldsymbol{k}^\top \boldsymbol{A}^\dagger \boldsymbol{h}^\top + \|\boldsymbol{h}\|^2.$

2. $\|\boldsymbol{q}_2\|^2 = \frac{\|\boldsymbol{h}\|^2}{\xi^2} \gamma_2.$

*Proof.* Similar to Lemma 6, we directly expand the two terms:

$$\|\boldsymbol{q}_1\|^2 = \left(-\frac{\eta \|\boldsymbol{t}\|^2}{\xi} \boldsymbol{k}^\top \boldsymbol{A}^\dagger - \boldsymbol{h}\right)\left(-\frac{\eta \|\boldsymbol{t}\|^2}{\xi} \boldsymbol{A}^{\dagger\top} \boldsymbol{k} - \boldsymbol{h}^\top\right) = \frac{\eta^2 \|\boldsymbol{t}\|^4}{\xi^2} \boldsymbol{k}^\top \boldsymbol{A}^\dagger \boldsymbol{A}^{\dagger\top} \boldsymbol{k} + \frac{2\eta \|\boldsymbol{t}\|^2}{\xi} \boldsymbol{k}^\top \boldsymbol{A}^\dagger \boldsymbol{h}^\top + \|\boldsymbol{h}\|^2.$$

$$\|\boldsymbol{q}_2\|^2 = \left(-\frac{\eta \|\boldsymbol{h}\|^2}{\xi} \boldsymbol{s}^\top - \boldsymbol{h}\right)\left(-\frac{\eta \|\boldsymbol{h}\|^2}{\xi} \boldsymbol{s} - \boldsymbol{h}^\top\right) = \frac{\eta^2 \|\boldsymbol{h}\|^4 \|\boldsymbol{s}\|^2}{\xi^2} + \|\boldsymbol{h}\|^2 \quad \text{since } \boldsymbol{h}\boldsymbol{s} = \boldsymbol{0}$$

$$= \frac{\|\boldsymbol{h}\|^2 (\eta^2 \|\boldsymbol{h}\|^2 \|\boldsymbol{s}\|^2 + \xi^2)}{\xi^2}$$

$$= \frac{\|\boldsymbol{h}\|^2}{\xi^2} \gamma_2.$$

$\square$

**Lemma 8** (Preliminary Expansion of Variance). *In the setting of Section 2, we have*

$$\mathbb{E}\left[\frac{1}{\tilde{n}}\left\|\boldsymbol{\beta}_{int}^\top \tilde{\boldsymbol{A}}\right\|_F^2\right] = \mathbb{E}\left[\frac{\tilde{\tau}^2 \alpha_z^2}{d}\boldsymbol{\beta}_*^\top \boldsymbol{Z}(\boldsymbol{Z}+\boldsymbol{A})^\dagger(\boldsymbol{Z}+\boldsymbol{A})^{\dagger\top}\boldsymbol{Z}\boldsymbol{\beta}_* + \frac{\tilde{\tau}^2 \alpha_A^2}{d}\boldsymbol{\beta}_*^\top \boldsymbol{A}(\boldsymbol{Z}+\boldsymbol{A})^\dagger(\boldsymbol{Z}+\boldsymbol{A})^{\dagger\top}\boldsymbol{A}^\top \boldsymbol{\beta}_*\right.$$
$$\left.+ \frac{2\tilde{\tau}^2 \alpha_A \alpha_z}{d}\boldsymbol{\beta}_*^\top \boldsymbol{Z}(\boldsymbol{Z}+\boldsymbol{A})^\dagger(\boldsymbol{Z}+\boldsymbol{A})^{\dagger\top}\boldsymbol{A}^\top \boldsymbol{\beta}_* + \frac{\tilde{\tau}^2}{d}\boldsymbol{\varepsilon}^\top(\boldsymbol{Z}+\boldsymbol{A})^\dagger(\boldsymbol{Z}+\boldsymbol{A})^{\dagger\top}\boldsymbol{\varepsilon}\right].$$

*Proof.* Since $\tilde{\boldsymbol{A}}$ is independent of the other terms, we replace $\tilde{\boldsymbol{A}}\tilde{\boldsymbol{A}}^\top$ with its expectation $\frac{\tilde{\tau}^2\tilde{n}}{d}\boldsymbol{I}$.

$$\mathbb{E}\left[\frac{1}{\tilde{n}}\left\|\boldsymbol{\beta}_{int}^\top \tilde{\boldsymbol{A}}\right\|_F^2\right] = \mathbb{E}\left[\frac{1}{\tilde{n}}\boldsymbol{\beta}_{int}^\top \tilde{\boldsymbol{A}}\tilde{\boldsymbol{A}}^\top \boldsymbol{\beta}_{int}\right] = \frac{1}{\tilde{n}}\frac{\tilde{\tau}^2\tilde{n}}{d}\mathbb{E}\left[\boldsymbol{\beta}_{int}^\top \boldsymbol{\beta}_{int}\right] = \frac{\tilde{\tau}^2}{d}\mathbb{E}\left[\|\boldsymbol{\beta}_{int}\|^2\right].$$

We now plug in the expression for $\boldsymbol{\beta}_{int}$. Since $\boldsymbol{\varepsilon}$ is a zero-mean vector and independent from other random variables, terms with only one $\boldsymbol{\varepsilon}$ have zero expectation. A straightforward expansion gives:

$$\frac{\tilde{\tau}^2}{d}\|\boldsymbol{\beta}_{int}\|_F^2 = \frac{\tilde{\tau}^2}{d}(\boldsymbol{\beta}_*^\top(\alpha_z \boldsymbol{Z} + \alpha_A \boldsymbol{A}) + \boldsymbol{\varepsilon}^\top)(\boldsymbol{Z}+\boldsymbol{A})^\dagger(\boldsymbol{Z}+\boldsymbol{A})^{\dagger\top}(\boldsymbol{\beta}_*^\top(\alpha_z \boldsymbol{Z} + \alpha_A \boldsymbol{A}) + \boldsymbol{\varepsilon}^\top)^\top.$$

After eliminating zero expectations as above, the expectation becomes:

$$\mathbb{E}\left[\frac{\tilde{\tau}^2}{d}\|\boldsymbol{\beta}_{int}\|_F^2\right] = \mathbb{E}\left[\frac{\tilde{\tau}^2 \alpha_z^2}{d}\boldsymbol{\beta}_*^\top \boldsymbol{Z}(\boldsymbol{Z}+\boldsymbol{A})^\dagger(\boldsymbol{Z}+\boldsymbol{A})^{\dagger\top}\boldsymbol{Z}\boldsymbol{\beta}_* + \frac{\tilde{\tau}^2 \alpha_A^2}{d}\boldsymbol{\beta}_*^\top \boldsymbol{A}(\boldsymbol{Z}+\boldsymbol{A})^\dagger(\boldsymbol{Z}+\boldsymbol{A})^{\dagger\top}\boldsymbol{A}^\top \boldsymbol{\beta}_*\right.$$
$$\left.+ \frac{2\tilde{\tau}^2 \alpha_A \alpha_z}{d}\boldsymbol{\beta}_*^\top \boldsymbol{Z}(\boldsymbol{Z}+\boldsymbol{A})^\dagger(\boldsymbol{Z}+\boldsymbol{A})^{\dagger\top}\boldsymbol{A}^\top \boldsymbol{\beta}_* + \frac{\tilde{\tau}^2}{d}\boldsymbol{\varepsilon}^\top(\boldsymbol{Z}+\boldsymbol{A})^\dagger(\boldsymbol{Z}+\boldsymbol{A})^{\dagger\top}\boldsymbol{\varepsilon}\right].$$

$\square$

### E.4 STEP 3: RANDOM MATRIX THEORY ESTIMATES

To do the estimates we recall the set up. In particular, we have that

$$\boldsymbol{Z} = \eta\boldsymbol{u}\boldsymbol{v}^\top, \quad \text{where } \theta = \frac{\eta}{\sqrt{n}} \text{ and } \|\boldsymbol{v}\| = 1,$$

and the entries of

$$A_{ij} = \mathcal{N}\left(0, \frac{\rho^2}{d}\right)$$

Recall the following definition $\boldsymbol{h} = \boldsymbol{v}^\top \boldsymbol{A}^\dagger$, $\boldsymbol{k} = \boldsymbol{A}^\dagger \boldsymbol{u}$, $\boldsymbol{t} = \boldsymbol{v}^\top(\boldsymbol{I} - \boldsymbol{A}^\dagger \boldsymbol{A})$, $\xi = 1 + \eta\boldsymbol{v}^\top \boldsymbol{A}^\dagger \boldsymbol{u}$, $\boldsymbol{s} = (\boldsymbol{I} - \boldsymbol{A}\boldsymbol{A}^\dagger)\boldsymbol{u}$, $\gamma_1 = \eta^2\|\boldsymbol{t}\|^2\|\boldsymbol{k}\|^2 + \xi^2$, $\gamma_2 = \eta^2\|\boldsymbol{s}\|^2\|\boldsymbol{h}\|^2 + \xi^2$ and

$$\boldsymbol{p}_1 = -\frac{\eta^2\|\boldsymbol{k}\|^2}{\xi}\boldsymbol{t}^\top - \eta\boldsymbol{k}, \qquad\qquad \boldsymbol{q}_1^\top = -\frac{\eta\|\boldsymbol{t}\|^2}{\xi}\boldsymbol{k}^\top \boldsymbol{A}^\dagger - \boldsymbol{h}.$$

$$\boldsymbol{p}_2 = -\frac{\eta^2\|\boldsymbol{s}\|^2}{\xi}\boldsymbol{A}^\dagger \boldsymbol{h}^\top - \eta\boldsymbol{k}, \qquad\qquad \boldsymbol{q}_2^\top = -\frac{\eta\|\boldsymbol{h}\|^2}{\xi}\boldsymbol{s}^\top - \boldsymbol{h},$$

To show that each of the four terms, bias, variance, data noise, and target alignment concentrate in the limit, we do this in two steps.

  (a) First, we compute the mean and variance for basic building blocks such as $\|\boldsymbol{h}\|^2$ and other variables. Section E.4.1.

  (b) Second, we provide bounds on the higher moments. Section E.4.2.

  (c) Next, we prove bounds on the moments of $\gamma_i$. Section E.4.3.

### E.4.1 STEP 3(A): SHOWING THAT BASIC BUILDING BLOCKS CONCENTRATE

We begin by bounding the mean and variance.

**Lemma 9** (Generalized version of Lemma 7 from Sonthalia and Nadakuditi (2023)). *Suppose $A_{ij}$ have mean 0 and variance $\rho^2/d$, the entries are uncorrelated, have finite fourth moment, the distribution is invariant under left and right orthogonal transformation and the empirical spectral distribution of $\frac{1}{\rho^2}AA^\top$ converges to the Marchenko-Pastur law. Additionally, if $u$ and $v$ are fixed unit norm vectors. Then we have that*

1. $\mathbb{E}[\|h\|^2] = \begin{cases} \frac{1}{\rho^2}\frac{c^2}{1-c} & c < 1 \\ \frac{1}{\rho^2}\frac{c}{c-1} & c > 1 \end{cases} + o\left(\frac{1}{\rho^2}\right)$ *and* $\mathrm{Var}(\|h\|^2) = O\left(\frac{1}{\rho^4 n}\right)$.

2. $\mathbb{E}[\|k\|^2] = \frac{1}{\rho^2}\frac{c}{1-c} + o\left(\frac{1}{\rho^2}\right)$ *and* $\mathrm{Var}(\|k\|^2) = O\left(\frac{1}{\rho^4 n}\right)$.

3. $\mathbb{E}[\|s\|^2] = 1 - \frac{1}{c}$ *and* $\mathrm{Var}(\|s\|^2) = O\left(\frac{1}{d}\right)$.

4. $\mathbb{E}[\|t\|^2] = 1 - c$ *and* $\mathrm{Var}(\|t\|^2) = O\left(\frac{1}{n}\right)$.

5. $\mathbb{E}\left[\frac{\xi}{\eta}\right] = \frac{1}{\eta}$ *and* $\mathrm{Var}\left(\frac{\xi}{\eta}\right) = O\left(\frac{1}{\max(n,d)}\frac{1}{\rho^2}\right)$.

6. $\mathbb{E}\left[\frac{\xi^2}{\eta^2}\right] = \frac{1}{\eta^2} + \frac{1}{\max(n,d)}\frac{c}{\rho^2|1-c|} + o\left(\frac{1}{\max(n,d)\rho^2}\right) = \frac{1}{\eta^2} + O\left(\frac{1}{\max(n,d)\rho^2}\right)$
   *and* $\mathrm{Var}\left(\frac{\xi^2}{\eta^2}\right) = O\left(\frac{1}{\max(d,n)\rho^4}\right)$.

*Note that here $\max(d,n)$, $d$, $n$ are interchangeable in the variance big-Oh terms since they only differ by an absolute constant $c$. We include the details for completion.*

*Proof.* Items $1-5$ come from the original statement, which assumes unit variance. Here our variance parameter $\rho$ simply induces a multiplicative change. We now focus on item $6$.

Let $\zeta = \xi/\eta = 1/\eta + v^\top A^\dagger u$. With $A = U\Sigma V^\top$ (SVD), $A \in \mathbb{R}^{d\times n}$ having i.i.d. $\mathcal{N}(0, \rho^2/d)$ entries, and $u, v$ fixed unit vectors, we have $\zeta = \frac{1}{\eta} + \sum_{i=1}^r \frac{1}{\sigma_i}b_i a_i$, where $r = \min(d,n)$, $a = V^\top v$, $b = U^\top u$ are uniformly random on $S^{n-1}$ and $S^{d-1}$ respectively since $U, V$ are random rotations.

Since $A$ has zero-mean entries, only the non-cross terms remain in the expectation, and the fourth moment is

$$\mathbb{E}[\zeta^4] = \frac{1}{\eta^4} + \frac{6}{\eta^2}\sum_{i,j}\mathbb{E}\left[\frac{1}{\sigma_i\sigma_j}\right]\mathbb{E}[b_i b_j]\mathbb{E}[a_i a_j] + \sum_{i,j,k,l}\mathbb{E}\left[\frac{1}{\sigma_i\sigma_j\sigma_k\sigma_l}\right]\mathbb{E}[b_i b_j b_k b_l]\mathbb{E}[a_i a_j a_k a_l].$$

Furthermore, non-zero expectation terms require paired indices (since odd moments of the uniformly random vector on the sphere equals 0). In particular, using exact spherical moments, we have $\mathbb{E}[a_i^4] = \frac{3}{n(n+2)}$, $\mathbb{E}[a_i^2] = \frac{1}{n}$, $\mathbb{E}[a_i^2 a_j^2] = \frac{1}{n(n+2)}$ $(i \neq j)$, $\mathbb{E}[b_i^4] = \frac{3}{d(d+2)}$, $\mathbb{E}[b_i^2] = \frac{1}{d}$, $\mathbb{E}[b_i^2 b_j^2] = \frac{1}{d(d+2)}$ $(i \neq j)$:

$$\mathbb{E}[\zeta^4] = \frac{1}{\eta^4} + \frac{6}{\eta^2}\sum_{i=1}^r\mathbb{E}\left[\frac{1}{\sigma_i^2}\right]\frac{1}{dn} + \sum_{i=1}^r\mathbb{E}\left[\frac{1}{\sigma_i^4}\right]\frac{9}{d(d+2)n(n+2)} + 3\sum_{i\neq k}\mathbb{E}\left[\frac{1}{\sigma_i^2\sigma_k^2}\right]\frac{1}{d(d+2)n(n+2)}$$

$$= \frac{1}{\eta^4} + \underbrace{\frac{9\sum_{i=1}^r\mathbb{E}[1/\sigma_i^4]}{d(d+2)n(n+2)}}_{I_1} + \underbrace{\frac{3\sum_{i\neq k}\mathbb{E}[1/(\sigma_i^2\sigma_k^2)]}{d(d+2)n(n+2)}}_{I_2} + \underbrace{\frac{6}{\eta^2}\frac{\sum_{i=1}^r\mathbb{E}[1/\sigma_i^2]}{dn}}_{I_3}.$$

**Leading Order Scaling and Mean.** Let $N = \max(d, n)$, assume $n, d \to \infty$ with $d/n \to c \neq 1$. Lemma 5 from Sonthalia and Nadakuditi (2023) implies that if $\boldsymbol{A}$ has unit variance entries, the moments of its inverse eigenvalue are expressions of $c$ and are hence $O(1)$. In our case, it will just scale with $\rho$ instead:

$$\mathbb{E}[1/\sigma_i^4] = O(1/\rho^4), \quad \mathbb{E}[1/(\sigma_i^2 \sigma_k^2)] = O(1/\rho^4), \quad \text{and} \quad \mathbb{E}[1/\sigma_i^8] = O(1/\rho^8) \quad \text{etc.}$$

In particular, we also need the following exact expectation from the same lemma:

$$\mathbb{E}\left[\frac{1}{\sigma_i^2}\right] = \frac{c}{\rho^2|1-c|} + o\left(\frac{1}{\rho^2}\right) = O\left(\frac{1}{\rho^2}\right). \tag{8}$$

Since the above $I_1, I_3$ have $r = \min(d, n)$ summands, this implies

$$I_1 = O\left(\frac{r}{N^4\rho^4}\right) = O\left(\frac{1}{N^3\rho^4}\right), \quad I_3 = O\left(\frac{r}{\eta^2 N^2 \rho^2}\right) = O\left(\frac{1}{N\rho^4}\right).$$

Similarly, $I_2$ has $r(r-1) \approx r^2$ summands, and

$$I_2 = O\left(\frac{r^2}{N^4\rho^4}\right) = O\left(\frac{1}{N^2\rho^4}\right)$$

$$\implies \mathbb{E}[\zeta^4] = \frac{1}{\eta^4} + I_1 + I_2 + I_3 = \frac{1}{\eta^4} + O\left(\frac{1}{\max(d,n)\rho^4}\right) \quad \text{since } I_3 \text{ dominates.} \tag{9}$$

With a similar expansion for the second moment and taking spherical moments, we get that

$$\mathbb{E}[\zeta^2] = \frac{1}{\eta^2} + \sum_{i,j} \mathbb{E}\left[\frac{1}{\sigma_i \sigma_j}\right] \mathbb{E}[b_i b_j] \mathbb{E}[a_i a_j] = \frac{1}{\eta^2} + \frac{\sum_{i=1}^r \mathbb{E}[1/\sigma_i^2]}{dn}$$

$$= \frac{1}{\eta^2} + \frac{\min(d,n)}{dn}\left(\frac{c}{\rho^2|1-c|} + o\left(\frac{1}{\rho^2}\right)\right) \quad \text{by Equation 8}$$

$$= \frac{1}{\eta^2} + \frac{1}{\max(d,n)}\frac{c}{\rho^2|1-c|} + o\left(\frac{1}{\max(d,n)\rho^2}\right).$$

This gives us the mean. Furthermore,

$$(\mathbb{E}[\zeta^2])^2 = \frac{1}{\eta^4} + \frac{2}{\eta^2}\frac{\sum_{i=1}^r \mathbb{E}[1/\sigma_i^2]}{dn} + \frac{(\sum_{i=1}^r \mathbb{E}[1/\sigma_i^2])^2}{d^2 n^2} = \frac{1}{\eta^4} + O\left(\frac{1}{\max(d,n)\rho^4}\right). \tag{10}$$

**Variance.** $\text{Var}(\zeta^2) = \mathbb{E}[\zeta^4] - (\mathbb{E}[\zeta^2])^2$. From Equations 9, 10, the overall scaling is determined by the dominant term:

$$\text{Var}\left(\left(\frac{\xi}{\eta}\right)^2\right) = O\left(\frac{1}{\max(d,n)\rho^4}\right).$$

$\square$

**Lemma 10** (General Terms). *In the setting of Section 2 we have the following expectations:*

1. *For $c < 1$, $\mathbb{E}[\boldsymbol{\beta}_*^\top \boldsymbol{u} \boldsymbol{k}^\top \boldsymbol{A}^\dagger \boldsymbol{\beta}_*] = \frac{c}{\rho^2(1-c)}(\boldsymbol{\beta}_*^\top \boldsymbol{u})^2 + o\left(\frac{1}{\rho^2}\right)$ and the variance is $O(1/(\rho^4 d))$.*

2. *For $c < 1$, $\mathbb{E}[\boldsymbol{k}^\top \boldsymbol{A}^\dagger \boldsymbol{A}^{\dagger\top} \boldsymbol{k}] = \frac{c^2}{\rho^4(1-c)^3} + o\left(\frac{1}{\rho^4}\right)$ and the variance is $O(1/(\rho^8 d))$.*

3. *For $c > 1$, $\mathbb{E}[\boldsymbol{\beta}_*^\top \boldsymbol{s} \boldsymbol{u}^\top \boldsymbol{\beta}_*] = \frac{c-1}{c}(\boldsymbol{\beta}_*^\top \boldsymbol{u})^2$ and the variance is $O(1/d)$.*

4. *For $c > 1$, $\mathbb{E}[\boldsymbol{\beta}_*^\top \boldsymbol{A} \boldsymbol{A}^\dagger \boldsymbol{u} \boldsymbol{s}^\top \boldsymbol{\beta}_*] = \frac{c-1}{c^2}(\boldsymbol{\beta}_*^\top \boldsymbol{u})^2 + o(1)$ and the variance is $O(1/d)$.*

5. *For $c > 1$, $\mathbb{E}[\boldsymbol{\beta}_*^\top \boldsymbol{h}^\top \boldsymbol{h} \boldsymbol{\beta}_*] = \frac{\|\boldsymbol{\beta}_*\|^2}{d}\frac{c}{\rho^2(c-1)} + o\left(\frac{1}{\rho^2 d}\right)$ and the variance is $O(1/(\rho^4 d^2))$.*

6. *For $c > 1$, $\mathbb{E}[\boldsymbol{h} \boldsymbol{A}^{\dagger\top} \boldsymbol{A}^\dagger \boldsymbol{h}^\top] = \frac{1}{\rho^4}\frac{c^3}{(c-1)^3} + o\left(\frac{1}{\rho^4}\right)$ and the variance is $O(1/(\rho^8 d))$.*

7. *For $c > 1$, $\mathbb{E}[\|\boldsymbol{k}\|^2] = \frac{1}{\rho^2} \frac{1}{c-1} + o\left(\frac{1}{\rho^2}\right)$ and the variance is $O(1/(\rho^4 n))$*

*Proof.* For all these terms, we evaluate the expectation using the SVD $\boldsymbol{A} = \boldsymbol{U\Sigma V}^\top$, with $\boldsymbol{A}^\dagger = \boldsymbol{V\Sigma}^\dagger \boldsymbol{U}^\top$, and important expectations from Lemma 5 of Sonthalia and Nadakuditi (2023) regarding the spectrum of $\boldsymbol{A}$: suppose $\tilde{\boldsymbol{A}}$ has unit variance (general $\rho^2$ is a multiplicative change), and let $\sigma_i(\tilde{\boldsymbol{A}})$ denote the $i$-th singular value. We have

$$\mathbb{E}\left[\frac{1}{\sigma_i^2(\tilde{\boldsymbol{A}})}\right] = \begin{cases} \frac{c}{1-c} + o(1) & c < 1 \\ \frac{c}{c-1} + o(1) & c > 1 \end{cases}, \qquad \mathbb{E}\left[\frac{1}{\sigma_i^4(\tilde{\boldsymbol{A}})}\right] = \begin{cases} \frac{c^2}{(1-c)^3} + o(1) & c < 1 \\ \frac{c^3}{(c-1)^3} + o(1) & c > 1 \end{cases}.$$

$$\mathbb{E}\left[\frac{1}{\sigma_i^2(\boldsymbol{A})}\right] = \begin{cases} \frac{1}{\rho^2} \frac{c}{1-c} + o\left(\frac{1}{\rho^2}\right) & c < 1 \\ \frac{1}{\rho^2} \frac{c}{c-1} + o\left(\frac{1}{\rho^2}\right) & c > 1 \end{cases}, \qquad \mathbb{E}\left[\frac{1}{\sigma_i^4(\boldsymbol{A})}\right] = \begin{cases} \frac{1}{\rho^4} \frac{c^2}{(1-c)^3} + o\left(\frac{1}{\rho^4}\right) & c < 1 \\ \frac{1}{\rho^4} \frac{c^3}{(c-1)^3} + o\left(\frac{1}{\rho^4}\right) & c > 1 \end{cases}.$$

$$\tag{11}$$

**For the first term**, we note that

$$\begin{aligned}
\boldsymbol{\beta}_*^\top \boldsymbol{u} \boldsymbol{k}^\top A^\dagger \boldsymbol{\beta}_* &= (\boldsymbol{\beta}_*^\top \boldsymbol{u}) \boldsymbol{u}^\top \boldsymbol{A}^{\dagger\top} \boldsymbol{A}^\dagger \boldsymbol{\beta}_* \\
&= (\boldsymbol{\beta}_*^\top \boldsymbol{u}) \boldsymbol{u}^\top \boldsymbol{U} \boldsymbol{\Sigma}^{\dagger\top} \boldsymbol{\Sigma}^\dagger \boldsymbol{U}^\top \boldsymbol{\beta}_* \\
&= (\boldsymbol{\beta}_*^\top \boldsymbol{u}) \sum_{i=1}^d (\boldsymbol{u}^\top \boldsymbol{U})_i (\boldsymbol{U}^\top \boldsymbol{\beta}_*)_i \frac{1}{\sigma_i^2(\boldsymbol{A})} \\
&= (\boldsymbol{\beta}_*^\top \boldsymbol{u}) \sum_{i=1}^d (\boldsymbol{u}^\top \boldsymbol{u}_i)(\boldsymbol{\beta}_*^\top \boldsymbol{u}_i) \frac{1}{\sigma_i^2(\boldsymbol{A})},
\end{aligned}$$

where $\boldsymbol{u}_i$ denotes the $i$-th column of $\boldsymbol{U}$. We further note that $\boldsymbol{u}^\top \boldsymbol{\beta}_* = \boldsymbol{u}^\top \boldsymbol{U} \boldsymbol{U}^\top \boldsymbol{\beta}_*$. Since permuting columns of an orthogonal matrix does not break orthogonality and $\boldsymbol{U}$ is uniformly random, we have that the marginals $\boldsymbol{u}_i$ are identical. Thus, we have that

$$\mathbb{E}[\boldsymbol{u}^\top \boldsymbol{u}_1 \boldsymbol{\beta}_*^\top \boldsymbol{u}_1] = \ldots = \mathbb{E}[\boldsymbol{u}^\top \boldsymbol{u}_d \boldsymbol{\beta}_*^\top \boldsymbol{u}_d] = \frac{1}{d}(\boldsymbol{u}^\top \boldsymbol{\beta}_*) \quad \text{since } \mathbb{E}[\boldsymbol{u}_i \boldsymbol{u}_i^\top] = \frac{1}{d}\boldsymbol{I}.$$

It follows from here that

$$\begin{aligned}
\mathbb{E}\left[\boldsymbol{\beta}_*^\top \boldsymbol{u} \boldsymbol{k}^\top \boldsymbol{A}^\dagger \boldsymbol{\beta}_*\right] &= (\boldsymbol{\beta}_*^\top \boldsymbol{u}) \sum_{i=1}^d \mathbb{E}[\boldsymbol{u}^\top \boldsymbol{u}_i \boldsymbol{\beta}_*^\top \boldsymbol{u}_i] \mathbb{E}\left[\frac{1}{\sigma_i^2(\boldsymbol{A})}\right] \\
&= \frac{1}{\rho^2}(\boldsymbol{\beta}_*^\top \boldsymbol{u})^2 \sum_{i=1}^d \frac{1}{d}\left(\frac{c}{1-c} + o(1)\right) \quad \text{by Equation 11} \\
&= \frac{1}{\rho^2} \frac{c}{1-c}(\boldsymbol{\beta}_*^\top \boldsymbol{u})^2 + o\left(\frac{1}{\rho^2}\right).
\end{aligned}$$

Since $\boldsymbol{A}$ is isotropic Gaussian, we have that $\boldsymbol{U}, \boldsymbol{V}$ are uniformly random orthogonal matrices. Thus, $\boldsymbol{u}^\top \boldsymbol{U}$ and $\boldsymbol{U}^\top \boldsymbol{\beta}_*$ are uniformly random vectors on the spheres of radius $\|\boldsymbol{u}\|$ and $\|\boldsymbol{\beta}_*\|$ respectively.

Hence, when we consider the squared terms to compute the variance, the term from the two uniform vectors will contribute $O(1/d^2)$. Together with the singular value term (now squared to have $O(1/\rho^4)$) and the summation, the variance is of order $O(1/(\rho^4 d))$.

**For the second term**, we have that by Equation 11,

$$\boldsymbol{k}^\top \boldsymbol{A}^\dagger \boldsymbol{A}^{\dagger\top} \boldsymbol{k} = \boldsymbol{u}^\top ((\boldsymbol{A}\boldsymbol{A}^\top)^\dagger)^2 \boldsymbol{u} = \boldsymbol{u}^\top \boldsymbol{U}((\boldsymbol{\Sigma}\boldsymbol{\Sigma}^\top)^\dagger)^2 \boldsymbol{U}^\top \boldsymbol{u} = \sum_{i=1}^d (\boldsymbol{u}^\top \boldsymbol{u}_i)^2 \frac{1}{\sigma_i^4(\boldsymbol{A})},$$

$$\mathbb{E}[\boldsymbol{k}^\top \boldsymbol{A}^\dagger \boldsymbol{A}^{\dagger\top} \boldsymbol{k}] = \sum_{i=1}^d \mathbb{E}[(\boldsymbol{u}^\top \boldsymbol{u}_i)^2] \mathbb{E}\left[\frac{1}{\sigma_i^4(A)}\right] = \sum_{i=1}^d \frac{1}{\rho^4} \frac{1}{d}\left(\frac{c^2}{(1-c)^3} + o(1)\right) = \frac{1}{\rho^4} \frac{c^2}{(1-c)^3} + o\left(\frac{1}{\rho^4}\right),$$

where we again use $\mathbb{E}[(\boldsymbol{u}^\top \boldsymbol{u}_i)^2] = 1/d$ since it is the entry of a uniformly random vector of length $\|\boldsymbol{u}\| = 1$.

Similarly, the variance is $O(1/(\rho^8 d))$ from the summation of $d$ independent variances each of $O(1/(\rho^8 d^2))$.

**For the third term**, we have that

$$\boldsymbol{\beta}_*^\top \boldsymbol{s}\boldsymbol{u}^\top \boldsymbol{\beta}_* = \boldsymbol{\beta}_*^\top (\boldsymbol{I} - \boldsymbol{A}\boldsymbol{A}^\dagger)\boldsymbol{u}(\boldsymbol{u}^\top \boldsymbol{\beta}_*) = (\boldsymbol{\beta}_*^\top \boldsymbol{u})^2 - (\boldsymbol{\beta}_*^\top \boldsymbol{u})\sum_{i=1}^n (\boldsymbol{\beta}_*^\top \boldsymbol{u}_i)(\boldsymbol{u}^\top \boldsymbol{u}_i).$$

Similarly, we take the expectation (in particular, $\mathbb{E}[(\boldsymbol{\beta}_*^\top \boldsymbol{u}_i)(\boldsymbol{u}^\top \boldsymbol{u}_i)] = 1/d(\boldsymbol{\beta}_*^\top \boldsymbol{u})$) and have

$$(\boldsymbol{\beta}_*^\top \boldsymbol{u})^2 \left[1 - \sum_{i=1}^n \frac{1}{d}\right] = \left(1 - \frac{1}{c}\right)(\boldsymbol{\beta}_*^\top \boldsymbol{u})^2.$$

The variance for this term is $O(1/d)$ from summation of $n = d/c$ terms of $O(1/d^2)$.

**For the fourth term**, we plug in $\boldsymbol{s} = (\boldsymbol{I} - \boldsymbol{A}\boldsymbol{A}^\dagger)\boldsymbol{u}$ and have

$$\boldsymbol{\beta}_*^\top \boldsymbol{A}\boldsymbol{A}^\dagger \boldsymbol{u}\boldsymbol{s}^\top \boldsymbol{\beta}_* = (\boldsymbol{\beta}_*^\top \boldsymbol{u})\boldsymbol{\beta}_*^\top \boldsymbol{A}\boldsymbol{A}^\dagger \boldsymbol{u} - (\boldsymbol{\beta}_*^\top \boldsymbol{A}\boldsymbol{A}^\dagger \boldsymbol{u})^2.$$

From previous calculations, we have that

$$\mathbb{E}[\boldsymbol{\beta}_*^\top \boldsymbol{A}\boldsymbol{A}^\dagger \boldsymbol{u}] = \mathbb{E}\left[\sum_{i=1}^n (\boldsymbol{\beta}_*^\top \boldsymbol{u}_i)(\boldsymbol{u}^\top \boldsymbol{u}_i)\right] = \frac{1}{c}(\boldsymbol{\beta}_*^\top \boldsymbol{u}).$$

Using Proposition 2 and this result, we can then show

$$\mathbb{E}[(\boldsymbol{\beta}_*^\top \boldsymbol{A}\boldsymbol{A}^\dagger \boldsymbol{u})^2] = \frac{1}{c^2}(\boldsymbol{\beta}_*^\top \boldsymbol{u})^2 + o(1).$$

It follows that

$$\mathbb{E}[\boldsymbol{\beta}_*^\top \boldsymbol{A}\boldsymbol{A}^\dagger \boldsymbol{u}\boldsymbol{s}^\top \boldsymbol{\beta}_*] = \frac{c-1}{c^2}(\boldsymbol{\beta}_*^\top \boldsymbol{u})^2 + o(1).$$

The variance for this term is $O(1/d)$, where the dominant term is a summation of $n = d/c$ terms of $O(1/d^2)$.

**For the fifth term**, we have

$$\boldsymbol{\beta}_*^\top \boldsymbol{h}^\top \boldsymbol{h}\boldsymbol{\beta}_* = (\boldsymbol{\beta}_*^\top \boldsymbol{A}^\dagger \boldsymbol{v})^2 = \sum_{i,j}^n (\boldsymbol{\beta}_*^\top \boldsymbol{U})_i (\boldsymbol{\beta}_*^\top \boldsymbol{U})_j \frac{1}{\sigma_i(\boldsymbol{A})\sigma_j(\boldsymbol{A})}(\boldsymbol{V}^\top \boldsymbol{v})_i (\boldsymbol{V}^\top \boldsymbol{v})_j.$$

Since $\boldsymbol{\beta}_*^\top \boldsymbol{U}$ (and $\boldsymbol{V}^\top \boldsymbol{v}$) are uniformly random and independent of everything else, we only have the diagonal terms when we take the expectation. By Equation 11,

$$\mathbb{E}[\boldsymbol{\beta}_*^\top \boldsymbol{h}^\top \boldsymbol{h}\boldsymbol{\beta}_*] = \sum_{i=1}^n \frac{\|\boldsymbol{\beta}_*\|^2}{d}\frac{1}{n}\frac{1}{\rho^2}\left(\frac{c}{c-1} + o(1)\right) = \frac{\|\boldsymbol{\beta}_*\|^2}{d}\frac{1}{\rho^2}\frac{c}{c-1} + o\left(\frac{1}{\rho^2 d}\right)$$

The variance for this term is $O(1/(\rho^4 d^2))$ from $O(d^2)$ terms of individual variances of $O(1/(\rho^4 d^4))$.

**For the sixth term**, by expansion and Equation 11, similar to above,

$$\mathbb{E}\left[\boldsymbol{h}\boldsymbol{A}^{\dagger\top} \boldsymbol{A}^\dagger \boldsymbol{h}^\top\right] = \sum_{i=1}^n \mathbb{E}\left[(\boldsymbol{V}^\top \boldsymbol{v})_i^2\right]\mathbb{E}\left[\frac{1}{\sigma_i^4(\boldsymbol{A})}\right] = \sum_{i=1}^n \frac{1}{n}\mathbb{E}\left[\frac{1}{\sigma_i^4(\boldsymbol{A})}\right] = \frac{1}{\rho^4}\frac{c^3}{(c-1)^3} + o\left(\frac{1}{\rho^4}\right).$$

The variance is $O\left(1/(\rho^8 d)\right)$.

**For the final term**, by expansion and Equation 11,

$$\mathbb{E}\left[\|\boldsymbol{k}\|^2\right] = \sum_{i=1}^n \mathbb{E}\left[(\boldsymbol{u}^\top \boldsymbol{U})_i^2\right]\mathbb{E}\left[\frac{1}{\sigma_i^2(\boldsymbol{A})}\right] = \frac{1}{\rho^2}\frac{n}{d}\frac{c}{c-1} + o\left(\frac{1}{\rho^2}\right) = \frac{1}{\rho^2}\frac{1}{c-1} + o\left(\frac{1}{\rho^2}\right)$$

The variance is $O\left(1/(\rho^4 n)\right)$. $\square$

**Lemma 11** (Zero Expectation). *In the setting of Section 2, we have the following expectations for*

1. $\forall c$, $\mathbb{E}[\boldsymbol{\beta}_*^\top \boldsymbol{uh}\boldsymbol{\beta}_*] = 0$ *and* $\mathrm{Var}(\boldsymbol{\beta}_*^\top \boldsymbol{uh}\boldsymbol{\beta}_*) = O(1/(\rho^2 d))$

2. *If* $c > 1$, $\mathbb{E}[\boldsymbol{\beta}_*^\top \boldsymbol{AA}^\dagger \boldsymbol{uh}\boldsymbol{\beta}_*] = 0$ *and* $\mathrm{Var}(\boldsymbol{\beta}_*^\top \boldsymbol{AA}^\dagger \boldsymbol{uh}\boldsymbol{\beta}_*) = O(1/(\rho^2 d^2))$

3. *If* $c > 1$, $\mathbb{E}[\boldsymbol{\beta}_*^\top \boldsymbol{sh}\boldsymbol{\beta}_*] = 0$ *and* $\mathrm{Var}(\boldsymbol{\beta}_*^\top \boldsymbol{sh}\boldsymbol{\beta}_*) = O(1/(\rho^2 d))$

4. $\forall c$, $\mathbb{E}[\boldsymbol{k}^\top \boldsymbol{A}^\dagger \boldsymbol{h}^\top] = 0$ *and* $\mathrm{Var}(\boldsymbol{k}^\top \boldsymbol{A}^\dagger \boldsymbol{h}^\top) = O(1/(\rho^6 d))$

5. *If* $c > 1$, $\mathbb{E}[\boldsymbol{hAA}^\dagger \boldsymbol{\beta}_*] = 0$ *and* $\mathrm{Var}(\boldsymbol{hAA}^\dagger \boldsymbol{\beta}_*) = O(1/(\rho^2 d))$

*Proof.* Similar to Lemma 10, for all these terms, we evaluate the expectation using the SVD $\boldsymbol{A} = \boldsymbol{U\Sigma V}^\top$, with $\boldsymbol{A}^\dagger = \boldsymbol{V\Sigma}^\dagger \boldsymbol{U}^\top$.

**For the first term**, we note that

$$\boldsymbol{\beta}_*^\top \boldsymbol{uh}\boldsymbol{\beta}_*^\top = (\boldsymbol{\beta}_*^\top \boldsymbol{u})\boldsymbol{v}^\top \boldsymbol{A}^\dagger \boldsymbol{\beta}_* = (\boldsymbol{\beta}_*^\top \boldsymbol{u})\boldsymbol{v}^\top \boldsymbol{V\Sigma}^\dagger \boldsymbol{U}^\top \boldsymbol{\beta}_* = (\boldsymbol{\beta}_*^\top \boldsymbol{u}) \sum_{i=1}^{\min(n,d)} (\boldsymbol{v}^\top \boldsymbol{V})_i (\boldsymbol{U}^\top \boldsymbol{\beta}_*)_i \frac{1}{\sigma_i(\boldsymbol{A})}.$$

Since $\boldsymbol{A}$ is isotropic Gaussian, again we have that $\boldsymbol{U}, \boldsymbol{V}$ are uniformly random orthogonal matrices. Thus, $\boldsymbol{v}^\top \boldsymbol{V}$ and $\boldsymbol{U}^\top \boldsymbol{\beta}_*$ are uniformly random vectors on a spheres of radius $\|\boldsymbol{v}\|$ and $\|\boldsymbol{\beta}_*\|$ respectively. In particular, they are independent and have mean zero, which implies

$$\mathbb{E}\left[\boldsymbol{\beta}_*^\top \boldsymbol{uh}\boldsymbol{\beta}_*^\top\right] = 0.$$

The variance will be $O(1/(\rho^2 d))$ as a summation of $O(d)$ terms of $O(1/(\rho^2 d^2))$.

**For the second term**, we note that

$$\boldsymbol{\beta}_*^\top \boldsymbol{AA}^\dagger \boldsymbol{u} = \sum_{i=1}^{\min(n,d)} (\boldsymbol{\beta}_*^\top \boldsymbol{U})_i (\boldsymbol{U}^\top \boldsymbol{u})_i \quad \text{and} \quad \boldsymbol{h}\boldsymbol{\beta}_* = \sum_{i=1}^{\min(n,d)} (\boldsymbol{v}^\top \boldsymbol{V})_i (\boldsymbol{U}^\top \boldsymbol{\beta}_*)_i \frac{1}{\sigma_i(\boldsymbol{A})}$$

Multiplying the two together yields

$$\boldsymbol{\beta}_*^\top \boldsymbol{AA}^\dagger \boldsymbol{uh}\boldsymbol{\beta}_* = \sum_{i,j}^{\min(n,d)} (\boldsymbol{\beta}_*^\top \boldsymbol{U})_i (\boldsymbol{U}^\top \boldsymbol{u})_i (\boldsymbol{v}^\top \boldsymbol{V})_j (\boldsymbol{U}^\top \boldsymbol{\beta}_*)_j \frac{1}{\sigma_i(\boldsymbol{A})}.$$

We note that $v^\top V$ is a uniformly random mean zero vector independent of everything else in the summation. Hence, the expectation is equal to zero, and similar to Lemma **??**, the variance of this term is $O(1/(\rho^2 d^2))$ (a summation of $O(d^2)$ terms of $O(1/(\rho^2 d^4))$).

**For the third term**, we have that

$$\boldsymbol{\beta}_*^\top \boldsymbol{sh}\boldsymbol{\beta}_* = \boldsymbol{\beta}_*^\top (\boldsymbol{I} - \boldsymbol{AA}^\dagger)\boldsymbol{uh}\boldsymbol{\beta}_* = \boldsymbol{\beta}_*^\top \boldsymbol{uh}\boldsymbol{\beta}_* - \boldsymbol{\beta}_*^\top \boldsymbol{AA}^\dagger \boldsymbol{uh}\boldsymbol{\beta}_*.$$

Then using the previous two parts, we get that each term has mean zero. Thus, we get the needed result. Using Lemma 34 and the first two terms, the variance of this term is $O(1/(\rho^2 d))$.

**For the fourth term**, we have that:

$$\boldsymbol{k}^\top \boldsymbol{A}^\dagger \boldsymbol{h}^\top = \boldsymbol{uU\Sigma}^{\dagger\top} \boldsymbol{\Sigma}^\dagger \boldsymbol{\Sigma}^{\dagger\top} \boldsymbol{V}^\top \boldsymbol{v} = \sum_{i=1}^{\min(n,d)} (\boldsymbol{u}^\top \boldsymbol{U})_i (\boldsymbol{V}^\top \boldsymbol{v})_i \frac{1}{\sigma_i(\boldsymbol{A})^3}.$$

Similarly, using the independence of $\boldsymbol{U}, \boldsymbol{\Sigma}, \boldsymbol{V}$ and uniformly random entries, we get mean zero and variance $O(1/(\rho^6 d))$.

**For the last term**, we have that:

$$\boldsymbol{hAA}^\dagger \boldsymbol{\beta}_* = \sum_{i=\min(n,d)}^{r} (\boldsymbol{V}^\top \boldsymbol{v})_i (\boldsymbol{U}^\top \boldsymbol{\beta}_*)_i \frac{1}{\sigma_i(\boldsymbol{A})}.$$

Using the independence of $\boldsymbol{U}, \boldsymbol{\Sigma}, \boldsymbol{V}$ and uniformly random entries, we get mean zero and variance $O(1/(\rho^2 d))$. $\qquad \square$

### E.4.2 STEP 3(B): BOUNDING THE HIGHER MOMENTS

To bound the higher moments, we will the following Gaussian hypercontractivity lemma.

**Lemma 12** (Gaussian Hypercontractivity Inequality). *Let $G \sim \mathcal{N}(0,1)$ be a standard Gaussian random variable. Let $f : \mathbb{R} \to \mathbb{R}$ be a degree $k$ polynomial. Then, for any $q \geq 2$, the $L_q$ norm of $f(G)$ is bounded by its $L_2$ norm as follows:*

$$\|f(G)\|_{L_q} \leq (q-1)^{k/2} \|f(G)\|_{L_2},$$

*where the $L_p$ norm of a random variable $X$ is defined as $\|X\|_{L_p} = (\mathbb{E}[|X|^p])^{1/p}$.*

*Proof.* Follows directly from (Mei et al., 2022, Lemma 20). $\qquad\square$

**Lemma 13** (Multivariate Gaussian Hypercontractivity). *Let $G = (G_1, \ldots, G_M) \sim \mathcal{N}(0, I_M)$ and let $P : \mathbb{R}^M \to \mathbb{R}$ be a polynomial of total degree $r$. Consider the Hermite expansion of $P$*

$$P(x) = \sum_{\alpha \in \mathbb{N}^m, |\alpha| \leq r} c_\alpha \boldsymbol{H}_\alpha(x).$$

*with coefficient random and independent of $G$. Then there exists a constant $C$ that is only dependent on $M, r$ such that for any $q \geq 2$,*

$$\|P(G)\|_{L_q} \leq C(q-1)^{r/2} \left( \sum_{|\alpha| \leq r} \|c_\alpha\|_{L_q}^2 \alpha! \right)^{1/2}$$

*Further, if for all $|\alpha| \leq r$, we have that $\|c_\alpha\|_{L_q}^2 \leq C_q^2 \|c_\alpha\|_{L_2}^2$, then*

$$\|P(G)\|_{L_q} \leq C(q-1)^{r/2} \|P(G)\|_{L_2}$$

*Where the $L_p$ norm is over all of the randomness. Furthermore,*

*Proof.* Let $H_k : \mathbb{R} \to \mathbb{R}$ be the probabilist Hermite polynomial. Given $\alpha \in \mathbb{N}^M$, define

$$\boldsymbol{H}_\alpha(x) := \prod_{j=1}^{M} H_{\alpha_j}(x_j)$$

Then since $P$ is degree $r$, then we can decompose

$$P(x) = \sum_{\alpha \in \mathbb{N}^m, |\alpha| \leq r} c_\alpha \boldsymbol{H}_\alpha(x).$$

Here $|\alpha| = \sum_j \alpha_j$. Since the Hermite polynomials are orthogonal, we can see that

$$\int_{\mathbb{R}^M} \boldsymbol{H}_\alpha(x) \boldsymbol{H}_{\tilde{\alpha}}(x) \gamma_M(x) = \delta_{\alpha\tilde{\alpha}} \prod_{j=1}^{M} \alpha_j!,$$

where $\gamma_M$ is the density for an $M$-dimensional standard normal distribution.

$$\begin{aligned}
\|P(x)\|_{L_2}^2 &= \mathbb{E}_{\boldsymbol{\Sigma}} \left[ \int_{\mathbb{R}^M} |P(x)|^2 \gamma_M(x) dx \right] \\
&= \sum_{|\alpha| \leq r} \sum_{|\tilde{\alpha}| \leq r} \mathbb{E}_{\boldsymbol{\Sigma}} [c_\alpha c_{\tilde{\alpha}}] \int \boldsymbol{H}_\alpha(x) \boldsymbol{H}_{\tilde{\alpha}}(x) \gamma_M(x) dx \\
&= \sum_{|\alpha| \leq r} \|c_\alpha\|_{L_2}^2 \alpha!
\end{aligned}$$

where $\alpha! := \prod_{j=1}^{M} \alpha_j!$.

Then using the 1D Gaussian Hypercontractivity (Lemma 12, we see that

$$\|\boldsymbol{H}_\alpha(x)\|_{L_q} = \prod_{j=1}^{M} \|H_{\alpha_j}(x_j)\|_{L_q}$$

$$\leq \prod_{j=1}^{M} (q-1)^{\alpha_j/2} \|H_{\alpha_j}(x_j)\|_{L_2}$$

$$= (q-1)^{|\alpha|/2} \prod_{j=1}^{M} \sqrt{\alpha_j!}$$

$$= (q-1)^{|\alpha|/2} \sqrt{\alpha!}$$

Thus, using the triangle inequality we get that

$$\|P(x)\|_{L_q} \leq \sum_{|\alpha|\leq r} \|c_\alpha \boldsymbol{H}_\alpha(x)\|_{L_q} = \sum_{|\alpha|\leq r} \|c_\alpha\|_{L_q} \|\boldsymbol{H}_\alpha(x)\|_{L_q}$$

Thus

$$\|P(x)\|_{L_q} \leq \sum_{|\alpha|\leq r} \|c_\alpha \boldsymbol{H}_\alpha(x)\|_{L_q} \leq \sum_{|\alpha|\leq r} \|c_\alpha\|_{L_q} (q-1)^{|\alpha|/2}\sqrt{\alpha!} \leq (q-1)^{r/2} \sum_{|\alpha|\leq r} \|c_\alpha\|_{L_q} \sqrt{\alpha!}$$

Then using Cauchy-Schwartz, we get that

$$\sum_{|\alpha|\leq r} \|c_\alpha\|_{L_q}\sqrt{\alpha!} \leq \left( \sum_{|\alpha|\leq r} \|c_\alpha\|_{L_q}^2 \alpha! \right)^{1/2} \left( \sum_{|\alpha|\leq r} 1 \right)^{1/2}.$$

Finally, we note that

$$C_{M,r} := \left( \sum_{|\alpha|\leq r} 1 \right)^{1/2}$$

is some universal constant that only depends on $M, r$. Thus, we get that

$$\|P(x)\|_{L_q} \leq C_{M,r} (q-1)^{r/2} \left( \sum_{|\alpha|\leq r} \|c_\alpha\|_{L_q}^2 \alpha! \right)^{1/2}$$

Using the assumption

$$\|c_\alpha\|_{L_q}^2 \leq C_q^2 \|c_\alpha\|_{L_2}^2$$

Then we get

$$\|P(x)\|_{L_q} \leq C_{M,r} C_q (q-1)^{r/2} \|P(x)\|_{L_2}$$

$\square$

**Lemma 14** (Product Spherical Hypercontractivity). *Let $l_1, l_2, l_3 \geq 0$, let $\Theta_1 \sim \mathrm{Unif}(S^{l_1})$, $\Theta_2 \sim \mathrm{Unif}(S^{l_2})$, $\Theta_3 \sim \mathrm{Unif}(S^{l_3})$ be independent, and let $H : \mathbb{R}^{l_1+1} \times \mathbb{R}^{l_2+1} \times \mathbb{R}^{l_3+1} \to \mathbb{R}$ be a multi-homogeneous polynomial of total degree $r$. Then for every $q \geq 2$,*

$$\|H(\Theta_1, \Theta_2, \Theta_3)\|_{L_q} \leq C_{r,q}(q-1)^{r/2} \|H(\Theta_1, \Theta_2, \Theta_3)\|_{L_2},$$

*where the norms are with respect to the product measure. For homogeneous polynomials, the constant is independent of the dimension.*

*Proof.* $H$ is multi-homogeneous of degrees $r_1, r_2, r_3$ with $r_1 + r_2 + r_3 = r$. Let $G_1 \sim \mathcal{N}(0, I_{l_1+1})$, $G_2 \sim \mathcal{N}(0, I_{l_2+1})$, $G_3 \sim \mathcal{N}(0, I_{l_3+1})$ be independent with polar decompositions $G_i = R_i \Theta_i$, where the $R_i$'s are independent of each other and of the $\Theta_i$'s. Then

$$H(G_1, G_2, G_3) = R_1^{r_1} R_2^{r_2} R_3^{r_3} H(\Theta_1, \Theta_2, \Theta_3),$$

so for any $p > 0$,

$$\mathbb{E}\left[|H(G_1, G_2, G_3)|^p\right] = \left(\prod_{i=1}^{3} \mathbb{E}\left[R_i^{pr_i}\right]\right) \mathbb{E}\left[|H(\Theta_1, \Theta_2, \Theta_3)|^p\right]$$

Then we have that

$$\|H(G_1, G_2, G_3)\|_{L_p} = \left(\prod_i (\mathbb{E}\left[R_i^{pr_i}\right])^{1/p}\right) \|H(\Theta_1, \Theta_2, \Theta_3)\|_{L_p}. \tag{12}$$

Apply Gaussian hypercontractivity (Lemma 12) to $H(G_1, G_2, G_3)$ (total degree $r$):

$$\|H(G_1, G_2, G_3)\|_{L_q} \leq C(q-1)^{r/2} \|H(G_1, G_2, G_3)\|_{L_2}, \qquad q \geq 2.$$

Using Equation 12 with $p = q$ and $p = 2$ yields

$$\|H(\Theta_1, \Theta_2, \Theta_3)\|_{L_q} \leq C(q-1)^{r/2} \left(\prod_i \frac{(\mathbb{E}\left[R_i^{2r_i}\right])^{1/2}}{(\mathbb{E}\left[R_i^{qr_i}\right])^{1/q}}\right) \|H(\Theta_1, \Theta_2, \Theta_3)\|_{L_2}.$$

For each $i$, since $q \geq 2$ and $R_i \geq 0$, monotonicity of $L_p$ norms implies $(\mathbb{E}\left[R_i^{qr_i}\right])^{1/(qr_i)} \geq (\mathbb{E}\left[R_i^{2r_i}\right])^{1/(2r_i)}$, hence

$$\frac{(\mathbb{E}\left[R_i^{2r_i}\right])^{1/2}}{(\mathbb{E}\left[R_i^{qr_i}\right])^{1/q}} \leq 1.$$

Thus the product is less than 1, so

$$\|H(\Theta_1, \Theta_2, \Theta_3)\|_{L_q} \leq C(q-1)^{r/2} \|H(\Theta_1, \Theta_2, \Theta_3)\|_{L_2}.$$

$\square$

**Lemma 15** (Product spherical hypercontractivity with random coefficients). *Let $l_1, l_2, l_3 \geq 0$ and let $\Theta_i \sim \mathrm{Unif}(S^{l_i})$ be independent. Let $r \in \mathbb{N}$ and let $H : \mathbb{R}^{l_1+1} \times \mathbb{R}^{l_2+1} \times \mathbb{R}^{l_3+1} \to \mathbb{R}$ be a multi-homogeneous polynomial of total degree at most $r$. Suppose the coefficients of $P$ are random on an auxiliary probability space and are independent of $(\Theta_1, \Theta_2, \Theta_3)$. If the random coefficients satisfy $\|c_\alpha\|_{L_q} \leq K_q \|c_\alpha\|_{L_2}$ in the Hermite basis expansion, then for all $q \geq 2$:*

$$\|H\|_{L_q} \leq C_{r,q} (q-1)^{r/2} \|H\|_{L_2}.$$

*Proof.* The proof is identical to that of Lemma 14, except we begin with the version of Gaussian hypercontractivity that handles random coefficients satisfying the stated assumption. $\square$

Recall

$$\boldsymbol{a} := \boldsymbol{V}^\top \boldsymbol{v} \in \mathbb{R}^n \quad \boldsymbol{b} := \boldsymbol{U}^\top \boldsymbol{u} \in \mathbb{R}^d, \quad \text{and} \quad \boldsymbol{u}_\beta = \boldsymbol{U}^\top \boldsymbol{\beta}_*$$

Then, since $\boldsymbol{u}, \boldsymbol{u}$ are fixed, and $\boldsymbol{U}, \boldsymbol{V}$ are independent Haar orthogonal matrices, we have that $\boldsymbol{a}, \boldsymbol{b}$ are all uniformly random vectors on their respective spheres. Additionally, using the assumption that $\boldsymbol{\beta}_*$ is uniformly random such that $\boldsymbol{\beta}_*^\top \boldsymbol{u}$ is constant. $\boldsymbol{u}_\beta$ is uniformly random on a sphere $\mathbb{S}^{d-2}$.

Consider the following centered versions and polynomial representations.

1. $Y_h := \|\boldsymbol{h}\|^2 - \mathbb{E}\left[\|\boldsymbol{h}\|^2\right] = \boldsymbol{a}^\top \left(\boldsymbol{\Sigma}^\dagger \boldsymbol{\Sigma}^{\dagger\top} - \mu_h\right) \boldsymbol{a}$

2. $Y_k := \|\boldsymbol{k}\|^2 - \mathbb{E}\left[\|\boldsymbol{k}\|^2\right] = \boldsymbol{b}^\top \left(\boldsymbol{\Sigma}^{\dagger\top} \boldsymbol{\Sigma}^\dagger - \mu_k\right) \boldsymbol{b}$

3. $Y_t := \|\boldsymbol{t}\|^2 - \mathbb{E}\left[\|\boldsymbol{t}\|^2\right] = \boldsymbol{a}^\top \left((I - \boldsymbol{\Sigma}^\dagger \boldsymbol{\Sigma}) - \mu_t\right)$

4. $Y_s := \|\boldsymbol{s}\|^2 - \mathbb{E}\left[\|\boldsymbol{s}\|^2\right] = \boldsymbol{b}^\top \left((I - \boldsymbol{\Sigma}\boldsymbol{\Sigma}^\dagger) - \mu_t\right) \boldsymbol{b}$

5. $Y_\xi := \frac{\xi}{\eta} - \mathbb{E}\left[\frac{\xi}{\eta}\right] = \boldsymbol{a}^\top \boldsymbol{\Sigma} \boldsymbol{b} = \boldsymbol{a}^\top \boldsymbol{\Sigma}^\dagger \boldsymbol{b}$

6. $\tilde{T}_1 := \boldsymbol{\beta}_*^\top \boldsymbol{u} \boldsymbol{k}^\top \boldsymbol{A}^\dagger \boldsymbol{\beta}_* - \mathbb{E}\left[\boldsymbol{\beta}_*^\top \boldsymbol{u} \boldsymbol{k}^\top \boldsymbol{A}^\dagger \boldsymbol{\beta}_*\right] = (\boldsymbol{\beta}_*^\top \boldsymbol{u}) \boldsymbol{b}^\top (\boldsymbol{\Sigma}^{\dagger\top} \boldsymbol{\Sigma}^\dagger) \boldsymbol{u}_\beta - \mu_{\tilde{T}_1}(\boldsymbol{b}^\top \boldsymbol{b})$

7. $\tilde{T}_2 := \boldsymbol{k}^\top \boldsymbol{A}^\dagger \boldsymbol{A}^{\dagger\top} \boldsymbol{k} - \mathbb{E}\left[\boldsymbol{k}^\top \boldsymbol{A}^\dagger \boldsymbol{A}^{\dagger\top} \boldsymbol{k}\right] = \boldsymbol{b}^\top \left(\left(\boldsymbol{\Sigma}^{\dagger\top} \boldsymbol{\Sigma}^\dagger\right)^2 - \mu_{\tilde{T}_2}\right) \boldsymbol{b}$

8. $\tilde{T}_3 := \boldsymbol{\beta}_*^\top \boldsymbol{s} \boldsymbol{u}^\top \boldsymbol{\beta}_* - \mathbb{E}\left[\boldsymbol{\beta}_*^\top \boldsymbol{s} \boldsymbol{u}^\top \boldsymbol{\beta}_*\right] = (\boldsymbol{\beta}_*^\top \boldsymbol{u}) \boldsymbol{u}_\beta^\top (I - \boldsymbol{\Sigma}\boldsymbol{\Sigma}^\dagger) \boldsymbol{b} - \mu_{\tilde{T}_3}(\boldsymbol{u}_\beta^\top \boldsymbol{u}_\beta)$

9. $\tilde{T}_4 := \boldsymbol{\beta}_*^\top \boldsymbol{A}\boldsymbol{A}^\dagger \boldsymbol{u}\boldsymbol{s}^\top \boldsymbol{\beta}_* - \mathbb{E}\left[\boldsymbol{\beta}_*^\top \boldsymbol{A}\boldsymbol{A}^\dagger \boldsymbol{u}\boldsymbol{s}^\top \boldsymbol{\beta}_*\right] = \boldsymbol{u}_\beta^\top \boldsymbol{\Sigma}\boldsymbol{\Sigma}^\dagger \boldsymbol{b}\boldsymbol{b}^\top (I - \boldsymbol{\Sigma}\boldsymbol{\Sigma}^\dagger)\boldsymbol{u}_\beta - \mu_{\tilde{T}_4}(\boldsymbol{b}^\top \boldsymbol{b})(\boldsymbol{u}_\beta^\top \boldsymbol{u}_\beta)$

10. $\tilde{T}_5 := \boldsymbol{\beta}_*^\top \boldsymbol{h}^\top \boldsymbol{h}\boldsymbol{\beta}_* - \mathbb{E}\left[\boldsymbol{\beta}_*^\top \boldsymbol{h}^\top \boldsymbol{h}\boldsymbol{\beta}_*\right] = \left(\boldsymbol{u}_\beta \boldsymbol{\Sigma}^{\dagger\top} \boldsymbol{a}\right)^2 - \mu_{\tilde{T}_5}(\boldsymbol{a}^\top \boldsymbol{a})(\boldsymbol{u}_\beta^\top \boldsymbol{u}_\beta)$

11. $\tilde{T}_6 := \boldsymbol{h}(\boldsymbol{A}^\dagger)^\top \boldsymbol{A}^\dagger \boldsymbol{h}^\top - \mathbb{E}\left[\boldsymbol{h}(\boldsymbol{A}^\dagger)^\top \boldsymbol{A}^\dagger \boldsymbol{h}^\top\right] = \boldsymbol{a}^\top \left(\left(\boldsymbol{\Sigma}^\dagger \boldsymbol{\Sigma}^{\dagger\top}\right)^2 - \mu_{\tilde{T}_6}\right) \boldsymbol{a}$

12. $\tilde{S}_1 := \boldsymbol{\beta}_*^\top \boldsymbol{u}\boldsymbol{h}\boldsymbol{\beta}_* - \mathbb{E}\left[\boldsymbol{\beta}_*^\top \boldsymbol{u}\boldsymbol{h}\boldsymbol{\beta}_*\right] = (\boldsymbol{\beta}_*^\top \boldsymbol{u})\, \boldsymbol{a}^\top \boldsymbol{\Sigma}^\dagger \boldsymbol{u}_\beta$

13. $\tilde{S}_2 := \boldsymbol{\beta}_*^\top \boldsymbol{A}\boldsymbol{A}^\dagger \boldsymbol{u}\boldsymbol{h}\boldsymbol{\beta}_* - \mathbb{E}\left[\boldsymbol{\beta}_*^\top \boldsymbol{A}\boldsymbol{A}^\dagger \boldsymbol{u}\boldsymbol{h}\boldsymbol{\beta}_*\right] = \boldsymbol{u}_\beta^\top \boldsymbol{\Sigma}\boldsymbol{\Sigma}^\dagger \boldsymbol{b}\boldsymbol{a}^\top \boldsymbol{\Sigma}^\dagger \boldsymbol{u}_\beta$

14. $\tilde{S}_3 := \boldsymbol{\beta}_*^\top \boldsymbol{s}\boldsymbol{h}\boldsymbol{\beta}_* - \mathbb{E}\left[\boldsymbol{\beta}_*^\top \boldsymbol{s}\boldsymbol{h}\boldsymbol{\beta}_*\right] = \boldsymbol{u}_\beta(I - \boldsymbol{\Sigma}\boldsymbol{\Sigma}^\dagger)\boldsymbol{b}\boldsymbol{a}^\top \boldsymbol{\Sigma}^\dagger \boldsymbol{u}_\beta$

15. $\tilde{S}_4 := \boldsymbol{k}^\top \boldsymbol{A}^\dagger \boldsymbol{h}^\top - \mathbb{E}\left[\boldsymbol{k}^\top \boldsymbol{A}^\dagger \boldsymbol{h}^\top\right] = \boldsymbol{b}^\top \boldsymbol{\Sigma}^{\dagger\top} \boldsymbol{\Sigma}^\dagger \boldsymbol{\Sigma}^{\dagger\top} \boldsymbol{a}$

Hence we see that these are all homogeneous polynomials in uniformly random spherical variables. Thus, we can use Lemma 14, we get bounds on the higher moments. In particular, since the coefficients are only dependent on constants and $\boldsymbol{\Sigma}$, we see that the coefficients are independent of $\boldsymbol{a}, \boldsymbol{b}, \boldsymbol{u}_\beta$. Then using a change of basis we see that that coefficients of the decomposition are also random and independent of the input variables. Finally, since the spectrum converges to the Marchenko-Pastur, we have that the coefficients have bounded moments. Hence the second assumption is satisfied.

### E.4.3 STEP 3(C): BOUNDING $\gamma_i$ MOMENTS.

**Lemma 16** (Moments of $\gamma_i/\eta^2$). *We have:*

*(i) For $\gamma_1/\eta^2$,*

$$\mathbb{E}\left[\frac{\gamma_1}{\eta^2}\right] = \frac{c}{\rho^2} + \frac{1}{\eta^2} + o\left(\frac{1}{\rho^2}\right), \quad \mathrm{Var}\left(\frac{\gamma_1}{\eta^2}\right) = O\left(\frac{1}{\rho^4 n}\right).$$

*(ii) For $\gamma_2/\eta^2$,*

$$\mathbb{E}\left[\frac{\gamma_2}{\eta^2}\right] = \frac{1}{\rho^2} + \frac{1}{\eta^2} + o\left(\frac{1}{\rho^2}\right), \quad \mathrm{Var}\left(\frac{\gamma_2}{\eta^2}\right) = O\left(\frac{1}{\rho^4 n}\right).$$

*Proof.* We decompose

$$\frac{\gamma_i}{\eta^2} = \zeta_i + \frac{\xi^2}{\eta^2}, \quad i = 1, 2, \quad \text{where } \zeta_1 = \|\boldsymbol{t}\|^2 \|\boldsymbol{k}\|^2, \quad \zeta_2 = \|\boldsymbol{s}\|^2 \|\boldsymbol{h}\|^2.$$

**Expectation Estimates:** We begin by noting that $\|\boldsymbol{t}\|^2$ depends only on $\boldsymbol{V}$ and is independent of $\boldsymbol{U}, \boldsymbol{\Sigma}$. $\|\boldsymbol{s}\|^2$ depends only on $\boldsymbol{U}$ and is independent of $\boldsymbol{V}, \boldsymbol{\Sigma}$. Additionally, $\|\boldsymbol{k}\|^2$ depends on $\boldsymbol{U}$ and $\boldsymbol{\Sigma}$, hence is independent of $\boldsymbol{V}$. Also $\|\boldsymbol{h}\|^2$ depends on $\boldsymbol{V}$ and $\boldsymbol{\Sigma}$ and is independent of $\boldsymbol{U}$, hence is independent of $\boldsymbol{U}$.

Thus, we have have that $\|\boldsymbol{t}\|^2$ and $\|\boldsymbol{k}\|^2$ are independent and $\|\boldsymbol{s}\|^2$ and $\|\boldsymbol{h}\|^2$ are independent. Thus, we see that

$$\mathbb{E}[\zeta_1] = \mathbb{E}[\|\boldsymbol{t}\|^2 \|\boldsymbol{k}\|^2] = \mathbb{E}[\|\boldsymbol{t}\|^2]\, \mathbb{E}[\|\boldsymbol{k}\|^2].$$

Using Lemma 9 again,

$$\mathbb{E}[\|\boldsymbol{t}\|^2] = 1 - c, \quad \mathbb{E}[\|\boldsymbol{k}\|^2] = \frac{1}{\rho^2} \frac{c}{1 - c} + o\left(\frac{1}{\rho^2}\right).$$

We plug them into the expectation and get:

$$\mathbb{E}[\zeta_1] = (1 - c)\left[\left(\frac{1}{\rho^2} \frac{c}{1 - c}\right) + o\left(\frac{1}{\rho^2}\right)\right] = \frac{c}{\rho^2} + o\left(\frac{1}{\rho^2}\right).$$

Finally, we also have that from Lemma 9,

$$\mathbb{E}\left[\frac{\xi^2}{\eta^2}\right] = \frac{1}{\eta^2} + O\left(\frac{1}{\rho^2 n}\right), \quad \mathrm{Var}\left(\frac{\xi^2}{\eta^2}\right) = O\left(\frac{1}{\rho^4 n}\right),$$

Hence,

$$\mathbb{E}\left[\frac{\gamma_1}{\eta^2}\right] = \mathbb{E}[\zeta_1] + \mathbb{E}\left[\frac{\xi^2}{\eta^2}\right] = \frac{c}{\rho^2} + \frac{1}{\eta^2} + o\left(\frac{1}{\rho^2}\right).$$

A similar argument applies for $\gamma_2/\eta^2$, using the corresponding results for $\|s\|^2, \|h\|^2$.

**Variance Estimates:**

Again using independence, we have that

$$\mathrm{Var}(\|t\|^2 \|k\|^2) = \mathrm{Var}(\|t\|^2)\mathrm{Var}(\|k\|^2) + \mathbb{E}[\|t\|^2]^2 \mathrm{Var}(\|k\|^2) + \mathbb{E}[\|k\|^2]^2 \mathrm{Var}(\|t\|^2)$$

$$= O\left(\frac{1}{n}\right) O\left(\frac{1}{\rho^4 n}\right) + (1-c)^2 O\left(\frac{1}{\rho^4 n}\right) + \frac{1}{\rho^4} \frac{c^2}{(1-c)^2} O\left(\frac{1}{n}\right)$$

$$= O\left(\frac{1}{\rho^4 n}\right).$$

We then use Lemma 34 to compute the variance of the sum:

$$\mathrm{Var}\left(\zeta_1 + \frac{\xi^2}{\eta^2}\right) \le \left(\sqrt{\mathrm{Var}(\zeta_1)} + \sqrt{\mathrm{Var}\left(\frac{\xi^2}{\eta^2}\right)}\right)^2$$

$$= \left(\sqrt{O\left(\frac{1}{\rho^4 n}\right)} + \sqrt{O\left(\frac{1}{\rho^4 n}\right)}\right)^2$$

$$= O\left(\frac{1}{\rho^4 n}\right).$$

This proof is similar to the other case. $\qquad\square$

**Lemma 17** (Moments of $(\gamma_i/\eta^2)^2$). *We have, as $n, d \to \infty$ with $d/n \to c \neq 1$,*

*(i) For $\gamma_1/\eta^2$,*

$$\mathbb{E}\left[\left(\frac{\gamma_1}{\eta^2}\right)^2\right] = \left(\frac{c}{\rho^2} + \frac{1}{\eta^2}\right)^2 + O\left(\frac{1}{\rho^4}\right), \quad \mathrm{Var}\left(\left(\frac{\gamma_1}{\eta^2}\right)^2\right) = O\left(\frac{1}{\rho^4 n}\right).$$

*(ii) For $\gamma_2/\eta^2$,*

$$\mathbb{E}\left[\left(\frac{\gamma_2}{\eta^2}\right)^2\right] = \left(\frac{1}{\rho^2} + \frac{1}{\eta^2}\right)^2 + O\left(\frac{1}{\rho^4}\right), \quad \mathrm{Var}\left(\left(\frac{\gamma_2}{\eta^2}\right)^2\right) = O\left(\frac{1}{\rho^4 n}\right).$$

*Proof.* Write, for $i \in \{1, 2\}$,

$$\frac{\gamma_i}{\eta^2} = \zeta_i + \frac{\xi^2}{\eta^2}, \quad \zeta_1 := \|t\|^2 \|k\|^2, \quad \zeta_2 := \|s\|^2 \|h\|^2.$$

**Means.** Using Lemma 16 and the fact that for any random variable

$$\mathbb{E}\left[Y^2\right] = \mathbb{E}[Y]^2 + \mathrm{Var}(Y)$$

we get the means.

**Variances.** Using

$$Y^2 = \mathbb{E}\left[Y\right]^2 + 2(\mathbb{E}\left[Y\right])\left(Y - \mathbb{E}\left[Y\right]\right) + \left(Y - \mathbb{E}\left[Y\right]\right)^2,$$

Thus, using Lemma 34 we have that

$$\text{Var}(Y^2) \leq \left( \sqrt{4 \left( \mathbb{E}\left[ X \right] \right)^2 \text{Var}(X_i)} + \sqrt{\text{Var}\left( (Y - \mathbb{E}\left[ Y \right])^2 \right)} \right)^2.$$

By spherical hypercontractivity for degree-$4$ polynomials,

$$\mathbb{E}\left[ \left( \frac{\gamma_i^2}{\eta^4} - \mathbb{E}\left[ \frac{\gamma_i^2}{\eta^4} \right] \right)^4 \right] \lesssim \text{Var}\left( \frac{\gamma_i^2}{\eta^4} \right)^2,$$

hence

$$\text{Var}\left( \left( \frac{\gamma_i^2}{\eta^4} - \mathbb{E}\left[ \frac{\gamma_i^2}{\eta^4} \right] \right)^2 \right) \mathbb{E}\left[ \left( \frac{\gamma_i^2}{\eta^4} - \mathbb{E}\left[ \frac{\gamma_i^2}{\eta^4} \right] \right)^4 \right] \lesssim \text{Var}\left( \frac{\gamma_i^2}{\eta^4} \right)^2.$$

Using $\mathbb{E}\left[ \frac{\gamma_i}{\eta^2} \right]^2 = O(1)$ and $\text{Var}\left( \frac{\gamma_i}{\eta^2} \right) = O(\rho^{-4}n^{-1})$ gives

$$\text{Var}\left( \frac{\gamma_i^2}{\eta^4} \right) = O\left( \frac{1}{\rho^4 n} \right),$$

as claimed. $\qquad\square$

**Lemma 18** (Finite Negative Moments of $\gamma_i$). *Fix $p > 0$. There exists an $N(p)$ such that for all $n, d \geq N(p)$, we have that for $c < 1$*

$$\mathbb{E}\left[ \gamma_1^{-p} \right] \leq \eta^{-2p} \mathbb{E}\left[ \sigma_1^{2p} \right] \mathbb{E}\left[ T^{-p} \right] \leq \frac{\rho^{2p}}{\eta^{2p}} M^p$$

*and for $c > 1$, we have that*

$$\mathbb{E}\left[ \gamma_2^{-p} \right] \leq \eta^{-2p} \mathbb{E}\left[ \sigma_1^{2p} \right] \mathbb{E}\left[ S^{-p} \right] \leq \frac{\rho^{2p}}{\eta^{2p}}, M^p$$

*where $\sigma_1$ is the largest singular value of $A$, $T := \|\boldsymbol{t}\|^2 \sim Beta\left( \frac{n-d}{2}, \frac{d}{2} \right)$, and $S := \|\boldsymbol{s}\|^2 \sim Beta\left( \frac{d-n}{2}, \frac{n}{2} \right)$.*

*Proof.* Recall our SVD $\boldsymbol{A} = \boldsymbol{U}\boldsymbol{\Sigma}\boldsymbol{V}^\top$ and that

$$\gamma_1 = \eta^2 \|\boldsymbol{t}\|^2 \|\boldsymbol{k}\|^2 + \xi^2 \quad \text{and} \quad \gamma_2 = \eta^2 \|\boldsymbol{s}\|^2 \|\boldsymbol{h}\|^2 + \xi^2.$$

Then we have that

$$\|\boldsymbol{k}\|^2 = \sum_{i=1}^{d} \frac{\boldsymbol{b}_i^2}{\sigma_i^2} \geq \frac{1}{\sigma_1^2} \|\boldsymbol{b}\|^2 = \frac{1}{\sigma_1^2}$$

Similarly,

$$\|\boldsymbol{h}\|^2 = \sum_{i=1}^{n} \frac{\boldsymbol{a}_i^2}{\sigma_i^2} \geq \frac{1}{\sigma_1^2} \|\boldsymbol{a}\|^2 = \frac{1}{\sigma_1^2}$$

Thus, we see that

$$\gamma_1 \geq \eta^2 \|\boldsymbol{t}\|^2 \frac{1}{\sigma_1^2} \quad \text{and} \quad \gamma_2 \geq \eta^2 \|\boldsymbol{s}\|^2 \frac{1}{\sigma_1^2}.$$

$\|\boldsymbol{t}\|^2$ depends only on $\boldsymbol{V}$ and is independent of $\boldsymbol{U}, \boldsymbol{\Sigma}$. $\|\boldsymbol{s}\|^2$ depends only on $\boldsymbol{U}$ and is independent of $\boldsymbol{V}, \boldsymbol{\Sigma}$. $\sigma_1$ depends only on $\boldsymbol{\Sigma}$ and is independent of $\boldsymbol{U}, \boldsymbol{V}$. Therefore, $\sigma_1$ is independent of $T := \|\boldsymbol{t}\|^2$ and of $S := \|\boldsymbol{s}\|^2$.

Thus, we get that

$$\frac{1}{\gamma_1^p} \leq \frac{1}{\eta^{2p}} \frac{\sigma_1^{2p}}{\|\boldsymbol{t}\|^{2p}} \quad \text{and} \quad \frac{1}{\gamma_2^p} \leq \frac{1}{\eta^{2p}} \frac{\sigma_1^{2p}}{\|\boldsymbol{s}\|^{2p}}$$

Then taking the expectation and using the independence, we get that

$$\mathbb{E}\left[ \frac{1}{\gamma_1^p} \right] \leq \frac{1}{\eta^{2p}} \mathbb{E}\left[ \frac{1}{\|\boldsymbol{t}\|^{2p}} \right] \mathbb{E}\left[ \sigma_1^{2p} \right] \quad \text{and} \quad \mathbb{E}\left[ \frac{1}{\gamma_2^p} \right] \leq \frac{1}{\eta^{2p}} \mathbb{E}\left[ \frac{1}{\|\boldsymbol{s}\|^{2p}} \right] \mathbb{E}\left[ \sigma_1^{2p} \right]$$

For $c < 1$ (where $d < n$), the right null space of $\boldsymbol{A}$ (dimension $n - d$) is a uniformly random $(n - d)$-dimensional subspace of $\mathbb{R}^n$. The squared norm $\|\boldsymbol{t}\|^2$ represents the squared length of the projection of the fixed unit vector $\boldsymbol{v} \in \mathbb{R}^n$ onto this random subspace. The distribution of such a squared projection norm is $\text{Beta}\left(\frac{n-d}{2}, \frac{d}{2}\right)$, as it can be represented as the ratio of two independent chi-squared random variables: $\sum_{i=1}^{n-d} G_i^2 / \sum_{i=1}^{n} G_i^2$, where $G_i \sim N(0,1)$ IID, which follows the desired Beta distribution. Similarly for $c > 1$.

Since the eigenvalue distribution converges to the compactly supported distribution. We can see that for sufficiently large $n, d$, we have that there exists an $M \geq 1$ such that $\sigma_1 \leq \rho M$ almost surely.

For $Y \sim \text{Beta}(\alpha, \beta)$ and $p < \alpha$,

$$\mathbb{E}[Y^{-p}] = \frac{\Gamma(\alpha - p)\,\Gamma(\alpha + \beta)}{\Gamma(\alpha)\,\Gamma(\alpha + \beta - p)}.$$

Moreover, using Stirling on the $\Gamma$ ratio,

$$\mathbb{E}[T^{-p}] \to_{n,d\to\infty} \left(\frac{\alpha_1 + \beta_1}{\alpha_1}\right)^p = \left(\frac{1}{1 - c}\right)^p \quad (c < 1),$$

and

$$\mathbb{E}[S^{-p}] \to_{n,d\to\infty} \left(\frac{\alpha_2 + \beta_2}{\alpha_2}\right)^p = \left(\frac{c}{c - 1}\right)^p \quad (c > 1).$$

Thus, there is an $M$ such that

$$\mathbb{E}\left[\frac{1}{\gamma_1^p}\right] \leq \left(\frac{\rho}{\eta}\right)^{2p} M^p \quad \text{and} \quad \mathbb{E}\left[\frac{1}{\gamma_2^p}\right] \leq \left(\frac{\rho}{\eta}\right)^{2p} M^p$$

$\square$

**Lemma 19** (Moments of $\eta^2/\gamma_i$). *We have:*

*(i) For $\eta^2/\gamma_1$,*

$$\mathbb{E}\left[\frac{\eta^2}{\gamma_1}\right] = \frac{\rho^2 \eta^2}{\eta^2 c + \rho^2} + o\left(\frac{1}{\rho^2}\right), \quad \text{Var}\left(\frac{\eta^2}{\gamma_1}\right) = O\left(\frac{1}{n}\right).$$

*(ii) For $\eta^2/\gamma_2$,*

$$\mathbb{E}\left[\frac{\eta^2}{\gamma_2}\right] = \frac{\rho^2 \eta^2}{\eta^2 + \rho^2} + o\left(\frac{1}{\rho^2}\right), \quad \text{Var}\left(\frac{\eta^2}{\gamma_2}\right) = O\left(\frac{1}{n}\right).$$

*Proof.* By Lemmas 32 and 16, the expectation of $\eta^2/\gamma_1$ can be computed by:

$$\mathbb{E}\left[\frac{\eta^2}{\gamma_1}\right] = \frac{1}{\mathbb{E}[\gamma_1/\eta^2]} 1 + o\left(\frac{1}{\rho^2 d}\right) = \frac{\rho^2 \eta^2}{\eta^2 c + \rho^2} + o\left(\frac{1}{\rho^2}\right).$$

By Lemmas 33 and 16, the variance of $\eta^2/\gamma_1$ can be computed by:

$$\text{Var}\left(\frac{\eta^2}{\gamma_1}\right) = \frac{1}{\mathbb{E}[\gamma_1/\eta^2]^4} O\left(\text{Var}\left(\frac{\gamma_1}{\eta^2}\right)\right) + o\left(\text{Var}\left(\frac{\gamma_1}{\eta^2}\right)\right)$$

$$= \frac{\rho^8 \eta^8}{(\eta^2 c + \rho^2)^4} O\left(\frac{1}{n}\right) + o\left(\frac{1}{n}\right)$$

$$= O\left(\frac{1}{n}\right) \quad \text{by the scalings of } \eta \text{ and } \rho.$$

The proof is similar for the other term. $\square$

**Lemma 20** (Moments of $\eta^4/\gamma_i^2$). *We have:*

*(i) For $\eta^4/\gamma_1^2$,*

$$\mathbb{E}\left[\frac{\eta^4}{\gamma_1^2}\right] = \frac{\rho^4 \eta^4}{(\eta^2 c + \rho^2)^2} + o(1), \quad \text{Var}\left(\frac{\eta^4}{\gamma_1^2}\right) = O\left(\frac{1}{n}\right).$$

*(ii) For $\eta^4/\gamma_2^2$,*

$$\mathbb{E}\left[\frac{\eta^4}{\gamma_2^2}\right] = \frac{\rho^4\eta^4}{(\eta^2+\rho^2)^2} + o\left(1\right), \quad \mathrm{Var}\left(\frac{\eta^4}{\gamma_2^2}\right) = O\left(\frac{1}{n}\right).$$

*Proof.* The expectation of $\eta^4/\gamma_1^2$ can be computed by Lemma 19. By definition we have that

$$\mathbb{E}\left[\frac{\eta^4}{\gamma_1^2}\right] = \left(\mathbb{E}\left[\frac{\eta^2}{\gamma_1}\right]\right)^2 + \mathrm{Var}\left(\frac{\eta^2}{\gamma_1}\right) = \left(\frac{\rho^2\eta^2}{\eta^2c+\rho^2} + o\left(\frac{1}{\rho^2}\right)\right)^2 + O\left(\frac{1}{n}\right).$$

The variance follows Lemma 33 and Lemma 17:

$$\mathrm{Var}\left(\frac{\eta^4}{\gamma_1^2}\right) = O\left(\frac{1}{n}\right),$$

since the mean is $O(1)$.

The proof is similar for the other term. $\qquad\square$

**Lemma 21.** *Suppose $\varepsilon \in \mathbb{R}^n$ whose entries have mean $0$, variance $\tau_\varepsilon$, and follow our noise assumptions. Then for any indepedent random matrix $\boldsymbol{Q} \in \mathbb{R}^{n\times n}$, we have*

$$\mathbb{E}_{\varepsilon,\boldsymbol{Q}}\left[\varepsilon^\top \boldsymbol{Q}\varepsilon\right] = \tau_\varepsilon^2\mathbb{E}\left[\mathrm{Tr}(\boldsymbol{Q})\right].$$

*Proof.* We have that

$$\varepsilon^\top \boldsymbol{Q}\varepsilon = \sum_{i=1}^n\sum_{j=1}^n \varepsilon_i\varepsilon_j Q_{ij}.$$

We take the expectation of this sum. By the independence assumption and assumption $\mathbb{E}[\varepsilon_i\varepsilon_j] = 0$ when $i \neq j$, we then have

$$\mathbb{E}_{\varepsilon,\boldsymbol{Q}}\left[\varepsilon^\top \boldsymbol{Q}\varepsilon\right] = \sum_{i=1}^n \mathbb{E}\left[\varepsilon_i^2\right]\mathbb{E}\left[Q_{ii}\right] = \tau_\varepsilon^2\mathbb{E}\left[\sum_{i=1}^n Q_{ii}\right] = \tau_\varepsilon^2\mathbb{E}\left[\mathrm{Tr}(\boldsymbol{Q})\right].$$

$\qquad\square$

### E.5 STEP 4: BOUNDING THE EXPECTATION OF PRODUCTS OF DEPENDENT TERMS

In Section E.2 we decomposed the error into four terms – Bias, Variance, Data Noise and Target alignment. In Section E.3, we wrote each of these terms as the sum and product of various "elementary building blocks". In Section E.4, we should that these elementary building blocks concentrate. In this section, since we have tight concentration (i.e., the higher moment bounds). We can use Lemma 36 and Lemma 37, which shows that the expectation of the product can be approximated by the product of the expectations. In this section, we do that calculation for our different terms.

#### E.5.1 STEP 4: BIAS

We begin with the bias term. Recall that for $c < 1$, the expected bias by Lemma 5 is equal to

$$\mathbb{E}[\textbf{Bias}] = \mathbb{E}\left[\left[\tilde{\alpha}_Z - \alpha_Z + \frac{\xi}{\gamma_1}(\alpha_Z - \alpha_A)\right]^2\tilde{\eta}^2(\boldsymbol{\beta}_*^\top\boldsymbol{u})^2 + \frac{\tilde{\eta}^2}{\eta^2}\frac{\xi^2}{\gamma_1^2}\tau_\varepsilon^2\|\boldsymbol{p}_1\|^2\right],$$

where the cross term equals $0$ due to $\varepsilon$ having mean zero entries. These two remaining expectations are given by Lemmas 22, 23, informally via:

$$\text{Lemma 22} \quad + \quad \tau_\varepsilon^2\frac{\tilde{\eta}^2}{\eta^2} \times \text{Lemma 23}.$$

For $c < 1$, we can plug in the value to get that the expected first term is given by

$$\tilde{\eta}^2(\boldsymbol{\beta}_*^\top\boldsymbol{u})^2\left[(\tilde{\alpha}_Z - \alpha_Z) + \frac{\rho^2}{\eta^2c+\rho^2}(\alpha_Z - \alpha_A)\right]^2 + o(1) + O\left(\frac{\eta}{n}\right)$$

and the second is given by

$$\tau_\varepsilon^2 \frac{\tilde{\eta}^2}{\eta^2} \left( \frac{c}{c-1} \frac{\eta^2}{\eta^2 c + \rho^2} + o(1) + O\left(\frac{1}{\rho^2 n}\right)\right).$$

Adding them, we then have the desired result:

$$\frac{\tilde{\eta}^2}{\tilde{n}} \left( \left[(\tilde{\alpha}_Z - \alpha_Z) + \frac{\rho^2}{\eta^2 c + \rho^2}(\alpha_Z - \alpha_A)\right]^2 (\boldsymbol{\beta}_*^\top \boldsymbol{u})^2 + \tau_\varepsilon^2 \frac{c}{1-c} \frac{1}{\eta^2 c + \rho^2}\right) + o\left(\frac{1}{\tilde{n}}\right) + O\left(\frac{\eta}{n^2}\right).$$

For $c > 1$, we instead have the following expanson:

$$\underbrace{\boldsymbol{\beta}_*^\top \left[(\tilde{\alpha}_Z - \alpha_Z)\boldsymbol{I} + \frac{\xi}{\gamma_2}(\alpha_Z \boldsymbol{I} - \alpha_A \boldsymbol{A}\boldsymbol{A}^\dagger)\right]\tilde{\boldsymbol{Z}}}_{\boldsymbol{t}_1} - \underbrace{\alpha_A \frac{\eta \|\boldsymbol{s}\|^2}{\gamma_2} \boldsymbol{\beta}_*^\top \boldsymbol{h}^\top \boldsymbol{u}^\top \tilde{\boldsymbol{Z}}}_{\boldsymbol{t}_2} + \underbrace{\frac{\tilde{\eta}}{\eta} \frac{\xi}{\gamma_2} \boldsymbol{\varepsilon}^\top \boldsymbol{p}_2 \tilde{\boldsymbol{v}}^\top}_{\boldsymbol{t}_3}$$

The bias equals the expectation of the norm of this vector. Taking the Frobenius norm, we have the six terms. Among the cross-terms, $\langle \boldsymbol{t}_1, \boldsymbol{t}_3 \rangle$ and $\langle \boldsymbol{t}_2, \boldsymbol{t}_3 \rangle$ have zero mean since $\boldsymbol{t}_3$ contains $\boldsymbol{\varepsilon}$ whose entries have mean 0. We now look at the other terms

$$\mathbb{E}\left[\|\boldsymbol{t}_3\|^2\right] = \mathbb{E}\left[\left\|\frac{\tilde{\eta}}{\eta} \frac{\xi}{\gamma_2} \boldsymbol{\varepsilon}^\top \boldsymbol{p}_2 \tilde{\boldsymbol{v}}^\top\right\|^2\right] = \tau_\varepsilon^2 \frac{\tilde{\eta}^2}{\eta^2} \mathbb{E}\left[\frac{\xi^2}{\gamma_2^2} \|\boldsymbol{p}_2\|^2\right] \quad \text{by Lemma 21}$$

The expectation is given by Lemma 23. Subsequently, Lemmas 22, 24, 25 give $\mathbb{E}[\|\boldsymbol{t}_1\|^2]$, $\mathbb{E}[\|\boldsymbol{t}_2\|^2]$, $\mathbb{E}[\langle \boldsymbol{t}_1, \boldsymbol{t}_3 \rangle]$ respectively. Informally, we can compute the bias via:

$$\mathbb{E}[\textbf{Bias}] = \mathbb{E}\left[\left\|\boldsymbol{\beta}_*^\top \left[(\tilde{\alpha}_Z - \alpha_Z)\boldsymbol{I} + \frac{\xi}{\gamma_2}(\alpha_Z \boldsymbol{I} - \alpha_A \boldsymbol{A}\boldsymbol{A}^\dagger)\right]\tilde{\boldsymbol{Z}} - \alpha_A \frac{\eta\|\boldsymbol{s}\|^2}{\gamma_2}\boldsymbol{\beta}_*^\top \boldsymbol{h}^\top \boldsymbol{u}^\top \tilde{\boldsymbol{Z}} + \frac{\tilde{\eta}}{\eta} \frac{\xi}{\gamma_2}\boldsymbol{\varepsilon}^\top \boldsymbol{p}_2 \tilde{\boldsymbol{v}}^\top\right\|^2\right]$$

$$= \mathbb{E}[\|\boldsymbol{t}_1\|^2] + \mathbb{E}[\|\boldsymbol{t}_2\|^2] + \mathbb{E}[\|\boldsymbol{t}_3\|^2] - 2\mathbb{E}[\langle \boldsymbol{t}_1, \boldsymbol{t}_3 \rangle^2]$$

$$= \text{Lemma 22} + \tau_\varepsilon^2 \frac{\tilde{\eta}^2}{\eta^2}\text{Lemma 23} + \text{Lemma 24} - 2 \times \text{Lemma 25}.$$

Similar to $c < 1$, adding them together and dividing by $\tilde{n}$, we get

$$\frac{\tilde{\eta}^2}{\tilde{n}} \left[(\boldsymbol{\beta}_*^\top \boldsymbol{u})^2 \left((\tilde{\alpha}_Z - \alpha_Z)^2 + \frac{\rho^2}{\eta^2 + \rho^2}\left(\alpha_Z - \frac{\alpha_A}{c}\right)\right)^2 + \alpha_A^2 \frac{\|\boldsymbol{\beta}_*\|^2}{d}\left(\frac{c-1}{c}\right)\frac{\eta^2\rho^2}{(\eta^2+\rho^2)^2} + \frac{\tau_\varepsilon^2}{c-1}\frac{\eta^2 c + \rho^2}{(\eta^2+\rho^2)^2}\right]$$

$$+ o\left(\frac{1}{\tilde{n}}\right) + O\left(\frac{\eta}{n^2}\right).$$

### E.5.2 STEP 4: VARIANCE

Recall that for the variance, we have the following expression (Section E.3.2).

$$\mathbb{E}\left[\frac{1}{\tilde{n}}\left\|\boldsymbol{\beta}_{int}^\top \tilde{\boldsymbol{A}}\right\|_F^2\right] = \mathbb{E}\left[\frac{\tilde{\tau}^2 \alpha_z^2}{d}\boldsymbol{\beta}_*^\top \boldsymbol{Z}(\boldsymbol{Z}+\boldsymbol{A})^\dagger(\boldsymbol{Z}+\boldsymbol{A})^{\dagger\top}\boldsymbol{Z}\boldsymbol{\beta}_* + \frac{\tilde{\tau}^2 \alpha_A^2}{d}\boldsymbol{\beta}_*^\top \boldsymbol{A}(\boldsymbol{Z}+\boldsymbol{A})^\dagger(\boldsymbol{Z}+\boldsymbol{A})^{\dagger\top}\boldsymbol{A}^\top \boldsymbol{\beta}_*\right.$$

$$\left.+ \frac{2\tilde{\tau}^2 \alpha_A \alpha_z}{d}\boldsymbol{\beta}_*^\top \boldsymbol{Z}(\boldsymbol{Z}+\boldsymbol{A})^\dagger(\boldsymbol{Z}+\boldsymbol{A})^{\dagger\top}\boldsymbol{A}^\top \boldsymbol{\beta}_* + \frac{\tilde{\tau}^2}{d}\boldsymbol{\varepsilon}^\top(\boldsymbol{Z}+\boldsymbol{A})^\dagger(\boldsymbol{Z}+\boldsymbol{A})^{\dagger\top}\boldsymbol{\varepsilon}\right].$$

In particular that the expectation will be the weighted sum of the expressions from Lemmas 26, 27, 28, 29. Informally,

$$\frac{\tilde{\rho}^2}{d}\left(\alpha_Z^2 \times \text{Lemma 26} + 2\alpha_Z \alpha_A \times \text{Lemma 28} + \alpha_A^2 \times \text{Lemma 27} + \text{Lemma 29}\right).$$

This yields that for $c < 1$, after simplification, the variance is

$$\frac{\tilde{\rho}^2}{d}\left[\alpha_A^2 \|\boldsymbol{\beta}_*\|^2 + (\boldsymbol{\beta}_*^\top \boldsymbol{u})^2\left[(\alpha_Z - \alpha_A)^2\frac{\eta^2(\eta^2+\rho^2)}{(\eta^2 c + \rho^2)^2}\frac{c^2}{1-c} + 2\alpha_A(\alpha_Z - \alpha_A)\frac{\eta^2 c}{\eta^2 c + \rho^2}\right]\right.$$

$$\left.+ \tau_\varepsilon^2\left(\frac{c}{1-c}\frac{d}{\rho^2} - \frac{\eta^2}{\rho^2(\eta^2 c + \rho^2)}\frac{c^2}{1-c}\right)\right] + o(1) + O\left(\frac{1}{n}\right).$$

For $c > 1$, we similarly simplify it to:

$$\frac{\tilde{\rho}^2}{d}\left[\|\boldsymbol{\beta}_*\|^2\left(\frac{\alpha_A^2}{c} - \frac{\alpha_A^2}{d}\frac{\eta^2}{\eta^2+\rho^2}\right) + (\boldsymbol{\beta}_*^\top\boldsymbol{u})^2\frac{c}{c-1}\frac{\eta^2}{\eta^2+\rho^2}\left(\alpha_Z - \frac{\alpha_A}{c}\right)^2\right.$$
$$\left.+\tau_\varepsilon^2\left(\frac{d}{\rho^2}\frac{1}{c-1} - \frac{\eta^2}{\rho^2(\eta^2+\rho^2)}\frac{c}{c-1}\right)\right] + o(1) + O\left(\frac{1}{n}\right).$$

### E.5.3   STEP 4: DATA NOISE

Recall that for the data noise, we have the following expression

$$\frac{\tilde{\alpha}_A^2\tilde{\rho}^2}{d}\|\boldsymbol{\beta}_*\|^2$$

Noting that $\|\boldsymbol{\beta}_*\|^2 = \Theta(1)$, we see that this term has no more randomness and we do not need to estimate anything.

### E.5.4   STEP 4: TARGET ALIGNMENT

Recall from Section E.3.4 that the alignment is given by

$$-\frac{2\tilde{\alpha}_A\tilde{\rho}^2}{d}\mathbb{E}\left[\alpha_z\boldsymbol{\beta}_*^\top(\boldsymbol{Z}+\boldsymbol{A})^{\dagger\top}\boldsymbol{Z}^\top\boldsymbol{\beta}_* + \alpha_A\boldsymbol{\beta}_*^\top(\boldsymbol{Z}+\boldsymbol{A})^{\dagger\top}\boldsymbol{A}^\top\boldsymbol{\beta}_*\right]$$

From Lemma 30, we have that

$$\mathbb{E}\left[\boldsymbol{\beta}_*^\top(\boldsymbol{Z}+\boldsymbol{A})^{\dagger\top}\boldsymbol{Z}^\top\boldsymbol{\beta}_*\right] = \begin{cases} \frac{\eta^2 c}{\rho^2+\eta^2 c}(\boldsymbol{\beta}_*^\top\boldsymbol{u})^2 + o(1) + O\left(\frac{1}{n}\right) & c < 1 \\ \frac{\eta^2}{\eta^2+\rho^2}(\boldsymbol{\beta}_*^\top\boldsymbol{u})^2 + o(1) + O\left(\frac{1}{n}\right) & c > 1 \end{cases}.$$

and from Lemma 31, we have that

$$\mathbb{E}\left[\boldsymbol{\beta}_*^\top(\boldsymbol{Z}+\boldsymbol{A})^{\dagger\top}\boldsymbol{A}^\top\boldsymbol{\beta}_*\right] = \begin{cases} \|\boldsymbol{\beta}_*\|^2 - \frac{\eta^2 c}{\rho^2+\eta^2 c}(\boldsymbol{\beta}_*^\top\boldsymbol{u})^2 + o\left(\frac{1}{\rho^2}\right) + O\left(\frac{1}{n}\right), & c < 1 \\ \frac{1}{c}\|\boldsymbol{\beta}_*\|^2 - \frac{\eta^2}{\eta^2+\rho^2}\left(\frac{\|\boldsymbol{\beta}_*\|^2}{d} + \frac{1}{c}(\boldsymbol{\beta}_*^\top\boldsymbol{u})^2\right) + o(1) + O\left(\frac{1}{n}\right), & c > 1 \end{cases}.$$

Thus for $c < 1$, the entire interaction term now becomes

$$-\frac{2\tilde{\alpha}_A\tilde{\rho}^2}{d}\left(\alpha_A\|\boldsymbol{\beta}_*\|^2 + (\alpha_Z - \alpha_A)(\boldsymbol{\beta}_*^\top\boldsymbol{u})^2\frac{\eta^2 c}{\rho^2+\eta^2 c} + o(1)\right).$$

For $c > 1$, instead we have

$$-\frac{2\tilde{\alpha}_A\tilde{\rho}^2}{d}\left(\frac{\alpha_A}{c}\|\boldsymbol{\beta}_*\|^2 - \frac{\alpha_A}{d}\frac{\eta^2}{\eta^2+\rho^2}\|\boldsymbol{\beta}_*\|^2 + \left(\alpha_Z - \frac{\alpha_A}{c}\right)\frac{\eta^2}{\eta^2+\rho^2}(\boldsymbol{\beta}_*^\top\boldsymbol{u})^2 + o(1)\right).$$

### E.5.5   BIAS: HELPER LEMMAS

**Lemma 22.** *In the same setting as Section 2, we have that for $c < 1$,*

$$\mathbb{E}\left[\left(\tilde{\alpha}_Z - \alpha_Z + \frac{\xi}{\gamma_1}(\alpha_Z - \alpha_A)\right)^2\tilde{\eta}^2(\boldsymbol{\beta}_*^\top\boldsymbol{u})^2\right]$$
$$= \tilde{\eta}^2(\boldsymbol{\beta}_*^\top\boldsymbol{u})^2\left[(\tilde{\alpha}_Z - \alpha_Z) + \frac{\rho^2}{\eta^2 c+\rho^2}(\alpha_Z - \alpha_A)\right]^2 + o(1) + O\left(\frac{\eta}{n}\right).$$

*For $c > 1$,*

$$\mathbb{E}\left[\left\|\boldsymbol{\beta}_*^\top\left[(\tilde{\alpha}_Z - \alpha_Z)\boldsymbol{I} + \frac{\xi}{\gamma_2}(\alpha_Z\boldsymbol{I} - \alpha_A\boldsymbol{A}\boldsymbol{A}^\dagger)\right]\tilde{\boldsymbol{Z}}\right\|^2\right]$$
$$= \tilde{\eta}^2(\boldsymbol{\beta}_*^\top\boldsymbol{u})^2\left[(\tilde{\alpha}_Z - \alpha_Z) + \frac{\rho^2}{\eta^2+\rho^2}\left(\alpha_Z - \frac{\alpha_A}{c}\right)\right]^2 + o(1) + O\left(\frac{\eta}{n}\right).$$

*Proof.* For $c < 1$, we first expand the square and get:

$$\left(\tilde{\alpha}_Z - \alpha_Z + \frac{\xi}{\gamma_1}(\alpha_Z - \alpha_A)\right)^2 = (\tilde{\alpha}_Z - \alpha_Z)^2 + \frac{1}{\eta^2}\frac{\eta^2\xi^2}{\gamma_1^2}(\alpha_Z - \alpha_A)^2 + \frac{2}{\eta}\frac{\eta\xi}{\gamma_1}(\alpha_Z - \alpha_A)(\tilde{\alpha}_Z - \alpha_Z).$$

By Lemmas 9 and 20, then we see that, using the square root of the covariance to bound the difference between the expectation of the product and the product of the expectation.

$$\mathbb{E}\left[\frac{\eta^2\xi^2}{\gamma_1^2}\right] = \mathbb{E}\left[\frac{\eta^4}{\gamma_1^2}\right]\mathbb{E}\left[\frac{\xi^2}{\eta^2}\right] + \sqrt{\mathrm{Var}\left(\frac{\eta^4}{\gamma_1^2}\right)\mathrm{Var}\left(\frac{\xi^2}{\eta^2}\right)}$$

$$= \left(\frac{\rho^4\eta^4}{(\eta^2 c + \rho^2)^2} + o(1)\right)\left(\frac{1}{\eta^2} + O\left(\frac{1}{\rho^2 n}\right)\right) + O\left(\frac{1}{n}\right)$$

$$= \frac{\rho^4\eta^2}{(\eta^2 c + \rho^2)^2} + o\left(\frac{1}{\eta^2}\right) + O\left(\frac{1}{n}\right).$$

$$\mathbb{E}\left[\frac{\eta\xi}{\gamma_1}\right] = \mathbb{E}\left[\frac{\eta^2}{\gamma_1}\right]\mathbb{E}\left[\frac{\xi}{\eta}\right] + \sqrt{\mathrm{Var}\left(\frac{\eta^2}{\gamma_1}\right)\mathrm{Var}\left(\frac{\xi}{\eta}\right)}$$

$$= \left(\frac{\rho^2\eta^2}{\eta^2 c + \rho^2} + o(1)\right)\left(\frac{1}{\eta}\right) + O\left(\frac{1}{n}\right)$$

$$= \frac{\rho^2\eta}{\eta^2 c + \rho^2} + o\left(\frac{1}{\eta}\right) + O\left(\frac{1}{n}\right).$$

Combining these terms together, we have that

$$\mathbb{E}\left[\left(\tilde{\alpha}_Z - \alpha_Z + \frac{\xi}{\gamma_1}(\alpha_Z - \alpha_A)\right)^2 \tilde{\eta}^2(\boldsymbol{\beta}_*^\top \boldsymbol{u})^2\right]$$

$$= (\boldsymbol{\beta}_*^\top \boldsymbol{u})^2 \left[\tilde{\eta}^2(\tilde{\alpha}_Z - \alpha_Z)^2 + \frac{\tilde{\eta}^2}{\eta^2}\left(\frac{\rho^4\eta^2}{(\eta^2 c + \rho^2)^2} + o\left(\frac{1}{\eta^2}\right) + O\left(\frac{1}{n}\right)\right)(\alpha_Z - \alpha_A)^2\right.$$

$$\left. + \frac{2\tilde{\eta}^2}{\eta}\left(\frac{\rho^2\eta}{\eta^2 c + \rho^2} + o\left(\frac{1}{\eta}\right) + O\left(\frac{1}{n}\right)\right)(\alpha_Z - \alpha_A)(\tilde{\alpha}_Z - \alpha_Z)\right]$$

$$= \tilde{\eta}^2(\boldsymbol{\beta}_*^\top \boldsymbol{u})^2\left(\left[(\tilde{\alpha}_Z - \alpha_Z) + \frac{\rho^2}{\eta^2 c + \rho^2}(\alpha_Z - \alpha_A)\right]^2 + o\left(\frac{1}{\eta^2}\right) + O\left(\frac{1}{\eta n}\right)\right).$$

We now consider $c > 1$. Recalling that $\tilde{\boldsymbol{Z}} = \tilde{\eta}\boldsymbol{u}\tilde{\boldsymbol{v}}^\top$, we let $c_1 = \tilde{\alpha}_Z - \alpha_Z$ and expand:

$$\left\|\boldsymbol{\beta}_*^\top\left[(\tilde{\alpha}_Z - \alpha_Z)\boldsymbol{I} + \frac{\xi}{\gamma_2}(\alpha_Z\boldsymbol{I} - \alpha_A\boldsymbol{A}\boldsymbol{A}^\dagger)\right]\tilde{\boldsymbol{Z}}\right\|^2$$

$$= \boldsymbol{\beta}_*^\top\left[(\tilde{\alpha}_Z - \alpha_Z)\boldsymbol{I} + \frac{\xi}{\gamma_2}(\alpha_Z\boldsymbol{I} - \alpha_A\boldsymbol{A}\boldsymbol{A}^\dagger)\right]\tilde{\boldsymbol{Z}}\tilde{\boldsymbol{Z}}^\top\left[(\tilde{\alpha}_Z - \alpha_Z)\boldsymbol{I} + \frac{\xi}{\gamma_2}(\alpha_Z\boldsymbol{I} - \alpha_A\boldsymbol{A}\boldsymbol{A}^\dagger)\right]^\top\boldsymbol{\beta}_*$$

$$= \tilde{\eta}^2\boldsymbol{\beta}_*^\top\left[(\tilde{\alpha}_Z - \alpha_Z)\boldsymbol{I} + \frac{\xi}{\gamma_2}(\alpha_Z\boldsymbol{I} - \alpha_A\boldsymbol{A}\boldsymbol{A}^\dagger)\right]\boldsymbol{u}\boldsymbol{u}^\top\left[(\tilde{\alpha}_Z - \alpha_Z)\boldsymbol{I} + \frac{\xi}{\gamma_2}(\alpha_Z\boldsymbol{I} - \alpha_A\boldsymbol{A}\boldsymbol{A}^\dagger)\right]^\top\boldsymbol{\beta}_*$$

$$= c_1^2\tilde{\eta}^2(\boldsymbol{\beta}_*^\top\boldsymbol{u})^2 + \tilde{\eta}^2\frac{\xi^2}{\gamma_2^2}\boldsymbol{\beta}_*^\top(\alpha_Z\boldsymbol{I} - \alpha_A\boldsymbol{A}\boldsymbol{A}^\dagger)\boldsymbol{u}\boldsymbol{u}^\top((\alpha_Z\boldsymbol{I} - \alpha_A\boldsymbol{A}\boldsymbol{A}^\dagger)^\top\boldsymbol{\beta}_* + 2c_1\tilde{\eta}^2\frac{\xi}{\gamma_2}\boldsymbol{\beta}_*^\top(\alpha_Z\boldsymbol{I} - \alpha_A\boldsymbol{A}\boldsymbol{A}^\dagger)\boldsymbol{u}\boldsymbol{u}^\top\boldsymbol{\beta}_*.$$

Not that for the second and third terms, we have that $\xi, \gamma_2$ only depend on the singular values of $\boldsymbol{A}$ and the rest only depend on the singular vectors. Hence, these terms are independent.

First note that when $d > n$, the number of singular values equals $n$, which is less than the dimension $d$. As a result,

$$\boldsymbol{A}\boldsymbol{A}^\dagger = \boldsymbol{U}\boldsymbol{\Sigma}\boldsymbol{V}^\top\boldsymbol{V}\boldsymbol{\Sigma}^\dagger\boldsymbol{U}^\top = \boldsymbol{U}\begin{bmatrix}\boldsymbol{I}_{n\times n} & \boldsymbol{0}_{n\times(d-n)} \\ \boldsymbol{0}_{(d-n)\times n} & \boldsymbol{0}_{(d-n)\times(d-n)}\end{bmatrix}\boldsymbol{U}^\top.$$

Then we have that

$$\mathbb{E}\left[\boldsymbol{\beta}_*^\top\boldsymbol{A}\boldsymbol{A}^\dagger\boldsymbol{\beta}_*^\top\right] = \sum_{i=1}^n \mathbb{E}\left[(\boldsymbol{\beta}_*^\top\boldsymbol{U})_i^2\right] = \frac{n}{d}\|\boldsymbol{\beta}_*\|^2 = \frac{1}{c}\|\boldsymbol{\beta}_*\|^2, \tag{13}$$

since $\boldsymbol{\beta}_*^\top \boldsymbol{U}$ is a uniformly random vector of length $\|\boldsymbol{\beta}_*\|$ in $\mathbb{R}^d$ after the rotation $\boldsymbol{U}$.

For the middle term, by Proposition 2 and the above Equation 13, we have

$$
\begin{aligned}
&\mathbb{E}\left[\boldsymbol{\beta}_*^\top(\alpha_Z\boldsymbol{I} - \alpha_A\boldsymbol{A}\boldsymbol{A}^\dagger)\boldsymbol{u}\boldsymbol{u}^\top((\alpha_Z\boldsymbol{I} - \alpha_A\boldsymbol{A}\boldsymbol{A}^\dagger)\boldsymbol{\beta}_*\right] \\
&= \alpha_Z^2(\boldsymbol{\beta}_*^\top\boldsymbol{u})^2 - 2\alpha_A\alpha_Z\mathbb{E}\left[\boldsymbol{\beta}_*^\top\boldsymbol{A}\boldsymbol{A}^\dagger\boldsymbol{u}\boldsymbol{u}^\top\boldsymbol{\beta}_*\right] + \alpha_A^2\mathbb{E}\left[(\boldsymbol{\beta}_*^\top\boldsymbol{A}\boldsymbol{A}^\dagger\boldsymbol{u})^2\right] \\
&= \left(\alpha_Z - \frac{\alpha_A}{c}\right)^2(\boldsymbol{\beta}_*^\top\boldsymbol{u})^2 + o(1).
\end{aligned}
$$

Similarly, for the last term, we have

$$
\mathbb{E}\left[\boldsymbol{\beta}_*^\top(\alpha_Z\boldsymbol{I} - \alpha_A\boldsymbol{A}\boldsymbol{A}^\dagger)\boldsymbol{u}\boldsymbol{u}^\top\boldsymbol{\beta}_*\right] = \left(\alpha_Z - \frac{\alpha_A}{c}\right)(\boldsymbol{\beta}_*^\top\boldsymbol{u})^2 + o(1).
$$

Thus putting these expectations together, we get

$$
\mathbb{E}\left[\tilde{\eta}^2(\boldsymbol{\beta}_*^\top\boldsymbol{u})^2\left[c_1^2 + \frac{\xi^2}{\gamma_2^2}\left(\alpha_Z - \frac{\alpha_A}{c}\right)^2 + 2c_1\frac{\xi}{\gamma_2}\left(\alpha_Z - \frac{\alpha_A}{c}\right)\right]\right] = \mathbb{E}\left[\tilde{\eta}^2(\boldsymbol{\beta}_*^\top\boldsymbol{u})^2\left[c_1 + \frac{\xi}{\gamma_2}\left(\alpha_Z - \frac{\alpha_A}{c}\right)\right]^2\right].
$$

Similar to the $c < 1$ case, we take the expectation for terms involving $\frac{\xi}{\gamma_2}$ and get:

$$
\tilde{\eta}^2(\boldsymbol{\beta}_*^\top\boldsymbol{u})^2\left[\left((\tilde{\alpha}_Z - \alpha_Z) + \frac{\rho^2}{\eta^2 + \rho^2}\left(\alpha_Z - \frac{\alpha_A}{c}\right)\right)^2 + o\left(\frac{1}{\eta^2}\right) + O\left(\frac{1}{\eta n}\right)\right].
$$

$\square$

**Lemma 23** (Expectations involving $p_1$ and $p_2$). *In the setting of Section 2, we have that*

1. *For $c = d/n < 1$:*

$$
\mathbb{E}\left[\frac{\xi^2}{\gamma_1^2}\|\boldsymbol{p}_1\|^2\right] = \frac{c}{1-c}\frac{\eta^2}{\eta^2 c + \rho^2} + o(1) + O\left(\frac{1}{\rho^2 n}\right).
$$

2. *For $c = d/n > 1$:*

$$
\mathbb{E}\left[\frac{\xi^2}{\gamma_2^2}\|\boldsymbol{p}_2\|^2\right] = \frac{\eta^2}{c-1}\frac{\eta^2 c + \rho^2}{(\eta^2 + \rho^2)^2} + o(1) + O\left(\frac{1}{\rho^2 n}\right).
$$

*Proof.* First, Lemma 6 tells us that

$$
\frac{\xi^2}{\gamma_1^2}\|\boldsymbol{p}_1\|^2 = \frac{\eta^2\|\boldsymbol{k}\|^2}{\gamma_1}.
$$

Then recall from Lemma 9 that

$$
\mathbb{E}[\|\boldsymbol{k}\|^2] = \frac{1}{\rho^2}\frac{c}{1-c} + o\left(\frac{1}{\rho^2}\right) \quad \text{and} \quad \text{Var}(\|\boldsymbol{k}\|^2) = O\left(\frac{1}{\rho^4 n}\right)
$$

and Lemma 19 tells us

$$
\mathbb{E}\left[\frac{\eta^2}{\gamma_1}\right] = \frac{\rho^2\eta^2}{\eta^2 c + \rho^2} + o\left(\frac{1}{\rho^2}\right) \quad \text{and} \quad \text{Var}\left(\frac{\eta^2}{\gamma_i}\right) = O\left(\frac{1}{n}\right)
$$

Again Section E.4.2 tells us that the assumption of Lemma 37 is satisfied and that

$$
\begin{aligned}
\mathbb{E}\left[\frac{\xi^2}{\gamma_1^2}\|\boldsymbol{p}_1\|^2\right] = \mathbb{E}\left[\frac{\eta^2\|\boldsymbol{k}\|^2}{\gamma_1}\right] &= \mathbb{E}\left[\frac{\eta^2}{\gamma_1}\right]\mathbb{E}\left[\|\boldsymbol{k}\|^2\right] + \sqrt{\text{Var}\left(\frac{\eta^2}{\gamma_1}\right)\text{Var}\left(\|\boldsymbol{k}\|^2\right)} \\
&= \left(\frac{\rho^2\eta^2}{\eta^2 c + \rho^2} + o\left(\frac{1}{\rho^2}\right)\right)\left(\frac{1}{\rho^2}\frac{c}{1-c} + o\left(\frac{1}{\rho^2}\right)\right) + O\left(\frac{1}{\rho^2 n}\right) \\
&= \frac{c}{1-c}\frac{\eta^2}{\eta^2 c + \rho^2} + o(1) + O\left(\frac{1}{\rho^2 n}\right).
\end{aligned}
$$

Using Lemma 6 for $p_2$,

$$\frac{\xi^2}{\gamma_2^2}\|p_2\|^2 = \frac{1}{\gamma_2^2}\left(\eta^4\|s\|^4 hA^{\dagger\top}A^\dagger h^\top + 2\eta^3\xi\|s\|^2 k^\top A^\dagger h^\top + \eta^2\xi^2\|k\|^2\right).$$

To begin, we start estimating

$$\mathbb{E}\left[\frac{\eta^4\|s\|^4}{\gamma_2^2}hA^{\dagger\top}A^\dagger h^\top\right].$$

Using our Spherical Hypercontractivity, we have that $\|s\|^2$ and $hA^{\dagger\top}A^\dagger h^\top$ satisfy the assumptions for Lemma 36. Then using Lemmas 9 and 10 we first have that

$$\mathbb{E}\left[\|s\|^2\right] = 1 - \frac{1}{c} \quad \text{and} \quad \text{Var}\left(\|s\|^2\right) = O\left(\frac{1}{d}\right)$$

$$\mathbb{E}\left[hA^{\dagger\top}A^\dagger h^\top\right] = \frac{1}{\rho^4}\frac{c^3}{(c-1)^3} + o\left(\frac{1}{\rho^4}\right) \quad \text{and} \quad \text{Var}\left(\beta_*^\top h^\top u^\top \beta_*\right) = O\left(\frac{1}{\rho^8 d}\right).$$

Thus, using Lemma 37, we have that

$$\mathbb{E}\left[\|s\|^4 hA^{\dagger\top}A^\dagger h^\top\right] = \left(\mathbb{E}\left[\|s\|^2\right]\right)^2 \mathbb{E}\left[hA^{\dagger\top}A^\dagger h^\top\right] + O\left(\max\left(\frac{1}{d}, \frac{1}{\rho^8 d}\right)\right)$$

$$= \left(1 - \frac{1}{c}\right)^2 \left(\frac{1}{\rho^4}\frac{c^3}{(c-1)^3} + o\left(\frac{1}{\rho^4}\right)\right) + O\left(\frac{1}{n}\right)$$

$$= \frac{1}{\rho^4}\frac{c}{c-1} + o\left(\frac{1}{\rho^4}\right) + O\left(\frac{1}{n}\right).$$

and using Lemma 36, since all the means are $O(1)$, we have that

$$\text{Var}\left(\|s\|^4 hA^{\dagger\top}A^\dagger h^\top\right) = O\left(\max\left(\text{Var}\left(\|s\|^2\right), \text{Var}\left(hA^{\dagger\top}A^\dagger h^\top\right)\right)\right) = O\left(\frac{1}{n}\right).$$

Then Lemma 20 gives mean and variance of $\frac{\eta^4}{\gamma_i^2}$. Since $\frac{\eta^4}{\gamma_i^2}$ does not satisfy the higher moment bound, and cannot be directly included in the product, we can include it via the classical bound:

$$\mathbb{E}\left[\frac{\eta^4\|s\|^4}{\gamma_2^2}hA^{\dagger\top}A^\dagger h^\top\right] = \mathbb{E}\left[\frac{\eta^4}{\gamma_2^2}\right]\mathbb{E}\left[\|s\|^4 hA^{\dagger\top}A^\dagger h^\top\right] + \sqrt{\text{Var}\left(\|s\|^4 hA^{\dagger\top}A^\dagger h^\top\right)\text{Var}\left(\frac{\eta^4}{\gamma_2^2}\right)}$$

$$\text{(14)}$$

$$= \left(\frac{\rho^4\eta^4}{(\eta^2+\rho^2)^2} + o(1)\right)\left(\frac{1}{\rho^4}\frac{c}{c-1} + o\left(\frac{1}{\rho^4}\right)\right) + O\left(\frac{1}{n}\right) \qquad \text{(15)}$$

$$= \frac{c}{c-1}\frac{\eta^4}{(\eta^2+\rho^2)^2} + o(1) + O\left(\frac{1}{n}\right). \qquad \text{(16)}$$

Similarly, we can do the same thing for the other term. For the middle term we note that from Lemma 11

$$\mathbb{E}\left[k^\top A^\dagger h^\top\right] = 0 \quad \text{and} \quad \text{Var}\left(k^\top A^\dagger h^\top\right) = O\left(\frac{1}{\rho^6 d}\right)$$

and Lemma 9 tells us

$$\mathbb{E}\left[\|s\|^2\right] = 1 - \frac{1}{c} \quad \text{and} \quad \text{Var}\left(\|s\|^2\right) = O\left(\frac{1}{d}\right)$$

and

$$\mathbb{E}\left[\frac{\xi}{\eta}\right] = \frac{1}{\eta} \quad \text{and} \quad \text{Var}\left(\frac{\xi}{\eta}\right) = O\left(\frac{1}{\rho^2 n}\right)$$

Thus using Lemma 37, we have that

$$\mathbb{E}\left[\frac{\xi}{\eta}\|s\|^2 k^\top A^\dagger h^\top\right] = 0 + O\left(\frac{1}{d}\right)$$

Thus using the standard covariance bound for the expectation of product versus product of expectation, we have that

$$\mathbb{E}\left[\frac{\eta^3 \xi \|\boldsymbol{s}\|^2}{\gamma_2^2} \boldsymbol{k}^\top \boldsymbol{A}^\dagger \boldsymbol{h}^\top\right] = 0 + \sqrt{\mathrm{Var}\left(\frac{\eta^4}{\gamma_2^2}\right) O\left(\frac{1}{n}\right)} = O\left(\frac{1}{n}\right).$$

For the last term, we have that, using Lemma 37

$$\mathbb{E}\left[\frac{\xi^2}{\eta^2}\|\boldsymbol{k}\|^2\right] = \frac{1}{\eta^2} \cdot \left(\frac{1}{\rho^2}\frac{1}{c-1} + o\left(\frac{1}{\rho^2}\right)\right) + O\left(\frac{1}{\rho^4 n}\right)$$

$$= \frac{1}{\eta^2 \rho^2}\frac{1}{c-1} + o\left(\frac{1}{\eta^2 \rho^2}\right) + O\left(\frac{1}{\rho^4 n}\right)$$

and from Lemma 36

$$\mathrm{Var}\left(\frac{\xi^2}{\eta^2}\|\boldsymbol{k}\|^2\right) = O\left(\frac{1}{\rho^4 n}\right)$$

Then using the standard bound, we have that

$$\mathbb{E}\left[\frac{\eta^2 \xi^2 \|\boldsymbol{k}\|^2}{\gamma_2^2}\right] = \mathbb{E}\left[\frac{\eta^4}{\gamma_2^2}\right] \mathbb{E}\left[\frac{\xi^2}{\eta^2}\|\boldsymbol{k}\|^2\right] + \sqrt{\mathrm{Var}\left(\frac{\eta^4}{\gamma_2^2}\right) O\left(\frac{1}{\rho^4 n}\right)}$$

$$= \left(\frac{\rho^4 \eta^4}{(\eta^2+\rho^2)^2} + o(1)\right)\left(\frac{1}{\eta^2 \rho^2}\frac{1}{c-1} + o\left(\frac{1}{\eta^2 \rho^2}\right) + O\left(\frac{1}{\rho^4 n}\right)\right) + O\left(\frac{1}{\rho^2 n}\right)$$

$$= \frac{1}{c-1}\frac{\eta^2 \rho^2}{(\eta^2+\rho^2)^2} + o\left(\frac{1}{\eta^2 \rho^2}\right) + O\left(\frac{1}{\rho^2 n}\right).$$

Finally, putting all three terms together we get

$$\mathbb{E}\left[\frac{\xi^2}{\gamma_2^2}\|\boldsymbol{p}_2\|^2\right] = \frac{c}{c-1}\frac{\eta^4}{(\eta^2+\rho^2)^2} + o(1) + \frac{1}{c-1}\frac{\eta^2 \rho^2}{(\eta^2+\rho^2)^2} + o\left(\frac{1}{\rho^2 \eta^2}\right) + O\left(\frac{1}{\rho^2 n}\right)$$

$$= \frac{\eta^2}{c-1}\frac{\eta^2 c + \rho^2}{(\eta^2+\rho^2)^2} + o(1) + O\left(\frac{1}{\rho^2 n}\right).$$

$\square$

From the above proofs, we make an important observation that the individual terms from Lemmas 9, 10, 11, 16 all have means $O(1)$ and variances $O(1/n)$. Hence, by Lemma 36, we can bound the variance of a product of terms by $O(1/n)$, given that these terms satisfy the lemma assumptions. Essentially, only $\eta^2/\gamma_i$ and $\eta^4/\gamma_i^2$ fail the assumption on higher moment bound, so we deal with them via the classical bound after carrying out the product. This simplification ensures proper concentration and will be used at times in the following proofs without reference.

### E.5.6 VARIANCE: HELPER LEMMAS

**Lemma 24.** *In the setting of Section 2, we have that for $c > 1$:*

$$\mathbb{E}\left[\left\|\alpha_A \frac{\eta\|\boldsymbol{s}\|^2}{\gamma_2}\boldsymbol{\beta}_*^\top \boldsymbol{h}^\top \boldsymbol{u}^\top \tilde{\boldsymbol{Z}}\right\|^2\right] = \tilde{\eta}^2 \alpha_A^2 \frac{\|\beta_*\|^2}{d}\left(\frac{c-1}{c}\right)\frac{\eta^2 \rho^2}{(\eta^2+\rho^2)^2} + O\left(\frac{1}{n}\right).$$

*Proof.* Since $\tilde{\boldsymbol{Z}} = \tilde{\eta}\boldsymbol{u}\tilde{\boldsymbol{v}}^\top$, we have that

$$\left\|\alpha_A \frac{\eta\|\boldsymbol{s}\|^2}{\gamma_2}\boldsymbol{\beta}_*^\top \boldsymbol{h}^\top \boldsymbol{u}^\top \tilde{\boldsymbol{Z}}\right\|^2 = \tilde{\eta}^2 \alpha_A^2 \frac{\eta^2 \|\boldsymbol{s}\|^4}{\gamma_2^2}\boldsymbol{\beta}_*^\top \boldsymbol{h}^\top \boldsymbol{h}\boldsymbol{\beta}_* = \alpha_A^2 \frac{\tilde{\eta}^2}{\eta^2}\frac{\eta^4\|\boldsymbol{s}\|^4}{\gamma_2^2}\boldsymbol{\beta}_*^\top \boldsymbol{h}^\top \boldsymbol{h}\boldsymbol{\beta}_*.$$

Similar to last lemma, using Lemmas 37, 9, 10, 20, we get

$$\mathbb{E}\left[\frac{\eta^4\|\boldsymbol{s}\|^4}{\gamma_2^2}\boldsymbol{\beta}_*^\top \boldsymbol{h}^\top \boldsymbol{h}\boldsymbol{\beta}_*\right] = \mathbb{E}\left[\frac{\eta^4}{\gamma_2^2}\right]\left(\mathbb{E}\left[\|\boldsymbol{s}\|^2\right]\right)^2 \mathbb{E}\left[\boldsymbol{\beta}_*^\top \boldsymbol{h}^\top \boldsymbol{h}\boldsymbol{\beta}_*\right] + \sqrt{\mathrm{Var}\left(\frac{\eta^4}{\gamma_2^2}\right) O\left(\frac{1}{n}\right)}$$

$$= \left(\frac{\rho^4 \eta^4}{(\rho^2+\eta^2)^2} + o(1)\right)\left(1 - \frac{1}{c}\right)^2\left(\frac{\|\boldsymbol{\beta}_*\|^2}{d}\frac{c}{\rho^2(c-1)} + o\left(\frac{1}{\rho^2 d}\right)\right) + O\left(\frac{1}{n}\right)$$

$$= \frac{\|\boldsymbol{\beta}_*\|^2}{d}\left(\frac{c-1}{c}\right)\frac{\eta^4 \rho^2}{(\eta^2+\rho^2)^2} + O\left(\frac{1}{n}\right).$$

Hence, it directly follows from here that

$$\mathbb{E}\left[\left\|\alpha_A \frac{\eta\|s\|^2}{\gamma_2}\beta_*^\top h^\top u^\top \tilde{Z}\right\|^2\right] = \alpha_A^2\frac{\tilde{\eta}^2}{\eta^2}\mathbb{E}\left[\frac{\eta^4\|s\|^4}{\gamma_2^2}\beta_*^\top h^\top h\beta_*\right]$$

$$= \tilde{\eta}^2\alpha_A^2\frac{\|\beta_*\|^2}{d}\left(\frac{c-1}{c}\right)\frac{\eta^2\rho^2}{(\eta^2+\rho^2)^2} + O\left(\frac{1}{n}\right).$$

$\square$

**Lemma 25.** *In the setting of Section 2, we have that for $c > 1$:*

$$\mathbb{E}\left[\frac{\eta\|s\|^2}{\gamma_2}\beta_*^\top\left[(\tilde{\alpha}_Z - \alpha_Z)I + \frac{\xi}{\gamma_2}(\alpha_Z I - \alpha_A AA^\dagger)\right]\tilde{Z}\tilde{Z}^\top uh\beta_*\right] = O\left(\frac{\eta}{n}\right).$$

*Proof.* Using $\tilde{Z} = \tilde{\eta}u\tilde{v}^\top$, we can expand this into three terms. We can take expectations in a similar way via Lemmas 37, 9, 10, 11: Let $c_1 = \tilde{\alpha}_Z - \alpha_Z$. Each term contains a zero expectation:

$$\mathbb{E}\left[\tilde{\eta}^2 c_1\frac{\eta\|s\|^2}{\gamma_2}\beta_*^\top uh\beta_*\right] = \frac{\tilde{\eta}^2}{\eta}c_1\left(\mathbb{E}\left[\frac{\eta^2}{\gamma_2}\right]\mathbb{E}\left[\|s\|^2\right]\mathbb{E}\left[\beta_*^\top uh\beta_*\right] + \sqrt{\mathrm{Var}\left(\frac{\eta^2}{\gamma_2}\right)O\left(\frac{1}{n}\right)}\right)$$

$$= \frac{\tilde{\eta}^2}{\eta}c_1\left(\sqrt{\mathrm{Var}\left(\frac{\eta^2}{\gamma_2}\right)O\left(\frac{1}{n}\right)}\right) = O\left(\frac{\eta}{n}\right).$$

$$\mathbb{E}\left[\tilde{\eta}^2\alpha_Z\frac{\eta\xi\|s\|^2}{\gamma_2^2}\beta_*^\top uh\beta_*\right] = \frac{\alpha_Z\tilde{\eta}^2}{\eta^2}\left(\mathbb{E}\left[\frac{\eta^4}{\gamma_2^2}\right]\mathbb{E}\left[\frac{\xi}{\eta}\right]\mathbb{E}\left[\|s\|^2\right]\mathbb{E}\left[\beta_*^\top uh\beta_*\right] + \sqrt{\mathrm{Var}\left(\frac{\eta^4}{\gamma_2^2}\right)O\left(\frac{1}{n}\right)}\right)$$

$$= \frac{\alpha_Z\tilde{\eta}^2}{\eta^2}\left(\sqrt{\mathrm{Var}\left(\frac{\eta^4}{\gamma_2^2}\right)O\left(\frac{1}{n}\right)}\right) = O\left(\frac{1}{n}\right).$$

$$\mathbb{E}\left[\tilde{\eta}^2\alpha_A\frac{\eta\xi\|s\|^2}{\gamma_2^2}\beta_*^\top AA^\dagger uh\beta_*\right] == \frac{\alpha_Z\tilde{\eta}^2}{\eta^2}\left(\mathbb{E}\left[\frac{\eta^4}{\gamma_2^2}\right]\mathbb{E}\left[\frac{\xi}{\eta}\right]\mathbb{E}\left[\|s\|^2\right]\mathbb{E}\left[\beta_*^\top AA^\dagger uh\beta_*\right] + \sqrt{\mathrm{Var}\left(\frac{\eta^4}{\gamma_2^2}\right)O\left(\frac{1}{n}\right)}\right)$$

$$= \frac{\alpha_Z\tilde{\eta}^2}{\eta^2}\left(\sqrt{\mathrm{Var}\left(\frac{\eta^4}{\gamma_2^2}\right)O\left(\frac{1}{n}\right)}\right) = O\left(\frac{1}{n}\right).$$

Thus the cross term concentrates around zero at a rate of $O(\eta/n)$. $\square$

**Lemma 26.** *In the same setting as Section 2, we have that*

$$\mathbb{E}\left[\beta_*^\top Z(Z+A)^\dagger(Z+A)^{\dagger\top}Z\beta_*\right] = \begin{cases} \frac{\eta^2(\eta^2+\rho^2)}{(\eta^2 c+\rho^2)^2}\frac{c^2}{1-c}(\beta_*^\top u)^2 + o(1) + O\left(\frac{1}{n}\right) & c < 1 \\ \frac{\eta^2}{\eta^2+\rho^2}\frac{c}{c-1}(\beta_*^\top u)^2 + o\left(\frac{1}{\rho^2}\right) + O\left(\frac{1}{\rho^2 n}\right) & c > 1 \end{cases}.$$

*Proof.* We start with $c < 1$ and expand this term using Proposition 1:

$$\beta_*^\top Z(Z+A)^\dagger(Z+A)^{\dagger\top}Z\beta_* = \frac{\eta^2\|h\|^2\xi^2}{\gamma_1^2}(\beta_*^\top u)^2 + \frac{\eta^4\|t\|^4}{\gamma_1^2}(k^\top A^\dagger A^{\dagger\top}k)(\beta_*^\top u)^2 + \frac{2\eta^3\|t\|^2\xi}{\gamma_1^2}k^\top A^\dagger h^\top(\beta_*^\top u)^2.$$

We then start plugging in the expectations of these three terms and the "cumulative" variance of the sum according to Lemma 37.

$$\mathbb{E}\left[\frac{\eta^2\|h\|^2\xi^2}{\gamma_1^2}(\beta_*^\top u)^2\right] = (\beta_*^\top u)^2\mathbb{E}\left[\frac{\eta^4}{\gamma_1^2}\right]\mathbb{E}\left[\frac{\xi^2}{\eta^2}\right]\mathbb{E}\left[\|h\|^2\right] + \sqrt{\mathrm{Var}\left(\frac{\eta^4}{\gamma_1^2}\right)O\left(\frac{1}{n}\right)}$$

$$= (\beta_*^\top u)^2\left(\frac{\rho^4\eta^4}{(\eta^2 c+\rho^2)^2} + o(1)\right)\left(\frac{1}{\eta^2} + O\left(\frac{1}{\rho^2 n}\right)\right)\left(\frac{1}{\rho^2}\frac{c^2}{1-c} + o\left(\frac{1}{\rho^2}\right)\right) + O\left(\frac{1}{n}\right)$$

$$= \frac{\eta^2\rho^2}{(\eta^2 c+\rho^2)^2}\frac{c^2}{1-c}(\beta_*^\top u)^2 + o(1) + O\left(\frac{1}{n}\right).$$

$$\mathbb{E}\left[\frac{\eta^4\|\boldsymbol{t}\|^4}{\gamma_1^2}(\boldsymbol{k}^\top \boldsymbol{A}^\dagger \boldsymbol{A}^{\dagger\top}\boldsymbol{k})(\boldsymbol{\beta}_*^\top \boldsymbol{u})^2\right] = (\boldsymbol{\beta}_*^\top \boldsymbol{u})^2 \mathbb{E}\left[\frac{\eta^4}{\gamma_1^2}\right]\left(\mathbb{E}\left[\|\boldsymbol{t}\|^2\right]\right)^2 \mathbb{E}\left[\boldsymbol{k}^\top \boldsymbol{A}^\dagger \boldsymbol{A}^{\dagger\top}\boldsymbol{k}\right] + \sqrt{\mathrm{Var}\left(\frac{\eta^4}{\gamma_1^2}\right)}O\left(\frac{1}{n}\right)$$

$$= (\boldsymbol{\beta}_*^\top \boldsymbol{u})^2\left(\frac{\rho^4\eta^4}{(\eta^2 c + \rho^2)^2} + o(1)\right)(1-c)^2\left(\frac{1}{\rho^4}\frac{c^2}{(1-c)^3} + o\left(\frac{1}{\rho^4}\right)\right) + O\left(\frac{1}{n}\right)$$

$$= \frac{\eta^4}{(\eta^2 c + \rho^2)^2}\frac{c^2}{1-c}(\boldsymbol{\beta}_*^\top \boldsymbol{u})^2 + o(1) + O\left(\frac{1}{n}\right).$$

and

$$\mathbb{E}\left[\frac{\eta^3\|\boldsymbol{t}\|^2\xi}{\gamma_1^2}\boldsymbol{k}^\top \boldsymbol{A}^\dagger \boldsymbol{h}^\top(\boldsymbol{\beta}_*^\top \boldsymbol{u})^2\right] = (\boldsymbol{\beta}_*^\top \boldsymbol{u})^2\left(\mathbb{E}\left[\frac{\eta^2}{\gamma_1}\right]\right)^2\mathbb{E}\left[\frac{\xi}{\eta}\right]\mathbb{E}\left[\|\boldsymbol{t}\|^2\right]\mathbb{E}\left[\boldsymbol{k}^\top \boldsymbol{A}^\dagger \boldsymbol{h}^\top\right] + O\left(\frac{1}{n}\right) = O\left(\frac{1}{n}\right).$$

Now we have the expectations and errors for the three terms. Combining them yields the Lemma statement.

For $c > 1$, we recall that $\boldsymbol{hs} = \boldsymbol{0}$, and Proposition 1 implies

$$\boldsymbol{\beta}_*^\top \boldsymbol{Z}(\boldsymbol{Z}+\boldsymbol{A})^\dagger(\boldsymbol{Z}+\boldsymbol{A})^{\dagger\top}\boldsymbol{Z}\boldsymbol{\beta}_* = \frac{\eta^2\|\boldsymbol{h}\|^2\xi^2}{\gamma_2^2}(\boldsymbol{\beta}_*^\top \boldsymbol{u})^2 + \frac{\eta^4\|\boldsymbol{h}\|^4\|\boldsymbol{s}\|^2}{\gamma_2^2}(\boldsymbol{\beta}_*^\top \boldsymbol{u})^2 + \frac{2\eta^3\|\boldsymbol{h}\|^2\xi}{\gamma_2^2}\boldsymbol{\beta}_*^\top \boldsymbol{uhsu}^\top \boldsymbol{\beta}_*$$

$$= \left(\frac{\eta^2\|\boldsymbol{h}\|^2(\xi^2 + \eta^2\|\boldsymbol{h}\|^2\|\boldsymbol{s}\|^2)}{\gamma_2^2}\right)(\boldsymbol{\beta}_*^\top \boldsymbol{u})^2$$

$$= \left(\frac{\eta^2\|\boldsymbol{h}\|^2\gamma_2}{\gamma_2^2}\right)(\boldsymbol{\beta}_*^\top \boldsymbol{u})^2$$

$$= \frac{\eta^2\|\boldsymbol{h}\|^2}{\gamma_2}(\boldsymbol{\beta}_*^\top \boldsymbol{u})^2.$$

Hence, we can take expectation:

$$\mathbb{E}[\boldsymbol{\beta}_*^\top \boldsymbol{Z}(\boldsymbol{Z}+\boldsymbol{A})^\dagger(\boldsymbol{Z}+\boldsymbol{A})^{\dagger\top}\boldsymbol{Z}\boldsymbol{\beta}_*] = \mathbb{E}\left[\frac{\eta^2}{\gamma_2}\right]\mathbb{E}\left[\|\boldsymbol{h}\|^2\right](\boldsymbol{\beta}_*^\top \boldsymbol{u})^2 + O\left(\frac{1}{n}\right)$$

$$= \frac{\eta^2}{\eta^2 + \rho^2}\frac{c}{c-1}(\boldsymbol{\beta}_*^\top \boldsymbol{u})^2 + o(1) + O\left(\frac{1}{n}\right).$$

$\square$

**Lemma 27.** *In the same setting as Section 2, we have that,*

$$\mathbb{E}\left[\boldsymbol{\beta}_*^\top \boldsymbol{A}(\boldsymbol{Z}+\boldsymbol{A})^\dagger(\boldsymbol{Z}+\boldsymbol{A})^{\dagger\top}\boldsymbol{A}\boldsymbol{\beta}_*\right] = \begin{cases} \|\boldsymbol{\beta}_*\|^2 + \frac{\eta^2(\eta^2+\rho^2)}{(\eta^2 c+\rho^2)^2}\frac{c^2}{1-c}(\boldsymbol{\beta}_*^\top \boldsymbol{u})^2 - \frac{2\eta^2 c}{\eta^2 c+\rho^2}(\boldsymbol{\beta}_*^\top \boldsymbol{u})^2 + o(1) + O\left(\frac{1}{n}\right) & c < 1 \\ \frac{\|\boldsymbol{\beta}_*\|^2}{c} - \frac{\eta^2}{\eta^2+\rho^2}\left(\frac{\|\boldsymbol{\beta}_*\|^2}{d} - \frac{(\boldsymbol{\beta}_*^\top \boldsymbol{u})^2}{c(c-1)}\right) + o(1) + O\left(\frac{1}{n}\right) & c > 1 \end{cases}.$$

*Proof.* We use similar expansions that follow from Lemma 2.

$$\boldsymbol{\beta}_*^\top \boldsymbol{A}(\boldsymbol{Z}+\boldsymbol{A})^\dagger(\boldsymbol{Z}+\boldsymbol{A})^{\dagger\top}\boldsymbol{A}\boldsymbol{\beta}_* = \|\boldsymbol{\beta}_*\|^2 + \frac{\eta^2\|\boldsymbol{h}\|^2\xi^2}{\gamma_1^2}(\boldsymbol{\beta}_*^\top \boldsymbol{u})^2 + \frac{\eta^4\|\boldsymbol{t}\|^4}{\gamma_1^2}(\boldsymbol{k}^\top \boldsymbol{A}^\dagger \boldsymbol{A}^{\dagger\top}\boldsymbol{k})(\boldsymbol{\beta}_*^\top \boldsymbol{u})^2$$

$$+ \frac{2\eta^3\|\boldsymbol{t}\|^2\xi}{\gamma_1^2}(\boldsymbol{\beta}_*^\top \boldsymbol{u})^2\boldsymbol{k}^\top \boldsymbol{A}^\dagger \boldsymbol{h}^\top - \frac{2\eta^2\|\boldsymbol{t}\|^2}{\gamma_1}\boldsymbol{\beta}_*^\top \boldsymbol{uk}^\top \boldsymbol{A}^\dagger \boldsymbol{\beta}_* - \frac{2\eta\xi}{\gamma_1}\boldsymbol{\beta}_*^\top \boldsymbol{uh}\boldsymbol{\beta}_*.$$

Lemma 26 gives the expectation of the first four terms:

$$\|\boldsymbol{\beta}_*^2\|^2 + \frac{\eta^2(\eta^2+\rho^2)}{(\eta^2 c+\rho^2)^2}\frac{c^2}{1-c}(\boldsymbol{\beta}_*^\top \boldsymbol{u})^2 + o(1) + O\left(\frac{1}{n}\right).$$

We have done the following expectations in Equations 18, 19:

$$\mathbb{E}\left[\frac{\eta\xi}{\gamma_1}\boldsymbol{\beta}_*^\top \boldsymbol{uh}\boldsymbol{\beta}_*\right] = O\left(\frac{1}{n}\right), \quad \mathbb{E}\left[\frac{\eta^2\|\boldsymbol{t}\|^2}{\gamma_1}\boldsymbol{\beta}_*^\top \boldsymbol{uk}^\top \boldsymbol{A}^\dagger \boldsymbol{\beta}_*\right] = \frac{\eta^2 c}{\eta^2 c+\rho^2} + o(1) + O\left(\frac{1}{n}\right).$$

Combining these results yields the lemma statement.

For $c > 1$, with $\boldsymbol{hs} = \boldsymbol{0}$, $\boldsymbol{s}^\top \boldsymbol{AA}^\dagger = \boldsymbol{0}$, $\boldsymbol{hAA}^\dagger = \boldsymbol{h}$, we have the following expansion by Lemma 2:

$$\boldsymbol{\beta}_*^\top \boldsymbol{A}(\boldsymbol{Z} + \boldsymbol{A})^\dagger(\boldsymbol{Z} + \boldsymbol{A})^{\dagger\top} \boldsymbol{A}\boldsymbol{\beta}_* = \boldsymbol{\beta}_*^\top \boldsymbol{AA}^\dagger \boldsymbol{\beta}_* + \frac{\eta^2 \|\boldsymbol{s}\|^2 \xi^2}{\gamma_2^2} \boldsymbol{\beta}_*^\top \boldsymbol{h}^\top \boldsymbol{h}\boldsymbol{\beta}_* + \frac{\eta^4 \|\boldsymbol{s}\|^4 \|\boldsymbol{h}\|^2}{\gamma_2^2} \boldsymbol{\beta}_*^\top \boldsymbol{h}^\top \boldsymbol{h}\boldsymbol{\beta}_*$$

$$+ \frac{\eta^4 \|\boldsymbol{h}\|^4 \|\boldsymbol{s}\|^2}{\gamma_2^2} \boldsymbol{\beta}_*^\top \boldsymbol{AA}^\dagger \boldsymbol{uu}^\top \boldsymbol{AA}^\dagger \boldsymbol{\beta}_* + \frac{\eta^2 \|\boldsymbol{h}\|^2 \xi^2}{\gamma_2^2} \boldsymbol{\beta}_*^\top \boldsymbol{AA}^\dagger \boldsymbol{uu}^\top \boldsymbol{AA}^\dagger \boldsymbol{\beta}_*$$

$$- \frac{2\eta^2 \|\boldsymbol{s}\|^2}{\gamma_2} \boldsymbol{\beta}_*^\top \boldsymbol{h}^\top \boldsymbol{h}\boldsymbol{\beta}_* - \frac{2\eta\xi}{\gamma_2} \boldsymbol{\beta}_*^\top \boldsymbol{AA}^\dagger \boldsymbol{uh}\boldsymbol{\beta}_*$$

$$- \frac{2\eta^3 \|\boldsymbol{s}\|^2 \|\boldsymbol{h}\|^2 \xi}{\gamma_2^2} \boldsymbol{\beta}_*^\top \boldsymbol{AA}^\dagger \boldsymbol{uh}\boldsymbol{\beta}_* + \frac{2\eta^3 \|\boldsymbol{s}\|^2 \|\boldsymbol{h}\|^2 \xi}{\gamma_2^2} \boldsymbol{\beta}_*^\top \boldsymbol{AA}^\dagger \boldsymbol{uh}\boldsymbol{\beta}_*.$$

We can combine the coefficients as:

$$\frac{\eta^2 \|\boldsymbol{s}\|^2 \xi^2}{\gamma_2^2} + \frac{\eta^4 \|\boldsymbol{s}\|^4 \|\boldsymbol{h}\|^2}{\gamma_2^2} - \frac{2\eta^2 \|\boldsymbol{s}\|^2}{\gamma_2} = \frac{\eta^2 \|\boldsymbol{s}\|^2 (\eta^2 \|\boldsymbol{s}\|^2 \|\boldsymbol{h}\|^2 + \xi^2) - 2\eta^2 \|\boldsymbol{s}\|^2 \gamma_2}{\gamma_2^2} = -\frac{\eta^2 \|\boldsymbol{s}\|^2}{\gamma_2},$$

$$\frac{\eta^4 \|\boldsymbol{h}\|^4 \|\boldsymbol{s}\|^2}{\gamma_2^2} + \frac{\eta^2 \|\boldsymbol{h}\|^2 \xi^2}{\gamma_2^2} = \frac{\eta^2 \|\boldsymbol{h}\|^2 (\eta^2 \|\boldsymbol{s}\|^2 \|\boldsymbol{h}\|^2 + \xi^2)}{\gamma_2^2} = \frac{\eta^2 \|\boldsymbol{h}\|^2 \gamma_2}{\gamma_2^2} = \frac{\eta^2 \|\boldsymbol{h}\|^2}{\gamma_2}.$$

Then we have that:

$$\boldsymbol{\beta}_*^\top \boldsymbol{A}(\boldsymbol{Z} + \boldsymbol{A})^\dagger(\boldsymbol{Z} + \boldsymbol{A})^{\dagger\top} \boldsymbol{A}\boldsymbol{\beta}_*$$

$$= \boldsymbol{\beta}_*^\top \boldsymbol{AA}^\dagger \boldsymbol{\beta}_* - \frac{\eta^2 \|\boldsymbol{s}\|^2}{\gamma_2} \boldsymbol{\beta}_*^\top \boldsymbol{h}^\top \boldsymbol{h}\boldsymbol{\beta}_* + \frac{\eta^2 \|\boldsymbol{h}\|^2}{\gamma_2} \boldsymbol{\beta}_*^\top \boldsymbol{AA}^\dagger \boldsymbol{uu}^\top \boldsymbol{AA}^\dagger \boldsymbol{\beta}_* - \frac{2\eta\xi}{\gamma_2} \boldsymbol{\beta}_*^\top \boldsymbol{AA}^\dagger \boldsymbol{uh}\boldsymbol{\beta}_*.$$

Recall from Equation 13 that $\mathbb{E}[\boldsymbol{\beta}_*^\top \boldsymbol{AA}^\dagger \boldsymbol{\beta}_*] = \|\boldsymbol{\beta}_*\|^2/c$. We then proceed similarly with the other expectations using Lemmas 9, 10, 11, 19:

$$\mathbb{E}\left[\frac{\eta^2 \|\boldsymbol{s}\|^2}{\gamma_2} \boldsymbol{\beta}_*^\top \boldsymbol{h}^\top \boldsymbol{h}\boldsymbol{\beta}_*\right] = \mathbb{E}\left[\frac{\eta^2}{\gamma_2}\right] \mathbb{E}\left[\|\boldsymbol{s}\|^2\right] \mathbb{E}\left[\boldsymbol{\beta}_*^\top \boldsymbol{h}^\top \boldsymbol{h}\boldsymbol{\beta}_*\right] + \sqrt{\mathrm{Var}\left(\frac{\eta^2}{\gamma_2}\right) O\left(\frac{1}{n}\right)}$$

$$= \left(\frac{\rho^2 \eta^2}{\eta^2 + \rho^2} + o\left(\frac{1}{\rho^2}\right)\right)\left(1 - \frac{1}{c}\right)\left(\frac{\|\boldsymbol{\beta}_*\|^2}{d} \frac{c}{\rho^2(c-1)} + o\left(\frac{1}{d\rho^2}\right)\right) + O\left(\frac{1}{n}\right)$$

$$= \frac{\|\boldsymbol{\beta}_*\|^2}{d} \frac{\eta^2}{\eta^2 + \rho^2} + o\left(\frac{1}{d}\right) + O\left(\frac{1}{n}\right).$$

$$\mathbb{E}\left[\frac{\eta^2 \|\boldsymbol{h}\|^2}{\gamma_2}(\boldsymbol{\beta}_*^\top \boldsymbol{AA}^\dagger \boldsymbol{u})^2\right] = \mathbb{E}\left[\frac{\eta^2}{\gamma_2}\right] \mathbb{E}\left[\|\boldsymbol{h}\|^2\right] \mathbb{E}\left[(\boldsymbol{\beta}_*^\top \boldsymbol{AA}^\dagger \boldsymbol{u})^2\right] + \sqrt{\mathrm{Var}\left(\frac{\eta^2}{\gamma_2}\right) O\left(\frac{1}{n}\right)}$$

$$= \left(\frac{\rho^2 \eta^2}{\eta^2 + \rho^2} + o\left(\frac{1}{\rho^2}\right)\right)\left(\frac{1}{\rho^2} \frac{c}{c-1} + o\left(\frac{1}{\rho^2}\right)\right)\left(\frac{1}{c^2}(\boldsymbol{\beta}_*^\top \boldsymbol{u})^2 + o(1)\right) + O\left(\frac{1}{n}\right)$$

$$= \frac{\eta^2}{\eta^2 + \rho^2} \frac{(\boldsymbol{\beta}_*^\top \boldsymbol{u})^2}{c(c-1)} + o(1) + O\left(\frac{1}{n}\right).$$

$$\mathbb{E}\left[\frac{\eta\xi}{\gamma_2} \boldsymbol{\beta}_*^\top \boldsymbol{AA}^\dagger \boldsymbol{uh}\boldsymbol{\beta}_*\right] = \mathbb{E}\left[\frac{\eta^2}{\gamma_2}\right] \mathbb{E}\left[\frac{\xi}{\eta}\right] \mathbb{E}\left[\boldsymbol{\beta}_*^\top \boldsymbol{AA}^\dagger \boldsymbol{uh}\boldsymbol{\beta}_*\right] + \sqrt{\mathrm{Var}\left(\frac{\eta^2}{\gamma_2}\right) O\left(\frac{1}{n}\right)}$$

$$= 0 + O\left(\frac{1}{n}\right). \tag{17}$$

We combine these results to produce the lemma statement. $\square$

**Lemma 28.** *In the same setting as Section 2, we have that*

$$\mathbb{E}\left[\boldsymbol{\beta}_*^\top \boldsymbol{Z}(\boldsymbol{Z} + \boldsymbol{A})^\dagger(\boldsymbol{Z} + \boldsymbol{A})^{\dagger\top} \boldsymbol{A}\boldsymbol{\beta}_*\right] = \begin{cases} -\left(\frac{\eta^2(\eta^2 + \rho^2)}{(\eta^2 c + \rho^2)^2} \frac{c^2}{1-c} - \frac{\eta^2 c}{\eta^2 c + \rho^2}\right)(\boldsymbol{\beta}_*^\top \boldsymbol{u})^2 + o(1) + O\left(\frac{1}{n}\right), & c < 1 \\ -\frac{\eta^2}{\eta^2 + \rho^2} \frac{1}{c-1}(\boldsymbol{\beta}_*^\top \boldsymbol{u})^2 + o(1) + O\left(\frac{1}{n}\right), & c > 1 \end{cases}$$

*Proof.* For $c < 1$, we expand it using Proposition 1, Lemma 2. Note that all of the relevant expectations have been evaluated in the proofs of Lemmas 26, 27,

$$\boldsymbol{\beta}_*^\top \boldsymbol{Z}(\boldsymbol{Z}+\boldsymbol{A})^\dagger(\boldsymbol{Z}+\boldsymbol{A})^{\dagger\top}\boldsymbol{A}\boldsymbol{\beta}_* = \frac{\eta\xi}{\gamma_1}\boldsymbol{\beta}_*^\top \boldsymbol{u}\boldsymbol{h}\boldsymbol{\beta}_* + \frac{\eta^2\|\boldsymbol{t}\|^2}{\gamma_1}\boldsymbol{\beta}_*^\top \boldsymbol{u}\boldsymbol{k}^\top \boldsymbol{A}^\dagger\boldsymbol{\beta}_* - \frac{2\eta^3\|\boldsymbol{t}\|^2\xi}{\gamma_1^2}(\boldsymbol{\beta}_*^\top \boldsymbol{u})^2\boldsymbol{h}\boldsymbol{A}^{\dagger\top}\boldsymbol{k}$$
$$- \frac{\eta^4\|\boldsymbol{t}\|^4}{\gamma_1^2}(\boldsymbol{k}^\top \boldsymbol{A}^\dagger \boldsymbol{A}^{\dagger\top}\boldsymbol{k})(\boldsymbol{\beta}_*^\top \boldsymbol{u})^2 - \frac{\eta^2\|\boldsymbol{h}\|^2\xi^2}{\gamma_1^2}(\boldsymbol{\beta}_*^\top \boldsymbol{u})^2.$$

The expectation of the last three terms is given by Lemma 26. The first two expectations come from Equations 18, 19 respectively. We can plug them in and compute the expectation:

$$\mathbb{E}\left[\boldsymbol{\beta}_*^\top \boldsymbol{Z}(\boldsymbol{Z}+\boldsymbol{A})^\dagger(\boldsymbol{Z}+\boldsymbol{A})^{\dagger\top}\boldsymbol{A}\boldsymbol{\beta}_*\right] = -\left(\frac{\eta^2(\eta^2+\rho^2)}{(\eta^2c+\rho^2)^2}\frac{c^2}{1-c} - \frac{\eta^2c}{\eta^2c+\rho^2}\right)(\boldsymbol{\beta}_*^\top \boldsymbol{u})^2 + o(1) + O\left(\frac{1}{n}\right).$$

For $c > 1$, again with $\boldsymbol{h}\boldsymbol{s} = \boldsymbol{0}$ and $\boldsymbol{s}^\top \boldsymbol{A} = \boldsymbol{0}$, $\boldsymbol{\beta}_*^\top \boldsymbol{Z}(\boldsymbol{Z}+\boldsymbol{A})^\dagger(\boldsymbol{Z}+\boldsymbol{A})^{\dagger\top}\boldsymbol{A}\boldsymbol{\beta}_*$ becomes:

$$\boldsymbol{\beta}_*^\top \frac{\eta\xi}{\gamma_2}\boldsymbol{u}\boldsymbol{h}\left(\boldsymbol{A}\boldsymbol{A}^\dagger + \frac{\eta\xi}{\gamma_2}\boldsymbol{s}\boldsymbol{h} - \frac{\eta^2\|\boldsymbol{s}\|^2}{\gamma_2}\boldsymbol{h}^\top \boldsymbol{h} - \frac{\eta^2\|\boldsymbol{h}\|^2}{\gamma_2}\boldsymbol{s}\boldsymbol{u}^\top \boldsymbol{A}\boldsymbol{A}^\dagger - \frac{\eta\xi}{\gamma_2}\boldsymbol{h}^\top \boldsymbol{u}^\top \boldsymbol{A}\boldsymbol{A}^\dagger\right)\boldsymbol{\beta}_*$$
$$+ \boldsymbol{\beta}_*^\top \frac{\eta^2\|\boldsymbol{h}\|^2}{\gamma_2}\boldsymbol{u}\boldsymbol{s}^\top \left(\boldsymbol{A}\boldsymbol{A}^\dagger + \frac{\eta\xi}{\gamma_2}\boldsymbol{s}\boldsymbol{h} - \frac{\eta^2\|\boldsymbol{s}\|^2}{\gamma_2}\boldsymbol{h}^\top \boldsymbol{h} - \frac{\eta^2\|\boldsymbol{h}\|^2}{\gamma_2}\boldsymbol{s}\boldsymbol{u}^\top \boldsymbol{A}\boldsymbol{A}^\dagger - \frac{\eta\xi}{\gamma_2}\boldsymbol{h}^\top \boldsymbol{u}^\top \boldsymbol{A}\boldsymbol{A}^\dagger\right)\boldsymbol{\beta}_*$$
$$= \boldsymbol{\beta}_*^\top \left[\frac{\eta\xi}{\gamma_2}\boldsymbol{u}\boldsymbol{h}\boldsymbol{A}\boldsymbol{A}^\dagger - \frac{\eta^3\xi\|\boldsymbol{s}\|^2\|\boldsymbol{h}\|^2}{\gamma_2^2}\boldsymbol{u}\boldsymbol{h} - \frac{\eta^2\|\boldsymbol{h}\|^2\xi^2}{\gamma_2^2}\boldsymbol{u}\boldsymbol{u}^\top \boldsymbol{A}\boldsymbol{A}^\dagger\right]\boldsymbol{\beta}_*$$
$$+ \boldsymbol{\beta}_*^\top \left[\frac{\eta^3\|\boldsymbol{h}\|^2\|\boldsymbol{s}\|^2\xi}{\gamma_2^2}\boldsymbol{u}\boldsymbol{h} - \frac{\eta^4\|\boldsymbol{h}\|^4\|\boldsymbol{s}\|^2}{\gamma_2^2}\boldsymbol{u}\boldsymbol{u}^\top \boldsymbol{A}\boldsymbol{A}^\dagger\right]\boldsymbol{\beta}_*$$
$$= \boldsymbol{\beta}_*^\top \left[\frac{\eta\xi}{\gamma_2}\boldsymbol{u}\boldsymbol{h}\boldsymbol{A}\boldsymbol{A}^\dagger - \frac{\eta^2\|\boldsymbol{h}\|^2\xi^2}{\gamma_2^2}\boldsymbol{u}\boldsymbol{u}^\top \boldsymbol{A}\boldsymbol{A}^\dagger - \frac{\eta^4\|\boldsymbol{h}\|^4\|\boldsymbol{s}\|^2}{\gamma_2^2}\boldsymbol{u}\boldsymbol{u}^\top \boldsymbol{A}\boldsymbol{A}^\dagger\right]\boldsymbol{\beta}_*$$
$$= (\boldsymbol{\beta}_*^\top \boldsymbol{u})\left(\frac{\eta\xi}{\gamma_2}\boldsymbol{h}\boldsymbol{A}\boldsymbol{A}^\dagger\boldsymbol{\beta}_* - \frac{\eta^2\|\boldsymbol{h}\|^2}{\gamma_2}\boldsymbol{u}^\top \boldsymbol{A}\boldsymbol{A}^\dagger\boldsymbol{\beta}_*\right) \quad \text{since } \gamma_2 = \eta^2\|\boldsymbol{s}\|^2\|\boldsymbol{h}\|^2 + \xi^2.$$

We need to evaluate two following expectations. Similar to $c < 1$,

$$\mathbb{E}\left[\frac{\eta\xi}{\gamma_2}\boldsymbol{h}\boldsymbol{A}\boldsymbol{A}^\dagger\boldsymbol{\beta}_*\right] = O\left(\frac{1}{n}\right).$$

$$\mathbb{E}\left[\frac{\eta^2\|\boldsymbol{h}\|^2}{\gamma_2}\boldsymbol{u}^\top \boldsymbol{A}\boldsymbol{A}^\dagger\boldsymbol{\beta}_*\right] = \mathbb{E}\left[\frac{\eta^2}{\gamma_2}\right]\mathbb{E}\left[\|\boldsymbol{h}\|^2\right]\mathbb{E}\left[\boldsymbol{\beta}_*^\top \boldsymbol{A}\boldsymbol{A}^\dagger\boldsymbol{u}\right] + \sqrt{\mathrm{Var}\left(\frac{\eta^2}{\gamma_2}\right)O\left(\frac{1}{n}\right)}$$
$$= \left(\frac{\rho^2\eta^2}{\eta^2+\rho^2} + o\left(\frac{1}{\rho^2}\right)\right)\left(\frac{1}{\rho^2}\frac{c}{c-1} + o\left(\frac{1}{\rho^2}\right)\right)\left(\frac{1}{c}(\boldsymbol{\beta}_*^\top \boldsymbol{u})\right) + O\left(\frac{1}{n}\right)$$
$$= \frac{\eta^2}{\eta^2+\rho^2}\frac{(\boldsymbol{\beta}_*^\top \boldsymbol{u})}{c-1} + o(1) + O\left(\frac{1}{n}\right).$$

Finally, we have that:

$$\mathbb{E}\left[\boldsymbol{\beta}_*^\top \boldsymbol{Z}(\boldsymbol{Z}+\boldsymbol{A})^\dagger(\boldsymbol{Z}+\boldsymbol{A})^{\dagger\top}\boldsymbol{A}\boldsymbol{\beta}_*\right] = -\frac{\eta^2}{\eta^2+\rho^2}\frac{1}{c-1}(\boldsymbol{\beta}_*^\top \boldsymbol{u})^2 + o(1) + O\left(\frac{1}{n}\right).$$

$\square$

**Lemma 29.** *In the same setting as Section 2, we have that,*

$$\mathbb{E}\left[\boldsymbol{\varepsilon}^\top (\boldsymbol{Z}+\boldsymbol{A})^\dagger(\boldsymbol{Z}+\boldsymbol{A})^{\dagger\top}\boldsymbol{\varepsilon}\right] = \begin{cases} \tau_\varepsilon^2\left(\frac{cd}{\rho^2(1-c)} - \frac{\eta^2}{\rho^2(\eta^2c+\rho^2)}\frac{c^2}{1-c}\right) + o\left(\frac{n}{\rho^2}\right) + O\left(\frac{1}{\rho^2 n}\right), & c < 1 \\ \tau_\varepsilon^2\left(\frac{d}{\rho^2(c-1)} - \frac{\eta^2}{\rho^2(\eta^2+\rho^2)}\frac{c}{c-1}\right) + o\left(\frac{n}{\rho^2}\right) + O\left(\frac{1}{\rho^2 n}\right), & c > 1 \end{cases}$$

*Proof.* For $c < 1$, we first expand this term using Theorem 6:

$$\boldsymbol{\varepsilon}^\top (\boldsymbol{Z} + \boldsymbol{A})^\dagger (\boldsymbol{Z} + \boldsymbol{A})^{\dagger\top} \boldsymbol{\varepsilon} = \boldsymbol{\varepsilon}^\top \left( \boldsymbol{A}^\dagger + \frac{\eta}{\xi} \boldsymbol{t}^\top \boldsymbol{k}^\top \boldsymbol{A}^\dagger - \frac{\xi}{\gamma_1} \boldsymbol{p}_1 \boldsymbol{q}_1^\top \right) \left( \boldsymbol{A}^\dagger + \frac{\eta}{\xi} \boldsymbol{t}^\top \boldsymbol{k}^\top \boldsymbol{A}^\dagger - \frac{\xi}{\gamma_1} \boldsymbol{p}_1 \boldsymbol{q}_1^\top \right)^\top \boldsymbol{\varepsilon}$$

$$= \boldsymbol{\varepsilon}^\top \boldsymbol{A}^\dagger \boldsymbol{A}^{\dagger\top} \boldsymbol{\varepsilon} + \frac{2\eta}{\xi} \boldsymbol{\varepsilon}^\top \boldsymbol{A}^\dagger \boldsymbol{A}^{\dagger\top} \boldsymbol{k} \boldsymbol{t} \boldsymbol{\varepsilon} - \frac{2\xi}{\gamma_1} \boldsymbol{\varepsilon}^\top \boldsymbol{A}^\dagger \boldsymbol{q}_1 \boldsymbol{p}_1^\top \boldsymbol{\varepsilon}$$

$$+ \frac{\eta^2}{\xi^2} \left( \boldsymbol{k}^\top \boldsymbol{A}^\dagger \boldsymbol{A}^{\dagger\top} \boldsymbol{k} \right) \boldsymbol{\varepsilon}^\top \boldsymbol{t}^\top \boldsymbol{t} \boldsymbol{\varepsilon} - \frac{2\eta}{\gamma_1} \boldsymbol{\varepsilon}^\top \boldsymbol{t}^\top \boldsymbol{k}^\top \boldsymbol{A}^\dagger \boldsymbol{q}_1 \boldsymbol{p}_1^\top \boldsymbol{\varepsilon} + \frac{\xi^2}{\gamma_1^2} \boldsymbol{\varepsilon}^\top \boldsymbol{p}_1 \boldsymbol{q}_1^\top \boldsymbol{q}_1 \boldsymbol{p}_1^\top \boldsymbol{\varepsilon}$$

Note that Lemma 21 and the fact that $\boldsymbol{t} \boldsymbol{A}^\dagger = \boldsymbol{0}$ imply that the second term has zero expectation:

$$\mathbb{E}_{\boldsymbol{\varepsilon}} \left[ \frac{2\eta}{\xi} \boldsymbol{\varepsilon}^\top \boldsymbol{A}^\dagger \boldsymbol{A}^{\dagger\top} \boldsymbol{k} \boldsymbol{t} \boldsymbol{\varepsilon} \right] = \frac{2\eta \tau_\varepsilon^2}{\xi} \boldsymbol{t} \boldsymbol{A}^\dagger \boldsymbol{A}^{\dagger\top} \boldsymbol{k} = 0.$$

Simiarly, we will later use:

$$\mathbb{E}_{\boldsymbol{\varepsilon}} \left[ \boldsymbol{\varepsilon}^\top \boldsymbol{A}^\dagger \boldsymbol{h}^\top \boldsymbol{t} \boldsymbol{\varepsilon} \right] = \tau_\varepsilon^2 \boldsymbol{t} \boldsymbol{A}^\dagger \boldsymbol{h}^\top = 0, \quad \mathbb{E}_{\boldsymbol{\varepsilon}} \left[ \boldsymbol{\varepsilon}^\top \boldsymbol{t}^\top \boldsymbol{k}^\top \boldsymbol{\varepsilon} \right] = \tau_\varepsilon^2 Tr(\boldsymbol{t}^\top \boldsymbol{k}^\top) = \tau_\varepsilon^2 Tr(\boldsymbol{k} \boldsymbol{t}) = 0.$$

Note that these equalities are exact without taking the expectation over other sources of randomness besides $\boldsymbol{\varepsilon}$.

We now expand the other terms one by one and compute their expectations along the way. We start by eliminating zero expectations and taking expectations w.r.t. $\boldsymbol{\varepsilon}$ using Lemma 21.

$$\mathbb{E} \left[ \frac{\eta^2}{\xi^2} \left( \boldsymbol{k}^\top \boldsymbol{A}^\dagger \boldsymbol{A}^{\dagger\top} \boldsymbol{k} \right) \boldsymbol{\varepsilon}^\top \boldsymbol{t}^\top \boldsymbol{t} \boldsymbol{\varepsilon} \right] = \mathbb{E} \left[ \frac{\eta^2 \|\boldsymbol{t}\|^2 \tau_\varepsilon^2}{\xi^2} \boldsymbol{k}^\top \boldsymbol{A}^\dagger \boldsymbol{A}^{\dagger\top} \boldsymbol{k} \right].$$

$$\mathbb{E} \left[ -\frac{2\xi}{\gamma_1} \boldsymbol{\varepsilon}^\top \boldsymbol{A}^\dagger \boldsymbol{q}_1 \boldsymbol{p}_1^\top \boldsymbol{\varepsilon} \right] = \mathbb{E} \left[ -\frac{2\xi}{\gamma_1} \boldsymbol{\varepsilon}^\top \boldsymbol{A}^\dagger \left( \frac{\eta \|\boldsymbol{t}\|^2}{\xi} \boldsymbol{A}^{\dagger\top} \boldsymbol{k} + \boldsymbol{h}^\top \right) \left( \frac{\eta^2 \|\boldsymbol{k}\|^2}{\xi} \boldsymbol{t} + \eta \boldsymbol{k}^\top \right) \boldsymbol{\varepsilon} \right]$$

$$= \mathbb{E} \left[ -\frac{2\eta^3 \|\boldsymbol{t}\|^2 \|\boldsymbol{k}\|^2}{\gamma_1 \xi} \boldsymbol{\varepsilon}^\top \boldsymbol{A}^\dagger \boldsymbol{A}^{\dagger\top} \boldsymbol{k} \boldsymbol{t} \boldsymbol{\varepsilon} - \frac{2\eta^2 \|\boldsymbol{t}\|^2}{\gamma_1} \boldsymbol{\varepsilon}^\top \boldsymbol{A}^\dagger \boldsymbol{A}^{\dagger\top} \boldsymbol{k} \boldsymbol{k}^\top \boldsymbol{\varepsilon} \right.$$

$$\left. - \frac{2\eta^2 \|\boldsymbol{k}\|^2}{\gamma_1} \boldsymbol{\varepsilon}^\top \boldsymbol{A}^\dagger \boldsymbol{h}^\top \boldsymbol{t} \boldsymbol{\varepsilon} - \frac{2\eta \xi}{\gamma_1} \boldsymbol{\varepsilon}^\top \boldsymbol{A}^\dagger \boldsymbol{h}^\top \boldsymbol{k}^\top \boldsymbol{\varepsilon} \right]$$

$$= \mathbb{E} \left[ -\frac{2\eta^2 \|\boldsymbol{t}\|^2 \tau_\varepsilon^2}{\gamma_1} \boldsymbol{k}^\top \boldsymbol{A}^\dagger \boldsymbol{A}^{\dagger\top} \boldsymbol{k} - \frac{2\eta \xi \tau_\varepsilon^2}{\gamma_1} \boldsymbol{k}^\top \boldsymbol{A}^\dagger \boldsymbol{h}^\top \right].$$

$$\mathbb{E} \left[ -\frac{2\eta}{\gamma_1} \boldsymbol{\varepsilon}^\top \boldsymbol{t}^\top \boldsymbol{k}^\top \boldsymbol{A}^\dagger \boldsymbol{q}_1 \boldsymbol{p}_1^\top \boldsymbol{\varepsilon} \right] = \mathbb{E} \left[ -\frac{2\eta}{\gamma_1} \boldsymbol{\varepsilon}^\top \boldsymbol{t}^\top \boldsymbol{k}^\top \boldsymbol{A}^\dagger \left( \frac{\eta \|\boldsymbol{t}\|^2}{\xi} \boldsymbol{A}^{\dagger\top} \boldsymbol{k} + \boldsymbol{h}^\top \right) \left( \frac{\eta^2 \|\boldsymbol{k}\|^2}{\xi} \boldsymbol{t} + \eta \boldsymbol{k}^\top \right) \boldsymbol{\varepsilon} \right]$$

$$= \mathbb{E} \left[ -\frac{2\eta^4 \|\boldsymbol{t}\|^2 \|\boldsymbol{k}\|^2}{\gamma_1 \xi^2} \left( \boldsymbol{k}^\top \boldsymbol{A}^\dagger \boldsymbol{A}^{\dagger\top} \boldsymbol{k} \right) \boldsymbol{\varepsilon}^\top \boldsymbol{t}^\top \boldsymbol{t} \boldsymbol{\varepsilon} - \frac{2\eta^3 \|\boldsymbol{k}\|^2}{\gamma_1 \xi} (\boldsymbol{k}^\top \boldsymbol{A}^\dagger \boldsymbol{h}^\top) \boldsymbol{\varepsilon}^\top \boldsymbol{t}^\top \boldsymbol{t} \boldsymbol{\varepsilon} \right.$$

$$\left. - \frac{2\eta^3 \|\boldsymbol{t}\|^2}{\gamma_1 \xi} \left( \boldsymbol{k}^\top \boldsymbol{A}^\dagger \boldsymbol{A}^{\dagger\top} \boldsymbol{k} \right) \boldsymbol{\varepsilon}^\top \boldsymbol{t}^\top \boldsymbol{k}^\top \boldsymbol{\varepsilon} - \frac{2\eta^2}{\gamma_1} (\boldsymbol{k}^\top \boldsymbol{A}^\dagger \boldsymbol{h}^\top) \boldsymbol{\varepsilon}^\top \boldsymbol{t}^\top \boldsymbol{k}^\top \boldsymbol{\varepsilon} \right]$$

$$= \mathbb{E} \left[ -\frac{2\eta^4 \|\boldsymbol{t}\|^4 \|\boldsymbol{k}\|^2 \tau_\varepsilon^2}{\gamma_1 \xi^2} \boldsymbol{k}^\top \boldsymbol{A}^\dagger \boldsymbol{A}^{\dagger\top} \boldsymbol{k} - \frac{2\eta^3 \|\boldsymbol{k}\|^2 \|\boldsymbol{t}\|^2 \tau_\varepsilon^2}{\gamma_1 \xi} \boldsymbol{k}^\top \boldsymbol{A}^\dagger \boldsymbol{h}^\top \right].$$

By the squared norms in Lemmas 6, 7, and Lemma 21,

$$
\begin{aligned}
\mathbb{E}\left[\frac{\xi^2}{\gamma_1^2}\boldsymbol{\varepsilon}^\top \boldsymbol{p}_1\boldsymbol{q}_1^\top \boldsymbol{q}_1\boldsymbol{p}_1^\top \boldsymbol{\varepsilon}\right] &= \frac{\xi^2\tau_\varepsilon^2}{\gamma_1^2}\|\boldsymbol{p}_1\|^2\|\boldsymbol{q}_1\|^2 \\
&= \frac{\xi^2\tau_\varepsilon^2}{\gamma_1^2}\left(\frac{\eta^2\|\boldsymbol{k}\|^2}{\xi^2}\gamma_1\right)\left(\frac{\eta^2\|\boldsymbol{t}\|^4}{\xi^2}\boldsymbol{k}\boldsymbol{A}^\dagger \boldsymbol{A}^{\dagger\top}\boldsymbol{k} + \frac{2\eta\|\boldsymbol{t}\|^2}{\xi}\boldsymbol{k}^\top \boldsymbol{A}^\dagger \boldsymbol{h}^\top + \|\boldsymbol{h}\|^2\right) \\
&= \frac{\tau_\varepsilon^2}{\gamma_1}\left(\eta^2\|\boldsymbol{k}\|^2\right)\left(\frac{\eta^2\|\boldsymbol{t}\|^4}{\xi^2}\boldsymbol{k}\boldsymbol{A}^\dagger \boldsymbol{A}^{\dagger\top}\boldsymbol{k} + \frac{2\eta\|\boldsymbol{t}\|^2}{\xi}\boldsymbol{k}^\top \boldsymbol{A}^\dagger \boldsymbol{h}^\top + \|\boldsymbol{h}\|^2\right) \\
&= \tau_\varepsilon^2\left(\frac{\eta^4\|\boldsymbol{t}\|^4\|\boldsymbol{k}\|^2}{\gamma_1\xi^2}\boldsymbol{k}\boldsymbol{A}^\dagger \boldsymbol{A}^{\dagger\top}\boldsymbol{k} + \frac{2\eta^3\|\boldsymbol{t}\|^2\|\boldsymbol{k}\|^2}{\gamma_1\xi}\boldsymbol{k}^\top \boldsymbol{A}^\dagger \boldsymbol{h}^\top + \frac{\eta^2\|\boldsymbol{k}\|^2\|\boldsymbol{h}\|^2}{\gamma_1}\right)
\end{aligned}
$$

We combine like terms and simplify the coefficients. For the term $\boldsymbol{k}^\top \boldsymbol{A}^\dagger \boldsymbol{A}^{\dagger\top}\boldsymbol{k}$,

$$
\begin{aligned}
\tau_\varepsilon^2\left(\frac{\eta^4\|\boldsymbol{t}\|^4\|\boldsymbol{k}\|^2}{\gamma_1\xi^2} - \frac{2\eta^4\|\boldsymbol{t}\|^4\|\boldsymbol{k}\|^2}{\gamma_1\xi^2} - \frac{2\eta^2\|\boldsymbol{t}\|^2}{\gamma_1} + \frac{\eta^2\|\boldsymbol{t}\|^2}{\xi^2}\right) &= \tau_\varepsilon^2\eta^2\|\boldsymbol{t}\|^2\left(\frac{\eta^2\|\boldsymbol{t}\|^2\|\boldsymbol{k}\|^2}{\gamma_1\xi^2} - \frac{2\eta^2\|\boldsymbol{t}\|^2\|\boldsymbol{k}\|^2}{\gamma_1\xi^2} - \frac{2}{\gamma_1} + \frac{1}{\xi^2}\right) \\
&= \tau_\varepsilon^2\eta^2\|\boldsymbol{t}\|^2\left(-\frac{\gamma_1-\xi^2}{\gamma_1\xi^2} - \frac{2}{\gamma_1} + \frac{1}{\xi^2}\right) \\
&= \tau_\varepsilon^2\eta^2\|\boldsymbol{t}\|^2\left(-\frac{\gamma_1-\xi^2}{\gamma_1\xi^2} - \frac{2\xi^2}{\gamma_1\xi^2} + \frac{\gamma_1}{\gamma_1\xi^2}\right) \\
&= -\tau_\varepsilon^2\frac{\eta^2\|\boldsymbol{t}\|^2}{\gamma_1}.
\end{aligned}
$$

For the term $\boldsymbol{k}^\top \boldsymbol{A}^\dagger \boldsymbol{h}^\top$,

$$
\tau_\varepsilon^2\left(\frac{2\eta^3\|\boldsymbol{t}\|^2\|\boldsymbol{k}\|^2}{\gamma_1\xi} - \frac{2\eta^3\|\boldsymbol{k}\|^2\|\boldsymbol{t}\|^2}{\gamma_1\xi} - \frac{2\eta\xi}{\gamma_1}\right) = -\tau_\varepsilon^2\frac{2\eta\xi}{\gamma_1}.
$$

Combining these terms together, we have:

$$
\mathbb{E}\left[\boldsymbol{\varepsilon}^\top(\boldsymbol{Z}+\boldsymbol{A})^\dagger(\boldsymbol{Z}+\boldsymbol{A})^{\dagger\top}\boldsymbol{\varepsilon}\right] = \mathbb{E}\left[\boldsymbol{\varepsilon}^\top \boldsymbol{A}^\dagger \boldsymbol{A}^{\dagger\top}\boldsymbol{\varepsilon} - \frac{\eta^2\|\boldsymbol{t}\|^2\tau_\varepsilon^2}{\gamma_1}\boldsymbol{k}^\top \boldsymbol{A}^\dagger \boldsymbol{A}^{\dagger\top}\boldsymbol{k} - \frac{2\eta\xi\tau_\varepsilon^2}{\gamma_1}\boldsymbol{k}^\top \boldsymbol{A}^\dagger \boldsymbol{h}^\top + \frac{\eta^2\|\boldsymbol{k}\|^2\|\boldsymbol{h}\|^2}{\gamma_1}\right].
$$

Similarly, using Lemmas 9, 10, 11, 19, 21, we have the following:

$$
\mathbb{E}\left[\boldsymbol{\varepsilon}^\top \boldsymbol{A}^\dagger \boldsymbol{A}^{\dagger\top}\boldsymbol{\varepsilon}\right] = \tau_\varepsilon^2\mathbb{E}\left[Tr(\boldsymbol{A}^\dagger \boldsymbol{A}^{\dagger\top})\right] = \tau_\varepsilon^2 n\mathbb{E}\left[\frac{1}{\lambda}\right] = \tau_\varepsilon^2\frac{cd}{\rho^2(1-c)} + o\left(\frac{d}{\rho^2}\right) \quad \text{by Equation 11.}
$$

$$
\begin{aligned}
\mathbb{E}\left[\frac{\eta^2\|\boldsymbol{t}\|^2}{\gamma_1}\boldsymbol{k}^\top \boldsymbol{A}^\dagger \boldsymbol{A}^{\dagger\top}\boldsymbol{k}\right] &= \mathbb{E}\left[\frac{\eta^2}{\gamma_1}\right]\mathbb{E}\left[\|\boldsymbol{t}\|^2\right]\mathbb{E}\left[\boldsymbol{k}^\top \boldsymbol{A}^\dagger \boldsymbol{A}^{\dagger\top}\boldsymbol{k}\right] + \sqrt{\text{Var}\left(\frac{\eta^2}{\gamma_1}\right)O\left(\frac{1}{n}\right)} \\
&= \left(\frac{\rho^2\eta^2}{\eta^2 c+\rho^2} + o\left(\frac{1}{\rho^2}\right)\right)(1-c)\left(\frac{1}{\rho^4}\frac{c^2}{(1-c)^3} + o\left(\frac{1}{\rho^4}\right)\right) + O\left(\frac{1}{n}\right) \\
&= \frac{\eta^2}{\eta^2 c+\rho^2}\frac{c^2}{\rho^2(1-c)^2}(\boldsymbol{\beta}_*^\top \boldsymbol{u})^2 + o(1) + O\left(\frac{1}{n}\right).
\end{aligned}
$$

$$
\mathbb{E}\left[\frac{\eta\xi}{\gamma_1}\boldsymbol{k}^\top \boldsymbol{A}^\dagger \boldsymbol{h}^\top\right] = \mathbb{E}\left[\frac{\eta^2}{\gamma_1}\right]\mathbb{E}\left[\frac{\xi}{\eta}\right]\mathbb{E}\left[\boldsymbol{k}^\top \boldsymbol{A}^\dagger \boldsymbol{h}^\top\right] + \sqrt{\text{Var}\left(\frac{\eta^2}{\gamma_1}\right)O\left(\frac{1}{n}\right)} = O\left(\frac{1}{n}\right).
$$

$$
\begin{aligned}
\mathbb{E}\left[\frac{\eta^2\|\boldsymbol{k}\|^2\|\boldsymbol{h}\|^2}{\gamma_1}\right] &= \mathbb{E}\left[\frac{\eta^2}{\gamma_1}\right]\mathbb{E}\left[\|\boldsymbol{k}\|^2\right]\mathbb{E}\left[\|\boldsymbol{h}\|\right] + \sqrt{\text{Var}\left(\frac{\eta^2}{\gamma_1}\right)O\left(\frac{1}{n}\right)} \\
&= \left(\frac{\rho^2\eta^2}{\eta^2 c+\rho^2} + o\left(\frac{1}{\rho^2}\right)\right)\left(\frac{1}{\rho^2}\frac{c^2}{1-c} + o\left(\frac{1}{\rho^2}\right)\right)\left(\frac{1}{\rho^2}\frac{c}{1-c} + o\left(\frac{1}{\rho^2}\right)\right) + O\left(\frac{1}{n}\right) \\
&= \frac{\eta^2}{\eta^2 c+\rho^2}\frac{c^3}{\rho^2(1-c)^2} + o(1) + O\left(\frac{1}{n}\right).
\end{aligned}
$$

After simple algebra, the result follows from here.

For $c > 1$, we can expand similarly using Theorem 6,

$$\boldsymbol{\varepsilon}^\top (\boldsymbol{Z} + \boldsymbol{A})^\dagger (\boldsymbol{Z} + \boldsymbol{A})^{\dagger\top} \boldsymbol{\varepsilon} = \boldsymbol{\varepsilon}^\top \left( \boldsymbol{A}^\dagger + \frac{\eta}{\xi} \boldsymbol{A}^{\dagger\top} \boldsymbol{h}^\top \boldsymbol{s}^\top - \frac{\xi}{\gamma_2} \boldsymbol{p}_2 \boldsymbol{q}_2^\top \right) \left( \boldsymbol{A}^{\dagger\top} + \frac{\eta}{\xi} \boldsymbol{s} \boldsymbol{h} \boldsymbol{A}^{\dagger\top} - \frac{\xi}{\gamma_2} \boldsymbol{q}_2 \boldsymbol{p}_2^\top \right) \boldsymbol{\varepsilon}$$

$$= \boldsymbol{\varepsilon}^\top \boldsymbol{A}^\dagger \boldsymbol{A}^{\dagger\top} \boldsymbol{\varepsilon} + \frac{2\eta}{\xi} \boldsymbol{\varepsilon}^\top \underbrace{\boldsymbol{A}^\dagger \boldsymbol{s}}_{0} \boldsymbol{h} \boldsymbol{A}^{\dagger\top} \boldsymbol{\varepsilon} - \frac{2\xi}{\gamma_2} \boldsymbol{\varepsilon}^\top \boldsymbol{A}^\dagger \boldsymbol{q}_2 \boldsymbol{p}_2^\top \boldsymbol{\varepsilon}$$

$$+ \frac{\eta^2 \|\boldsymbol{s}\|^2}{\xi^2} \boldsymbol{\varepsilon}^\top \boldsymbol{A}^\dagger \boldsymbol{h}^\top \boldsymbol{h} \boldsymbol{A}^{\dagger\top} \boldsymbol{\varepsilon} - \frac{2\eta}{\gamma_2} \boldsymbol{\varepsilon}^\top \boldsymbol{A}^\dagger \boldsymbol{h}^\top \boldsymbol{s}^\top \boldsymbol{q}_2 \boldsymbol{p}_2^\top \boldsymbol{\varepsilon} + \frac{\xi^2}{\gamma_2^2} \boldsymbol{\varepsilon}^\top \boldsymbol{p}_2 \boldsymbol{q}_2^\top \boldsymbol{q}_2 \boldsymbol{p}_2^\top \boldsymbol{\varepsilon}.$$

We expand the other terms one by one, marking those with zero expectations:

$$\mathbb{E}\left[ \frac{\eta^2 \|\boldsymbol{s}\|^2}{\xi^2} \boldsymbol{\varepsilon}^\top \boldsymbol{A}^\dagger \boldsymbol{h}^\top \boldsymbol{h} \boldsymbol{A}^{\dagger\top} \boldsymbol{\varepsilon} \right] = \mathbb{E}\left[ \frac{\eta^2 \|\boldsymbol{s}\|^2 \tau_\varepsilon^2}{\xi^2} \boldsymbol{h} \boldsymbol{A}^{\dagger\top} \boldsymbol{A}^\dagger \boldsymbol{h}^\top \right].$$

$$\mathbb{E}\left[ -\frac{2\xi}{\gamma_2} \boldsymbol{\varepsilon}^\top \boldsymbol{A}^\dagger \boldsymbol{q}_2 \boldsymbol{p}_2^\top \boldsymbol{\varepsilon} \right] = \mathbb{E}\left[ -\frac{2\xi}{\gamma_2} \boldsymbol{\varepsilon}^\top \boldsymbol{A}^\dagger \left( \frac{\eta \|\boldsymbol{h}\|^2}{\xi} \boldsymbol{s} + \boldsymbol{h}^\top \right) \left( \frac{\eta^2 \|\boldsymbol{s}\|^2}{\xi} \boldsymbol{h} \boldsymbol{A}^{\dagger\top} + \eta \boldsymbol{k}^\top \right) \boldsymbol{\varepsilon} \right]$$

$$= \mathbb{E}\left[ -\frac{2\xi}{\gamma_2} \boldsymbol{\varepsilon}^\top \boldsymbol{A}^\dagger \boldsymbol{h}^\top \left( \frac{\eta^2 \|\boldsymbol{s}\|^2}{\xi} \boldsymbol{h} \boldsymbol{A}^{\dagger\top} + \eta \boldsymbol{k}^\top \right) \boldsymbol{\varepsilon} \right]$$

$$= \mathbb{E}\left[ -\frac{2\eta^2 \|\boldsymbol{s}\|^2}{\gamma_2} \boldsymbol{\varepsilon}^\top \boldsymbol{A}^\dagger \boldsymbol{h}^\top \boldsymbol{h} \boldsymbol{A}^{\dagger\top} \boldsymbol{\varepsilon} - \frac{2\eta\xi}{\gamma_2} \boldsymbol{\varepsilon}^\top \boldsymbol{A}^\dagger \boldsymbol{h}^\top \boldsymbol{k}^\top \boldsymbol{\varepsilon} \right]$$

$$= \mathbb{E}\left[ -\frac{2\eta^2 \|\boldsymbol{s}\|^2 \tau_\varepsilon^2}{\gamma_2} \boldsymbol{h} \boldsymbol{A}^{\dagger\top} \boldsymbol{A}^\dagger \boldsymbol{h}^\top - \frac{2\eta\xi \tau_\varepsilon^2}{\gamma_2} \boldsymbol{k}^\top \boldsymbol{A}^\dagger \boldsymbol{h}^\top \right].$$

$$\mathbb{E}\left[ -\frac{2\eta}{\gamma_2} \boldsymbol{\varepsilon}^\top \boldsymbol{A}^\dagger \boldsymbol{h}^\top \boldsymbol{s}^\top \boldsymbol{q}_2 \boldsymbol{p}_2^\top \boldsymbol{\varepsilon} \right] = \mathbb{E}\left[ -\frac{2\eta}{\gamma_2} \boldsymbol{\varepsilon}^\top \boldsymbol{A}^\dagger \boldsymbol{h}^\top \boldsymbol{s}^\top \left( \frac{\eta \|\boldsymbol{h}\|^2}{\xi} \boldsymbol{s} + \boldsymbol{h}^\top \right) \left( \frac{\eta^2 \|\boldsymbol{s}\|^2}{\xi} \boldsymbol{h} \boldsymbol{A}^{\dagger\top} + \eta \boldsymbol{k}^\top \right) \boldsymbol{\varepsilon} \right]$$

$$= \mathbb{E}\left[ -\frac{2\eta}{\gamma_2} \boldsymbol{\varepsilon}^\top \boldsymbol{A}^\dagger \boldsymbol{h}^\top \left( \frac{\eta \|\boldsymbol{h}\|^2 \|\boldsymbol{s}\|^2}{\xi} \right) \left( \frac{\eta^2 \|\boldsymbol{s}\|^2}{\xi} \boldsymbol{h} \boldsymbol{A}^{\dagger\top} + \eta \boldsymbol{k}^\top \right) \boldsymbol{\varepsilon} \right]$$

$$= \mathbb{E}\left[ -\frac{2\eta^4 \|\boldsymbol{s}\|^4 \|\boldsymbol{h}\|^2}{\gamma_2 \xi^2} \boldsymbol{\varepsilon}^\top \boldsymbol{A}^\dagger \boldsymbol{h}^\top \boldsymbol{h} \boldsymbol{A}^{\dagger\top} \boldsymbol{\varepsilon} - \frac{2\eta^3 \|\boldsymbol{s}\|^2 \|\boldsymbol{h}\|^2}{\gamma_2 \xi} \boldsymbol{\varepsilon}^\top \boldsymbol{A}^\dagger \boldsymbol{h}^\top \boldsymbol{k}^\top \boldsymbol{\varepsilon} \right]$$

$$= \mathbb{E}\left[ -\frac{2\eta^4 \|\boldsymbol{s}\|^4 \|\boldsymbol{h}\|^2 \tau_\varepsilon^2}{\gamma_2 \xi^2} \boldsymbol{h} \boldsymbol{A}^{\dagger\top} \boldsymbol{A}^\dagger \boldsymbol{h}^\top - \frac{2\eta^3 \|\boldsymbol{s}\|^2 \|\boldsymbol{h}\|^2 \tau_\varepsilon^2}{\gamma_2 \xi} \boldsymbol{k}^\top \boldsymbol{A}^\dagger \boldsymbol{h}^\top \right].$$

Using the squared norms from Lemmas 6, 7,

$$\mathbb{E}\left[ \frac{\xi^2}{\gamma_2^2} \boldsymbol{\varepsilon}^\top \boldsymbol{p}_2 \boldsymbol{q}_2^\top \boldsymbol{q}_2 \boldsymbol{p}_2^\top \boldsymbol{\varepsilon} \right] = \mathbb{E}\left[ \frac{\xi^2}{\gamma_2^2} \tau_\varepsilon^2 \|\boldsymbol{p}_2\|^2 \|\boldsymbol{q}_2\|^2 \right]$$

$$= \mathbb{E}\left[ \frac{\xi^2 \tau_\varepsilon^2}{\gamma_2^2} \left( \frac{\|\boldsymbol{h}\|^2}{\xi^2} \gamma_2 \right) \left( \frac{\eta^4 \|\boldsymbol{s}\|^4}{\xi^2} \boldsymbol{h} \boldsymbol{A}^{\dagger\top} \boldsymbol{A}^\dagger \boldsymbol{h}^\top + \frac{2\eta^3 \|\boldsymbol{s}\|^2}{\xi} \boldsymbol{k}^\top \boldsymbol{A}^\dagger \boldsymbol{h}^\top + \eta^2 \|\boldsymbol{k}\|^2 \right) \right]$$

$$= \mathbb{E}\left[ \tau_\varepsilon^2 \left( \frac{\eta^4 \|\boldsymbol{h}\|^2 \|\boldsymbol{s}\|^4}{\gamma_2 \xi^2} \boldsymbol{h} \boldsymbol{A}^{\dagger\top} \boldsymbol{A}^\dagger \boldsymbol{h}^\top + \frac{2\eta^3 \|\boldsymbol{h}\|^2 \|\boldsymbol{s}\|^2}{\gamma_2 \xi} \boldsymbol{k}^\top \boldsymbol{A}^\dagger \boldsymbol{h}^\top + \frac{\eta^2 \|\boldsymbol{h}\|^2 \|\boldsymbol{k}\|^2}{\gamma_2} \right) \right].$$

Similarly, we combine the coefficients: For the term $\boldsymbol{h} \boldsymbol{A}^{\dagger\top} \boldsymbol{A}^\dagger \boldsymbol{h}^\top$,

$$\tau_\varepsilon^2 \left( \frac{\eta^4 \|\boldsymbol{s}\|^4 \|\boldsymbol{h}\|^2}{\gamma_2 \xi^2} - \frac{2\eta^4 \|\boldsymbol{s}\|^4 \|\boldsymbol{h}\|^2}{\gamma_2 \xi^2} - \frac{2\eta^2 \|\boldsymbol{s}\|^2}{\gamma_2} + \frac{\eta^2 \|\boldsymbol{s}\|^2}{\xi^2} \right) = \tau_\varepsilon^2 \eta^2 \|\boldsymbol{s}\|^2 \left( \frac{\eta^2 \|\boldsymbol{s}\|^2 \|\boldsymbol{h}\|^2}{\gamma_2 \xi^2} - \frac{2\eta^2 \|\boldsymbol{s}\|^2 \|\boldsymbol{h}\|^2}{\gamma_2 \xi^2} - \frac{2}{\gamma_2} + \frac{1}{\xi^2} \right)$$

$$= \tau_\varepsilon^2 \eta^2 \|\boldsymbol{s}\|^2 \left( -\frac{\gamma_2 - \xi^2}{\gamma_2 \xi^2} - \frac{2}{\gamma_2} + \frac{1}{\xi^2} \right)$$

$$= \tau_\varepsilon^2 \eta^2 \|\boldsymbol{s}\|^2 \left( -\frac{\gamma_2 - \xi^2}{\gamma_2 \xi^2} - \frac{2\xi^2}{\gamma_2 \xi^2} + \frac{\gamma_2}{\gamma_2 \xi^2} \right)$$

$$= -\tau_\varepsilon^2 \frac{\eta^2 \|\boldsymbol{s}\|^2}{\gamma_2}.$$

For the term $\boldsymbol{k}^\top \boldsymbol{A}^\dagger \boldsymbol{h}^\top$,

$$\tau_\varepsilon^2 \left( \frac{2\eta^3 \|\boldsymbol{s}\|^2 \|\boldsymbol{h}\|^2}{\gamma_2 \xi} - \frac{2\eta^3 \|\boldsymbol{s}\|^2 \|\boldsymbol{h}\|^2}{\gamma_2 \xi} - \frac{2\eta\xi}{\gamma_2} \right) = -\tau_\varepsilon^2 \frac{2\eta\xi}{\gamma_2}.$$

Combining these terms together, we have:

$$\mathbb{E}\left[ \varepsilon^\top (\boldsymbol{Z} + \boldsymbol{A})^\dagger (\boldsymbol{Z} + \boldsymbol{A})^{\dagger\top} \varepsilon \right] = \mathbb{E}\left[ \varepsilon^\top \boldsymbol{A}^\dagger \boldsymbol{A}^{\dagger\top} \varepsilon - \frac{\eta^2 \|\boldsymbol{s}\|^2 \tau_\varepsilon^2}{\gamma_2} \boldsymbol{h} \boldsymbol{A}^{\dagger\top} \boldsymbol{A}^\dagger \boldsymbol{h}^\top - \frac{2\eta\xi\tau_\varepsilon^2}{\gamma_2} \boldsymbol{k}^\top \boldsymbol{A}^\dagger \boldsymbol{h}^\top + \frac{\eta^2 \|\boldsymbol{k}\|^2 \|\boldsymbol{h}\|^2}{\gamma_2} \right].$$

Similarly, replicating the proof with the $c > 1$ counterparts, we have the following:

$$\mathbb{E}\left[ \varepsilon^\top \boldsymbol{A}^\dagger \boldsymbol{A}^{\dagger\top} \varepsilon \right] = \tau_\varepsilon^2 \mathbb{E}\left[ Tr(\boldsymbol{A}^\dagger \boldsymbol{A}^{\dagger\top}) \right] = \tau_\varepsilon^2 n \mathbb{E}\left[ \frac{1}{\lambda} \right] = \tau_\varepsilon^2 \frac{d}{\rho^2(c-1)} + o\left( \frac{d}{\rho^2} \right).$$

$$\mathbb{E}\left[ \frac{\eta^2 \|\boldsymbol{s}\|^2}{\gamma_2} \boldsymbol{h} \boldsymbol{A}^{\dagger\top} \boldsymbol{A}^\dagger \boldsymbol{h}^\top \right] = \frac{\eta^2}{\eta^2 + \rho^2} \frac{c^2}{\rho^2(c-1)^2} + o(1) + O\left( \frac{1}{n} \right).$$

$$\mathbb{E}\left[ \frac{\eta\xi}{\gamma_2} \boldsymbol{k}^\top \boldsymbol{A}^\dagger \boldsymbol{h}^\top \right] = O\left( \frac{1}{n} \right).$$

$$\mathbb{E}\left[ \frac{\eta^2 \|\boldsymbol{k}\|^2 \|\boldsymbol{h}\|^2}{\gamma_2} \right] = \frac{\eta^2}{\eta^2 + \rho^2} \frac{c}{\rho^2(c-1)^2} + o(1) + O\left( \frac{1}{n} \right).$$

After simple algebra, the result follows.

$\square$

### E.5.7 TARGET ALIGNMENT: HELPER LEMMAS

**Lemma 30.** *In the same setting as Section 2, we have that*

$$\mathbb{E}\left[ \boldsymbol{\beta}_*^\top (\boldsymbol{Z} + \boldsymbol{A})^{\dagger\top} \boldsymbol{Z}^\top \boldsymbol{\beta}_* \right] = \begin{cases} \frac{\eta^2 c}{\rho^2 + \eta^2 c} (\boldsymbol{\beta}_*^\top \boldsymbol{u})^2 + o(1) + O\left( \frac{1}{n} \right) & c < 1 \\ \frac{\eta^2}{\eta^2 + \rho^2} (\boldsymbol{\beta}_*^\top \boldsymbol{u})^2 + o(1) + O\left( \frac{1}{n} \right) & c > 1 \end{cases}.$$

*Proof.* For $c < 1$, from Proposition 1, we get that

$$\boldsymbol{\beta}_*^\top (\boldsymbol{Z} + \boldsymbol{A})^{\dagger\top} \boldsymbol{Z}^\top \boldsymbol{\beta}_* = \frac{\eta\xi}{\gamma_1} \boldsymbol{\beta}_*^\top \boldsymbol{h}^\top \boldsymbol{u}^\top \boldsymbol{\beta}_* + \frac{\eta^2 \|\boldsymbol{t}\|^2}{\gamma_1} \boldsymbol{\beta}_*^\top \boldsymbol{A}^{\dagger\top} \boldsymbol{k} \boldsymbol{u}^\top \boldsymbol{\beta}_*.$$

To begin, we start estimating

$$\mathbb{E}\left[ \frac{\xi}{\eta} \boldsymbol{\beta}_*^\top \boldsymbol{h}^\top \boldsymbol{u}^\top \boldsymbol{\beta}_* \right].$$

Using our Spherical Hypercontractivity, we have that $\frac{\xi}{\eta}$ and $\boldsymbol{\beta}_*^\top \boldsymbol{h}^\top \boldsymbol{u}^\top \boldsymbol{\beta}_*$ satisfy the assumptions for Lemma 36. Then using Lemma 9 we have that

$$\mathbb{E}\left[ \frac{\xi}{\eta} \right] = \frac{1}{\eta} \quad \text{and} \quad \text{Var}\left( \frac{1}{\eta} \right) = O\left( \frac{1}{\rho^2 d} \right)$$

and Lemma 11, we have that

$$\mathbb{E}\left[ \boldsymbol{\beta}_*^\top \boldsymbol{h}^\top \boldsymbol{u}^\top \boldsymbol{\beta}_* \right] = 0 \quad \text{and} \quad \text{Var}\left( \boldsymbol{\beta}_*^\top \boldsymbol{h}^\top \boldsymbol{u}^\top \boldsymbol{\beta}_* \right) = O\left( \frac{1}{\rho^2 d} \right)$$

Thus, using Lemma 37, we have that

$$\mathbb{E}\left[ \frac{\xi}{\eta} \boldsymbol{\beta}_*^\top \boldsymbol{h}^\top \boldsymbol{u}^\top \boldsymbol{\beta}_* \right] = 0 + O\left( \frac{1}{\rho^2 d} \right)$$

and using Lemma 36, since all the means are $O(1)$, we have that

$$\text{Var}\left( \frac{\xi}{\eta} \boldsymbol{\beta}_*^\top \boldsymbol{h}^\top \boldsymbol{u}^\top \boldsymbol{\beta}_* \right) = O\left( \max\left( \text{Var}\left( \frac{\xi}{\eta} \right), \text{Var}\left( \boldsymbol{\beta}_*^\top \boldsymbol{h}^\top \boldsymbol{u}^\top \boldsymbol{\beta}_* \right) \right) \right) = O\left( \frac{1}{\rho^2 n} \right).$$

Then Lemma 19 gives mean and variance of $\frac{\eta^2}{\gamma_i}$. Since $\frac{\eta^2}{\gamma_i}$ does not satisfy the higher moment bound, and cannot be directly included in the product, we can include it via the classical bound:

$$\mathbb{E}\left[\frac{\eta\xi}{\gamma_1}\boldsymbol{\beta}_*^\top \boldsymbol{h}^\top \boldsymbol{u}^\top \boldsymbol{\beta}_*\right] = \mathbb{E}\left[\frac{\eta^2}{\gamma_1}\right]\mathbb{E}\left[\frac{\xi}{\eta}\boldsymbol{\beta}_*^\top \boldsymbol{h}^\top \boldsymbol{u}^\top \boldsymbol{\beta}_*\right] + \sqrt{\mathrm{Var}\left(\frac{\xi}{\eta}\boldsymbol{\beta}_*^\top \boldsymbol{h}^\top \boldsymbol{u}^\top \boldsymbol{\beta}_*\right)\mathrm{Var}\left(\frac{\eta^2}{\gamma_1}\right)} = O\left(\frac{1}{n}\right).$$

(18)

For the second term, we begin with

$$\mathbb{E}\left[\|\boldsymbol{t}\|^2 \boldsymbol{\beta}_*^\top \boldsymbol{A}^{\dagger\top}\boldsymbol{k}\boldsymbol{u}^\top \boldsymbol{\beta}_*\right].$$

Lemma 9 tells us that

$$\mathbb{E}[\|\boldsymbol{t}\|^2] = 1 - c \quad \text{and} \quad \mathrm{Var}\left(\|\boldsymbol{t}\|^2\right) = O\left(\frac{1}{n}\right)$$

and Lemma 10 tells us

$$\mathbb{E}\left[\boldsymbol{\beta}_*^\top \boldsymbol{A}^{\dagger\top}\boldsymbol{k}\boldsymbol{u}^\top \boldsymbol{\beta}_*\right] = \frac{1}{\rho^2}\frac{c}{1-c}(\boldsymbol{\beta}_*^\top \boldsymbol{u})^2 + o\left(\frac{1}{\rho^2}\right) \quad \text{and} \quad \mathrm{Var}\left(\boldsymbol{\beta}_*^\top \boldsymbol{A}^{\dagger\top}\boldsymbol{k}\boldsymbol{u}^\top \boldsymbol{\beta}_*\right) = O\left(\frac{1}{\rho^4 d}\right).$$

Thus using Lemmas 37 and Lemma 36, we get that

$$\mathbb{E}\left[\|\boldsymbol{t}\|^2 \boldsymbol{\beta}_*^\top \boldsymbol{A}^{\dagger\top}\boldsymbol{k}\boldsymbol{u}^\top \boldsymbol{\beta}_*\right] = (\boldsymbol{\beta}_*^\top \boldsymbol{u})^2 \frac{c}{\rho^2} + o\left(\frac{1}{\rho^2}\right) + O\left(\frac{1}{n}\right) \quad \text{and} \quad \mathrm{Var}\left(\|\boldsymbol{t}\|^2 \boldsymbol{\beta}_*^\top \boldsymbol{A}^{\dagger\top}\boldsymbol{k}\boldsymbol{u}^\top \boldsymbol{\beta}_*\right) = O\left(\frac{1}{n}\right)$$

Recalling the mean and variance for $\frac{\eta^2}{\gamma_1}$ from 19, we have that

$$\mathbb{E}\left[\frac{\eta^2\|\boldsymbol{t}\|^2}{\gamma_1}\boldsymbol{\beta}_*^\top \boldsymbol{A}^{\dagger\top}\boldsymbol{k}\boldsymbol{u}^\top \boldsymbol{\beta}_*\right] = \mathbb{E}\left[\frac{\eta^2}{\gamma_1}\right]\mathbb{E}\left[\|\boldsymbol{t}\|^2\boldsymbol{\beta}_*^\top \boldsymbol{A}^{\dagger\top}\boldsymbol{k}\boldsymbol{u}^\top \boldsymbol{\beta}_*\right] + \sqrt{O\left(\frac{1}{n}\right)\mathrm{Var}\left(\frac{\eta^2}{\gamma_1}\right)}$$

$$= \left(\frac{\rho^2\eta^2}{\eta^2 c + \rho^2} + o\left(\frac{1}{\rho^2}\right)\right)\left((\boldsymbol{\beta}_*^\top \boldsymbol{u})^2\frac{c}{\rho^2} + o\left(\frac{1}{\rho^2}\right) + O\left(\frac{1}{n}\right)\right) + O\left(\frac{1}{n}\right)$$

$$= (\boldsymbol{\beta}_*^\top \boldsymbol{u})^2\frac{\eta^2 c}{\eta^2 c + \rho^2} + o(1) + O\left(\frac{1}{n}\right).$$

(19)

Combining these two terms yields the first result.

Similarly, for $c > 1$, Proposition 1 gives the expansion:

$$\boldsymbol{\beta}_*^\top (\boldsymbol{Z} + \boldsymbol{A})^{\dagger\top}\boldsymbol{Z}^\top \boldsymbol{\beta}_* = \boldsymbol{\beta}_*^\top\left(\frac{\eta\xi}{\gamma_2}\boldsymbol{u}\boldsymbol{h} + \frac{\eta^2\|\boldsymbol{h}\|^2}{\gamma_2}\boldsymbol{u}\boldsymbol{s}^\top\right)^\top \boldsymbol{\beta}_* = \frac{\eta\xi}{\gamma_2}\boldsymbol{\beta}_*^\top \boldsymbol{h}^\top \boldsymbol{u}^\top \boldsymbol{\beta}_* + \frac{\eta^2\|\boldsymbol{h}\|^2}{\gamma_2}\boldsymbol{\beta}_*^\top \boldsymbol{s}\boldsymbol{u}^\top \boldsymbol{\beta}_*.$$

For the first term, we begin with

$$\mathbb{E}\left[\frac{\xi}{\eta}\boldsymbol{\beta}_*^\top \boldsymbol{h}^\top \boldsymbol{u}^\top \boldsymbol{\beta}_*\right].$$

Recalling form Lemma 11, we see that

$$\mathbb{E}\left[\boldsymbol{\beta}_*^\top \boldsymbol{h}^\top \boldsymbol{u}^\top \boldsymbol{\beta}_*\right] = 0 \quad \text{and} \quad \mathrm{Var}\left(\boldsymbol{\beta}_*^\top \boldsymbol{h}^\top \boldsymbol{u}^\top \boldsymbol{\beta}_*\right) = O\left(\frac{1}{\rho^2 d}\right).$$

Thus again using Lemma 36 and Lemma 37, we see that

$$\mathbb{E}\left[\frac{\xi}{\eta}\boldsymbol{\beta}_*^\top \boldsymbol{h}^\top \boldsymbol{u}^\top \boldsymbol{\beta}_*\right] = 0 + O\left(\frac{1}{\rho^2 d}\right) \quad \text{and} \quad \mathrm{Var}\left(\frac{\xi}{\eta}\boldsymbol{\beta}_*^\top \boldsymbol{h}^\top \boldsymbol{u}^\top \boldsymbol{\beta}_*\right) = O\left(\frac{1}{\rho^2 d}\right).$$

Next using the standard covariance bound on the expectation of the product. We see that

$$\mathbb{E}\left[\frac{\eta\xi}{\gamma_1}\boldsymbol{\beta}_*^\top \boldsymbol{h}^\top \boldsymbol{u}^\top \boldsymbol{\beta}_*\right] = 0 + O\left(\frac{1}{\rho^2 d}\right) + O\left(\frac{1}{n}\right) = O\left(\frac{1}{n}\right).$$

For the second term, we begin with

$$\mathbb{E}\left[\|\boldsymbol{h}\|^2 \boldsymbol{\beta}_*^\top \boldsymbol{s}\boldsymbol{u}\boldsymbol{\beta}_*\right].$$

Recall from Lemma 9 we have that

$$\mathbb{E}[\|\boldsymbol{h}\|^2] = \frac{1}{\rho^2}\frac{c}{c-1} + o\left(\frac{1}{\rho^2}\right) \quad \text{and} \quad \operatorname{Var}(\|\boldsymbol{h}\|^2) = O\left(\frac{1}{\rho^4 n}\right)$$

and from Lemma 10

$$\mathbb{E}\left[\boldsymbol{\beta}_*^\top \boldsymbol{s}\boldsymbol{u}\boldsymbol{\beta}_*\right] = \left(1 - \frac{1}{c}\right)(\boldsymbol{\beta}_*^\top \boldsymbol{u})^2 \quad \text{and} \quad \operatorname{Var}\left(\boldsymbol{\beta}_*^\top \boldsymbol{s}\boldsymbol{u}\boldsymbol{\beta}_*\right) = O\left(\frac{1}{d}\right).$$

Thus using Lemma 36 and Lemma 37, we get that

$$\mathbb{E}\left[\|\boldsymbol{h}\|^2 \boldsymbol{\beta}_*^\top \boldsymbol{s}\boldsymbol{u}\boldsymbol{\beta}_*\right] = \frac{(\boldsymbol{\beta}_*^\top \boldsymbol{u})^2}{\rho^2} + o\left(\frac{1}{\rho^2}\right) + O\left(\frac{1}{d}\right) \quad \text{and} \quad \operatorname{Var}\left(\|\boldsymbol{h}\|^2 \boldsymbol{\beta}_*^\top \boldsymbol{s}\boldsymbol{u}\boldsymbol{\beta}_*\right) = O\left(\frac{1}{d}\right).$$

Recalling the mean and variance for $\frac{\eta^2}{\gamma_2}$ from Lemma 19 and using the classical covariance bound for the expectation of the product, we get that

$$
\begin{aligned}
\mathbb{E}\left[\frac{\eta^2\|\boldsymbol{h}\|^2}{\gamma_2}\boldsymbol{\beta}_*^\top \boldsymbol{s}\boldsymbol{u}^\top \boldsymbol{\beta}_*\right] &= \mathbb{E}\left[\frac{\eta^2}{\gamma_2}\right]\mathbb{E}\left[\|\boldsymbol{h}\|^2 \boldsymbol{\beta}_*^\top \boldsymbol{s}\boldsymbol{u}^\top \boldsymbol{\beta}_*\right] + \sqrt{O\left(\frac{1}{n}\right)\operatorname{Var}\left(\frac{\eta^2}{\gamma_2}\right)} \\
&= \left(\frac{\rho^2\eta^2}{\eta^2 + \rho^2} + o\left(\frac{1}{\rho^2}\right)\right)\left(\frac{(\boldsymbol{\beta}_*^\top \boldsymbol{u})^2}{\rho^2} + o\left(\frac{1}{\rho^2}\right) + O\left(\frac{1}{d}\right)\right) + O\left(\frac{1}{n}\right) \\
&= \frac{\eta^2}{\eta^2 + \rho^2}(\boldsymbol{\beta}_*^\top \boldsymbol{u})^2 + o(1) + O\left(\frac{1}{n}\right).
\end{aligned}
$$

Then adding the two together, we get the result for $c > 1$ as well. $\qquad\square$

**Lemma 31.** *In the same setting as Section 2, we have that, for $c < 1$*

$$\mathbb{E}\left[\boldsymbol{\beta}_*^\top (\boldsymbol{Z} + \boldsymbol{A})^{\dagger\top}\boldsymbol{A}^\top \boldsymbol{\beta}_*\right] = \|\boldsymbol{\beta}_*\|^2 - \frac{\eta^2 c}{\rho^2 + \eta^2 c}(\boldsymbol{\beta}_*^\top \boldsymbol{u})^2 + o\left(\frac{1}{\rho^2}\right) + O\left(\frac{1}{n}\right).$$

*and for $c > 1$*

$$\mathbb{E}\left[\boldsymbol{\beta}_*^\top (\boldsymbol{Z} + \boldsymbol{A})^{\dagger\top}\boldsymbol{A}^\top \boldsymbol{\beta}_*\right] = \frac{1}{c}\|\boldsymbol{\beta}_*\|^2 - \frac{\eta^2}{\eta^2 + \rho^2}\left(\frac{\|\boldsymbol{\beta}_*\|^2}{d} + \frac{1}{c}(\boldsymbol{\beta}_*^\top \boldsymbol{u})^2\right) + o(1) + O\left(\frac{1}{n}\right).$$

*Proof.* For $c < 1$, using the expectation from Lemma 30, we get

$$
\begin{aligned}
\mathbb{E}\left[\boldsymbol{\beta}_*^\top (\boldsymbol{Z} + \boldsymbol{A})^{\dagger\top}\boldsymbol{A}^\top \boldsymbol{\beta}_*\right] &= \mathbb{E}\left[\boldsymbol{\beta}_*^\top \left(\boldsymbol{I} - \boldsymbol{Z}(\boldsymbol{Z} + \boldsymbol{A})^\dagger\right)^\top \boldsymbol{\beta}_*\right] \\
&= \|\boldsymbol{\beta}_*\|^2 - \frac{\eta^2 c}{\rho^2 + \eta^2 c}(\boldsymbol{\beta}_*^\top \boldsymbol{u})^2 + o\left(1\right) + O\left(\frac{1}{n}\right).
\end{aligned}
$$

For $c > 1$, using Lemma 2, we get

$$\boldsymbol{\beta}_*^\top (\boldsymbol{Z} + \boldsymbol{A})^{\dagger\top}\boldsymbol{A}^\top \boldsymbol{\beta}_* = \boldsymbol{\beta}_*^\top \left(\boldsymbol{A}\boldsymbol{A}^\dagger + \frac{\eta\xi}{\gamma_2}\boldsymbol{h}^\top \boldsymbol{s}^\top - \frac{\eta^2\|\boldsymbol{s}\|^2}{\gamma_2}\boldsymbol{h}^\top \boldsymbol{h} - \frac{\eta^2\|\boldsymbol{h}\|^2}{\gamma_2}\boldsymbol{A}\boldsymbol{A}^\dagger \boldsymbol{u}\boldsymbol{s}^\top - \frac{\eta\xi}{\gamma_2}\boldsymbol{A}\boldsymbol{A}^\dagger \boldsymbol{u}\boldsymbol{h}\right)^\top \boldsymbol{\beta}_*.$$

We then compute the expectation of each term above. To begin, we have that

$$\mathbb{E}\left[\boldsymbol{\beta}_*^\top \boldsymbol{A}\boldsymbol{A}^\dagger \boldsymbol{\beta}_*\right] = \frac{1}{c}\|\boldsymbol{\beta}_*\|^2 \quad \text{by Equation 13.}$$

Next, we recall from Lemma 11 that

$$\mathbb{E}[\boldsymbol{\beta}_*^\top \boldsymbol{h}^\top \boldsymbol{s}^\top \boldsymbol{\beta}_*] = 0 \text{ and } \operatorname{Var}(\boldsymbol{\beta}_*^\top \boldsymbol{h}^\top \boldsymbol{s}^\top \boldsymbol{\beta}_*) = O\left(\frac{1}{\rho^2 d}\right).$$

and from Lemma 9 that

$$\mathbb{E}\left[\frac{\xi}{\eta}\right] = \frac{1}{\eta} + o\left(\frac{1}{\rho^2}\right) \quad \text{and} \quad \text{Var}\left(\frac{\xi}{\eta}\right) = O\left(\frac{1}{\rho^2 n}\right)$$

Thus, using Lemmas 36 and Lemma 37, we have that

$$\mathbb{E}\left[\frac{\xi}{\eta}\boldsymbol{\beta}_*^\top \boldsymbol{h}^\top \boldsymbol{s}^\top \boldsymbol{\beta}_*\right] = O\left(\frac{1}{\rho^2 n}\right) \quad \text{and} \quad \text{Var}\left(\frac{\xi}{\eta}\boldsymbol{\beta}_*^\top \boldsymbol{h}^\top \boldsymbol{s}^\top \boldsymbol{\beta}_*\right) = O\left(\frac{1}{\rho^2 n}\right).$$

Then recalling the mean and variance of $\eta^2/\gamma_2$ from 19, using the standard covariance bound on the difference between the product of the expectation and the expectation of the product, we get that

$$\mathbb{E}\left[\frac{\eta\xi}{\gamma_2}\boldsymbol{\beta}_*^\top \boldsymbol{h}^\top \boldsymbol{s}^\top \boldsymbol{\beta}_*\right] = O\left(\frac{1}{n}\right) \quad \text{and} \quad \mathbb{E}\left[\frac{\eta\xi}{\gamma_2}\boldsymbol{\beta}_*^\top \boldsymbol{A}\boldsymbol{A}^\dagger \boldsymbol{u}\boldsymbol{h}\boldsymbol{\beta}_*\right] = O\left(\frac{1}{n}\right).$$

Furthermore, for the next three terms, recall from Lemma 10 that

$$\mathbb{E}[\boldsymbol{\beta}_*^\top \boldsymbol{h}^\top \boldsymbol{h}\boldsymbol{\beta}_*] = \frac{\|\boldsymbol{\beta}_*\|^2}{d}\frac{c}{\rho^2(c-1)} + o\left(\frac{1}{\rho^2 d}\right) \quad \text{and} \quad \text{Var}\left(\boldsymbol{\beta}_*^\top \boldsymbol{h}^\top \boldsymbol{h}\boldsymbol{\beta}_*\right) = O\left(\frac{1}{\rho^2 d^2}\right)$$

and

$$\mathbb{E}\left[\boldsymbol{\beta}_*^\top \boldsymbol{A}\boldsymbol{A}^\dagger \boldsymbol{u}\boldsymbol{s}^\top \boldsymbol{\beta}_*\right] = \frac{c-1}{c^2}(\boldsymbol{\beta}_*^\top \boldsymbol{u})^2 + o(1) \quad \text{and} \quad \text{Var}\left(\boldsymbol{\beta}_*^\top \boldsymbol{A}\boldsymbol{A}^\dagger \boldsymbol{u}\boldsymbol{s}^\top \boldsymbol{\beta}_*\right) = O\left(\boldsymbol{\beta}_*^\top \boldsymbol{A}\boldsymbol{A}^\dagger \boldsymbol{u}\boldsymbol{s}^\top \boldsymbol{\beta}_*\frac{1}{d}\right)$$

and from Lemma 11

$$\mathbb{E}[\boldsymbol{\beta}_*^\top \boldsymbol{A}\boldsymbol{A}^\dagger \boldsymbol{u}\boldsymbol{h}\boldsymbol{\beta}_*] = 0 \quad \text{and} \quad \text{Var}\left(\boldsymbol{\beta}_*^\top \boldsymbol{A}\boldsymbol{A}^\dagger \boldsymbol{u}\boldsymbol{h}\boldsymbol{\beta}_*\right) = O\left(\frac{1}{\rho^2 d^2}\right).$$

Then recalling from Lemma 9, we have that

$$\mathbb{E}[\|\boldsymbol{s}\|^2] = 1 - \frac{1}{c} \quad \text{and} \quad \text{Var}(\|\boldsymbol{s}\|^2) = O\left(\frac{1}{d}\right).$$

Then using Lemma 36 and Lemma 37, we have that for third term

$$\mathbb{E}[\|\boldsymbol{s}\|^2\boldsymbol{\beta}_*^\top \boldsymbol{h}^\top \boldsymbol{h}\boldsymbol{\beta}_*] = \frac{1}{\rho^2 d}\|\boldsymbol{\beta}_*\|^2 + o\left(\frac{1}{\rho^2 d}\right) + O\left(\frac{1}{d}\right) \quad \text{and} \quad \text{Var}\left(\|\boldsymbol{s}\|^2\boldsymbol{\beta}_*^\top \boldsymbol{h}^\top \boldsymbol{h}\boldsymbol{\beta}_*\right) = O\left(\frac{1}{d}\right)$$

for the fourth term

$$\mathbb{E}[\|\boldsymbol{h}\|^2\boldsymbol{\beta}_*^\top \boldsymbol{A}\boldsymbol{A}^\dagger \boldsymbol{u}\boldsymbol{s}^\top] = \left(\frac{1}{\rho^2}\frac{c}{c-1} + o\left(\frac{1}{\rho^2}\right)\right)\left(\frac{c-1}{c^2}(\boldsymbol{\beta}_*^\top \boldsymbol{u})^2 + o(1)\right) + O\left(\frac{1}{\rho^2 d}\right)$$

$$= \frac{(\boldsymbol{\beta}_*^\top \boldsymbol{u})^2}{\rho^2 c} + o(1) + O\left(\frac{1}{\rho^2 d}\right)$$

with variance

$$\text{Var}(\|\boldsymbol{h}\|^2\boldsymbol{\beta}_*^\top \boldsymbol{A}\boldsymbol{A}^\dagger \boldsymbol{u}\boldsymbol{s}^\top) = O\left(\frac{1}{\rho^2 d}\right).$$

For the first term, we have that

$$\mathbb{E}\left[\frac{\xi}{\eta}\boldsymbol{\beta}_*^\top \boldsymbol{A}\boldsymbol{A}^\dagger \boldsymbol{u}\boldsymbol{h}\boldsymbol{\beta}_*\right] = 0 + O\left(\frac{1}{\rho^2 d}\right) \quad \text{and} \quad \text{Var}\left(\frac{\xi}{\eta}\boldsymbol{\beta}_*^\top \boldsymbol{A}\boldsymbol{A}^\dagger \boldsymbol{u}\boldsymbol{h}\boldsymbol{\beta}_*\right) = O\left(\frac{1}{\rho^2 d}\right)$$

Adding the last three terms and using Lemma 34 twice, we get that

$$\mathbb{E}\left[\boldsymbol{\beta}_*^\top \left(\|\boldsymbol{s}\|^2\boldsymbol{h}^\top \boldsymbol{h} + \|\boldsymbol{h}\|^2\top\boldsymbol{A}\boldsymbol{A}^\dagger \boldsymbol{u}\boldsymbol{s}^\top + \frac{\xi}{\eta}\boldsymbol{A}\boldsymbol{A}^\dagger \boldsymbol{u}\boldsymbol{h}\right)\boldsymbol{\beta}_*\right] = \frac{1}{\rho^2 d}\|\boldsymbol{\beta}_*\|^2 + \frac{(\boldsymbol{\beta}_*^\top \boldsymbol{u})^2}{\rho^2 c} + 0 + o(1) + O\left(\frac{1}{d}\right)$$

With variance

$$\text{Var}\left(\boldsymbol{\beta}_*^\top \left(\|\boldsymbol{s}\|^2\boldsymbol{h}^\top \boldsymbol{h} + \|\boldsymbol{h}\|^2\top\boldsymbol{A}\boldsymbol{A}^\dagger \boldsymbol{u}\boldsymbol{s}^\top + \frac{\xi}{\eta}\boldsymbol{A}\boldsymbol{A}^\dagger \boldsymbol{u}\boldsymbol{h}\right)\boldsymbol{\beta}_*\right) = O\left(\frac{1}{d}\right)$$

Then recalling the mean and variance of $\eta^2/\gamma_2$ from Lemma 19, and using the covariance bound for the expectation of products, we get that

$$\mathbb{E}\left[\frac{\eta^2}{\gamma_2}\boldsymbol{\beta}_*^\top \left(\|\boldsymbol{s}\|^2\boldsymbol{h}^\top \boldsymbol{h} + \|\boldsymbol{h}\|^2\top\boldsymbol{A}\boldsymbol{A}^\dagger \boldsymbol{u}\boldsymbol{s}^\top + \frac{\xi}{\eta}\boldsymbol{A}\boldsymbol{A}^\dagger \boldsymbol{u}\boldsymbol{h}\right)\boldsymbol{\beta}_*\right] = \frac{\eta^2}{\eta^2 + \rho^2}\left(\frac{\|\boldsymbol{\beta}_*\|^2}{d} + \frac{1}{c}(\boldsymbol{\beta}_*^\top \boldsymbol{u})^2\right) + o(1) + O\left(\frac{1}{n}\right).$$

Adding all five terms, we get that

$$\mathbb{E}\left[\boldsymbol{\beta}_*^\top (\boldsymbol{Z} + \boldsymbol{A})^{\dagger\top}\boldsymbol{A}^\top \boldsymbol{\beta}_*\right] = \frac{1}{c}\|\boldsymbol{\beta}_*\|^2 - \frac{\eta^2}{\eta^2 + \rho^2}\left(\frac{\|\boldsymbol{\beta}_*\|^2}{d} + \frac{1}{c}(\boldsymbol{\beta}_*^\top \boldsymbol{u})^2\right) + o(1) + O\left(\frac{1}{n}\right).$$

$$\square$$

### E.6 STEP 5: UPSCALING AND ASYMPTOTIC RISK FORMULAS

In the previous step we derived downscaled expressions for the four constituent terms of the risk: **Bias, Variance, Data Noise, and Target Alignment**. We stop our abuse of notation and are explicit again about douwnscaled vs. upscaled.

**Bias (downscaled).** For $c < 1$, the bias term is

$$\frac{\tilde{\eta}^2}{\tilde{n}} \left( \left[ (\tilde{\alpha}_Z - \alpha_Z) + \frac{\rho^2}{\eta^2 c + \rho^2}(\alpha_Z - \alpha_A) \right]^2 (\boldsymbol{\beta}_*^\top \boldsymbol{u})^2 + \tau_{\varepsilon,r}^2 \frac{c}{1-c}\frac{1}{\eta^2 c + \rho^2} \right) + o\left(\frac{1}{\tilde{n}}\right) + o\left(\frac{1}{n}\right).$$

For $c > 1$, the bias term is

$$\frac{\tilde{\eta}^2}{\tilde{n}} \left[ (\boldsymbol{\beta}_*^\top \boldsymbol{u})^2 \left( (\tilde{\alpha}_Z - \alpha_Z) + \frac{\rho^2}{\eta^2 + \rho^2}\left(\alpha_Z - \frac{\alpha_A}{c}\right) \right)^2 + \alpha_A^2 \frac{\|\boldsymbol{\beta}_*\|^2}{d}\left(\frac{c-1}{c}\right)\frac{\eta^2 \rho^2}{(\eta^2 + \rho^2)^2} \right.$$
$$\left. + \frac{\tau_{\varepsilon,r}^2}{c-1}\frac{\eta^2 c + \rho^2}{(\eta^2 + \rho^2)^2} \right] . + o\left(\frac{1}{\tilde{n}}\right) + o\left(\frac{1}{n}\right)$$

**Variance (downscaled).** For $c < 1$, the variance term is

$$\frac{\tilde{\rho}^2}{d} \left[ \alpha_A^2 \|\boldsymbol{\beta}_*\|^2 + (\boldsymbol{\beta}_*^\top \boldsymbol{u})^2 \left( (\alpha_Z - \alpha_A)^2 \frac{\eta^2(\eta^2 + \rho^2)}{(\eta^2 c + \rho^2)^2}\frac{c^2}{1-c} + 2\alpha_A(\alpha_Z - \alpha_A)\frac{\eta^2 c}{\eta^2 c + \rho^2} \right) \right.$$
$$\left. + \tau_{\varepsilon,r}^2 \left( \frac{c}{1-c}\frac{d}{\rho^2} - \frac{\eta^2}{\rho^2(\eta^2 c + \rho^2)}\frac{c^2}{1-c} \right) \right].$$

For $c > 1$, the variance term is

$$\frac{\tilde{\rho}^2}{d} \left[ \|\boldsymbol{\beta}_*\|^2 \left( \frac{\alpha_A^2}{c} - \frac{\alpha_A^2}{d}\frac{\eta^2}{\eta^2 + \rho^2} \right) + (\boldsymbol{\beta}_*^\top \boldsymbol{u})^2 \frac{c}{c-1}\frac{\eta^2}{\eta^2 + \rho^2}\left(\alpha_Z - \frac{\alpha_A}{c}\right)^2 + \tau_{\varepsilon,r}^2 \left( \frac{d}{\rho^2}\frac{1}{c-1} - \frac{\eta^2}{\rho^2(\eta^2 + \rho^2)}\frac{c}{c-1} \right) \right].$$

**Data noise (downscaled).** The data noise term is

$$\frac{\tilde{\alpha}_A^2 \tilde{\rho}^2}{d} \|\boldsymbol{\beta}_*\|^2.$$

**Target alignment (downscaled).** For $c < 1$, the alignment term is

$$-\frac{2\tilde{\alpha}_A \tilde{\rho}^2}{d}\left( \alpha_A \|\boldsymbol{\beta}_*\|^2 + (\alpha_Z - \alpha_A)(\boldsymbol{\beta}_*^\top \boldsymbol{u})^2 \frac{\eta^2 c}{\rho^2 + \eta^2 c} \right).$$

For $c > 1$, the alignment term is

$$-\frac{2\tilde{\alpha}_A \tilde{\rho}^2}{d}\left( \frac{\alpha_A}{c}\|\boldsymbol{\beta}_*\|^2 - \frac{\alpha_A}{d}\frac{\eta^2}{\eta^2 + \rho^2}\|\boldsymbol{\beta}_*\|^2 + \left(\alpha_Z - \frac{\alpha_A}{c}\right)\frac{\eta^2}{\eta^2 + \rho^2}(\boldsymbol{\beta}_*^\top \boldsymbol{u})^2 \right).$$

These formulas are expressed in terms of the concentrated building blocks, but still at the "microscopic" scale in which $\eta$ is $O(\sqrt{d})$, $\rho = \Theta(1)$, and $\tau_{\varepsilon,r}^2 = O(1/d)$.

In this section we return to the macroscopic, or upscaled, version of the problem. Specifically, we multiply each term by $d$ and reparametrize according to

$$\theta^2 = \frac{d}{n}\eta^2, \qquad \tilde{\theta}^2 = \frac{d}{\tilde{n}}\tilde{\eta}^2, \qquad \tau_\varepsilon^2 = d\,\tau_{\varepsilon,r}^2,$$

while keeping $\rho, \tilde{\rho}$ fixed. This normalization ensures that the effective spike strength $\theta$, isotropic noise level $\rho$, and label noise $\tau_{\varepsilon,r}$ are all of order one. In this scaling, the risk is $d$ times larger than in the downscaled representation, and the resulting formulas cleanly separate the contributions of the four terms.

The terms change as follows

**Front factors (after multiplying by $d$).**

$$\frac{\tilde{\eta}^2}{\tilde{n}} \xrightarrow{\times d} \tilde{\theta}^2, \qquad \frac{\tilde{\rho}^2}{d} \xrightarrow{\times d} \tilde{\rho}^2, \qquad \frac{\tilde{\alpha}_A^2 \tilde{\rho}^2}{d} \xrightarrow{\times d} \tilde{\alpha}_A^2 \tilde{\rho}^2, \qquad d\tau_{\varepsilon,r}^2 \to \tau_{\varepsilon}^2. \tag{20}$$

**Denominator identities.**

$$\eta^2 c + \rho^2 = \theta^2 + \rho^2, \qquad \eta^2 + \rho^2 = \frac{\theta^2 + c\rho^2}{c}. \tag{21}$$

**Frequently used ratios and their upscaled forms.**

$$\frac{\rho^2}{\eta^2 c + \rho^2} = \frac{\rho^2}{\theta^2 + \rho^2}, \tag{22}$$

$$\frac{\eta^2 c}{\eta^2 c + \rho^2} = \frac{\theta^2}{\theta^2 + \rho^2}, \tag{23}$$

$$\frac{\eta^2}{\eta^2 + \rho^2} = \frac{\theta^2}{\theta^2 + c\rho^2}, \tag{24}$$

$$\frac{\rho^2}{\eta^2 + \rho^2} = \frac{c\rho^2}{\theta^2 + c\rho^2}, \tag{25}$$

$$\frac{\eta^2 \rho^2}{(\eta^2 + \rho^2)^2} = \frac{\theta^2 \rho^2}{(\theta^2 + c\rho^2)^2} c, \tag{26}$$

$$\frac{\eta^2(\eta^2 + \rho^2)}{(\eta^2 c + \rho^2)^2} \frac{c^2}{1-c} = \frac{\theta^2(\theta^2 + c\rho^2)}{(\theta^2 + \rho^2)^2} \frac{1}{1-c}. \tag{27}$$

**Noise terms with aspect-ratio factors.** After multiplying by $d$ and substituting $\tau_{\varepsilon}^2 = d\tau_{\varepsilon,r}^2$:

$$\tau_{\varepsilon,r}^2 \left( \frac{c}{1-c} \frac{d}{\rho^2} - \frac{\eta^2}{\rho^2(\eta^2 c + \rho^2)} \frac{c^2}{1-c} \right) \longrightarrow \tau_{\varepsilon}^2 \left( \frac{1}{\rho^2} \frac{c}{1-c} - \frac{\theta^2}{\rho^2(\theta^2 + \rho^2)} \frac{c}{1-c} \right), \tag{28}$$

$$\tau_{\varepsilon,r}^2 \left( \frac{d}{\rho^2} \frac{1}{c-1} - \frac{\eta^2}{\rho^2(\eta^2 + \rho^2)} \frac{c}{c-1} \right) \longrightarrow \tau_{\varepsilon}^2 \left( \frac{1}{\rho^2} \frac{1}{c-1} - \frac{\theta^2}{\rho^2(\theta^2 + c\rho^2)} \frac{c}{c-1} \right). \tag{29}$$

**Alignment-specific identities.**

$$\frac{\eta^2 c}{\rho^2 + \eta^2 c} = \frac{\theta^2}{\rho^2 + \theta^2}, \qquad \frac{\eta^2}{\eta^2 + \rho^2} = \frac{\theta^2}{\theta^2 + c\rho^2}. \tag{30}$$

We now state the explicit upscaled limits for each component. As before, we present results separately in the underparametrized regime ($c < 1$) and the overparametrized regime ($c > 1$). Each term has a little $o(1)$ error term.

**Bias.** For $c < 1$, the bias contribution is

$$\tilde{\theta}^2 \left( \left[ (\tilde{\alpha}_Z - \alpha_Z) + \frac{\rho^2}{\theta^2 + \rho^2}(\alpha_Z - \alpha_A) \right]^2 (\boldsymbol{\beta}_*^\top \boldsymbol{u})^2 + \frac{\tau_{\varepsilon}^2}{d} \frac{c}{1-c} \frac{1}{\theta^2 + \rho^2} \right).$$

For $c > 1$, the bias is

$$\tilde{\theta}^2 \left[ (\boldsymbol{\beta}_*^\top \boldsymbol{u})^2 \left( (\tilde{\alpha}_Z - \alpha_Z) + \frac{\rho^2}{\frac{\theta^2}{c} + \rho^2} \left( \alpha_Z - \frac{\alpha_A}{c} \right) \right)^2 + \alpha_A^2 \frac{\|\boldsymbol{\beta}_*\|^2}{d} \left( \frac{c-1}{c} \right) \frac{\frac{\theta^2}{c}\rho^2}{\left( \frac{\theta^2}{c} + \rho^2 \right)^2} + \frac{\tau_{\varepsilon}^2}{d} \frac{1}{c-1} \frac{\theta^2 + \rho^2}{\left( \frac{\theta^2}{c} + \rho^2 \right)^2} \right].$$

**Variance.** For $c < 1$, the variance contribution is

$$\tilde{\rho}^2 \left[ \alpha_A^2 \|\boldsymbol{\beta}_*\|^2 + (\boldsymbol{\beta}_*^\top \boldsymbol{u})^2 \left( (\alpha_Z - \alpha_A)^2 \frac{\theta^2(\theta^2 + c\rho^2)}{(\theta^2 + \rho^2)^2} \frac{1}{1-c} + 2\alpha_A(\alpha_Z - \alpha_A) \frac{\theta^2}{\theta^2 + \rho^2} \right) \right.$$
$$\left. + \tau_\varepsilon^2 \left( \frac{1}{\rho^2} \frac{c}{1-c} - \frac{1}{d} \frac{\theta^2}{\rho^2(\theta^2 + \rho^2)} \cdot \frac{c}{1-c} \right) \right].$$

For $c > 1$, the variance is

$$\tilde{\rho}^2 \left[ \|\boldsymbol{\beta}_*\|^2 \left( \frac{\alpha_A^2}{c} - \frac{\alpha_A^2}{d} \frac{\theta^2}{\theta^2 + c\rho^2} \right) + (\boldsymbol{\beta}_*^\top \boldsymbol{u})^2 \frac{c}{c-1} \frac{\theta^2}{\theta^2 + c\rho^2} \left( \alpha_Z - \frac{\alpha_A}{c} \right)^2 + \tau_\varepsilon^2 \left( \frac{1}{\rho^2} \frac{1}{c-1} - \frac{1}{d} \frac{\theta^2}{\rho^2(\theta^2 + c\rho^2)} \cdot \frac{c}{c-1} \right) \right].$$

**Data Noise.** The data noise term is independent of $c$:

$$\tilde{\alpha}_A^2 \, \tilde{\rho}^2 \, \|\boldsymbol{\beta}_*\|^2.$$

**Target Alignment.** For $c < 1$, the target alignment contribution is

$$-2\tilde{\alpha}_A \tilde{\rho}^2 \left( \alpha_A \|\boldsymbol{\beta}_*\|^2 + (\alpha_Z - \alpha_A) (\boldsymbol{\beta}_*^\top \boldsymbol{u})^2 \frac{\theta^2}{\rho^2 + \theta^2} \right).$$

For $c > 1$, the alignment term is

$$-2\tilde{\alpha}_A \tilde{\rho}^2 \left( \frac{\alpha_A}{c} \|\boldsymbol{\beta}_*\|^2 - \frac{\alpha_A}{d} \frac{\theta^2}{\theta^2 + c\rho^2} \|\boldsymbol{\beta}_*\|^2 + \left( \alpha_Z - \frac{\alpha_A}{c} \right) \frac{\theta^2}{\theta^2 + c\rho^2} (\boldsymbol{\beta}_*^\top \boldsymbol{u})^2 \right).$$

Lastly, replacing $\tilde{\rho}, \rho$ with $\tilde{\tau}, \tau$ and using $d/n \to c$ yield the detailed expressions in Theorem 5, up to simple algebra (rearranging terms and simplifying the fractions).

## F PROBABILITY LEMMAS

**Proposition 2.** *If $\boldsymbol{u}, \boldsymbol{v} \in \mathbb{R}^d$ are fixed unit norm vector and $\boldsymbol{A} \in \mathbb{R}^{d \times n}$ is a Gaussian matrix with i.i.d. $\mathcal{N}(0,1)$ entries. If $d > n$, then we have that*

$$\mathbb{E}[(\boldsymbol{u}^\top \boldsymbol{A} \boldsymbol{A}^\dagger \boldsymbol{v})^2] = \frac{n}{d(d+2)} \left[ (\boldsymbol{u}^\top \boldsymbol{v})^2(n+2) + \frac{(1 - (\boldsymbol{u}^\top \boldsymbol{v})^2)(d-n)}{d-1} \right] = \frac{1}{c^2} (\boldsymbol{u}^\top \boldsymbol{v})^2 + o(1),$$

$$\mathrm{Var}\left( \boldsymbol{u}^\top \boldsymbol{A} \boldsymbol{A}^\dagger \boldsymbol{v})^2 \right) = O\left( \frac{1}{d} \right).$$

*Proof.* Let $\boldsymbol{P} := \boldsymbol{A}\boldsymbol{A}^\dagger$. This is the orthogonal projection matrix onto the column space of $\boldsymbol{A}$, denoted $C(\boldsymbol{A}) = \mathrm{Range}(\boldsymbol{A})$. The subspace $C(\boldsymbol{A})$ is an $n$-dimensional subspace of $\mathbb{R}^d$. Because the entries $A_{ij}$ are i.i.d. $\mathcal{N}(0,1)$, the distribution of the random subspace $C(\boldsymbol{A})$ is isotropic (or rotationally invariant). Consequently, the distribution of the random projection matrix $\boldsymbol{P}$ is also rotationally invariant. That is, for any fixed $d \times d$ orthogonal matrix $\boldsymbol{Q}$, the distribution of $\boldsymbol{Q}\boldsymbol{P}\boldsymbol{Q}^\top$ is the same as the distribution of $\boldsymbol{P}$.

We are interested in $\mathbb{E}[(\boldsymbol{u}^\top \boldsymbol{P} \boldsymbol{v})^2]$. Let $\theta$ be the angle between $\boldsymbol{u}$ and $\boldsymbol{v}$, such that $\cos(\theta) = \boldsymbol{u}^\top \boldsymbol{v}$ (since they are unit vectors). Due to the rotational invariance of the distribution of $\boldsymbol{P}$, we can choose an orthonormal basis without loss of generality. Let $\boldsymbol{Q}$ be an orthogonal matrix such that $\boldsymbol{u}' = \boldsymbol{Q}\boldsymbol{u} = \boldsymbol{e}_1 = (1, 0, \ldots, 0)^\top$ and $\boldsymbol{v}' = \boldsymbol{Q}\boldsymbol{v}$ lies in the span of $\boldsymbol{e}_1$ and $\boldsymbol{e}_2$. Specifically, $\boldsymbol{v}' = \cos(\theta)\boldsymbol{e}_1 + \sin(\theta)\boldsymbol{e}_2$. Let $\boldsymbol{P}' = \boldsymbol{Q}\boldsymbol{P}\boldsymbol{Q}^\top$. $\boldsymbol{P}'$ has the same distribution as $\boldsymbol{P}$. Then,

$$\boldsymbol{u}^\top \boldsymbol{P} \boldsymbol{v} = (\boldsymbol{Q}^\top \boldsymbol{u}')^\top \boldsymbol{P}(\boldsymbol{Q}^\top \boldsymbol{v}') = (\boldsymbol{u}')^\top (\boldsymbol{Q}\boldsymbol{P}\boldsymbol{Q}^\top)\boldsymbol{v}' = (\boldsymbol{u}')^\top \boldsymbol{P}' \boldsymbol{v}'$$

Substituting $\boldsymbol{u}' = \boldsymbol{e}_1$ and $\boldsymbol{v}' = \cos(\theta)\boldsymbol{e}_1 + \sin(\theta)\boldsymbol{e}_2$:

$$\boldsymbol{u}^\top \boldsymbol{P} \boldsymbol{v} = \boldsymbol{e}_1^\top \boldsymbol{P}'(\cos(\theta)\boldsymbol{e}_1 + \sin(\theta)\boldsymbol{e}_2)$$
$$= \cos(\theta)(\boldsymbol{e}_1^\top \boldsymbol{P}' \boldsymbol{e}_1) + \sin(\theta)(\boldsymbol{e}_1^\top \boldsymbol{P}' \boldsymbol{e}_2)$$
$$= \cos(\theta) P_{11}' + \sin(\theta) P_{12}'$$

where $P'_{ij}$ are the elements of $\boldsymbol{P}'$. Since $\boldsymbol{P}'$ has the same distribution as $\boldsymbol{P}$, we can drop the prime for calculating expectations involving the elements. Let $\boldsymbol{X} = \boldsymbol{u}^\top \boldsymbol{P} \boldsymbol{v}$. We then need $\mathbb{E}[X^2]$.

$$
\begin{aligned}
\mathbb{E}[X^2] &= \mathbb{E}[(\cos(\theta)P_{11} + \sin(\theta)P_{12})^2] \\
&= \mathbb{E}[\cos^2(\theta)P_{11}^2 + \sin^2(\theta)P_{12}^2 + 2\cos(\theta)\sin(\theta)P_{11}P_{12}] \\
&= \cos^2(\theta)\mathbb{E}[P_{11}^2] + \sin^2(\theta)\mathbb{E}[P_{12}^2] + 2\cos(\theta)\sin(\theta)\mathbb{E}[P_{11}P_{12}]
\end{aligned}
$$

**Calculation of Moments.** We need to compute $\mathbb{E}[P_{11}^2]$, $\mathbb{E}[P_{12}^2]$, and $\mathbb{E}[P_{11}P_{12}]$.

Consider a reflection matrix $\boldsymbol{R}$ that maps $\boldsymbol{e}_2$ to $-\boldsymbol{e}_2$ and leaves other basis vectors unchanged (i.e., $\boldsymbol{R} = \mathrm{diag}(1, -1, 1, \ldots, 1)$). Since the distribution of $\boldsymbol{P}$ is isotropic, it is invariant under reflection. Let $\boldsymbol{P}^* = \boldsymbol{R}\boldsymbol{P}\boldsymbol{R}^\top = \boldsymbol{R}\boldsymbol{P}\boldsymbol{R}$. $\boldsymbol{P}^*$ has the same distribution as $\boldsymbol{P}$. The components are related:

$$
P_{11}^* = (RPR)_{11} = R_{11}P_{11}R_{11} = P_{11}
$$

and

$$
P_{12}^* = (RPR)_{12} = R_{11}P_{12}R_{22} = (1)P_{12}(-1) = -P_{12}.
$$

Therefore,

$$
\mathbb{E}[P_{11}P_{12}] = \mathbb{E}[P_{11}^* P_{12}^*] = \mathbb{E}[P_{11}(-P_{12})] = -\mathbb{E}[P_{11}P_{12}].
$$

This implies $2\mathbb{E}[P_{11}P_{12}] = 0$, so $\mathbb{E}[P_{11}P_{12}] = 0$.

The diagonal element $P_{11} = \boldsymbol{e}_1^\top \boldsymbol{P} \boldsymbol{e}_1 = \|\boldsymbol{P}\boldsymbol{e}_1\|_2^2$ represents the squared norm of the projection of the fixed unit vector $\boldsymbol{e}_1$ onto the random $n$-dimensional subspace $C(\boldsymbol{A})$. This variable follows a Beta distribution:

$$
P_{11} \sim \mathrm{Beta}\left(\frac{n}{2}, \frac{d-n}{2}\right)
$$

The mean and variance of a $\mathrm{Beta}(\alpha, \beta)$ distribution are $\frac{\alpha}{\alpha+\beta}$ and $\frac{\alpha\beta}{(\alpha+\beta)^2(\alpha+\beta+1)}$, respectively. Here, $\alpha = n/2$ and $\beta = (d-n)/2$, so $\alpha + \beta = d/2$.

$$
\mathbb{E}[P_{11}] = \frac{n/2}{d/2} = \frac{n}{d}
$$

Next

$$
\mathrm{Var}(P_{11}) = \frac{(n/2)((d-n)/2)}{(d/2)^2(d/2+1)} = \frac{n(d-n)/4}{(d^2/4)((d+2)/2)} = \frac{n(d-n) \cdot 8}{4d^2(d+2)} = \frac{2n(d-n)}{d^2(d+2)}
$$

Now we find $\mathbb{E}[P_{11}^2]$ using $\mathbb{E}[P_{11}^2] = \mathrm{Var}(P_{11}) + (\mathbb{E}[P_{11}])^2$:

$$
\begin{aligned}
\mathbb{E}[P_{11}^2] &= \frac{2n(d-n)}{d^2(d+2)} + \left(\frac{n}{d}\right)^2 \\
&= \frac{2n(d-n) + n^2(d+2)}{d^2(d+2)} \\
&= \frac{2nd - 2n^2 + n^2 d + 2n^2}{d^2(d+2)} \\
&= \frac{2nd + n^2 d}{d^2(d+2)} \\
&= \frac{n(n+2)}{d(d+2)}.
\end{aligned}
$$

We use the property that $\boldsymbol{P}$ is a projection matrix, so $\boldsymbol{P}^2 = \boldsymbol{P}$. The trace is $\mathrm{Tr}(\boldsymbol{P}) = n$. Also $\mathrm{Tr}(\boldsymbol{P}^2) = \mathrm{Tr}(\boldsymbol{P}) = n$. We can write $\mathrm{Tr}(\boldsymbol{P}^2) = \mathrm{Tr}(\boldsymbol{P}\boldsymbol{P}^\top)$ since $\boldsymbol{P}$ is symmetric.

$$
\mathrm{Tr}(\boldsymbol{P}^2) = \sum_{i=1}^{d}\sum_{j=1}^{d}(P_{ij})^2
$$

Taking the expectation:

$$\mathbb{E}[\mathrm{Tr}(\boldsymbol{P}^2)] = \mathbb{E}\left[\sum_{i,j} P_{ij}^2\right] = \sum_{i,j} \mathbb{E}[P_{ij}^2] = n$$

By rotational symmetry, $\mathbb{E}[P_{ii}^2]$ is the same for all $i$, and $\mathbb{E}[P_{ij}^2]$ is the same for all $i \neq j$.

$$\sum_{i=1}^{d} \mathbb{E}[P_{ii}^2] + \sum_{i \neq j} \mathbb{E}[P_{ij}^2] = n.$$

There are $d$ diagonal terms and $d(d-1)$ off-diagonal terms.

$$d\,\mathbb{E}[P_{11}^2] + d(d-1)\,\mathbb{E}[P_{12}^2] = n$$

Substitute the value for $\mathbb{E}[P_{11}^2]$ (assuming $d > 1$):

$$d\left(\frac{n(n+2)}{d(d+2)}\right) + d(d-1)\,\mathbb{E}[P_{12}^2] = n$$

$$\frac{n(n+2)}{d+2} + d(d-1)\,\mathbb{E}[P_{12}^2] = n$$

$$d(d-1)\,\mathbb{E}[P_{12}^2] = n - \frac{n(n+2)}{d+2} = \frac{n(d+2) - n(n+2)}{d+2} = \frac{nd + 2n - n^2 - 2n}{d+2} = \frac{n(d-n)}{d+2}$$

$$\mathbb{E}[P_{12}^2] = \frac{n(d-n)}{d(d-1)(d+2)}$$

Substitute the moments back into the expression for $\mathbb{E}[X^2]$:

$$\mathbb{E}[X^2] = \cos^2(\theta)\mathbb{E}[P_{11}^2] + \sin^2(\theta)\mathbb{E}[P_{12}^2] + 2\cos(\theta)\sin(\theta) \cdot 0$$

Using $\cos(\theta) = \boldsymbol{u}^\top \boldsymbol{v}$, $\cos^2(\theta) = (\boldsymbol{u}^\top \boldsymbol{v})^2$, and $\sin^2(\theta) = 1 - \cos^2(\theta) = 1 - (\boldsymbol{u}^\top \boldsymbol{v})^2$:

$$\mathbb{E}[(u^\top AA^\dagger v)^2] = (\boldsymbol{u}^\top \boldsymbol{v})^2 \left(\frac{n(n+2)}{d(d+2)}\right) + (1 - (\boldsymbol{u}^\top \boldsymbol{v})^2)\left(\frac{n(d-n)}{d(d-1)(d+2)}\right)$$

$$= \frac{n}{d(d+2)}\left[(\boldsymbol{u}^\top \boldsymbol{v})^2(n+2) + \frac{(1 - (\boldsymbol{u}^\top \boldsymbol{v})^2)(d-n)}{d-1}\right]$$

$$= \frac{1}{c^2}(\boldsymbol{u}^\top \boldsymbol{v})^2 + O\left(\frac{1}{d}\right).$$

**Calculation of Variance.** Recall that reflection $\boldsymbol{R} = \mathrm{diag}(1, -1, 1, \ldots, 1)$ implies $\boldsymbol{P} \stackrel{d}{=} \boldsymbol{R}\boldsymbol{P}\boldsymbol{R}$ (equal in distribution) and thus $\mathbb{E}[P_{11}P_{12}] = 0$, and in general any mixed moment with an odd power of $P_{12}$ vanishes. Therefore, we have the following expansion:

$$\mathbb{E}[X^4] = \cos^4\theta\,\mathbb{E}[P_{11}^4] + 6\cos^2\theta\sin^2\theta\,\mathbb{E}[P_{11}^2 P_{12}^2] + \sin^4\theta\,\mathbb{E}[P_{12}^4]. \tag{31}$$

We start with $\mathbb{E}[P_{11}^4]$. Since $P_{11} \sim \mathrm{Beta}(\alpha, \beta)$ with $\alpha = \frac{n}{2}$, $\beta = \frac{d-n}{2}$. We need the higher moments for the Beta distribution: for $m \geq 1$,

$$\mathbb{E}[P_{11}^m] = \frac{\alpha^{(m)}}{(\alpha+\beta)^{(m)}} = \frac{(\frac{n}{2})^{(m)}}{(\frac{d}{2})^{(m)}}, \qquad x^{(m)} := x(x+1)\cdots(x+m-1).$$

In particular, we have the following third and fourth moments:

$$\mathbb{E}[P_{11}^3] = \frac{(\frac{n}{2})^{(3)}}{(\frac{d}{2})^{(3)}} = \frac{1}{c^3} + O\left(\frac{1}{d}\right), \quad \mathbb{E}[P_{11}^4] = \frac{(\frac{n}{2})^{(4)}}{(\frac{d}{2})^{(4)}} = \frac{1}{c^4} + O\left(\frac{1}{d}\right).$$

We now move on to $\mathbb{E}[P_{11}^2 P_{12}^2]$. From idempotency, $(P^2)_{11} = P_{11}$ gives the row identity $P_{11} = \sum_{k=1}^{d} P_{1k}^2$. Multiplying by $P_{11}^2$ and taking expectations, we have that

$$\mathbb{E}[P_{11}^3] = \mathbb{E}[P_{11}^4] + \sum_{k=2}^{d} \mathbb{E}[P_{11}^2 P_{1k}^2] = \mathbb{E}[P_{11}^4] + (d-1)\,\mathbb{E}[P_{11}^2 P_{12}^2].$$

$$\mathbb{E}[P_{11}^2 P_{12}^2] = \frac{\mathbb{E}[P_{11}^3] - \mathbb{E}[P_{11}^4]}{d-1} = \frac{1}{d-1}\left(\frac{(\frac{n}{2})^{(3)}}{(\frac{d}{2})^{(3)}} - \frac{(\frac{n}{2})^{(4)}}{(\frac{d}{2})^{(4)}}\right) = \frac{1}{d-1}\left(\frac{1}{c^3} - \frac{1}{c^4} + O\left(\frac{1}{d}\right)\right) = O\left(\frac{1}{d}\right).$$

We still need to evaluate or upper bound $\mathbb{E}[P_{12}^4]$. From $P_{11} = \sum_{k=1}^{d} P_{1k}^2$ we have $\sum_{k=2}^{d} P_{1k}^2 = P_{11} - P_{11}^2$. By Cauchy–Schwarz,

$$\sum_{k=2}^{d} P_{1k}^4 = \left(\sum_{k=2}^{d} P_{1k}^2\right)^2 = (P_{11} - P_{11}^2)^2.$$

Taking expectations, we get:

$$(d-1)\mathbb{E}[P_{12}^4] \le \mathbb{E}[(P_{11} - P_{11}^2)^2] = \mathbb{E}[P_{11}^2] - 2\mathbb{E}[P_{11}^3] + \mathbb{E}[P_{11}^4].$$

$$\mathbb{E}[P_{12}^4] \le \frac{1}{d-1}\left(\frac{1}{c^2} - \frac{2}{c^3} + \frac{1}{c^4}\right) + O\left(\frac{1}{d^2}\right) = O\left(\frac{1}{d}\right).$$

We can now plug these expectation bounds into Equation 31:

$$\mathbb{E}[X^4] = \cos^4\theta \frac{(\frac{n}{2})^{(4)}}{(\frac{d}{2})^{(4)}} + O\left(\frac{1}{d}\right) 6\cos^2\theta\sin^2\theta + O\left(\frac{1}{d}\right)\sin^4\theta$$

$$= \frac{1}{c^4}(\boldsymbol{u}^\top \boldsymbol{v})^4 + O\left(\frac{1}{d}\right).$$

Recall from the prior proof that:

$$\mathbb{E}[X^2] = \cos^2\theta \frac{n(n+2)}{d(d+2)} + \sin^2\theta \frac{n(d-n)}{d(d-1)(d+2)} = \frac{1}{c^2}(\boldsymbol{u}^\top \boldsymbol{v})^2 + O\left(\frac{1}{d}\right).$$

Finally, we have that the variance is of order:

$$\mathrm{Var}(X^2) = \mathbb{E}[X^4] - \left(\mathbb{E}[X^2]\right)^2 = O\left(\frac{1}{d}\right).$$

$\square$

**Lemma 32.** *Let $a \ne 0$ be a constant and suppose that $\zeta = a + o(f(n))$ as $n \to \infty$. Then,*

$$\frac{1}{\zeta} = \frac{1}{a} + o(f(n)).$$

*Proof.* Write $\zeta = a + r_n$ with $r_n = o(f(n))$. Then

$$\frac{1}{\zeta} = \frac{1}{a + r_n} = \frac{1}{a} \cdot \frac{1}{1 + \frac{r_n}{a}}.$$

Using the expansion

$$\frac{1}{1+u} = 1 - u + O(u^2) \quad \text{as } u \to 0,$$

with $u = r_n/a$, we obtain

$$\frac{1}{\zeta} = \frac{1}{a}\left(1 - \frac{r_n}{a} + O\big((r_n/a)^2\big)\right) = \frac{1}{a} - \frac{r_n}{a^2} + O(r_n^2).$$

Since $r_n = o(f(n))$ and $f(n) \to 0$, we have $r_n^2 = o(f(n))$. Therefore

$$\frac{1}{\zeta} = \frac{1}{a} + o(f(n)),$$

which is the desired expansion. $\square$

**Lemma 33** (Variance of a reciprocal). *Let $X$ be a random variable satisfying*

$$E[X] = a > 0 \quad \text{and} \quad \text{Var}(X) = \sigma^2 = o(1),$$

*and assume that $X$ is bounded away from zero with high probability. That is, there exists $C \in (0, a)$ such that*

$$\Pr[X \geq C] = 1 - o(1)$$

*If there exists an $M$ such that*

$$\mathbb{E}\left[X^{-8}\right] \leq M \quad \text{and} \quad \mathbb{E}\left[(X - \mathbb{E}[X])^4\right] = O(\sigma^4)$$

*Then*

$$\text{Var}\left(\frac{1}{X}\right) = \frac{1}{a^4} \text{Var}(X) + o\left(\text{Var}(X)\right),$$

*so in particular, $\text{Var}(1/X) = o(1)$.*

*Proof.* Let $Y := X - a$. Then

$$\mathbb{E}[Y] = 0, \qquad \mathbb{E}\left[Y^2\right] = \sigma^2, \qquad \mathbb{E}\left[Y^4\right] = O(\sigma^4).$$

By Taylor's theorem with Lagrange remainder for $f(x) = 1/x$, there exists $\theta = \theta(X) \in (0, 1)$ such that

$$\frac{1}{X} = \frac{1}{a} - \frac{Y}{a^2} + Z, \qquad Z := \frac{Y^2}{(a + \theta Y)^3} \geq 0.$$

Write $\Delta := \frac{1}{X} - \frac{1}{a} = -\frac{Y}{a^2} + Z$. Then

$$\text{Var}\left(\frac{1}{X}\right) = \mathbb{E}\left[\Delta^2\right] - \left(\mathbb{E}\left[\Delta\right]\right)^2.$$

We will show

$$\mathbb{E}\left[\Delta^2\right] = \frac{\sigma^2}{a^4} + o\left(\sigma^2\right) \quad \text{and} \quad \left(\mathbb{E}\left[\Delta\right]\right)^2 = o\left(\sigma^2\right).$$

Let $G := \{X \geq C\}$ and $B := \{X < C\}$. Since $C < a$ and $\mathbb{E}\left[Y^2\right] = \sigma^2$, Chebyshev gives the quantitative bound

$$\Pr\left[B\right] = \Pr\left[\,|Y| \geq a - C\,\right] \leq \frac{\mathbb{E}\left[Y^2\right]}{(a - C)^2} = \frac{\sigma^2}{(a - C)^2} = O\left(\sigma^2\right) = o(1).$$

**Second moment** $\mathbb{E}\left[\Delta^2\right]$. We split over $G$ and $B$.

*On $G$.* Since $a + \theta Y = \theta X + (1 - \theta)a \geq C$, we have

$$|Z| \leq \frac{Y^2}{C^3}, \qquad Z^2 \leq \frac{Y^4}{C^6}.$$

Therefore

$$\mathbb{E}\left[\left(-\frac{Y}{a^2} + Z\right)^2 \mathbf{1}_G\right] = \frac{1}{a^4} \mathbb{E}\left[Y^2 \mathbf{1}_G\right] - \frac{2}{a^2} \mathbb{E}\left[YZ\,\mathbf{1}_G\right] + \mathbb{E}\left[Z^2 \mathbf{1}_G\right].$$

We bound each term as follows.

$$\mathbb{E}\left[Z^2 \mathbf{1}_G\right] \leq \frac{1}{C^6} \mathbb{E}\left[Y^4\right] = O\left(\sigma^4\right),$$

and, using $\mathbf{1}_G \leq 1$ and Lyapunov/monotonicity of $L^p$ norms,

$$\mathbb{E}\left[|YZ|\,\mathbf{1}_G\right] \leq \frac{1}{C^3} \mathbb{E}\left[|Y|^3\right] \leq \frac{1}{C^3} \left(\mathbb{E}\left[Y^4\right]\right)^{3/4} = O\left(\sigma^3\right) = o\left(\sigma^2\right).$$

Moreover,

$$\mathbb{E}\left[Y^2 \mathbf{1}_G\right] = \sigma^2 - \mathbb{E}\left[Y^2 \mathbf{1}_B\right], \qquad \mathbb{E}\left[Y^2 \mathbf{1}_B\right] \leq \left(\mathbb{E}\left[Y^4\right]\right)^{1/2} \Pr\left[B\right]^{1/2} = O\left(\sigma^2\right) \Pr\left[B\right]^{1/2} = o\left(\sigma^2\right).$$

Hence

$$\mathbb{E}\left[\left(-\frac{Y}{a^2}+Z\right)^2\mathbf{1}_G\right]=\frac{\sigma^2}{a^4}+o(\sigma^2).$$

*On $B$.* Using the algebraic identity

$$\left(\frac{1}{X}-\frac{1}{a}\right)^2=\frac{Y^2}{a^2X^2},$$

Cauchy–Schwarz and Hölder (with exponents $2,2$) give

$$\mathbb{E}\left[\Delta^2\mathbf{1}_B\right]=\frac{1}{a^2}\mathbb{E}\left[\frac{Y^2}{X^2}\mathbf{1}_B\right]\le\frac{1}{a^2}\left(\mathbb{E}\left[Y^4\right]\right)^{1/2}\left(\mathbb{E}\left[X^{-4}\mathbf{1}_B\right]\right)^{1/2}\le\frac{1}{a^2}O(\sigma^2)\left(\mathbb{E}\left[X^{-8}\right]\right)^{1/4}\Pr\left[B\right]^{1/4}.$$

Under the lemma's assumption $\mathbb{E}\left[X^{-8}\right]\le M$, we get

$$\mathbb{E}\left[\Delta^2\mathbf{1}_B\right]=O(\sigma^2)\Pr\left[B\right]^{1/4}=o(\sigma^2).$$

Combining the $G$ and $B$ parts,

$$\mathbb{E}\left[\Delta^2\right]=\frac{\sigma^2}{a^4}+o(\sigma^2).$$

**Mean correction $(\mathbb{E}\left[\Delta\right])^2$.** Since $\mathbb{E}\left[Y\right]=0$, we have

$$\mathbb{E}\left[\Delta\right]=\mathbb{E}\left[Z\right]=\mathbb{E}\left[Z\,\mathbf{1}_G\right]+\mathbb{E}\left[Z\,\mathbf{1}_B\right].$$

On $G$, $Z\le Y^2/C^3$, so

$$\mathbb{E}\left[Z\,\mathbf{1}_G\right]\le\frac{1}{C^3}\mathbb{E}\left[Y^2\mathbf{1}_G\right]\le\frac{1}{C^3}\sigma^2.$$

On $B$, The inequality

$$Z=\frac{Y^2}{(a+\theta Y)^3}\le\frac{X^2}{Y^3}$$

holds on set $B$ because on this set as $X<a$, meaning the point $a+\theta Y$ lies between $X$ and $a$, so $a+\theta Y>X$. Thus, using Cauchy–Schwarz and Hölder,

$$\mathbb{E}\left[Z\,\mathbf{1}_B\right]\le\mathbb{E}\left[\frac{Y^2}{X^3}\mathbf{1}_B\right]\le\left(\mathbb{E}\left[Y^4\right]\right)^{1/2}\left(\mathbb{E}\left[X^{-6}\mathbf{1}_B\right]\right)^{1/2}\le O(\sigma^2)\left(\mathbb{E}\left[X^{-12}\right]\right)^{1/4}\Pr\left[B\right]^{1/4}=o(\sigma^2).$$

Thus $|\mathbb{E}\left[\Delta\right]|=O(\sigma^2)$ and therefore

$$(\mathbb{E}\left[\Delta\right])^2=O(\sigma^4)=o(\sigma^2).$$

Putting the two steps together,

$$\operatorname{Var}\left(\frac{1}{X}\right)=\mathbb{E}\left[\Delta^2\right]-(\mathbb{E}\left[\Delta\right])^2=\frac{\sigma^2}{a^4}+o(\sigma^2)=\frac{1}{a^4}\operatorname{Var}(X)+o(\operatorname{Var}(X)).$$

$\square$

**Lemma 34** (Variance of a sum). *Let $A$ and $B$ be any random variables with finite variances $V(A)=\operatorname{Var}(A)$ and $V(B)=\operatorname{Var}(B)$. Then,*

$$\operatorname{Var}(A+B)\le\left(\sqrt{V(A)}+\sqrt{V(B)}\right)^2.$$

*Proof.* Recall that

$$\operatorname{Var}(A+B)=\operatorname{Var}(A)+\operatorname{Var}(B)+2\,\operatorname{Cov}(A,B).$$

By the Cauchy–Schwarz inequality, we have

$$|\operatorname{Cov}(A,B)|\le\sqrt{V(A)V(B)}.$$

Thus,

$$\operatorname{Var}(A+B)\le V(A)+V(B)+2\sqrt{V(A)V(B)}=\left(\sqrt{V(A)}+\sqrt{V(B)}\right)^2.$$

$\square$

**Lemma 35** (Variance of one product). *Let $A, B$ be real random variables with means $a = \mathbb{E}[A]$, $b = \mathbb{E}[B]$ and finite variances. Assume*

$$\mathbb{E}\left[(A-a)^4\right] \leq K_A \operatorname{Var}(A)^2, \qquad \mathbb{E}\left[(B-b)^4\right] \leq K_B \operatorname{Var}(B)^2.$$

*Then, with $C_4 := (K_A K_B)^{1/4}$,*

$$\sqrt{\operatorname{Var}(AB)} \ \leq \ |a|\sqrt{\operatorname{Var}(B)} \ + \ |b|\sqrt{\operatorname{Var}(A)} \ + \ C_4\sqrt{\operatorname{Var}(A)\operatorname{Var}(B)}.$$

*Moreover, as $\operatorname{Var}(A), \operatorname{Var}(B) \to 0$,*

$$\operatorname{Var}(AB) = O\left(a^2\operatorname{Var}(B)\right) + O\left(b^2\operatorname{Var}(A)\right) + o(\operatorname{Var}(A) + \operatorname{Var}(B)).$$

*It directly follows that if all the means are $O(1)$,*

$$\operatorname{Var}(AB) = O\left(\operatorname{Var}(B)\right) + O\left(\operatorname{Var}(A)\right).$$
$$\operatorname{Var}(ABC) = O\left(\operatorname{Var}(C)\right) + O\left(\operatorname{Var}(B)\right) + O\left(\operatorname{Var}(A)\right) \quad \text{and so on by induction.}$$

*Proof.* Write
$$AB - ab = a\,\tilde{B} + b\,\tilde{A} + \tilde{A}\tilde{B}.$$
Using $\operatorname{Var}(U+V) = \operatorname{Var}(U) + \operatorname{Var}(V) + 2\operatorname{Cov}(U,V)$ and $|\operatorname{Cov}(U,V)| \leq \sqrt{\operatorname{Var}(U)\operatorname{Var}(V)}$, we get

$$\operatorname{Var}(AB) = \operatorname{Var}\left(a\tilde{B} + b\tilde{A} + \tilde{A}\tilde{B}\right)$$

$$\leq \left(|a|\sqrt{\operatorname{Var}\left(\tilde{B}\right)} \ + \ |b|\sqrt{\operatorname{Var}\left(\tilde{A}\right)} \ + \ \sqrt{\operatorname{Var}\left(\tilde{A}\tilde{B}\right)}\right)^2.$$

Since $\operatorname{Var}(\tilde{A}) = \operatorname{Var}(A)$ and $\operatorname{Var}(\tilde{B}) = \operatorname{Var}(B)$, it remains to bound $\operatorname{Var}\left(\tilde{A}\tilde{B}\right)$. By Cauchy–Schwarz (Hölder with $p = q = 2$),

$$\operatorname{Var}\left(\tilde{A}\tilde{B}\right) \ \leq \ \mathbb{E}\left[\tilde{A}^2\tilde{B}^2\right] \ \leq \ \left(\mathbb{E}\left[\tilde{A}^4\right]\right)^{1/2}\left(\mathbb{E}\left[\tilde{B}^4\right]\right)^{1/2}.$$

Since we assume fourth–moment control $\mathbb{E}\left[\tilde{A}^4\right] \leq K_A \operatorname{Var}(A)^2$ and $\mathbb{E}\left[\tilde{B}^4\right] \leq K_B \operatorname{Var}(B)^2$, then

$$\sqrt{\operatorname{Var}\left(\tilde{A}\tilde{B}\right)} \ \leq \ (K_A K_B)^{1/4}\sqrt{\operatorname{Var}(A)\operatorname{Var}(B)}.$$

Hence

$$\operatorname{Var}(AB) \ \leq \ \left(|a|\sqrt{\operatorname{Var}(B)} + |b|\sqrt{\operatorname{Var}(A)} + C_4\sqrt{\operatorname{Var}(A)\operatorname{Var}(B)}\right)^2, \qquad C_4 := (K_A K_B)^{1/4}.$$

For the moreover part, using the exact variance–covariance expansion,

$$\operatorname{Var}(AB) = a^2\operatorname{Var}(B) + b^2\operatorname{Var}(A) + 2ab\operatorname{Cov}(A,B) + \operatorname{Var}\left(\tilde{A}\tilde{B}\right) + 2a\operatorname{Cov}\left(\tilde{B}, \tilde{A}\tilde{B}\right) + 2b\operatorname{Cov}\left(\tilde{A}, \tilde{A}\tilde{B}\right),$$

we bound the three remainder terms using Cauchy–Schwarz and the fourth–moment control:

$$\operatorname{Var}\left(\tilde{A}\tilde{B}\right) \leq \mathbb{E}\left[\tilde{A}^2\tilde{B}^2\right] \leq \left(\mathbb{E}\left[\tilde{A}^4\right]\right)^{1/2}\left(\mathbb{E}\left[\tilde{B}^4\right]\right)^{1/2} \leq C_4^2\operatorname{Var}(A)\operatorname{Var}(B),$$

$$\left|\operatorname{Cov}\left(\tilde{B}, \tilde{A}\tilde{B}\right)\right| \leq \sqrt{\operatorname{Var}\left(\tilde{B}\right)}\sqrt{\operatorname{Var}\left(\tilde{A}\tilde{B}\right)} \leq C_4\operatorname{Var}(B)\sqrt{\operatorname{Var}(A)},$$

$$\left|\operatorname{Cov}\left(\tilde{A}, \tilde{A}\tilde{B}\right)\right| \leq \sqrt{\operatorname{Var}\left(\tilde{A}\right)}\sqrt{\operatorname{Var}\left(\tilde{A}\tilde{B}\right)} \leq C_4\operatorname{Var}(A)\sqrt{\operatorname{Var}(B)}.$$

As $\operatorname{Var}(A), \operatorname{Var}(B) \to 0$, each of these is $o(\operatorname{Var}(A) + \operatorname{Var}(B))$.

For the covariance term, Cauchy–Schwarz and the inequality $2uv \leq \varepsilon u^2 + \varepsilon^{-1} v^2$ (for any $\varepsilon > 0$) with $u := |a|\sqrt{\mathrm{Var}(B)}$, $v := |b|\sqrt{\mathrm{Var}(A)}$ give

$$| 2ab\,\mathrm{Cov}(A,B)| \leq 2|ab|\sqrt{\mathrm{Var}(A)\,\mathrm{Var}(B)} \leq \varepsilon\, a^2 \mathrm{Var}(B) + \varepsilon^{-1} b^2 \mathrm{Var}(A)\,.$$

Therefore,

$$\mathrm{Var}(AB) \leq (1+\varepsilon)\, a^2 \mathrm{Var}(B) + (1+\varepsilon^{-1})\, b^2 \mathrm{Var}(A) + o(\mathrm{Var}(A) + \mathrm{Var}(B))\,.$$

Choosing, e.g., $\varepsilon = 1$ yields

$$\mathrm{Var}(AB) = O\big(a^2 \mathrm{Var}(B)\big) + O\big(b^2 \mathrm{Var}(A)\big) + o(\mathrm{Var}(A) + \mathrm{Var}(B))\,,$$

which proves the moreover statement. $\qquad\square$

**Lemma 36** (Variance of general product). *Let $m \geq 2$ and let $X_1, \ldots, X_m$ be real random variables with nonzero means $\mu_i := \mathbb{E}[X_i] \neq 0$ and variances $f_i(n) := \mathrm{Var}(X_i) \to 0$ as $n \to \infty$. Assume that for some integer $M \geq m$ (it is enough to take $M = m$),*

$$\mathbb{E}\big[\, |X_i - \mu_i|^{2M}\,\big] \;=\; O\big(\mathrm{Var}(X_i)^M\big) \qquad \text{for each } i = 1, \ldots, m. \tag{32}$$

*Then*

$$\mathrm{Var}\bigg(\prod_{i=1}^{m} X_i\bigg) \;=\; O\bigg(\Big(\sum_{i=1}^{m} \sqrt{f_i(n)}\Big)^2\bigg) \;=\; O\Big(\max_{1 \leq i \leq m} f_i(n)\Big)\,.$$

*Proof.* Write $\Delta_i := X_i - \mu_i$ so that $\mathbb{E}[\Delta_i] = 0$ and $\|\Delta_i\|_{L_2} = \sigma_i$. By assumption Equation 32 with $M \geq m$ and monotonicity of $L_p$ norms,

$$\|\Delta_i\|_{L_{2k}} \;=\; O\Big(\sqrt{f_i(n)}\Big) \qquad \text{for every } 1 \leq k \leq m,\ i = 1, \ldots, m.$$

Expand the product multilinearly:

$$\prod_{i=1}^{m} X_i - \prod_{i=1}^{m} \mu_i = \sum_{\emptyset \neq S \subseteq [m]} \bigg(\prod_{j \in S^c} \mu_j\bigg) \bigg(\prod_{i \in S} \Delta_i\bigg)\,.$$

Taking $L_2$ norms and using the triangle inequality,

$$\bigg\|\prod_{i=1}^{m} X_i - \prod_{i=1}^{m} \mu_i\bigg\|_{L_2} \;\leq\; \sum_{\emptyset \neq S \subseteq [m]} \bigg(\prod_{j \in S^c} |\mu_j|\bigg) \bigg\|\prod_{i \in S} \Delta_i\bigg\|_{L_2}\,.$$

For a fixed nonempty $S$ with $|S| = k$, apply Hölder with exponents all equal to $2k$:

$$\bigg\|\prod_{i \in S} \Delta_i\bigg\|_{L_2} \;\leq\; \prod_{i \in S} \|\Delta_i\|_{L_{2k}} \;=\; O\bigg(\prod_{i \in S} \sqrt{f_i}\bigg)\,,$$

where we used $\|\Delta_i\|_{L_{2k}} = O(\sqrt{f_i})$ for $k \leq m$.

Let $c_i := \sqrt{f_i(n)}$. Summing over subsets $S$ shows

$$\bigg\|\prod_{i=1}^{m} X_i - \prod_{i=1}^{m} \mu_i\bigg\|_{L_2} \;\leq\; A\Big(\prod_{i=1}^{m}(1 + c_i) - 1\Big) \;\leq\; A\,(e^{\Xi} - 1)\,,$$

where $\Xi := \sum_{i=1}^{m} c_i$ and $A$ is a constant depending only on $m$, $\{\mu_i\}$, and the moment constants (not on $n$). Hence

$$\mathrm{Var}\Big(\prod_{i=1}^{m} X_i\Big) \;\leq\; \bigg\|\prod_{i=1}^{m} X_i - \prod_{i=1}^{m} \mu_i\bigg\|_{L_2}^2 \;=\; O(\Xi^2) \;=\; O\bigg(\Big(\sum_{i=1}^{m} \sqrt{f_i(n)}\Big)^2\bigg)\,.$$

Since $m$ is fixed, $(\sum_{i=1}^{m} \sqrt{f_i})^2 \leq m^2 \max_i f_i$, giving the claimed bound. $\qquad\square$

**Corollary 1** (Higher moments of the centered product). *Fix $p \geq 1$. Under the hypotheses of Lemma 36, then*

$$\left\| \prod_{i=1}^{m} X_i - \prod_{i=1}^{m} \mathbb{E}[X_i] \right\|_{L_{2p}} \leq C_{p,m} \sum_{\emptyset \neq S \subseteq [m]} \left( \prod_{j \in S^c} |\mathbb{E}[X_j]| \right) \prod_{i \in S} \sqrt{f_i} = o(1),$$

*and hence $\mathbb{E} \left| \prod_{i=1}^{m} X_i - \mathbb{E} \prod_{i=1}^{m} X_i \right|^{2p} = o(1).$*

**Lemma 37** (Expectation of Product vs. Product of Expectations). *Fix $k \geq 2$. Let $X_1, \ldots, X_k$ be random variables. Assume:*

1. *Uniformly bounded means: $\sup_{n,i} |\mathbb{E}[X_i]| \leq M < \infty$.*

2. *Vanishing variances: $\mathrm{Var}(X_i) = f_i(n)$ with $f_i(n) \to 0$ as $n \to \infty$ for each $i$.*

3. *Moment control up to order $k$: For each $i$ and every $p \in \{2, \ldots, k\}$,*

$$\mathbb{E}\left[ |X_i - \mathbb{E}[X_i]|^p \right] \leq C_p \, \mathrm{Var}(X_i)^{p/2},$$

   *with constants $C_p$.*

*Then for finite $k$, we have:*

$$\left| \mathbb{E}\left[ \prod_{i=1}^{k} X_i \right] - \prod_{i=1}^{k} \mathbb{E}X_i \right| = O\left( \left( \sum_{i=1}^{k} \sqrt{f_i(n)} \right)^2 \right) = O\left( \max_{1 \leq i \leq k} f_i(n) \right).$$

*Proof.* Set $\Delta_i := X_i - \mathbb{E}[X_i]$, so $\mathbb{E}\Delta_i = 0$, $\mathrm{Var}(X_i) = \mathrm{Var}(\Delta_i) = f_i(n)$, and by assumption

$$\|\Delta_i\|_{L_p} := \left( \mathbb{E}\left[ |\Delta_i|^p \right] \right)^{1/p} \leq C_p^{1/p} f_i(n)^{1/2}, \qquad p = 2, \ldots, k.$$

Using the multilinearity of expectation,

$$\prod_{i=1}^{k} X_i = \prod_{i=1}^{k} \left( \mathbb{E}[X_i] + \Delta_i \right) = \sum_{S \subseteq [k]} \left( \prod_{i \in S} \Delta_i \right) \left( \prod_{j \notin S} \mathbb{E}[X_j] \right),$$

Thus,

$$\prod_{i=1}^{k} X_i - \prod_{i=1}^{k} \mathbb{E}[X_i] = \sum_{\emptyset \neq S \subseteq [k]} \left[ \prod_{i \in S} \Delta_i \right] \prod_{j \notin S} \mathbb{E}[X_j].$$

Then taking the expectation and noting that $\prod_{j \notin S} \mathbb{E}[X_j]$ is a constant, we get

$$\mathbb{E}\left[ \prod_{i=1}^{k} X_i \right] - \prod_{i=1}^{k} \mathbb{E}[X_i] = \sum_{\emptyset \neq S \subseteq [k]} \mathbb{E}\left[ \prod_{i \in S} \Delta_i \right] \prod_{j \notin S} \mathbb{E}[X_j].$$

If $S = \{\ell\}$ then $\mathbb{E}\left[ \prod_{i \in S} \Delta_i \right] = \mathbb{E}[\Delta_\ell] = 0$. Hence every singleton term vanishes exactly, and the sum begins at $|S| = 2$. From the bounded means assumption,

$$\left| \prod_{j \notin S} \mathbb{E}[X_j] \right| \leq M^{k - |S|}, \qquad \forall S \subseteq [k].$$

Fix a nonempty subset $S$ with $|S| = m \geq 2$. By generalized Hölder with all exponents equal to $m$ (so $\sum_{i \in S} \frac{1}{m} = 1$),

$$\left| \mathbb{E}\left[ \prod_{i \in S} \Delta_i \right] \right| \leq \prod_{i \in S} \|\Delta_i\|_{L_m} \leq \prod_{i \in S} \left( C_m^{1/m} f_i(n)^{1/2} \right) = C_m \prod_{i \in S} \sqrt{f_i(n)}.$$

Therefore, for every $S$ with $|S| = m \geq 2$,

$$\left| \mathbb{E}\left[ \prod_{i \in S} \Delta_i \right] \prod_{j \notin S} \mathbb{E}X_j \right| \leq M^{k - m} C_m \prod_{i \in S} \sqrt{f_i(n)}.$$

Let $c_i := \sqrt{f_i(n)} \geq 0$. Denote by

$$e_m(c_1, \ldots, c_k) := \sum_{\substack{S \subseteq [k] \\ |S|=m}} \prod_{i \in S} c_i$$

the $m$-th elementary symmetric polynomial. Summing the bound from, we get

$$\left| \mathbb{E}\left[\prod_{i=1}^k X_i\right] - \prod_{i=1}^k \mathbb{E}X_i \right| \leq \sum_{m=2}^k M^{k-m} C_m \, e_m(c_1, \ldots, c_k).$$

Let $M_\star := \max_{2 \leq m \leq k} M^{k-m} C_m$. Since $e_m \geq 0$ for $c_i \geq 0$,

$$\sum_{m=2}^k M^{k-m} C_m \, e_m \leq M_\star \sum_{m=2}^k e_m(c_1, \ldots, c_k).$$

Recall the identity

$$\prod_{i=1}^k (1 + c_i) = \sum_{m=0}^k e_m(c_1, \ldots, c_k) = 1 + \sum_{m=1}^k e_m(c_1, \ldots, c_k),$$

so that $\sum_{m=2}^k e_m = \prod_{i=1}^k (1 + c_i) - 1 - \sum_{i=1}^k c_i$. Hence

$$\left| \mathbb{E}\left[\prod_{i=1}^k X_i\right] - \prod_{i=1}^k \mathbb{E}X_i \right| \leq M_\star \left( \prod_{i=1}^k (1 + c_i) - 1 - \sum_{i=1}^k c_i \right).$$

Let $\Xi := \sum_{i=1}^k c_i \to 0$ as $n \to \infty$. Since $\log(1 + u) \leq u$ for $u \geq 0$,

$$\prod_{i=1}^k (1 + c_i) = \exp\left( \sum_{i=1}^k \log(1 + c_i) \right) \leq \exp(\Xi).$$

Thus, the difference is at most $M_\star(e^\Xi - 1 - \Xi)$. By Taylor's theorem, $e^\Xi = 1 + \Xi + \frac{1}{2}\Xi^2 e^\xi$ for some $\xi \in [0, \Xi]$, so $e^\Xi - 1 - \Xi = \frac{1}{2}\Xi^2 e^\xi \leq \frac{1}{2}\Xi^2 e^\Xi$ (since $\xi \leq \Xi$ and $e^\xi \leq e^\Xi$). Therefore,

$$\left| \mathbb{E}\left[\prod_{i=1}^k X_i\right] - \prod_{i=1}^k \mathbb{E}X_i \right| \leq \frac{M_\star}{2} \Xi^2 e^\Xi = O(\Xi^2),$$

as $\Xi \to 0$ and $e^\Xi \to 1$. Since $\Xi = O\left( \sum_{i=1}^k \sqrt{f_i(n)} \right)$, we get the result. $\qquad\square$

**Lemma 38** (Moment preservation under monomial $\leftrightarrow$ Hermite change of basis). *Fix $M \in \mathbb{N}$ and degree $r \in \mathbb{N}$. Let*

$$\mathcal{M} := \{x^\gamma : \gamma \in \mathbb{N}^M, |\gamma| \leq r\}, \qquad \mathcal{H} := \{\boldsymbol{H}_\alpha : \alpha \in \mathbb{N}^M, |\alpha| \leq r\},$$

*with $\boldsymbol{H}_\alpha(x) = \prod_{j=1}^M H_{\alpha_j}(x_j)$ the probabilists' Hermite basis. For any (random) coefficients $\{a_\gamma\}_{|\gamma| \leq r}$ define the random polynomial $P(x) = \sum_{|\gamma| \leq r} a_\gamma x^\gamma$. Then there is a deterministic, invertible matrix $T = T(M, r)$ such that the Hermite coefficients $c = \{c_\alpha\}_{|\alpha| \leq r}$ in $P(x) = \sum_{|\alpha| \leq r} c_\alpha \boldsymbol{H}_\alpha(x)$ satisfy*

$$c = T a.$$

*Consequently, for any $p \geq 1$,*

$$\|c_\alpha\|_{L_p} \leq \sum_{|\gamma| \leq r} |T_{\alpha\gamma}| \, \|a_\gamma\|_{L_p} \quad \text{for all } \alpha,$$

*so if each $a_\gamma \in L_p$ then each $c_\alpha \in L_p$. Moreover, since $T$ is invertible, the converse also holds: if each $c_\alpha \in L_p$ then each $a_\gamma \in L_p$.*

*Proof.* In one dimension, each monomial admits a finite Hermite expansion $x^m = \sum_{j=0}^{\lfloor m/2 \rfloor} t_{m,j} H_{m-2j}(x)$ with deterministic coefficients $t_{m,j}$; in several dimensions, take tensor products to obtain $x^\gamma = \sum_{|\alpha| \leq |\gamma|} T_{\alpha\gamma} \boldsymbol{H}_\alpha(x)$. Ordering multi-indices by total degree yields a block upper-triangular, deterministic, invertible matrix $T = T(M, r)$. Linearity gives $c = T a$. For $p \geq 1$, Minkowski's inequality yields $\|c_\alpha\|_{L_p} = \left\| \sum_\gamma T_{\alpha\gamma} a_\gamma \right\|_{L_p} \leq \sum_\gamma |T_{\alpha\gamma}| \, \|a_\gamma\|_{L_p}$, so finiteness of all $\|a_\gamma\|_{L_p}$ implies finiteness of all $\|c_\alpha\|_{L_p}$. Invertibility gives the converse using $a = T^{-1}c$ and the same argument with $T^{-1}$. $\qquad\square$

# G PROOF OF SPECIFIC CASES AND OVERFITTING

## G.1 PROOF OF THEOREM 1.

*Proof.* We set $\alpha_Z = \alpha_A = \tilde{\alpha}_Z = \tilde{\alpha}_A = \alpha$, $\tilde{\theta} = \theta$, $\tilde{\tau} = \tau$ in the above Theorem 5 and note that it greatly simplifies each term. Algebra shows that for $c < 1$

$$\text{Bias} = \tau_\varepsilon^2 \frac{c}{1-c} \frac{\theta^2}{d(\theta^2 + \tau^2)}, \quad \text{Variance} = \alpha^2 \tau^2 \|\boldsymbol{\beta}_*\|^2 + \tau_\varepsilon^2 \frac{c}{1-c} \left[1 - \frac{\theta^2}{d(\theta^2 + \tau^2)}\right],$$

$$\text{Data Noise} = \alpha^2 \tau^2 \|\boldsymbol{\beta}_*\|^2, \quad \text{Target Alignment} = -2\alpha^2 \tau^2 \|\boldsymbol{\beta}_*\|^2,$$

While for $c > 1$, we can first send $d, n \to \infty$ and many terms become asymptotically 0. In the end, we get that:

$$\text{Bias} = \alpha^2 \theta^2 (\boldsymbol{\beta}_*^\top \boldsymbol{u})^2 \left(1 - \frac{1}{c}\right)^2 \left(\frac{\tau^2 c}{\theta^2 + \tau^2 c}\right)^2, \quad \text{Data Noise} = \alpha^2 \tau^2 \|\boldsymbol{\beta}_*\|^2,$$

$$\text{Variance} = \alpha^2 \tau^2 \|\boldsymbol{\beta}_*\|^2 \frac{1}{c} + \alpha^2 \tau^2 (\boldsymbol{\beta}_*^\top \boldsymbol{u})^2 \frac{\theta^2}{\theta^2 + \tau^2 c} \left(1 - \frac{1}{c}\right) + \tau_\varepsilon^2 \frac{1}{c-1}.$$

$$\text{Target Alignment} = -2\alpha^2 \tau^2 \left(\left(1 - \frac{1}{c}\right) \frac{\theta^2}{\theta^2 + \tau^2 c} (\boldsymbol{\beta}_*^\top \boldsymbol{u})^2 + \|\boldsymbol{\beta}_*\|^2 \frac{1}{c}\right),$$

Adding these terms together, we see with simple algebra that many terms cancel or can be combined, establishing the stated formula. □

## G.2 PROOF OF THEOREM 2.

*Proof.* We set $\alpha_Z = \tilde{\alpha}_Z$, $\alpha_A = \tilde{\alpha}_A$, $\tilde{\theta} = \theta$, $\tilde{\tau} = \tau$, and send $d, n \to \infty$ in Theorem 5. Recall that $\Delta_c = \alpha_Z - \frac{\alpha_A}{c}$ and $\Delta_1 = \alpha_Z - \alpha_A$. Then some algebra shows that for $c < 1$,

$$\text{Bias} = \theta^2 (\boldsymbol{\beta}_*^\top \boldsymbol{u})^2 \Delta_1^2 \left(\frac{\tau^2}{\theta^2 + \tau^2}\right)^2, \quad \text{Data Noise} = \alpha_A^2 \tau^2 \|\boldsymbol{\beta}_*\|^2,$$

$$\text{Target Alignment} = -2\alpha_A^2 \tau^2 \|\boldsymbol{\beta}_*\|^2 - 2\alpha_A \tau^2 (\boldsymbol{\beta}_*^\top \boldsymbol{u})^2 \Delta_1 \frac{\theta^2}{\theta^2 + \tau^2},$$

$$\text{Variance} = \alpha_A^2 \tau^2 \|\boldsymbol{\beta}_*\|^2 + \tau_\varepsilon^2 \frac{c}{1-c} + \tau^2 (\boldsymbol{\beta}_*^\top \boldsymbol{u})^2 \left[\frac{1}{1-c} \frac{\theta^4 + \theta^2 \tau^2 c}{(\theta^2 + \tau^2)^2} \Delta_1^2 + 2\alpha_A \Delta_1 \frac{\theta^2}{\theta^2 + \tau^2}\right].$$

For $c > 1$, we have that

$$\text{Bias} = \theta^2 (\boldsymbol{\beta}_*^\top \boldsymbol{u})^2 \Delta_c^2 \left(\frac{\tau^2 c}{\theta^2 + \tau^2 c}\right)^2, \quad \text{Data Noise} = \alpha_A^2 \tau^2 \|\boldsymbol{\beta}_*\|^2,$$

$$\text{Target Alignment} = -2\alpha_A^2 \tau^2 \frac{\|\boldsymbol{\beta}_*\|^2}{c} - 2\alpha_A \tau^2 (\boldsymbol{\beta}_*^\top \boldsymbol{u})^2 \Delta_c \frac{\theta^2}{\theta^2 + \tau^2 c},$$

$$\text{Variance} = \alpha_A^2 \tau^2 \frac{\|\boldsymbol{\beta}_*\|^2}{c} + \tau_\varepsilon^2 \frac{1}{c-1} + \tau^2 (\boldsymbol{\beta}_*^\top \boldsymbol{u})^2 \frac{c}{1-c} \frac{\theta^2}{\theta^2 + \tau^2 c} \Delta_c^2.$$

We proceed by adding these terms together and the results follow from algebra. □

## G.3 PROOF OF THEOREM 3.

*Proof.* We set $\tilde{\theta} = \theta$ and $\tilde{\tau} = \tau$ in Theorem 5 and have the regime of equal operator norm $\theta^2 = \gamma \tau^2$. Since we are interested in the limit $c \to \infty$, we only consider the overparameterized case $c > 1$. We first take the limit $d, n \to \infty$ and have that:

$$\text{Bias} = \tau^2 (\boldsymbol{\beta}_*^\top \boldsymbol{u})^2 \left(\sqrt{\gamma}(\tilde{\alpha}_Z - \alpha_Z) + \left(\alpha_Z - \frac{\alpha_A}{c}\right) \frac{c\sqrt{\gamma}}{\gamma + c}\right)^2, \quad \text{Data Noise} = \tilde{\alpha}_A^2 \tau^2 \|\boldsymbol{\beta}_*\|^2,$$

$$\text{Target Alignment} = -2\tilde{\alpha}_A \tau^2 \left(\left(\alpha_Z - \frac{\alpha_A}{c}\right) \frac{\gamma}{\gamma + c} (\boldsymbol{\beta}_*^\top \boldsymbol{u})^2 + \alpha_A \frac{\|\boldsymbol{\beta}_*\|^2}{c}\right),$$

$$\text{Variance} = \tau^2\alpha_A^2\frac{\|\boldsymbol{\beta}_*\|^2}{c} + \tau^2(\boldsymbol{\beta}_*^\top\boldsymbol{u})^2\frac{c}{(c-1)}\frac{\gamma}{\gamma+c}\left(\alpha_Z - \frac{\alpha_A}{c}\right)^2 + \tau_\varepsilon^2\left(\frac{1}{c-1}\right).$$

The rest follows from simple calculus: if $\tilde{\alpha}_Z \neq \alpha_Z$, $\gamma = \omega_c(1)$, and $\boldsymbol{\beta}_*^\top\boldsymbol{u} \neq 0$, the bias will diverge and other terms are controlled, yielding catastrophic. If $\tilde{\alpha}_Z = \alpha_Z$, $\omega_c(1) \leq \gamma \leq o_c(c^2)$, and $\boldsymbol{\beta}_*^\top\boldsymbol{u} \neq 0$, a similar thing happens. In other cases, all of these terms are controlled and become finite values in the limit $\lim_{c\to\infty}\mathcal{R}_c - \tau_\varepsilon^2$, giving us tempered overfitting.

$$\lim_{c\to\infty}\mathcal{R}_c = \begin{cases} \tilde{\alpha}_A^2\tau^2\|\boldsymbol{\beta}_*\|^2 & \boldsymbol{\beta} \perp u \\ \tau^2\left[\gamma\tilde{\alpha}_Z^2(\boldsymbol{\beta}_*^\top\boldsymbol{u})^2 + \tilde{\alpha}_A^2\|\boldsymbol{\beta}_*\|^2\right] & \boldsymbol{\beta} \not\perp u, \gamma = \Theta_c(1) \\ \infty & \alpha_Z \neq \tilde{\alpha}_Z, \boldsymbol{\beta}_* \not\perp u, \gamma = \omega(1) \\ \infty & \alpha_Z = \tilde{\alpha}_Z, \boldsymbol{\beta}_* \not\perp u, \omega(1) \leq \gamma \leq o(c^2) \\ \tau^2\left[\left(\frac{\phi}{(\phi+1)^2}\alpha_Z^2 - 2\tilde{\alpha}_A\alpha_Z\right)(\boldsymbol{\beta}_*^\top\boldsymbol{u})^2 + \alpha_A^2\|\boldsymbol{\beta}_*\|^2\right] & \alpha_Z = \tilde{\alpha}_Z, \boldsymbol{\beta}_* \not\perp u, \gamma = \phi c^2 \\ \tau^2\left[(\alpha_Z^2 - 2\tilde{\alpha}_A\alpha_Z)(\boldsymbol{\beta}_*^\top\boldsymbol{u})^2 + \alpha_A^2\|\boldsymbol{\beta}_*\|^2\right] & \alpha_Z = \tilde{\alpha}_Z, \boldsymbol{\beta}_* \not\perp u, \gamma = \omega(c^2) \end{cases}$$

$\square$

### G.4 PROOF OF THEOREM 4.

*Proof.* We start with the first part and assume that $\alpha_Z \neq \tilde{\alpha}_Z$. Similarly, we have that $\tilde{\theta} = \theta$ and $\tilde{\tau} = \tau$ in Theorem 5. To achieve equal Frobenius norm, we set $\theta^2 = d\tau^2$ and send $d, n \to \infty$ so several terms would vanish.

In particular, for $c < 1$, we have that

$$\text{Bias} = \theta^2(\boldsymbol{\beta}_*^\top\boldsymbol{u})^2\left(\tilde{\alpha}_Z - \alpha_Z + (\alpha_Z - \alpha_A)\frac{\tau^2}{\theta^2 + \tau^2}\right)^2 = \tau^2(\boldsymbol{\beta}_*^\top\boldsymbol{u})^2\left(\sqrt{d}(\tilde{\alpha}_Z - \alpha_Z) + (\alpha_Z - \alpha_A)\frac{\sqrt{d}}{d+1}\right)^2,$$

It is clear that this term becomes $\infty$ since the term inside the parentheses scales with $d$. Note that the variance and data noise are non-negative, and target alignment is controlled. We have that $\mathcal{R}_c = \infty$ for $c \in (0, 1)$.

For $c > 1$, the same logic follows, and we also note that:

$$\text{Bias} = \theta^2(\boldsymbol{\beta}_*^\top\boldsymbol{u})^2\left(\tilde{\alpha}_Z - \alpha_Z + \left(\alpha_Z - \frac{\alpha_A}{c}\right)\frac{\tau^2 c}{\theta^2 + \tau^2 c}\right)^2 = \tau^2(\boldsymbol{\beta}_*^\top\boldsymbol{u})^2\left(\sqrt{d}(\tilde{\alpha}_Z - \alpha_Z) + \left(\alpha_Z - \frac{\alpha_A}{c}\right)\frac{\sqrt{d}c}{d+c}\right)^2,$$

which scales with $d$ with other terms controlled. Hence, $\mathcal{R}_c = \infty$ for all $c \neq 1$.

Now assume that $\alpha_Z = \tilde{\alpha}_Z$. Since we are interested in $c \to \infty$, we only consider $c > 1$. First, from algebra and taking the limit for $d, n$, we have that:

$$\text{Bias} = \tau^2(\boldsymbol{\beta}_*^\top\boldsymbol{u})^2\left(\left(\alpha_Z - \frac{\alpha_A}{c}\right)\frac{c\sqrt{d}}{d+c}\right)^2 \to 0, \quad \text{Data Noise} = \tilde{\alpha}_A^2\tau^2\|\boldsymbol{\beta}_*\|^2,$$

$$\text{Target Alignment} = -2\tilde{\alpha}_A\tau^2\left(\left(\alpha_Z - \frac{\alpha_A}{c}\right)(\boldsymbol{\beta}_*^\top\boldsymbol{u})^2 + \alpha_A\frac{\|\boldsymbol{\beta}_*\|^2}{c}\right),$$

$$\text{Variance} = \tau^2\alpha_A^2\frac{\|\boldsymbol{\beta}_*\|^2}{c} + \tau^2(\boldsymbol{\beta}_*^\top\boldsymbol{u})^2\frac{c}{(c-1)}\left(\alpha_Z - \frac{\alpha_A}{c}\right)^2 + \tau_\varepsilon^2\left(\frac{1}{c-1}\right).$$

We now take $c \to \infty$ and many terms vanish in this limit, yielding:

$$\lim_{c\to\infty}\mathcal{R}_c = -2\tilde{\alpha}_A\alpha_Z\tau^2(\boldsymbol{\beta}_*^\top\boldsymbol{u})^2 + \tau^2(\boldsymbol{\beta}_*^\top\boldsymbol{u})^2\alpha_Z^2 + \tilde{\alpha}_A^2\tau^2\|\boldsymbol{\beta}_*\|^2 = \tau^2\left[(\boldsymbol{\beta}_*^\top\boldsymbol{u})^2(\alpha_Z^2 - 2\tilde{\alpha}_A\alpha_Z) + \|\boldsymbol{\beta}_*\|^2\tilde{\alpha}_A^2\right].$$

$\square$

**Proposition 3** (Non–existence of a canceling scale parameter). *Let $\alpha_A, \alpha_Z > 0$ be fixed scalars, let $u, \beta_* \in \mathbb{R}^d$ be fixed vectors, and set*

$$a := \|\beta_*\|^2 > 0, \qquad b := (\beta_*^\top u)^2 \in [0, a].$$

*For every positive real number $\phi$ define*

$$f(\phi) = \alpha_A^2\, a \; + \; \left(\alpha_Z^2\Big(1 + \frac{1}{\phi}\Big) - 2\alpha_Z\alpha_A\right) b.$$

*Then*

$$f(\phi) > 0 \quad \text{for all } \phi > 0.$$

*Consequently the equation $f(\phi) = 0$ has no solution with $\phi \in (0, \infty)$.*

*Proof.* If $b = 0$ (i.e. $\beta_*$ is orthogonal to $u$) we have $f(\phi) = \alpha_A^2 a > 0$, so no positive $\phi$ can cancel the expression. Hence assume $b > 0$.

Writing $r := b/a \in (0, 1]$ we obtain

$$f(\phi) = a\left[\alpha_A^2 + \alpha_Z(\alpha_Z - 2\alpha_A)\, r + \frac{\alpha_Z^2 r}{\phi}\right]. \tag{$*$}$$

Since $r \le 1$,

$$\alpha_A^2 + \alpha_Z(\alpha_Z - 2\alpha_A)\, r \;\ge\; \alpha_A^2 + \alpha_Z(\alpha_Z - 2\alpha_A) \;=\; (\alpha_A - \alpha_Z)^2 \;\ge 0.$$

Thus the square bracket in $(*)$ is the sum of a non–negative term and a strictly positive term.

$\square$

