# OpenReview forum: "Risk Phase Transitions in Spiked Regression: Alignment Driven Benign and Catastrophic Overfitting"
_ICLR.cc/2026/Conference — ICLR 2026 Poster_

### Official Review · Reviewer_ywoE · 2025-10-22

**Soundness:** 2
**Presentation:** 2
**Contribution:** 2
**Rating:** 2
**Confidence:** 3

**Summary:**

This paper analyzes the performance of min-norm interpolators under a spiked covariance assumption for the data in high-dimensional linear regression. The authors consider the asymptotic proportional regime where both $n$ (number of samples) and $d$ (input dimension) diverge to infinity with a fixed aspect ratio $c = d/n$, and specifically examine scenarios where the spike in the covariance is correlated with the teacher vector.
After deriving the limiting value of the excess risk, the authors classify different conditions leading to benign, tempered, or catastrophic overfitting in the overparameterized regime. The main theoretical finding is that increasing spike correlation can, counterintuitively, induce catastrophic overfitting.
The authors provide limited empirical evidence suggesting that analogous phenomena appear in 3-layer ReLU networks under similar data conditions.

**Strengths:**

The paper presents several technical strengths:

- The paper shows a high degree of mathematical rigor: The theoretical analysis appears sound, with the novel results properly recovering the known results from Hastie et al. 2020 as special cases. The generalization to spiked covariance models beyond isotropic settings is a natural extension of existing literature.

- The systematic classification of overfitting behaviors (benign, tempered, catastrophic) across different parameter regimes provides a detailed characterization of the problem space. Table 1 is especially effective in summarizing the different cases.

- The paper uncovers counterintuitive yet interesting behaviors, particularly that alignment with the spike is not universally beneficial and that increasing spike strength can worsen generalization before improving it.

**Weaknesses:**

While I recognize the substantial theoretical effort invested in this work, I believe the paper in its current form has some significant presentation and validation issues that limit its readability and impact, especially for the broader machine learning community not only interested in theory.

My main concerns are the following:

1. The paper lacks in accessibility. The paper prioritizes mathematical formalism over intuitive explanations, with limited discussion of *why* these behaviors emerge or *when* practitioners should expect them.
For a venue like ICLR that serves a diverse ML community, the paper should consider a more accessible presentation alongside mathematical rigor.

   *Specific recommendation*: Each major theorem should be followed by an intuitive explanation section that uses simple examples or visualizations to explain the underlying mechanisms. For instance, why does increasing spike strength lead to catastrophic overfitting before benign overfitting? What is the intuitive mechanism?

2. The data model (first display on page 1) represents a significant simplification - a rank-one spike plus isotropic noise. While mathematically tractable, it is unclear whether this captures realistic data structures encountered in modern machine learning. Real data show both hierarchical structure and other types of correlation with the ground truth rule. Additionally, real data have been shown to follow power law covariance structure.

   Moreover, even from a purely theoretical perspective, the way non-linearity and model misspecification are captured through the two coefficients $\alpha_Z$ and $\alpha_A$ (which differentially weight the spike and bulk components) is quite restrictive. This specific parametric form of non-linearity—where the target depends linearly on spike and bulk components but with different coefficients—represents only a narrow class of possible misspecifications.

   *Specific recommendation*: (a) Include a discussion about the relationship between the spiked covariance model and real-world data, with empirical evidence showing when real data approximately satisfies these assumptions. (b) Discuss the limitations of the $(\alpha_Z, \alpha_A)$ parameterization for capturing general forms of model misspecification, and acknowledge what types of practical scenarios this does and does not cover. (c) Consider extending the analysis to more realistic spectral structures or more general misspecification models, even if only as future work.

3. The experimental section requires substantial expansion and can be greatly improved to better justify the studied model:
   - Only one experimental setup is shown (3-layer ReLU networks, Figure 4)
   - No experimental details are provided in the main paper, forcing readers to search Appendix B for minimal additional information
   - The connection between the linear theory and nonlinear experiments is not rigorously established
   - No experiments on real datasets are provided

   *Specific recommendation*: Add experiments on: (a) deeper networks with different architectures, (b) different activation functions, (c) real-world datasets where spike structure can be validated, (d) ablation studies showing which theoretical predictions transfer to practice

4. Even considering Appendix B, the experimental details are insufficient for reproduction. Critical information missing includes: exact network initialization, optimization hyperparameters beyond learning rate and epochs, how the spike direction and alignment are constructed, how results are aggregated across trials, and the random seeds used.

   *Specific recommendation*: Provide complete experimental details in a dedicated section or supplementary code repository, following ICLR reproducibility guidelines.

5. The term "phase transition" in the title and throughout suggests sharp, discontinuous changes in behavior.
However, Figure 3 and the mathematical expressions suggest smooth, continuous transitions from positive to negative coefficients except for the case of $c=1$ which has been extensively discussed in the literature.
This is more accurately described as a trade-off or crossover phenomenon rather than a phase transition in the statistical physics sense.

   *Specific recommendation*: Either provide evidence of sharp transitions (e.g., derivatives of risk with respect to parameters showing discontinuities) or adjust the terminology throughout the paper to more accurately reflect the continuous nature of these transitions.

Additional minor concerns:

6. The proof sketch in Section 3.4 that explains the risk decomposition is deferred to near the end of the paper. This interpretable decomposition should appear much earlier to help readers understand the subsequent results.

7. The paper would benefit from a section explicitly discussing what practitioners should take away from this work. When should they worry about alignment? How can they diagnose whether their setting corresponds to the catastrophic regime? What are actionable recommendations that one gains by understanding this model?

**Questions:**

Most of my questions for the authors are connected to what was previously written in the Strengths and Weaknesses sections.

1. Have the authors tried to verify the theoretical predictions with other network architectures beyond 3-layer ReLU networks? For example, does the same behavior on synthetic data appear for Convolutional networks or Residual networks?
   After establishing the phenomenon on synthetic data, do similar alignment-dependent phase transitions appear on real-world datasets for the same models?

2. Can the authors provide guidance on how practitioners can determine whether their data exhibits spiked covariance structure? What are the practical tools or diagnostics that would indicate when this theory applies? Has this been shown on some specific datasets for example? This would greatly enhance the practical relevance of the work.

3. The paper shows one experiment with nonlinear networks but provides no theoretical justification for why the linear analysis should transfer. Can the authors provide:
   - Theoretical analysis (even informal or pointing to relevant references) of why these phenomena persist in nonlinear models?
   - More extensive empirical validation showing the boundaries of when the theory does and does not apply?

4. Given that the $(\alpha_Z, \alpha_A)$ parameterization represents a specific form of misspecification, can the authors comment on what types of real-world problems might naturally exhibit this structure? Are there examples from machine learning practice where targets are known to have this differential dependence on principal and bulk components?

---

> ### Author Response · Authors · 2025-11-17
> **Response Part 1**
>
> We thank the reviewer for acknowledging the strength of our paper.
>
> - Our mathematical rigor,
>
> - comprehensive summary of overfitting behaviors,
>
> - new observations that spike alignment is not universally beneficial.
>
> We appreciate the positive feedback and will address the concerns in turn. We wanted to emphasize that the goal of our theory paper is to offer a precise breakdown of generalization properties in spiked models and break seemingly reliable intuitions practitioners might tend to accept for granted.
>
> > Lack of accessibility
>
> We thank the reviewer for their comment. Due to the space limitations it was difficult to include more intuition. However, we would also like to present the following intuition that we will add to the paper.
>
> **Planned change.** After each main theorem we will add a short intuition. The core picture comes from our decomposition into *bias*, *variance*, *data noise*, and *target-alignment* terms. The data noise is not too important. The bias and variance are non-negative (harmful), while the alignment term is negative (beneficial). The competition between **(i)** bias/variance and **(ii)** alignment determines the regime.
>
> For example, for the well specified case, (See Appendix F, lines 3568-3575). Here we see that alignment completely cancels the variance, but not the bias. **Hence it is the bias that drives the catastrophic overfitting**.
>
> Note **for fixed $c$** the bias scales like $\theta^2\tau^4/(\theta^2+\tau^2)^2$. Hence bigger spikes result in smaller bias. On the other hand, the alignment scales like $\theta^2\tau^2/(\theta^2 + \tau^2)$. Hence this is constant in the size of the spike. Hence larger spikes benefit.
>
> > Data model simplification
>
> Recent one-step feature-learning analyses show that learned features are *linear polynomials* of the inputs and the learned spikes (e.g., Moniri et al., 2024). Motivated by this, we study the *simplest non-trivial polynomial target* that separates bulk and spike contributions. Specifically, if $a$ is the bulk and $z$ is the spike, we consider the multivariate linear polynomial $y = \alpha_A a + \alpha_Z z$. In particular, for the single spike this is exactly the same form of polynomial in Moniri et al. 2024. However, there are some differences.
>
> **Extensions to ridge and multispike are feasible but technically heavy.**
>
> - *Ridge.* As in Li & Sonthalia 2024, one can analyze the resolvent away from zero. This leads to modified versions of Lemmas 9–10. The remainder of the proof structure is unchanged but requires more much more careful bookkeeping.
>
> - *Multiple spikes.* This paper relies on Meyer, 1967, which tells us the spectral structure of the pseudoinverse after a rank 1 perturbation. This result if generalized to any finite rank perturbation in Wei, 2001. We can then use this result instead of the Meyer result with the same proof structure. We then bounds contributions along each singular direction. The resulting formulas are less compact, but the program follows through, however, again with significantly more bookkeeping.
>
> Due to the significant effort involved rigorously doing these calculations. We leave this as future work. However, we believe that an extension to ridge with multiple spikes is feasible.
>
> > Transfer of the phenomenon.
>
> This paper https://arxiv.org/abs/1810.01075 is a reason that such a phenomenon may transfer. Here the paper shows that these bulk + spike spectra appear for the initial layers naturally during training (see Figure 3a, 7a, 14c, 15b) when training ResNets on real data
>
> Additionally papers such as https://arxiv.org/abs/2402.10127 show that such spikes can propagate deeper into the network (at least initialization). Also this work https://arxiv.org/abs/2310.03010v2 show that they can persist during training as well (see figure 2.5). Hence we believe that this spike plus bulk phenomenon is a persistent consistently seen structure during neural network training.
>
> > Experiments details
>
> We would like to highlight that the footnote on page 2 links to anonymous github repository with our code. In particular the three layer relu network, we direct the reviewer to the notebook titled equal_operator_norm.ipynb. The last section of the notebook titled Alignment plots have the complete code for the experiment.

---

> > ### Author Response · Authors · 2025-11-17
> > **Response Part 2**
> >
> > > Additional Experiments
> >
> > We thank the reviewer suggesting these experiments. We have conducted the following two new experiments.
> >
> > We switched the model from a 3 layer Relu network to a network with skip connections, dropout and LayerNorm. Additionally, we use GeLU activation.
> >
> > We then took the MNIST data and then created an artificial spike and added it to the data and trained the network. The results and code are in the anonymous github (https://anonymous.4open.science/r/Alignment-Spike-45B6/rebuttal/Equal_Operator_Norm-2.ipynb)
> >
> > We considered $\alpha_A, \alpha_Z \in \{1,4\}$ and got 4 plots. We see that for $(\alpha_A, \alpha_Z) = (1,1)$ anti-alignment is beneficial, for (4,1) alignment, for (1,4), anti-alignment and for (4,4) alignment.
> >
> > Hence we believe our theory setup is an important first step in understanding this phenomenon.
> >
> > > Phase Transitions.
> >
> > The reviewer is correct that not all of our transitions are abrupt, but many of them are.
> >
> > Looking at table 1, for equal operator norm,  $\beta \perp u$ versus $\beta \perp u$ is a sharp transition between catastrophic and tempered.
> >
> > Further, suppose $\gamma = c^\kappa$. Then for the well specified, equal operator norm, and $\beta || u$ case, we see that if $0 < \kappa < 2$ it is catastrophic (i.e. infinity) where for $\kappa > 2$ is zero. This is also a sharp phase transition.
> >
> > Similarly in the misspecified case,  if $\gamma = c^\kappa$, we again see that the risk is infinite for $\kappa < 2$ and finite for $\kappa = 2$ which is a sharp phase transition.
> >
> > Hence our usage of the word phase transition.
> >
> > > Practitioners
> >
> > Our take away is that you should always consider the alignment. Additionally, prior work one step gradient descent shows that we get these rank 1 spikes and how big they are. It is usually dependent on step size and activation. We believe our work determines which spikes are not good in these one step papers.

---

### Official Review · Reviewer_KtDe · 2025-10-29

**Soundness:** 3
**Presentation:** 2
**Contribution:** 3
**Rating:** 8
**Confidence:** 3

**Summary:**

The authors study a min-L2-norm regression task.
The data x is a Gaussian along a direction u plus an isotropic bulk, while the labels y are a linear model of the projection of x on direction u plus a linear model of the bulk part.
The predictor is a linear model learned with min-L2-norm regression.

They study:
Generalization: MSE evaluated on a new data point (including noise, and possibly with different coefficients in the target function - to model covariate shift). They consider the excess risk to remove the noise plateau.
Alignment: how beneficial it is to have the task weights aligned to the hidden direction u for the generalization?
Overfitting: when c -> infinity (n << d, few samples, perfect fitting of dataset is possible), does the generalization go to zero (benign), to a constant (tempered) to infinity (catastrophic)?

They consider both the scaling in which the spike dominates, as well as the one in which the spike is of the same order of the noise (spectrally).

Their main result is Theorem 5, providing a full risk decomposition for the setting under consideration.
In Section 3 they study generalization, alignment and overfitting in the case of, respectively:
- Section 3.1: well specified problem, in which the target function is linear in x (and not linear in the spike and bulk contributions of x)
- Section 3.2: missspecification but no covariate shift: the target function is not linear in x, but testing is done with the same target function as the training data
- Section 3.3: missspecification and covariate shift: the target is non-linear in x as in the previous case, and testing is done with a structurally identical target, but with spike-bulk contributions altered.

Table 1 classifies all the regimes studied by the authors.

**Strengths:**

The authors provide a complete classification of overfitting in linear regression, studying in particular how alignment between data structure and task helps or hinders generalisation, in a toy but fully controllable learning setting. This creates a nice benchmark to understand the phenomenon in more complicated settings (non-linear estimators for e.g.).

**Weaknesses:**

It is unclear to me how specific the results are to the data model. The authors do not discuss which elements may generalize to more complicated data, and which are surely specific.

The presentation is a bit heavy in Section 3. The authors present the classification of behaviors, but provide little intuition on why the behaviors observed make sense, or why it challenges common beliefs. This results in a bit of a dry description of the phenomenology, from which is difficult to gather a more general take home message (which is more nicely summarised in lines 60-70).
Maybe this is not possible, but I invite the authors to see whether this element can be improved.

**Questions:**

In your work you consider min-norm interpolation, i.e. the limit lambda -> 0^+ of ridge regression with regularization strength lambda. How hard would it be to generalize your results to finite lambda? Would that make sense in the context of studying overfitting?

Typos:
- line 45-46: z and a should be defined clearly given X in line 34. Are they the columns of Z, A? This gets defined only in line 180
- line 45-46: what is the dimensionality of y? It seems a scalar, but is boldfaced. Same for epsilon in eq 2.
- line 103: "we shall ..." seems to miss a verb
- Figure captions sometimes lack the values of parameters (such as d and n etc) at which the curves/experiments are plotted

---

> ### Author Response · Authors · 2025-11-17
>
> Thank you for the positive evaluation and for highlighting our contributions
>
> - a clear, controllable benchmark for alignment-dependent generalization
>
> - an explicit regime map (benign / tempered / catastrophic) supported by simulations.
>
> Below we address the two main weaknesses and the question.
>
>
> > It is unclear to me how specific the results are to the data model. The authors do not discuss which elements may generalize to more complicated data, and which are surely specific.
>
> We discuss some of the assumptions here. We shall add some of this discussion to the paper.
>
> We do need the entries of $v$ to IID Normal. As long as the entries are from any distribution with mean 0 and variance 1, the proof goes through. In particular, if we don’t assume that $v$ is random and that it is just some fixed deterministic vector on the sphere of radius $n$ the proof goes through.
>
> Suppose $A = U\Sigma V^T \in \mathbb{R}^{d \times n}$, we need $U$ to Haar distributed. However, we do not need $V$ to be Haar distributed, *AND* $v$ to be isotropic Gaussian. In particular, what we need is $V^Tv$ to be rotationally invariant. So we can achieve this via assuming $V$ is Haar distributed. Or by assuming that $v$ is rotationally invariant and independent of $V$.
>
> We do not need the limiting distribution to Marchenko-Pastur, as long we have a limiting distribution that should be sufficient.
>
> We can also consider the following extensions. Both extensions are feasible but technically heavy.
>
> - *Ridge.* As in Li & Sonthalia NeurIPS 2024, one can analyze the resolvent away from zero. This leads to modified versions of Lemmas 9–10. The remainder of the proof structure is unchanged but requires much more careful bookkeeping.
>
> - *Multiple spikes.* This paper relies on Meyer, 1967, which tells us the spectral structure of the pseudoinverse after a rank 1 perturbation. This result is generalized to any finite rank perturbation in Wei, 2001. We can then use this result instead of the Meyer result with the same proof structure. We then bound contributions along each singular direction. The resulting formulas are less compact, but the program follows through, however, again with significantly more bookkeeping.
>
> Due to the significant effort involved rigorously doing these calculations. We leave this as future work. However, we believe that an extension to a ridge with multiple spikes is feasible.
>
> > Intuition
>
> We thank the reviewer for their comment. Due to the space limitations it was difficult to include more intuition. However, we would also like to present the following intuition that we will add to the paper.
>
> **Planned change.** After each main theorem we will add a short intuition. The core picture comes from our decomposition into *bias*, *variance*, *data noise*, and *target-alignment* terms. The first two are non-negative (harmful), while the alignment term is negative (beneficial). The competition between **(i)** bias/variance and **(ii)** alignment determines the regime.
>
> For example for the well specified case, (See Appendix F, lines 3568-3575). Here we see that alignment completely cancels the variance, but not the bias. **Hence it is the bias that drives the catastrophic overfitting**.
>
> Note **for fixed $c$** the bias scales like $\theta^2\tau^4/(\theta^2+\tau^2)^2$. Hence bigger spikes result in smaller bias. On the other hand, the alignment scales like $\theta^2\tau^2/(\theta^2 + \tau^2)$. Hence this is constant in the size of the spike. Hence larger spikes benefit.
>
>
> > Typos
>
> Thank you shall fix these.

---

> > ### Comment · Reviewer_KtDe · 2025-11-25
> >
> > I thank the authors for engaging with my comments. My concerns were marginal and have been addressed. I keep my original score.

---

> > > ### Author Response · Authors · 2025-11-27
> > >
> > > We thank the reviewer for the careful reading and for the positive evaluation of our work. Your comments have been helpful for us to improve our paper. We will incorporate the suggested changes in the revised version.

---

### Official Review · Reviewer_uSkU · 2025-10-31

**Soundness:** 2
**Presentation:** 2
**Contribution:** 1
**Rating:** 2
**Confidence:** 4

**Summary:**

This paper studies the generalization error of linear regression using spiked covariance data models, characterizing the impacts of spike strength, target-spike alignment, target model misspecification, and train-test covariate shift. The theoretical output of the paper is an exact expression for the generalization error in the considered setting. This allows the authors to identify the regimes where spike strength and target-spike alignment lead to improved generalization.

**Strengths:**

1. Rigorous asymptotic analysis of the generalization error of linear regression in a setting involving multiple interesting variables (i.e., spike strength, target-spike alignment, target model misspecification, and train-test covariate shift).

2. The writing of the paper (including figures and theoretical results) is of good quality. Therefore, it is easy to follow.

3. The studied setting is somewhat interesting, and it can be useful for understanding some effects of data distribution on the generalization error.

4. The paper includes experimental extension (results) for multilayer neural networks, which is beneficial, although the experimental results are also limited to the synthetic data setting considered in the paper.

**Weaknesses:**

1. The paper lacks highly relevant literature and contextualization relative to the prior work. Specifically, there are at least three papers (about generalization errors with spiked covariance) that are quite related to this work but not mentioned in the paper:
- [1] studied "generalization for least squares regression with spiked covariances", which is also the fundamental topic of this work. Also, the settings are quite similar for [1] and this paper.
- [2,3] analyzed two-layer neural networks with spiked covariance, which leads to spiked covariance for the features learned by the model (as discussed in the introduction of this work as well), so these works are also relevant. Specifically, [2] focused on the effect of spike magnitude and spike-target alignment, which is also the focus of this paper.
Therefore, without proper contextualization relative to these prior works, this paper is weak in terms of positioning and novelty.

2. The fundamental theoretical challenge relative to the prior work should be discussed. Specifically, considering the closeness of this work to related setups in [1], Hastie et al. (2022), Sonthalia & Nadakuditi (2023), and [4], the difference in the proof/derivation techniques should be explicit. Since this is currently missing, the results in this work can be considered trivial generalizations/extensions of the other results.

3. The motivation for this specific data model (in line 45) is unclear. While being the main object of study and the source of most of the claimed results, the authors do not explicitly motivate the data model $\mathbf{y} = \alpha_Z \boldsymbol{\beta}^T \mathbf{z} + \alpha_A \boldsymbol{\beta}^T \mathbf{a} + \boldsymbol{\epsilon}$, other than saying it is a non-linear function of $\mathbf{x} = \mathbf{z} + \mathbf{a}$ for different $\alpha_Z, \alpha_A$ cases (introducing mis-specification).

4. Considering that the target is a non-linear function of the input (a.k.a. mis-specificed case in the paper), studying the performance of a linear model becomes irrelevant (not that interesting). What is the point of studying this case?

5. The benefit of spike-target alignment on the generalization is known in the literature [2]. On the other hand, most of the cases where the alignment hurts the generalization in this paper can be attributed to the lack of proper regularization. For example, in most of the figures where the alignment is harmful, the authors set the spike magnitude $\theta = O(\sqrt{c})$ with respect to $c = d/n$ (ratio of dimension to samples) and let $c \to \infty$. In this case, the model is highly overparameterized, and the norm of the input is large (due to the spike magnitude), but there is no explicit regularization beyond the implicit nature of the min-norm solution.

6. Spiked covariance with a single spiked direction (Assumption 1) is considered, limiting the practical relevance of the found explicit expression for the generalization error. Also, the practical relevance of the considered target $\mathbf{y}$ model seems unclear.


**Related work that are not mentioned in the paper:**

*[1] Li, Jiping, and Rishi Sonthalia. "Generalization for least squares regression with simple spiked covariances." arXiv preprint arXiv:2410.13991 (2024).*

*[2] Demir, Samet, and Zafer Dogan. "Random features outperform linear models: Effect of strong input-label correlation in spiked covariance data." arXiv preprint arXiv:2409.20250 (2024).*

*[3] Demir, Samet, and Zafer Dogan. "Asymptotic Analysis of Two-Layer Neural Networks after One Gradient Step under Gaussian Mixtures Data with Structure." International Conference on Learning Representations (2025).*

*[4] Li, Xinyue and Rishi Sonthalia. "Least squares regression can exhibit under-parameterized double
descent." Advances in Neural Information Processing Systems (2024).*

**Questions:**

1. What is the positioning of this paper in comparison to the papers I mentioned in weaknesses 1 and 2?

2. What is the motivation of the specific data model (in line 45)? If it is to introduce a non-linear function of $\mathbf{x}$, why don't you use $\sigma(\boldsymbol{\beta}^T \mathbf{x})$ for some nonlinear $\sigma: R \to R$?

3. Is it possible to connect the data model to the one I just mentioned?

4. If the target is a non-linear function of the input (as stated in lines 49-50), how can we expect the linear regression to perform well? What is the point of studying this case?

5. What happens if you apply ridge regression (with a regularization constant that increases with $c$ (ratio of dimension to samples)) in Figure 1a, for example?

6. Could the authors explain the practical relevance of their setting (specifically, their data model of input and target)? Also, is it possible to identify real-world datasets/scenarios that approximately satisfy the assumptions and experiment with them?

7. Is it possible to extend the assumption to multiple spikes?

8. Is the setting (together with the assumptions) of this work enough to characterize the generalization error for neural networks trained with one gradient step (Dandi et al. 2024; Moniri et al. 2023; and [3] above)? If so, it would be a strength?

---

> ### Author Response · Authors · 2025-11-17
> **Response Part 1**
>
> Thank you for the careful reading and for highlighting the strengths of our submission:
>
> - the rigorous asymptotic analysis of generalization under spiked covariances,
>
> - the clarity of exposition/figures,
>
> - the broader interest of studying spike strength, alignment, misspecification, and covariate shift.
>
> We appreciate these positive points and address your questions and concerns below.
>
> We will respond to the review as follows. First, we shall have general responses. Following this we respond to each specific point raised by the reviewer. We hope that these additional clarifications address all of the reviewers' concerns.
>
> # General Response
>
> ---
>
> ## Adding ridge regularization
>
> We considered the setting of figure 1a ($\theta^2 = c \tau^2$), added a ridge penalty, and numerically computed the error. We tried a variety of different regularization strength scalings.
>
> $\lambda = 0, \lambda = 1, \lambda = c, \lambda = c^2, \lambda = dc$
>
> In all cases, we observed catastrophic overfitting. Please go see the anonymous github repository  (https://anonymous.4open.science/r/Alignment-Spike-45B6/Equal_Operator_Norm.ipynb) (also linked page 2 of the paper). Please go to the rebuttal folder and see the ridge-catastrophic images.
>
> ## Setting and motivation
>
> Recent one-step feature-learning analyses (including the papers pointed out by the reviewer) show that learned features are *linear polynomials* of the inputs and the learned spikes (e.g., Moniri et al., 2023). Motivated by this, we study the *simplest non-trivial polynomial target* that separates bulk and spike contributions. Specifically, if $a$ is the bulk and $z$ is the spike, we consider the multivariate linear polynomial $y = \alpha_A a + \alpha_Z z$. In particular, for the single spike this is exactly the same form of polynomial in Moniri et al. 2024. However, there are some subtle differences: in their setting, the bulk feature $F\_0=\sigma(WX)$ is not Gaussian like in this paper. However, the *spike structure* matches ours.
>
> While $\sigma(\beta^T x)$ is a compelling nonlinear target, it treats bulk and spike identically. Our aim was to *separate* these effects cleanly, hence the polynomial-in-(bulk, spike) model. Formally, the *degree-1 Hermite truncation* of $\sigma(\beta^T x)$ yields
>
> $$ \sigma(\beta^\top x) \approx \mu+\alpha_A\,\beta^\top a+\alpha_Z\,z $$
>
> where $\alpha_A=\mathbb{E}[\sigma’(\beta^T x)]$ and $\alpha_Z=(\beta \cdot  v)\,\mathbb{E}[\sigma’(\beta^T x)]. $
>
>
> This is *exactly* our $(\alpha_A,\alpha_Z)$ form, but with a *fixed ratio* $\alpha_Z/\alpha_A$. Our analysis allows this ratio to vary, capturing *misspecification* and more general feature-learning effects beyond a single index.
>
>
> **Multi-index.** Suppose we consider a multi-index model instead. That is
>
> $$ y=\sum_{j=1}^k \sigma_j(\beta_j^T x) $$
>
> The same rank 1 approximation now lets us realize *arbitrary*  $(\alpha_A,\alpha_Z)$.

---

> > ### Author Response · Authors · 2025-11-17
> > **Response Part 2**
> >
> > ##  Positioning and prior work
> >
> > We thank the reviewer for pointing us to these papers. We shall add the following discussion.
> >
> > - *Xinyue Li & Sonthalia (2024).* They study (i) an extension of Sonthalia & Nadakuditi (2023) with ridge and (ii) a mixture model. Our Remark 1 shows how to recover the SN’23 ridgeless setting. Hence, we change the target function while they change the loss.
> >
> > - *Jiping Li & Sonthalia (2024).* Their models correspond to *two specific choices* of $(\alpha_A,\alpha_Z)$. We *greatly generalize* both the setup and the overfitting analysis beyond those fixed cases.
> >
> > We thank the reviewer for pointing us to Demir–Dogan [2,3]. *We will cite and position our results accordingly.*
> >
> > - *Scope and object of study.*  [3] analyzes after one gradient step on Gaussian-mixture/spiked data. In contrast, we provide closed-form generalization risk for the minimum-norm linear interpolator. **In particular, this papers serve as motivation for studying the problem here.** Our setting is the analytically tractable “base case” that isolates spike–alignment mechanisms.
> >
> > On the other hand, [2] looks when substituting one activation function for another results in the same generalization error. However, it does not characterize the error, **which is the goal of this paper**.
> >
> > - *Alignment and spike magnitude effects.*
> >   [2] emphasizes that random features can outperform linear models when input–label correlation is strong; alignment is largely *beneficial* in their regimes once nonlinearity is introduced. Our results show that *before* introducing nonlinear features, alignment can exhibit *non-monotone behavior*, including a *catastrophic band* at intermediate spike strengths and that sufficiently large spikes eventually move the system back to tempered/benign regimes. This clarifies when/why nonlinear features in [2] help they act (in part) like implicit regularization that avoids our catastrophic band.
> >
> > [2,3] illuminate *nonlinear* advantages under spiked correlations; we provide the *exact linear baseline* that (i) explains when alignment hurts/helps, (ii) quantifies spike-strength transitions, and (iii) supplies thresholds that predict where nonlinear/random-feature gains in [2,3] are most pronounced. We have added citations and a short related-work paragraph making these links and distinctions clear.
> >
> > **Technical novelty.** Prior works pointed out by the reviewer control only *first/second moments* via concentration to MP limits. Our analysis needs *higher-moment control* to evaluate the risk *in closed form* under anisotropy, alignment, misspecification, and covariate shift. We obtain these using *Mixed Spherical Hypercontractivity* (Sec. D.4.2), including a *new hypercontractivity result on product manifolds with random polynomial coefficients* (Lemmas 13 and 15). We also bound a variety of *new quadratic forms* (Lemmas 10–11) and require *negative-moment bounds* not needed previously (Lemmas 18–20).
> >
> > ## Extensions: multiple spikes and ridge regularization
> >
> > Both extensions are feasible but technically heavy.
> >
> > - *Ridge.* As in Li & Sonthalia, one can analyze the resolvent away from zero. This leads to modified versions of Lemmas 9–10. The remainder of the proof structure is unchanged but requires more much more careful bookkeeping.
> >
> > - *Multiple spikes.* This paper as well as (LS24, SN23) really on Meyer, 1967, which tells us the spectral structure of the pseudoinverse after a rank 1 perturbation. This result is generalized to any finite rank perturbation in Wei, 2001. We can then use this result instead of the Meyer result with the same proof structure. We then bound contributions along each singular direction. The resulting formulas are less compact, but the program follows through, however, again with significantly more bookkeeping.
> >
> > Due to the significant effort involved rigorously doing these calculations. We leave this as future work. However, we believe that an extension to ridge with multiple spikes is feasible.
> >
> > ---
> >
> > ## Summary.
> >
> > Our target class is precisely the *rank-1 Hermite projection* of broad multi-index models and, crucially, *decouples* bulk and spike to reveal the alignment-driven *phase transitions* we characterize. Additionally, these are exactly the type of polynomials seen in prior feature learning in papers. The proofs introduce *new higher-moment and hypercontractive bounds* specific to the spiked, interpolating regime. We believe these clarifications and connections address the reviewer’s concerns and strengthen the paper’s positioning.

---

> > > ### Author Response · Authors · 2025-11-17
> > > **Response Part 3**
> > >
> > > # Answers to specific points.
> > >
> > > > Weakness 1
> > >
> > > Please see our response on positioning and prior work. We will add this discussion to the paper.
> > >
> > > > Weakness 2
> > >
> > > Please see our technical novelty response. In particular, we need stronger concentration results (bounding higher moments and negative moments) compared to prior work. To do this we use hypercontractivity. The reason for this is that we need to bound 1 over a quantity. Due to the complexity of the setup, we have significantly more divisions. Hence we need stronger concentration.
> > >
> > > > Weakness 3
> > >
> > > Please see our response on the setting. This motivation comes from prior work on one step feature analysis
> > >
> > > > Weakness 4
> > >
> > > Again please see our response on the setting for our motivations. Additionally, characterizing the effect of misspecification is interesting
> > >
> > > For example
> > >
> > > - The well specified case has the interesting non-monotonic behavior
> > >
> > > - The misspecified case, even if we don’t have noise in $y$ (i.e. $\tau_\epsilon = 0$). Then we still have double descent (see formula on lines 348-351). This is not present in the well specified case.
> > >
> > > - In the misspecified and covariate shift case, we see that if the training data was misspecified but the test data is well specified, we can benignly overfit but only in the perfect alignment and large spike setting. Additionally this is only if specification for the spike was incorrect while the bulk specification was correct. For the case where we have the incorrect relationship with the bulk, we tend to catastrophically overfit. Thus, suggesting that correct bulk modelling can be more important than correct spike modeling.
> > >
> > >
> > > > Weakness 5
> > >
> > > Please see our ridge regularized numerical example. The reviewer is correct that the norm of the input is large, but so the norm of the output. Hence it is not clear that we should expect the norm of the learned weights to be large.
> > >
> > > In particular, we can consider the two extremes, zero regularization, as shown in the paper we can catastrophically overfit, however increasing the size of the spike . For infinite regularization, we learn the zero predictor, which as the norm of the response goes to infinity, we again catastrophically overfit. However, in this case even if we make the size of the spike bigger, we can’t counteract the catastrophic overfitting.
> > >
> > > This does lead to an interesting question, of what is the correct scale of regularization and does appropriate regularization mitigate this. Experimentally, we tried a few different scales and did not find one that does mitigate this catastrophic overfitting.
> > >
> > > The code will be added to the anonymous github repository linked on page 2 of the paper.
> > >
> > > > Weakness 6
> > >
> > > Please see our discussion on extensions and setting
> > >
> > > > Question 1
> > >
> > > Please see our positioning response
> > >
> > > > Question 2
> > >
> > > Please see our setting response
> > >
> > > > Question 3
> > >
> > > Yes it is. Please see our setting response
> > >
> > > > Question 4
> > >
> > > Please see the response to weakness 4. While it is true we can’t always exactly do well. It is important to understand how well we can do.
> > >
> > > > Question 5
> > >
> > > Please see response to weakness 5
> > >
> > > > Question 6
> > >
> > > Please see our setting and motivation response. Additionally, please see the MNIST experiments in response to Reviewer ywoE
> > >
> > > > Question 7
> > >
> > > Yes, but it is very involved to do the bookkeeping. See response to extensions
> > >
> > > > Question 8
> > >
> > > We can’t exactly give the generalization for the Moniri et al 2023 setting. Unfortunately, the bulk has a slightly different density (generalized MP vs standard MP here) and there is a dependence between the $v$ component of the spike and the bulk that we do not model.
> > >
> > > However, we believe it is possible to get the generalization error by building on this paper. If we think of $v$ as fixed (not random) then our proof goes through. In particular, in Moniri et al, we have that $v = Xy$. In a high dimensional setting (with appropriate scaling) this does concentrate to a fixed $v$. Hence this approach may be feasible.
> > >
> > > Hence, we believe this to be a concrete, non-trivial set towards getting this generalization error.

---

### Official Review · Reviewer_bT6s · 2025-11-12

**Soundness:** 3
**Presentation:** 3
**Contribution:** 3
**Rating:** 6
**Confidence:** 3

**Summary:**

This paper studied the asymptotic generalization error of the minimum-norm interpolation solution of linear regression when the data has a rank-one signal in the population covariance. Under the proportional limit, the authors derived a systematic description of the generalization error with respect to the aspect ratio, the signal strength, the alignment between target direction and data signal, model misspecification, and train–test covariate shift. This detailed analysis gives us a full picture of benign, tempered, and catastrophic overfitting regimes. For instance, when the target is aligned with the data signal, increasing signal strength can have a transition from
tempered overfitting to catastrophic overfitting, then to tempered overfitting, and then to benign overfitting,

**Strengths:**

This paper is well-written, and the statements are clear to me. This paper provides a comprehensive understanding of how varying signal strengths and target-input alignments impact risk in overparameterized settings. The transition between benign and catastrophic overfitting is an important and interesting topic for generalization theory and the ML community. It also includes extensive simulations for both linear and nonlinear networks.

**Weaknesses:**

The main limitation of this paper is the model assumption. This paper only studies a linear model, which may not generalize well to nonlinear models for the transition between benign and catastrophic overfitting. Besides, it only considers the minimum-norm interpolation solution. It would be more beneficial to generalize these results to the ridge regression case, where altering the regularization parameter may yield more diverse outcomes. Additionally, the regimes of benign, tempered, and catastrophic overfitting are highly intricate from a theoretical perspective. It would be beneficial to provide some heuristic explanations of why these phenomena occur differently in various cases.

**Questions:**

1. What is $(\tilde{X},\tilde{y})$ in Theorem 1? Do you only consider one test data point $(\tilde{x},\tilde{y})$? Please clarify the notion $\tilde{Z},\tilde{A}$ in all theorems. For instance, in Theorem 5, are you considering $\tilde{Z},\tilde{A}$ as vectors $\tilde{z},\tilde{a}$?

2. Do you assume $\alpha_A\neq \tilde{\alpha}_A$ in Theorems 3 and 4? Why is there no $\alpha_A$ in the limit of $\mathcal{R}_c$ in Theorem 4? It would be better to clearly state the equal operator norm and equal Frobenius norm conditions in Theorems 3 and 4, respectively.

3. Can you compare the test errors of training 3-layer ReLU networks in section 3.5 and linear regression on the same training and test datasets? Are the test losses for 3-layer ReLU networks always smaller than linear regression model?

4. Typo: line 1592: (Lemma 12

5. Where do you use Lemmas 13 and 15? Can you explain the application of the Gaussian hypercontractivity in the proofs?

6. Why do you need to assume $v$ in (1) has i.i.d. standard normal entries? In Assumption 2, do you need to assume independent entries of $A$, or is it uncorrelated enough? There should be a detailed discussion on the three conditions of Assumption 2. Can we use the first two conditions to imply the last condition of the Marchenko–Pastur law?

7. In (2), can you consider $\beta_*$ as a fixed vector? Why do we need to assume that it is uniformly distributed in some subspace?

8. In this paper, you focus on the overfitting regime. How about the underfitting case when $c\to 0$?

9. Is there a heuristic explanation of Figure 2a? Why could there be a region where the aligned risk is lower than the anti-aligned risk, but the aligned risk becomes strictly larger outside this region?

---

> ### Author Response · Authors · 2025-11-17
> **Response Part 1**
>
> Thank you for the constructive review and for highlighting the strengths of our work.
>
> - clear statements,
>
> - comprehensive characterization of benign/tempered/catastrophic regimes,
>
> - simulations for both linear and nonlinear networks.
>
> We address the concerns and questions below and indicate concrete clarifications/edits we will add.
>
> > Ridge Extension
>
> As in Li & Sonthalia NeurIPS 2024, one can analyze the resolvent away from zero. This leads to modified versions of Lemmas 9–10, whose formulas will become much more complicated. The remainder of the proof structure is unchanged but requires much more careful bookkeeping. Enough so that we think of it as future work.
>
> Experimentally, we considered the setting of figure 1a ($\theta^2 = c \tau^2$), added a ridge penalty, and numerically computed the error. We tried a variety of different regularization strength scalings.
>
> $\lambda = 0, \lambda = 1, \lambda = c, \lambda = c^2, \lambda = dc$
>
> In all cases, we observed catastrophic overfitting. Please go see the anonymous github repository (https://anonymous.4open.science/r/Alignment-Spike-45B6/Equal_Operator_Norm.ipynb) (also linked page 2 of the paper). Please go to the rebuttal folder and see the ridge-catastrophic images.
>
> > Intuition
>
> The most basic intuitions are
>
> large spikes in the *input data* are beneficial. However, moderate sized spikes can be harmful as they are not strong enough to completely “resolve” from the noise bulk and confuse the model (table 1)
> We want relatively balanced dependence between spike and bulk in the *targets*
>  (table 2)
>
> > Tilde variables
>
> We thank the reviewer for this question. We shall clarify this in the paper.
>
> $\tilde{Z}$ and $\tilde{A}$ are matrices of IID test data points. Specifically, we have $\tilde{n}$ data points. The proof then proceeds to compute the empirical test risk for these specific $\tilde{n}$ data points (averaged over $\tilde{A}$ and output noise). See appendix D.6 5 for the formulas.
>
> > Theorems 3 and 4
>
> We do not need $\alpha_A \neq \tilde{\alpha}_A$. The reason for this and why it doesn't appear in the limit of theorem 4 is that in all cases, we have $\alpha_A/c$, and in the $c \to \infty$ limit, it does play a crucial role.
>
> > Comparing Errors
>
> This is a great question. We computed the errors for the settings in Figures 4a, 4b, 4c. In all three cases the neural network consistently outperforms regression by a factor of at least 2.
>
> | Angle Degrees | 4a NN Error | 4a Regression Error | 4b NN Error | 4b Regression Error | 4c NN Error | 4c Regression Error |
> |-----:|---------:|---------:|---------:|---------:|---------:|---------:|
> | 0  | 1.2374 | 4.3810 | 0.3435 | 1.8119 | 11.5463 | 28.0090 |
> | 5  | 1.1240 | 4.5257 | 0.3454 | 2.0251 | 11.6999 | 27.8677 |
> | 10 | 1.1526 | 4.4140 | 0.3672 | 2.0649 | 11.4317 | 27.8909 |
> | 15 | 1.2120 | 4.4630 | 0.3909 | 1.9175 | 10.7866 | 27.2104 |
> | 20 | 1.1860 | 4.5461 | 0.4434 | 1.9958 | 10.1571 | 24.7542 |
> | 25 | 1.2111 | 4.0969 | 0.4894 | 2.0306 | 9.6305  | 24.3950 |
> | 30 | 1.1731 | 4.3161 | 0.5121 | 2.2355 | 8.9685  | 20.3954 |
> | 35 | 1.1831 | 3.9682 | 0.6155 | 2.1362 | 8.0515  | 20.8092 |
> | 40 | 1.1935 | 4.0369 | 0.6481 | 2.2856 | 7.6719  | 18.4733 |
> | 45 | 1.1998 | 3.1463 | 0.6728 | 2.2970 | 6.4977  | 16.1898 |
> | 50 | 1.2612 | 3.3239 | 0.7647 | 2.2603 | 5.3943  | 13.9266 |
> | 55 | 1.1569 | 3.0527 | 0.8353 | 2.3655 | 4.8540  | 11.0815 |
> | 60 | 1.1407 | 2.8119 | 0.8636 | 2.2217 | 3.7700  | 9.1020  |
> | 65 | 1.0733 | 2.5347 | 0.9101 | 2.2680 | 2.9010  | 7.9944  |
> | 70 | 1.0766 | 2.7368 | 0.9694 | 2.0943 | 2.4093  | 5.5262  |
> | 75 | 1.0323 | 2.4289 | 1.0003 | 2.4820 | 1.8578  | 4.1461  |
> | 80 | 1.0269 | 2.4491 | 1.0973 | 2.4277 | 1.4516  | 3.2216  |
> | 85 | 1.0218 | 2.1982 | 1.4179 | 2.2120 | 1.1435  | 2.4008  |
> | 90 | 1.0317 | 2.5067 | 0.9874 | 2.3849 | 1.0170  | 2.5209  |
>
> > Typo
>
> Thank you, we shall fix this.
>
> > Hypercontractivity
>
> We use Lemma 13 to prove Lemma 15, which is then used in the proof. Specifically, our error decompositions are products of terms that concentrate. That is error terms of the form $T_1 T_2 T_3 T_4$ where each of the $T_i$ concentrates, but the $T_i$’s are potentially dependent. (see Lemma 23 for one such example).
>
> To get that the product concentrates, having control on the variances of the $T_i$’s is insufficient. We need to control the 4th, 6th and 8th moments as well to get concentration (in particular, if we have the product of k terms, we need to go up to the 2k-th moment). This is exactly what Lemma 36 says.
>
> In particular, in Section D.4.2 (lines 1686-1710) we show that all terms that concentrate are polynomials of the form in Lemma 15. Hence, the variance bounds that we established help us generalize to higher moment bounds due to hypercontractivity. This further allows us to get bounds on the variance of the product of such terms.

---

> > ### Author Response · Authors · 2025-11-17
> > **Response Part 2**
> >
> > > Assumptions
> >
> > We do need the entries of $v$ to IID Normal. As long as the entries are from any distribution with mean 0 and variance 1, the proof goes through. In particular, if we don’t assume that $v$ is random and that it is just some fixed deterministic vector on the sphere of radius $n$, the proof still goes through.
> >
> > More specifically, suppose $A = U\Sigma V^T \in \mathbb{R}^{d \times n}$, we need $U$ to be Haar distributed. However, we do not need $V$ to be Haar distributed, *AND* $v$ to be isotropic Gaussian. In particular, what we want for $V^Tv$ is to be rotationally invariant, which we can achieve by assuming $V$ is Haar distributed or that $v$ is rotationally invariant and independent of $V$.
> >
> > Uncorrelated is sufficient. In particular, we just need $\mathbb{E}[AA^T] = \tau^2 I$.
> >
> > We don’t think conditions 1+2 imply the third condition. It is very much possible that they do. However, since we do not have a proof, we opted to explicitly state it as an assumption. If the reviewer knows of a result or paper that does show that implication, we would be happy to update the paper and remove an assumption.
> >
> >
> > > Assumption on $\beta_{star}$
> >
> > This is a great question and was a subtle issue that we had to deal with. Unfortunately, without the randomness in $\beta_{star}$ we could not get the Hypercontractivity arguments to go through cleanly. In particular, we need $\beta_{star}$, $Uu$, and $V^Tv$ to have randomness to get the concentration.
> >
> > > underfitting regime
> >
> > This is a great question: we take the limit and see what happens. For example, in theorem 1 we see that the excess error always goes to 0. This is expected for the well-specified, no covariate shift case, as we should get the true solution when getting infinitely much data.
> >
> > For theorem 3, we can consider the equal operator norm case $\theta^2 = \gamma \tau^2$. Here we see that if $\gamma$ is a constant then, we get tempered overfitting. However, if $\gamma = c^\kappa$ for any $\kappa > 0$, we get benign overfitting.
> >
> > > Figure 2a
> >
> > Unfortunately, we do not have a heuristic reason for that case.
> >
> > One possible speculative reason is the following. Specifically, the bias and variance are positive (hence harmful), whereas the target alignment is negative (hence beneficial). However, these terms scale differently, and asymptotically the error is dominated by the bias. For a certain amount of overparameterization, the bias might not have become big enough. So it could be like an over-parametered bias-variance-alignment tradeoff.

---

### Author Response · Authors · 2025-12-01
**To New AC**

Thank you to all the four reviewers for their careful, thoughtful, and technically engaged feedback on our submission. We are encouraged that the reviewers consistently highlight several strengths of our paper, which we provide a summary below:

- The precise asymptotic risk analysis of spiked regression (all the reviewers)

- The clear comprehensive classification of benign/tempered/catastrophic overfitting regimes (all the reviewers)

- The general relevance of target alignment and spike structure to advancing understanding of overparameterized models (Reviewers bT6s, ywoE, KtDe)

- Experimental extension to nonlinear models (Reviewers bT6s, uSkU)

Across the reviews, we noticed that the main concerns mainly cluster into four themes:

- Positioning and motivation of the data/target model

- Technical novelty beyond prior spiked and benign overfitting works

- Scope and robustness of the theoretical assumptions

- Empirical support and intuition for ML practitioners.

**Summary of Revisions**. Through new experiments (ridge regularization, deeper synthetic architectures, and MNIST with a modern network), new theoretical clarifications (mixed spherical hypercontractivity, higher- and negative-moment bounds, and detailed comparisons to prior spiked/benign overfitting works), and substantially expanded intuition and motivation, we believe we have **resolved all major concerns raised by the reviewers** regarding positioning of the model, technical novelty, robustness of assumptions, and empirical/practical relevance.

---

> ### Author Response · Authors · 2025-12-01
> **Global Response**
>
> **Here we show that we answered all of the reviewers questions. We hope that the AC agrees that we have sufficiently resolved the reviewer concerns.**
>
> **Ridge Regularization** There was also concern if the results transferred to ridge regularized setting. We empirically conducted an experiment with ridge regularizer and showed that the same catastrophic overfitting occurred. This addresses major concerns of Reviewers uSkU and ywoE.
>
> **Setting and Motivation**: Our setting models the simplest non-trivial polynomial target that separates bulk and spike contributions. Specifically, if $a$ is the bulk and $z$ is the spike, we consider the multivariate linear polynomial $y = \alpha_A a + \alpha_Z z$. In particular, for the single spike case, this is exactly the same form of polynomial in Moniri et al. 2024, while the bulk feature $F\_0=\sigma(WX)$ is not Gaussian like in our paper. Additionally, our setting can be viewed as a low-order Hermite approximation of nonlinear multi-index models, captured by our  $(\alpha_A, \alpha_Z)$-parameterization. This essentially makes our model generalizable to a broad class of nonlinear targets, and we can track its behaviors using linear tools from random matrix theory.
>
> This addresses a major concern of reviewer uSkU
>
> **MNIST**:  We also ran additional experiments on more realistic networks with skip connections, LayerNorm and GeLU. We also conducted this on MNIST data set. Here we see the same case, where changing $\alpha_A, \alpha_Z$ changes whether spike alignment is beneficial or not. This addressed the major concern of reviewer ywoE.
>
> **Empirical Support and Intuition**: The intricate phase transitions come from a tug-of-war between two positive terms (Bias, Variance) and one negative term (Target Alignment) in the error decomposition in Theorem 5. For instance, for the well specified case, (Appendix F, lines 3568-3575), target alignment completely cancels the variance, but not the bias, which then drives the catastrophic overfitting phase.
>
> This addressed the major concern of reviewer ywoE.
>
> **Technical Novelty**: Notably, prior works control only *first/second moments* via concentration to MP limits, but our analysis needs stronger concentration results (bounding higher moments and negative moments), i.e. *higher-moment control* to evaluate the risk *in closed form* under anisotropy, alignment, misspecification, and covariate shift.
>
> We obtain these by introducing *Mixed Spherical Hypercontractivity* (Sec. D.4.2). In particular, we have:
>
>  - A *new hypercontractivity result on product manifolds with random polynomial coefficients* (Lemmas 13 and 15).
>
>  - Bound on a variety of *new quadratic forms* (Lemmas 10–11)
>
>  - *Negative-moment bounds* absent from all previous works (Lemmas 18–20).

---

### Meta-Review · Area_Chair_nW8R · 2026-01-08

**Summary:**

The paper studies the generalization error (risk) of min-norm solutions to linear regression when the predictors come from a spiked covariance model and the response may be aligned to the dominant predictor direction. The paper considers a much more general setting than the prior works and is able to derive exact theoretical expressions for the risk, that are confirmed in simulations. The results show various interesting regimes such as benign/catastrophic overfitting arising in various situations.

There was large disagreement between the reviewers with scores ranging from 2 to 8. At the same time all reviews (both positive and negative) were of high quality. The concerns included model assumptions, motivation of some modeling choices, lack of citations to some of the prior literature, and "too much math" for a venue like ICLR.

**Reviewer Concerns:**

The authors provided a details rebuttal and ran some additional experiments, but unfortunately for some reason did not update the manuscript. Without an updated manuscript, I feel it is unlikely that the critical reviewers would have increased their scores...

Personally, I feel the two critical reviewers were unfairly critical. The paper seems technically very strong and derives lots of novel results. I actually agree that ICLR is not the best venue for such work, and would recommend to consider journals like either JMLR or Annals of Statistics, where I think this kind of work would be more appreciated. But ultimately it is the choice of authors of where to submit, and this work is definitely on-topic for ICLR.

So personally I tend to side with the positive reviewers.

One comment that I would like to add, is that there are more papers that seem very relevant and could/should be cited in the Introduction. Kobak et al. 2020 https://jmlr.org/papers/v21/19-844.html was one of the first papers to study min-norm solutions in a spiked model in a practical setting. I feel this work has to be cited and briefly discussed. Freeman 2025 https://arxiv.org/abs/2510.19206 is a very recent concurrent work that I recently came across; it would be great to cite it as concurrent and briefly discuss how it relates.

Another comment is that I would like to see some comments in the main text about the mathematical apparatus used in the Appendix for the proofs. What apparatus do the proofs use, and is it similar to some of the prior papers? It would be helpful to the reader to point that out. Note that I did not read the proofs.

**Reviewer Scores:**

The original scores were 2/2/6/8. All reviewers were high-quality. Unfortunately the authors did not update their manuscript, so I feel it is unlikely that the two critical reviewers would have raised their scores by much. Perhaps one of them would raise it to 4. So I feel the final scores would have been something like 2/4/6/8.

At the same time, I liked the paper and am siding with the positive reviewers. In particular, one critical reviewer criticized the paper as essentially too mathematical for ICLR, and while I can see this point, I feel it is fine to accept such papers. So despite two strongly negative reviews, I am recommending acceptance.

I really hope that in the event of acceptance, the authors will invest substantial work in revising the manuscript according to all the comments!!

---

### Decision · Program_Chairs · 2026-01-26

Accept (Poster)